# The Herpetofauna of the Insular Systems of Mexico

Víctor Hugo González-Sánchez [1,*], Jerry D. Johnson [2], Oscar Frausto-Martínez [1], Luis M. Mejía Ortíz [1], Alberto Pereira-Corona [1], María del Pilar Blanco-Parra [1,3,4], Pierre Charruau [5] and Carlos Alberto Níño-Torres [1,3,4]

[1] División de Desarrollo Sustentable, Universidad Autónoma del Estado de Quintana Roo, Cozumel 77600, Mexico; carlosalni@gmail.com (C.A.N.-T.)
[2] Department of Biological Sciences, The University of Texas at El Paso, El Paso, TX 79968-0500, USA
[3] Fundación Internacional Para la Naturaleza y la Sustentabilidad (FINS), Calle Larún Mz75-L4, Andara, Chetumal 77014, Mexico
[4] Consejo Nacional de Humanidades, Ciencia y Técnología, Ciudad de Mexico 03940, Mexico
[5] Departamento de Conservación de la Biodiversidad El Colegio de la Frontera Sur—Unidad Villahermosa, Carretera Villahermosa-Reforma km 15.5, Ranchería Guineo 2a Sección, Villahermosa 86280, Mexico
* Correspondence: biologovhgs@gmail.com

**Abstract:** The herpetofauna of the insular systems of Mexico is composed of 226 species, of which 14 are anurans, two are salamanders, and 210 are reptiles, comprised of two crocodilians, 195 squamates, and 13 turtles. Although the surface of the Mexican islands is only 0.26% of the Mexican territorial extension, these 226 species constitute 16.1% of Mexico's documented herpetofauna of 1405 species. We classified the Mexican islands into five physiographic regions: the islands of Pacific Baja California; the islands of the Gulf of California; the islands of the Tropical Pacific; the islands of the Gulf of Mexico; and the islands of the Mexican Caribbean. The highest species richness among these regions is in the Gulf of California, with 108 species, and the lowest richness is 40 for the islands of the Pacific Baja California and 46 for those of the Gulf of Mexico. We identified introduced species, risk of wildfires, climate change, and urban/tourist development as the main environmental threats impinging on these species. In addition, we assessed the conservation status of the native species by comparing the SEMARNAT (NOM-059), IUCN Red List, and the Environmental Vulnerability Score (EVS) systems. The comparison of these systems showed that the NOM-059 and the IUCN systems seriously underestimate the degree of threat for insular endemics, being particularly concerning for those insular species that are known only from their respective type localities. The EVS system proved to be practical and indicated that 94 species have a high vulnerability status, 62 a medium status, and 56 a low status. The Relative Herpetofaunal Priority system, which contrasts the number of endemic and threatened species among different physiographic areas, indicates that the regions with the highest priority are the Islands of the Gulf of California, followed by the islands of the Tropical Pacific. Finally, we discussed the completeness of the Mexican Natural Protected Areas on the insular systems of the country; the result is outstanding since Mexico is already close to achieving the goal of having all their islands under some degree of federal protection.

**Keywords:** islands; cays; archipelagos; endemism; EVS; Sea of Cortes; Gulf of Mexico; Mexican Caribbean

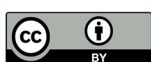

## 1. Introduction

Islands have become important model systems for scientific research in ecology, evolutionary biology, and biogeography by allowing the isolation of particular ecological factors and processes and the exploration of their effects [1]. In fact, it can be said that one milestone event in the coming-of-age of ecological science as a discipline with a theoretical/conceptual base was the publication of MacArthur and Wilson's Theory of Island Biogeography in 1967 [2]. In addition, many of the theories generated by island biogeography

have been extensively used (but not without controversy) in the understanding of the dynamics of discontinuous habitats or "insular like systems" [3], and have great importance in biodiversity conservation and management, since scientists and conservationists attempt to manage the impacts of habitat loss and fragmentation [1].

Islands cover 2.7% of the Earth's surface [4]. Despite their reduced surface compared with that of the mainland, they are hotspots for biodiversity conservation, combining the attributes of unique biodiversity, recent species extinction, and high risk of future species losses [1]. Often, islands are the refuge of lineages (relict species) that cannot survive the biotic pressure of most environments and only persist in habitats that their competitors or predators might have not reached [5]. Some remarkable examples of ancient relicts on the islands of Sonora are *Aspidoscelis ceralbensis*, *Sceloporus angustus*, and *Sceloporus grandaevus* [6].

Islands are also scenarios of the in-situ evolution of species with limited defensive or competitive abilities [5]. Thus, it is not surprising that islands harbor 61% of all species listed by the IUCN as extinct and 37% of species listed as critically endangered [7] and that most known animal extinctions occurred on islands [4]. This is critical for reptiles, of whom 90% of extinctions are insular species [8]. Moreover, the protection of island ecosystems constitutes a considerable challenge, not only ecologically, but also because of their fragmented nature, scattered across the globe and, generally below the horizon of media networks [1].

Mexico has a terrestrial extension of 1,964,375 km$^2$, of which 1,959,248 km$^2$ correspond to the continental surface, and only 5127 km$^2$ (0.26%) to islands [9]. This surface area is barely higher than the territorial extension of the smaller Mexican entities such as Ciudad de México (1485 km$^2$), Tlaxcala (3991 km$^2$), and Morelos (4958 km$^2$), and barely smaller than Colima (5625 km$^2$) and Aguascalientes (5589 km$^2$) [10].

Remarkably, Mexico encompasses 231,813 km$^2$ of territorial sea and 3,149,920 km$^2$ of exclusive economic zone [9]. The easternmost and westernmost territories of Mexico are islands: Isla Mujeres and Roca Elefante (Isla Guadalupe), respectively [11]. More importantly, the archipelagos Revillagigedo and Alacranes, and Isla Guadalupe play a key role because of their remote locations; they are the farthest extensions of the Mexican exclusive economic zone. Thus, despite their discrete contribution to terrestrial extension, the insular territories of Mexico are strategic in the conformation of the country's maritime limits [12,13].

The total of insular elements registered for Mexico, according with Instituto Nacional de Estadística y Geografía (INEGI) is 4111, most of them remaining unnamed. At least 3210 are true islands, whereas 1203 are in oceanic waters. The oceanic islands are 29.3% of the Mexican insular elements and cover 4529.7 km$^2$, which represent more than half of the Mexican insular surface. On the other hand, the coastal elements cover 3136.7 km$^2$. The great majority of the insular surface is in the Mexican northwest Pacific, especially in the Sea of Cortes. The islands of this interior sea constitute half of the Mexican insular surface [14] (Figure 1).

In a broad sense, any isolated habitat can be considered an island [5], but for the purpose of this work, we refer to the Mexican insular territory as the many insular elements that are part of the national territory, as stated in article 42 of the Politic Constitution of the Mexican United States, and article 121 of the United Nations Convention on the Law of the Sea (UNCLOS). These articles include the definitions of islands, reefs, and cays, as follows:

Island: A natural extension of land, surrounded by water, and situated above the high tide level. It includes small portions of land permanently surrounded by water, or scarped massive structures that are permanently emerged.

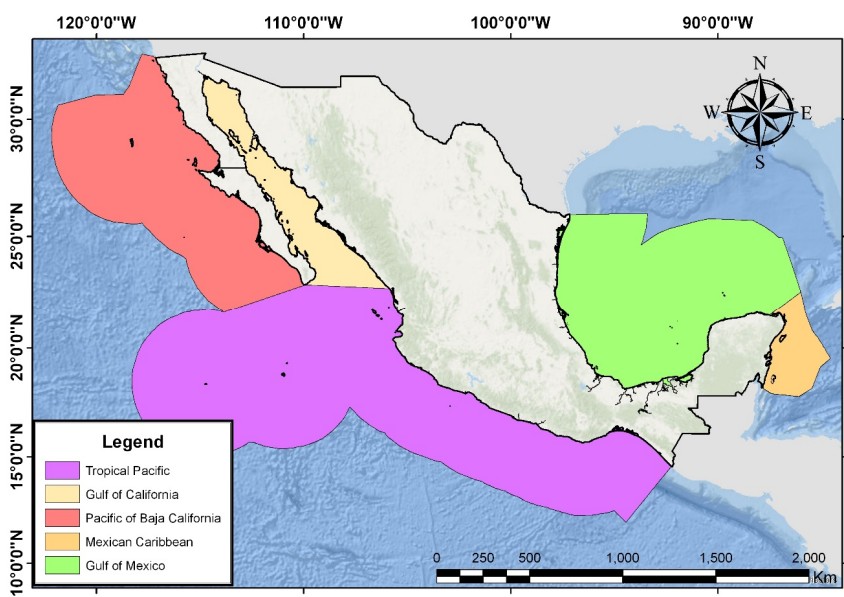

**Figure 1.** Mexican Insular Territory Regions, modified from Instituto Nacional de Estadística y Geografía (INEGI) (2015).

Insular reef: Rocky structures, generally of coralline origin, which emerge from the sea surface or are located at a very shallow level. Usually located near the shoreline.

Cay: Extension of land surrounded by sea water, located above the high tide level, derived of the accumulation of non-consolidated materials of calcarean nature, rocky or sandy texture with a permanent tropical vegetal cover, located mainly in the warm waters of the Caribbean Sea and Gulf of Mexico, whose formation dynamics are tightly linked to the coral reef systems. Also included in this category are the insular-like bodies, formed by aggregations of hydrophilic vegetation (i.e., mangroves) surrounded by sea water, which usually grow above banks of soft sediments with muddy and waterlogged soils, sometimes barely under the high tide level, and which are common in the littoral and lagunar systems of the Gulf of Mexico and the Caribbean Sea.

We also adopt the terms islet (for a small portion of land, often unnamed and surrounded permanently by water, that generally was part of a larger island or continent and, due to erosion, ended up separated from it), rock (a small and escarped rocky structure permanently emerged), and insular bar (formed from an underwater bar in the process of moving toward the coast, and its subsequent outcrop above sea level), according to the definitions provided by Aguirre-Muñoz et al. [15]. Another necessary term is inland islands, which we consider as those masses surrounded by freshwater and mainly located within a river or lake on the mainland. We do not include the herpetofauna of the inland islands in this study, but sometimes this term is referenced in the text. We do include, however, the Tamalcab island, due to its federal jurisdiction (see the protected areas section), and since it cannot be considered entirely as inland island since it is within a coastal lagoon.

Based on their origin, the insular elements are categorized as follows:

- Volcanic islands: Originated by the volcanic activity in the ocean, often (but not always) related with plate tectonics [16]. This is the case with most of the most remote islands, such the Revillagigedo Archipelago.
- Tecto-orogenic islands: Frequently, they are the result of the collision of lithospheric plates, that lead to the uplift of oceanic mountains due to compressional forces [16]. This origin is the case for the majority of the Mexican islands connected to the mainland by the continental platform, like the Coronado Archipielago, San Marcos, San

Ildefonso, and Cerralvo, although some of these islands were created in conjunction with volcanic or erosional processes [17].

- Sedimentary islands: Formed due to the accumulation of sand, mud, and other sediments dragged by the current of rivers to the oceans and other tidal processes; these events only occur in shallow shelf areas that allow the accumulation of sediments [16]. Examples are the islands of Laguna de Términos, the Montague Island in the Delta of Rio Colorado, or the barrier islands on the coast of Tamaulipas.
- Coralline islands: Result of the accumulation of coralline material. Besides the zoogenic factor, two other conditions are critical, such as shallow marine platforms and water temperature; thus, these islands are more frequent in warm oceans [16]. Cays of Banco Chinchorro and Arrecife Alacranes are examples of this kind.

Besides these four categories, Sockman [16] distinguished five types of islands by secondary processes of isolation; of these, the most relevant for the purposes of our work are the dislocation islands, which are formed as a result of horizontal or vertical tectonic shifts. The vertical tectonic shifts can lead to the lifting of blocks, giving rise to a horst island (such as Cozumel). While horizontal tectonic movement leads to the separation and migration of peripheral parts of a lithospheric plate, which generates drift islands, these are only drifted fragments of the mainland forming part of the respective continental crust. Clear examples are many of the islands in the Gulf of California.

Given their location, the insular elements will be placed into two main groups: (1) coastal or continental, i.e., formed on the continental platform, and (2) oceanic, i.e., those located in the marine zone, above basements formed beyond the neritic zone of the continental platform, deeper than 200 m [14,18]. A third kind are the inland islands (*terrestres*), or those located within the mainland [18], which are not considered in this paper.

1. Oceanic islands have never had a connection with continental landmasses, so they emerged from the seabed as an isolated land mass, and are the result principally of tectonic uplift, volcanic activity, or reef coralline formations (when these formations grow because of these two other processes) [5,19]. Oceanic island biotas originated almost solely by dispersal, with an obvious tendency toward a more depauperate biota the more isolated the island is, but this isolation might lead to speciation and, in archipelagos, to specific radiation [5]. Remarkably, the insular elements of the marine zone are 3.8% of the Mexico's total islands tota, but they constitute 29.3% of the surface of the insular territory due to their great size [14].

2. By contrast, most continental islands were joined in the past to the continental landmasses, and were separated due to tectonic dynamics or, mostly due to sea level rise following lowered levels during the Pleistocene glaciations. Thus, the rocks of these islands are like those of their parent mainland. When the island is separated from the mainland, it contains a fraction of the continental biota; usually the biodiversity of the island declines as an adjustment of a smaller and isolated environment [19]. The splitting of a species' population by this process often leads to speciation due to vicariance.

We note that we do not consider as true islands those that have lost completely their isolated condition from the mainland, such Cancún, which urban development has converted now into a peninsula, or San Juan de Ulúa, destroyed by a military fort built on it and attached to the mainland by several adjacent constructions (such as a shipyard and others) built above land won from the sea.

In addition, we omit those small islands that are an extension of a larger island, that is, that are separated by shallow waters, as well as too close to each other. This situation would be the case for Isla la Pasión, which can be considered an integral part of Cozumel (this island is not related to the "Isla de la Pasión" also known as "Isla Clipperton," which is a French possession in the Pacific Ocean), Isla San Juanito, which is associated with María Madre Island, and Islote Pelón and Las Monas islets, which are part of Isla Isabel,

among many others. In these examples, the smaller island cannot be considered a "true" island, independent of the larger island.

Also, it is important to remark that article 48 of the Constitution of the United Mexican States explains that all islands, cays, and reefs inside the Mexican territory or territorial seas are under federal administration, with the exception of those that were administrated by any state before the date of the promulgation of the Constitution (1917) [12]. Cozumel is an example of this latter case, as it is administered by the state of Quintana Roo (although this state was created in 1974, Quintana Roo has existed as a federal territory since 1902) and it is explicitly mentioned as an integral part (among Cancun, Mujeres, Blanca, Contoy, and Holbox) of Quintana Roo in the constitution of that state. Of the 17 Mexican states with marine littoral regions, the great majority does not make any mention of the insular elements, besides Quintana Roo, only the constitutions of Sonora (Tiburón, San Esteban, and Lobos) and Baja California Sur (Natividad, San Roque, Asunción, Magdalena, Margarita y Creciente, Cerralvo, Santa Catalina or Catalana, San Juan Nepomuceno, Espíritu Santo, San José de Santa Cruz, del Carmen, Coronados, San Marcos, and Tortugas) mention specific islands as a part of their territory. Whereas, Campeche, Veracruz, and Nayarit's constitutions only refer to "adjacent islands" or those "corresponding to the article 48 of the General Constitution of the Republic," independent of these constitutions, those states cannot exert sovereignty over these islands if they did not have effective jurisdiction over these islands prior to the promulgation of the regent federal constitution in 1917 [13]. In this sense, the validity of Quintana Roo's claims over their islands is questioned, since the state constitution is more recent than the Mexican Constitution. As a matter of fact, those islands are strongly associated with the history, culture, and economy of Quintana Roo, and the Mexican Federation has never disputed the administrative sovereignty of those islands [20]. Additionally, according to article 1 of the Ley General de Bienes Nacionales, the islands, cays, and reefs in the adjacent seas are part of the nation's patrimony [21]. Very few islands in Mexico are private; the most notable is Cerralvo, but also Carmen (in the Gulf of California, not the barrier island of Campeche), San José, Macapule, Vinorama, and Huivulai in the Gulf of California. Cayo Venado and La Pasión (the islet next to Cozumel) in the Caribbean have an unclear property status, with certain persons claiming ownership. Espiritu Santo and Partida are under ejidal regimen, whereas Tiburón is under community administration (by the Seri people) [22]. As can be noted, most of the Mexican islands are under federal administration. Consequently, any reference to the islands of any state must be interpreted as a reference to proximity (i.e., "islands of the coast of Veracruz," "islands of the Pacific of Baja California," "cays of Quintana Roo," and "barrier islands of Tamaulipas"), but that does not necessarily imply that the island "belongs" to that state.

Finally, we employ other terms such as "Territorial Sea" and "Exclusive Economic Zone," and statements about sovereignty of marine areas, as stated in the Convention on the Law of the Sea (UNCLOS).

*Comment*

The lead author began to draft this manuscript in late 2017, after writing a comprehensive listing of the Mexican Yucatan Herpetofauna and providing listings from the regional insular systems for the Mexican states of Campeche, Yucatan, and Quintana Roo. However, while this paper was being prepared for submission, Pliego-Sánchez et al. [23] published a paper with similar goals and objectives to ours. For a while, we were dubitative if our paper should or should not be published. However, after a two-year dormancy and revising both manuscripts cold-headed, we found that both papers, although similar in goals and objectives, differ in their approaches and taxonomy, and that there are still so many differences that makes our work still worthy of publishing.

1.  First, the taxonomic approach differs widely: Pliego-Sánchez et al. [23] follows the names indicated in the Amphibia Web and ReptileDatabase. On the other hand, as

this paper was intended originally to be part of the Mesoamerican herpetofauna conservation series (MCS), it largely follows the nomenclature provided in the mesoamericanhepetology portal. Also, we do not consider subspecies as a valid taxonomic entity, while Pliego-Sánchez et al. [23] does. The reasons for that have been discussed in previous papers by Johnson et al. [24,25]. This introduces important differences in the listings and diversity values we provide. However, after this paper was submitted, Ramírez-Bautista [26] published an updated taxonomic list for the Mexican herpetofauna. We made the effort to match that taxonomy.

2. Second, the regionalizations are different: Pliego Sanchez et al. [23] (2021) uses a regionalization based in Morrone [27], and they divide the insular systems into neartic and neotropical ones, and then subdivided them into Californian, Baja Californian, Sonoran, Pacific Lowlands, Veracruzan, and Yucatan Peninsula Provinces. We, on the other hand, classify the insular systems by using physiographic regions, largely based in INEGI (with some modifications, as mentioned in the physiographic regions description).

3. Third, we present a different analytic approache: the Coefficient of Biogeographic Resemblance (CBR), whereas, Pliego-Sanchez et al. [23] present a taxonomic turnover value (Bsim dissimilarity) and regressions of species' richness against area and distance to mainland.

4. Fourth, as with other papers we present a detailed discussion regarding how this herpetofauna is represented in the Mexican System of Natural Protected Areas.

5. Fifth, we provide the full listing for every island with herpetofaunal records, which is absent from the Pliego-Sanchez et al. [23] paper (we do not know the reason).

An important thought that motivated us to take this paper off from dormancy is that there are many examples of similar cases when different authors presented biological listings of some regions but differ in their approach. For example, McCranie [28] presented his work on Honduran herpetofauna and shows several taxonomic differences with the Solís et a al. [29] paper for the reptiles and amphibians of that country. In the same way, Lemos-Spinal and Dixon [30] presented a book on herpetofauna of the Mexican state of Hidalgo and show differences with a similar book from Ramírez-Bautista et al. [31]. Also, there are several checklists made for this state in the 2010s [31–33]. Other examples include the classical books of Campbell [34] and Julian Lee [35,36], which deal with the Herpetofauna of the Yucatan Peninsula and are similar in intention and structure, but have many differences in their writing and narratives. With these examples in mind, we consider that the Pliego-Sanchez et al. [23] paper and ours have many differences in taxonomy, approach, analysis, and discussion, and that both papers provide a complementary view of the knowledge of the Mexican insular herpetofauna that are worthy of being read by any herpetologist with an interest in Mexican insular herpetofauna.

## 2. Materials and Methods

We compiled a list of the insular reptiles and amphibians in Mexico from an exhaustive revision of the available literature for the region. This review is supplemented with records obtained from the Global Biodiversity Information Facility (GBIF), iNaturalist, and Vertnet platforms. Of these, we considered only records with research quality and properly georeferenced, as well as those from the databases of CONABIO in Mexico (National Commission for the Understanding and Use of Biodiversity).

### 2.1. Our Taxonomic Position

Scientific names are based on Wilson et al. [24,25] and Johnson et al. [37], along with the most recent lists in the Reptile Database (http://reptile-database.reptarium.cz/search accessed on 30 April 2023), Amphibian Species of the World (http://research.amnh.org/vz/herpetology/amphibia/ accessed on 30 April 2023), the taxonomic list of Mesoamerican Herpetology (http://mesoamericanherpetology.com/taxonomic-

list.html) (accessed on 30 April 2023), and Ramírez-Bautista et al. [26]. Common names, when appropriate, follow Liner and Casas-Andreu [38]. For the purpose of this study, we adopt the posture of Johnson et al. [37] and do not consider the subspecies taxon. We understand that nomenclatural changes will happen regularly during future taxonomic revisions.

### 2.2. Our Toponomic Position

The toponomy of islands is particularly complex; often islands are known by several names and have synonyms or are known with different names throughout other periods of history [39], or by an original indigenous name [40], or are named differently by the inhabitants of the islands than their neighbors from the mainland [41]. This situation is predominantly problematic with small islands, since they are primarily uninhabited, and fishermen and local people refer to them only as a reference point. Thus, they often give different names to the same island, depending on whether the morphology of the island resembles something (Mamut), if there is a specific animal species abundant there (the many cays, rocks, islets named "Lobos"), or by a local event. The main reference we use is the Catalogo del Territorio Insular Mexicano by the Instituto Nacional de Estadística y Geografía (INEGI) [14]. In the case of widely known synonyms, we put them into parentheses in the tables. We do use, however, some exceptions to the INEGI nomenclature, since we give prevalence to the Spanish denominations. The synonyms are preferred in cases where homonymies are present to avoid confusion. A historical particularity occurs in the case of Cerralvo because we chose to preserve the historical character of the name.

### 2.3. System for Determining Distributional Status

We used the same system for the determination of the distributional status of the members of the herpetofauna of the insular systems of Mexico, as was originated by Alvarado-Díaz et al. [42] for use with the herpetofauna of Michoacán, and has been widely used since by the many other herpetofaunistical listings which have also utilized this system. This system, as modified for the purposes of this paper, consists of the following four categories: IE = Insular Endemic (Please be aware that this indicates the species is also exclusive to the region); CE = endemic to Mexico; NE = not endemic to Mexico; and NN = non-native to Mexico.

### 2.4. Systems for Determining Conservation Status

To assess the conservation status of the herpetofauna of the insular regions of Mexico, we employed the same systems (i.e., SEMARNAT, IUCN, and EVS) as used by Alvarado-Díaz et al. [42]. Several other herpetofaunistical papers [33,43,44] use these three systems and provide detailed descriptions and do not need to be repeated here.

### 2.5. Insular Physiographic Regions

We recognize five insular physiographic regions in Mexico, as indicated in Figure 1, and as described below.

#### 2.5.1. Islands of the Pacific of Baja California (Islas del Pacífico de Baja California)

These islands are located in the Pacific, off the western coast of the Baja California Peninsula (Figure 2). Most of them are of continental origin, but two, San Martín and Guadalupe, are of volcanic origin [45]. Located here is the northernmost island of Mexico, Coronado Norte (32°26′20.426″ N) [14], belonging to the Coronado Archipielago. This island is located a few kilometers (~8) south of the Mexico–USA line of the territorial sea. Additionally, Isla Guadalupe (118°17′ W, ~260 km from the continent) is the westernmost territory of Mexico, specifically the Roca Elefante at 118°27′24″ [11]. Also existing in this region are the Rocas Alijos, or Escollos Alijos, which are a group of small islets of volcanic origin [46], recently renamed as Islas Alijos [14,47], located in this region at ~370 km from

the Baja California Peninsula [47,48], with three principal emerged rocks and several smaller submerged (subtidal) or only temporarily emergent ones. These rocks are rich in submarine invertebrates and microbiota and serve as refuges for seabirds and marine mammals, but reptiles are absent from this place [46].

These islands are very dissimilar in altitude; by far, Guadalupe and Cedros have the higher altitudes of the region (and of all the Mexican islands, only followed closely by Isla Socorro, from the tropical islands of the Pacific), with 1298 m and 1204 m, respectively. With the exception of Santa Margarita (566 m) and Magdalena (338 m), all other islands have their highest points at approximately 200 m or less [45].

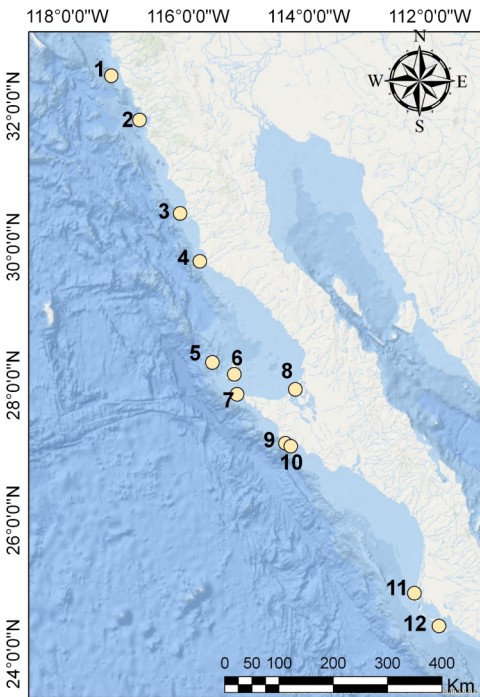

**Figure 2.** Islands of the Pacific of Baja California: (1) Coronado Archipielago, (2) Todos Santos Archipielago, (3) San Martín, (4) San Jerónimo, (5) San Benito Archipielago, (6) Cedros, (7) Natividad, (8) Islands from Laguna Ojo de Liebre (Las Brozas, Piedra and Pata), (9) San Roque, (10) Asunción, (11) Magdalena, and (12) Santa Margarita.

Due to their rocky origin and small size, the islands Asunción and San Roque lack almost any vegetation, whereas the coastal and desert shrub and dunes vegetation are the dominant vegetation types of most islands (Figure 3), with the exception of Cedros [45]. Nonetheless, the introduced ice plant (*Mesembryanthemum crystallinum*) and mallow (*Malva parviflora*) have been consistently observed on the islands of the region [4,49] In numerous instances, they exhibit a high level of dominance (pers. observ.).

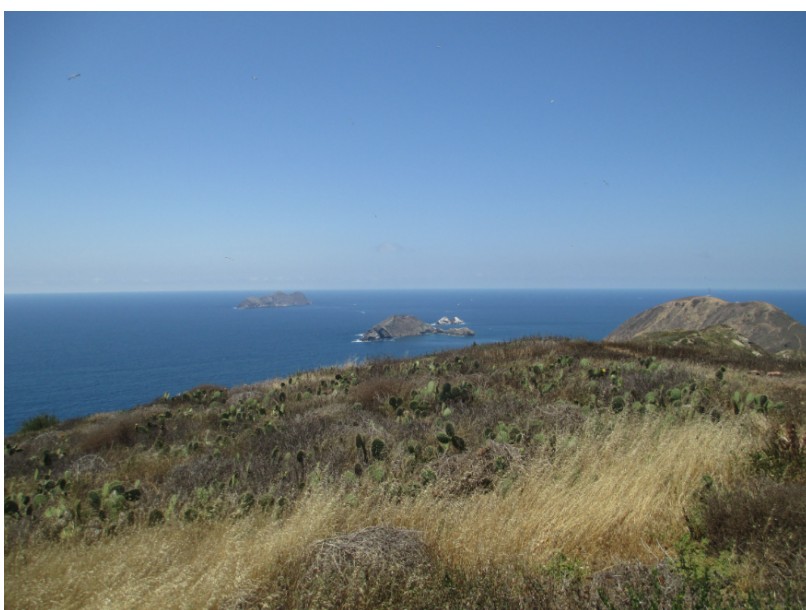

**Figure 3.** View from Coronado Sur. The Coronado are an archipiélago of four islands of continental origin. The dominant vegetation is coastal shrub, with *Opuntia* spp. But the iceplant is highly dominant, which is explainable due to decades of overgrazing by feral goats, who cleared extensive areas from native vegetation (fortunately, nowadays no feral goats live there).

2.5.2. Islands of the Gulf of California (Islas del Golfo de California)

These islands are confined by the Sea of Cortes (Mar de Cortéz), an interior sea, from the basin of Rio Colorado to the Los Cabos region, with a length of ~1100 km and an average width of between 108 and 234 km (Figure 4). These islands are limited on the west by the Baja California Peninsula, and on the east by the states of Sonora, Sinaloa, and Nayarit [50]. The Sea of Cortés has one of the most extended and complex geological origins of any Mexican region, but, in summary, it can be said that it formed by the tearing of the Baja California Peninsula from continental Mexico due to continental drift, principally during the Miocene and the Pleistocene, processes that still continue today [17,51]. Thus, most of the insular elements of the area are nothing other than extensions of land that became isolated in the sea by this process. However, some others are the result of volcanic activity, such as Tortuga, Espiritu Santo, and Coronado [17,52]. A few in the upper portion of the Gulf of California are the result of sedimentary depositions from the Delta of the Río Colorado, such as Montague, Pelícano, and Gore. These islands are very young; most of them from the Holocene [17].



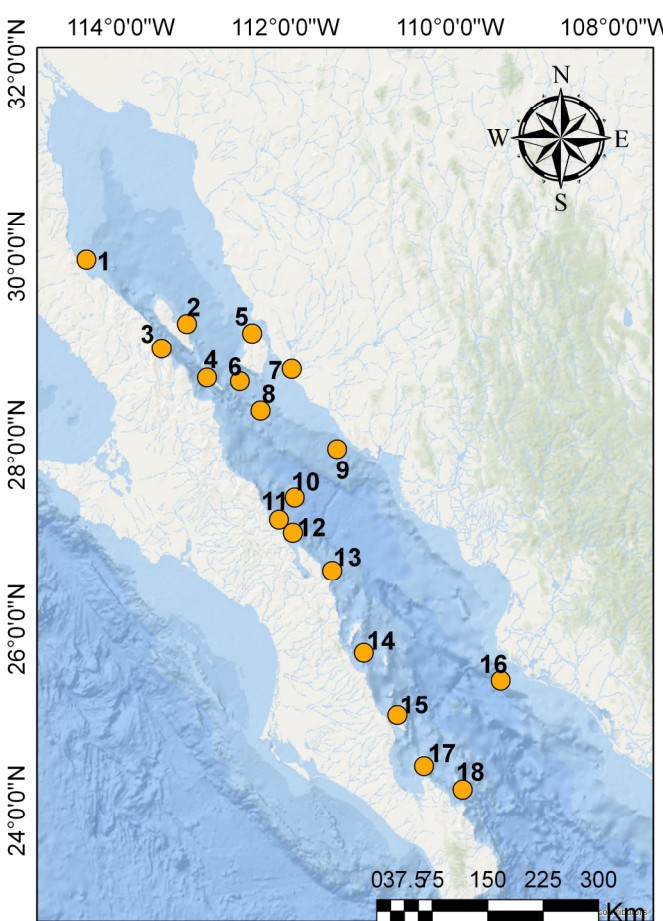

**Figure 4.** Islands of the Gulf of California: (1) Las encantadas Archipielago (El Muerto, Coloradito, Encantada, Blancos, San Luis, San Luis Gonzaga), (2) Isla Angel de la Guarda and associated islands (Tiburón, Mejía, Granito and Estanque), (3) Islands of Bahía de los Ángeles (Bota, Cabeza de Caballo, Cerraja, Flecha, Ventana, Mitlán, Pata, Piojo, Coronado (Smith)), (4) Archipielago de San Lorenzo (San Lorenzo, Las Ánimas, Lagartija, Salsipuedes, Rasa, Partida Norte, Roca Cardonosa, Blanca), (5) Tiburón and asociated islands (Tiburón, Datil, Cholludo and Patos), (6) San Esteban, (7) Alcatraz, (8) San Pedro Mártir, (9) San Pedro Nolasco, (10) Tortuga, (11) San Marcos, (12) Santa Ines, (13) San Ildelfonso, (14) Archipielago de Loreto (Coronados, Carmen, Danzante, Montserrat, Santa Catalina, Coyote, Tijeras), (15) Archipielago de Loreto (San Francisco, El Pardito, Las Animas, San Jose, San Diego, Santa Cruz), (16) Farallón de San Ignacio, (17) Arrecife de Espíritu Santo (Gallina, Gallo, Ba-llena, Espíritu Santo, Partida Sur), and (18) Cerralvo.

This region, according to INEGI [14], is the second one with the most insular elements in Mexico, with a total of 1003 (including islands, islets, and rocks), but it is, by far, the one that contributes most to the Mexican insular surface, with ~50% of this surface, due to the size of its islands. It must be noted, however, that the area indicated by INEGI [14] is extended 100 km south of Los Cabos to include the Islas Marias, which we include in the Tropical islands of the Pacific region. Most of the islands in this region are barren, without water, and uninhabited, and the sporadic and sparse rainfall allows for little vegetation, primarily desert shrubs, and cacti [52] (Figure 5).

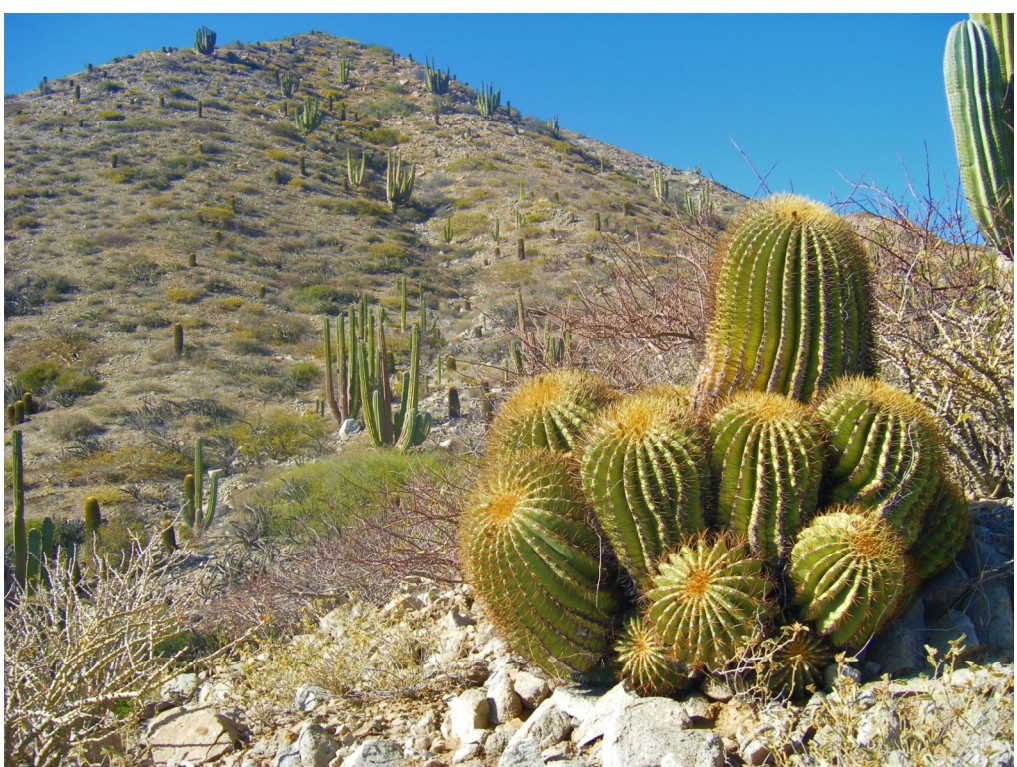

**Figure 5.** Santa Catalina is a rocky island of continental origin on which tropical dry forest is the dominant vegetation. With the remarkable presence of the Santa Catalina barrel cactus (*Ferocactus diguetii*, in the first plane), an endemic plant of the Sea of Cortes whose tallest specimens inhabit this island. Here live several endemic reptiles, one of the most intriguing is the rattleless rattlesnake (*Crotalus catalinensis*). Photo by Rubén Alonso Carbajal-Márquez.

### 2.5.3. Tropical Islands of the Pacific (Islas Tropicales del Pacífico)

We group in this region all islands on the Mexican Pacific south of the tropic of Cancer (but below the Los Cabos region), up to the Suchiate Basin, on the coast of Chiapas (Figure 6). We mention, however, only tangentially the elements of the coasts of Chiapas and Oaxaca (grouped by the INEGI as islands of the Tehuantepec Gulf), since the majority of them are elements of coastal lagoons [14], or which have unknown herpetofaunas.

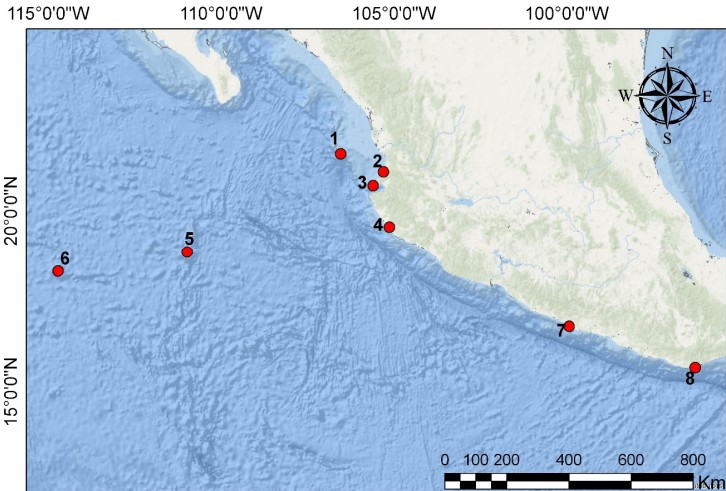

**Figure 6.** Islands of the Tropical Pacific: (1) Marías Archipielago, (2) La Peña (Coral), (3) Islas Marietas, (4) Islas de Bahía de Chamela, (5) Revillagigedo Archipielago (Socorro, San Benedicto, Roca Partida), (6) Clarión (Part of Revillagigedo's), (7) La Roqueta, and (8) Ixtapa.

The most conspicuous insular elements in this region are of volcanic origin. This includes the important islands of the Revillagigedo Archipielago, which are part of a Quaternary submarine volcanic mountain range, from which the emerged peaks constitute the Revillagigedo Islands. Notably, Socorro (Figure 7) is the largest island of the region, with an altitude of 1130 m, and dimensions of 39 km long and 14 km wide, and is the emerged portion of Evermann Volcano, whose last eruption was in 1993. Clarión, San Benedicto, and Roca Partida, are also emerged peaks. San Benedicto's last eruptions were in 1948 and 1952 [53].

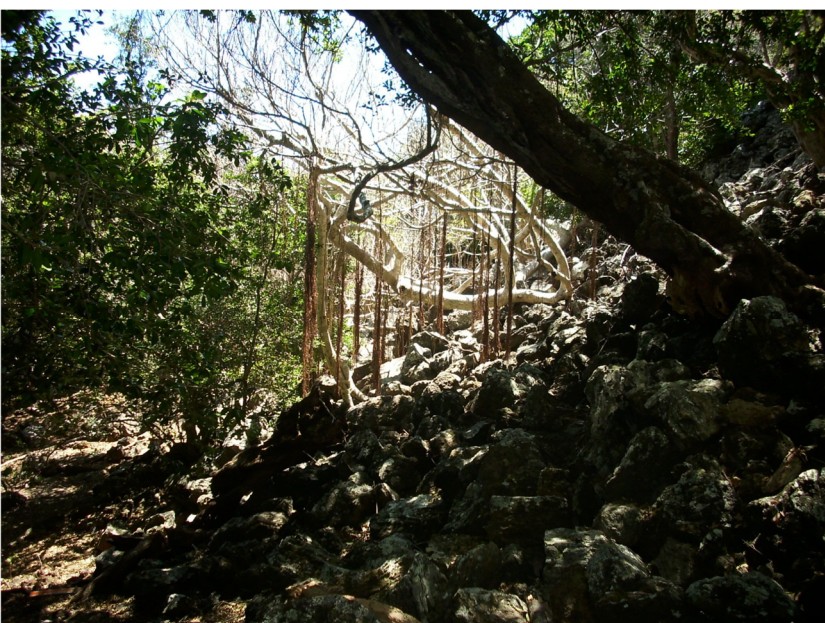

**Figure 7.** Socorro, just like the other Revillagigedo Archipielago islands, is a tropical oceanic volcanic island (the Evermann volcano originated this island). It possesses an outstanding diversity of endemic plants. Thus, most of its biomes are unique when compared to the rest of the world. Unfortunately, most of the island has eroded or turned into grassland across generations of herding sheep (the sheep were removed from this island during the mid-2000s). But some relicts of native biomes still survive on the higher slides of the volcano, like this path of manchineel (*Hippomane mancinella*) forest. Photo by Victor Hugo González-Sánchez.

Isla Isabel is another example of a volcanic island, but unlike other Pacific volcanic islands such as Socorro, Bárcena, and Guadalupe which formed from intraplate-type volcanic rocks, Isabel formed in extended continental crust. The crater of the emerged Isabel is very conspicuous. Also, the nearby Islote Pelón is the remnant of a tuff cone. The islets Las Monas probably are the remnant of another explosion crater [54].

The Islas Marías are a group of four islands: San Juanito (sometimes referred to as San Juanico), María Madre, María Magdalena, and María Cleofas. In the literature, generally only the last three are referenced, usually as the "Tres Marías," since San Juanito is considered by some authors as an extension of Isla María Madre [55]. The origin of these islands is still a matter of controversy, but it seems that it is tightly associated with that of the Los Cabos region, in the Baja California Peninsula [56].

2.5.4. Gulf Islands (Islas del Golfo de México)

This region extends from the mouth of the Río Bravo/Grande on the Mexico–USA border with the Caribbean, close to the Yucatán–Quintana Roo border, thus encompassing the shorelines of Tamaulipas, Veracruz, Tabasco, Campeche, and Yucatán (Figure 8). The principal characteristic of this area is the wide and extensive continental platform, known as the "sonda de Campeche". According to INEGI [14], 1216 insular elements have been

catalogued in this region, the vast majority of which are located on the continental platform.

Most islands in this region are the result of the influence of riparian systems, and/or barrier systems in coastal lagoons. Most notably, the Laguna de Términos has many rivers and streams flowing into it near Ciudad del Carmen. The sand and mud the currents transport to sea have led to the accumulation of this material and with time have created the many islands and coastal lagoons in the region of Laguna de Términos, the most important of them being Isla del Carmen [57]. Despite the important alteration occasioned by the city of Ciudad del Carmen, the Laguna de Términos system is one of the most important biodiversity areas in Mesoamerica [58]. Very often this dynamic creates a long chain of enlarged islands that form a barrier between the coastal lagoon and the sea, this generates an enclosed or nearly enclosed system with an environment of low-level forces such the wind or currents; this action favors the accumulation of sediments of many sources, leading to the creation of sandy islands inside this enclosed system [59].

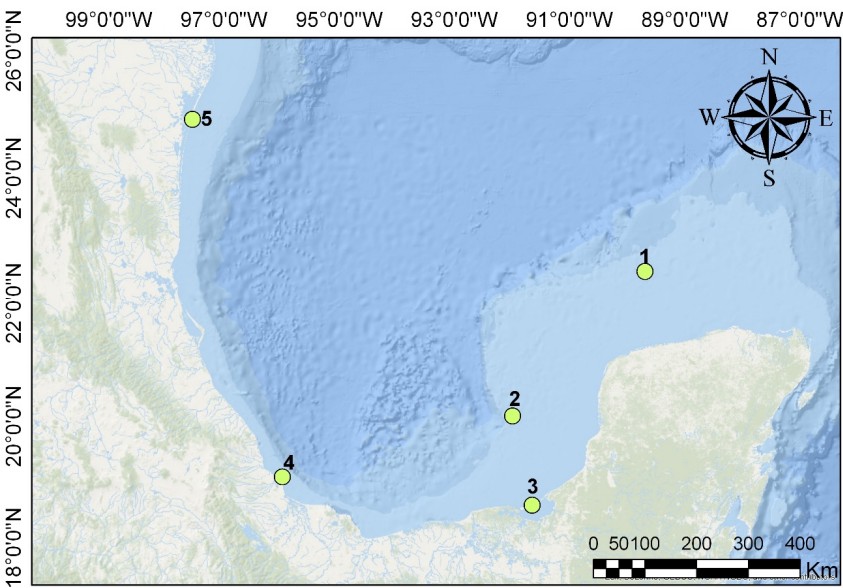

**Figure 8.** Islands of the Gulf of Mexico: (1) Arrechife Alacranes, (2) Cayo Arcas, (3) Carmen and islands of the Laguna de Terminos, (4) Sacrificios and Cays of the Sistema Arrecifal Veracruzano, and (5) Islands of Laguna Madre.

Another remarkable example is the coast of Tamaulipas, particularly where the 2000 km²-long Laguna Madre of Mexico hosts an estimated six hundred such islands; many of these islands are unstable and have different degrees of consolidation or ages. Also, many can be considered "relictual" elements, thus the cartography and/or categorization of these islands is difficult [59]. As can be inferred, most of them remain unnamed [14].

The coralline islands are present in this region as well, although they can be found all along the Gulf of Mexico sinus. The most remarkable is the Arrecife Alacranes (Figure 9), a false atoll, or better said, a reef platform formation created by the accumulation of calcareous material of coralline origin that comprises an area of ~300 km², with five clearly distinguishable sandy islands with an area of 530,407.78 m² (0.53 km²), or 1.7% of the whole reef system area [60]. The vegetation on the Alacranes seems to follow an interesting gradient of vegetal succession from sea grasses in the shallow waters of the litoral region to succulent and herbaceous plants dominating the coastal dunes, to shrubs in the interior [57].

Another important coralline system comprises the reefs and cays off the coast of Veracruz, most importantly, the islands belonging to the Sistema Arrecifal Veracruzano. These consist of six calcareous sandy cays with a total area of 12.24 ha, with the Isla de

Sacrificios being the largest one with an area of 5.24 ha. As with most of the coralline islands, the dominant vegetation is coastal dune and halophilic vegetation like mangrove [61].

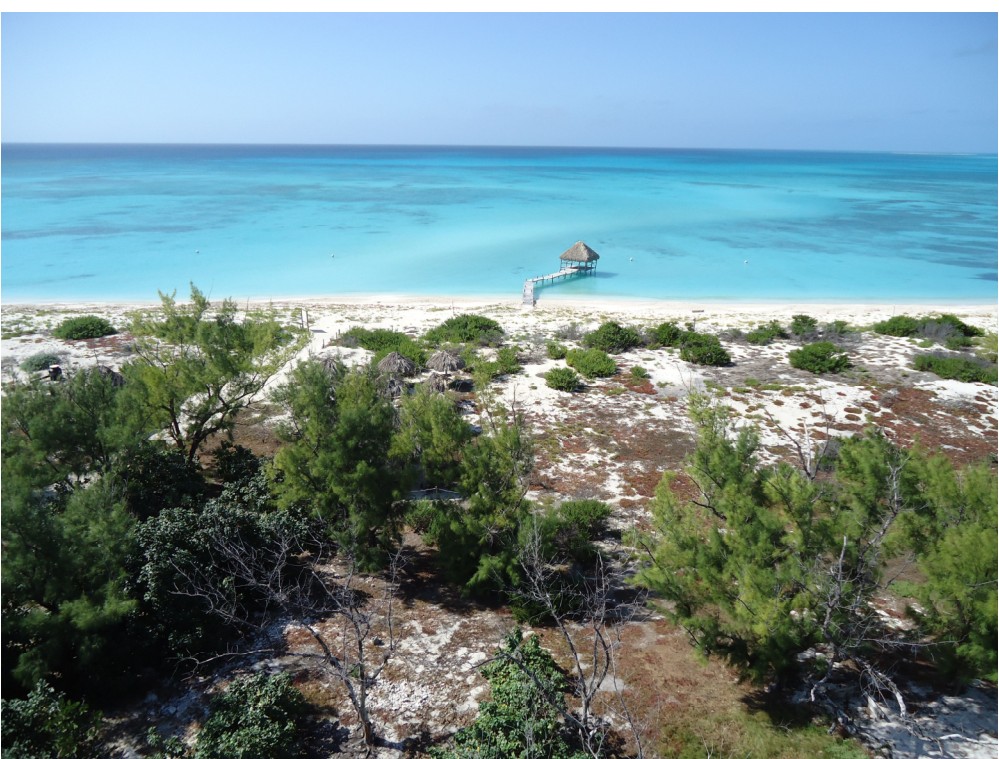

**Figure 9.** Isla Pérez (Arrecife Alacranes) is an outstanding example of oceanic islands of coralline origin. The vegetation on these islands is interesting since walking through a linear transect from the shore to the center allows us to see how the island is covered by several plant associations that can be considered to be different successional stages, from the sea grasses (*Thalassia* and *Halodule*) in the shallow waters near the coast, to the herbaceous and succulent plants dominating the coastal dunes (*Sesuvium*, *Portulaca*, *Sporobolus*, *Cenchrus*, *Chamaesyce*, and *Batis*), and to the coastal scrubs dominated by species of the genera Tournefortia and Suriana. Some introduced plants can be found there, such as species of *Opuntia* and *Casuarina.* Photo by Rigel Sansores.

2.5.5. Caribbean Islands (Islas del Caribe Mexicano)

The Caribbean Sea is a semi-enclosed sea of the Atlantic Ocean in Mexico and is restricted to the eastern coast of the Yucatan Peninsula from Holbox to the Xcalak Peninsula, located in the Bacalar Chico Channel [14] (Figure 10). This region is almost equivalent to the entire length of the shoreline of the Mexican state of Quintana Roo [57]. An estimate of ~845 insular elements have been catalogued in this region [14], as expected since this area is host to an important fraction of the Mesoamerican reef system [62] (Ardisson et al., 2011). The great majority of these insular elements are cays or reefs (30 and 51%, respectively), with an estimated proportion of ~88% of these insular elements remaining unnamed. Most of them are within the coastal lagoons of Bahía de Chetumal, La Ascensión, and Espiritu Santo bays [14]. Isla Mujeres, located at 86°32′36″ W is the easternmost land of Mexico [11]. Of special relevance is Cozumel, the largest island of the region and third largest in the country, with an area of 467 km$^2$ [14], and perhaps the most biodiverse island in Mexico, but also one of the most threatened due to urbanization pressure for touristic development [57].

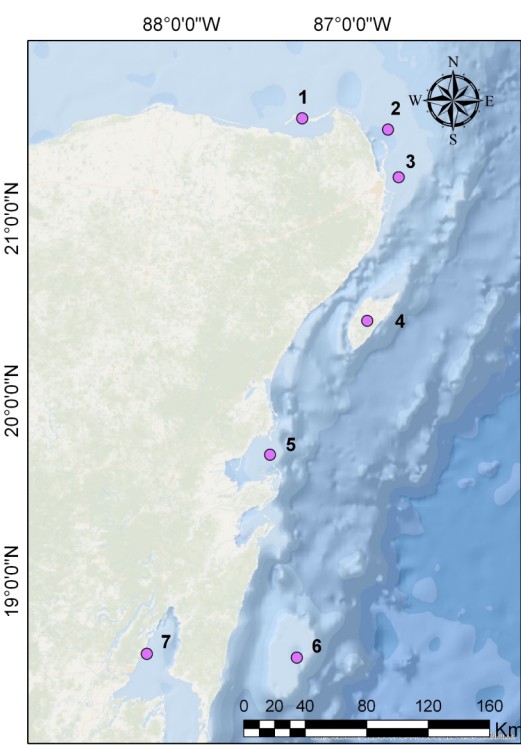

**Figure 10.** Islands of the Mexican Caribbean: (1) Holbox, (2) Contoy, (3) Mujeres, (4) Cozumel, (5) Cayo Culebras, (6) Banco Chinchorro, and (7) Tamalcab.

The origins of the principal islands of the Quintana Roo coast are diverse; Contoy, Isla Mujeres, and Cancún are the result of accumulation of carbonated sand in eolian ridges or dunes during the Pleistocene. Cozumel is an emergent portion of a horst block pushed upward between two normal fault lines, whereas Banco Chinchorro (Figure 11) is a semicircular reef formation or false atoll unique in kind in Mexico due its origins and characteristics. The particulars of these islands are discussed in more detail in González-Sánchez et al. [57]. The dominant vegetation on most of these islands and cays is the coastal dune and mangrove, but on the larger islands there exist the palmar and tropical forests [14], but also see González-Sánchez et al. [57].

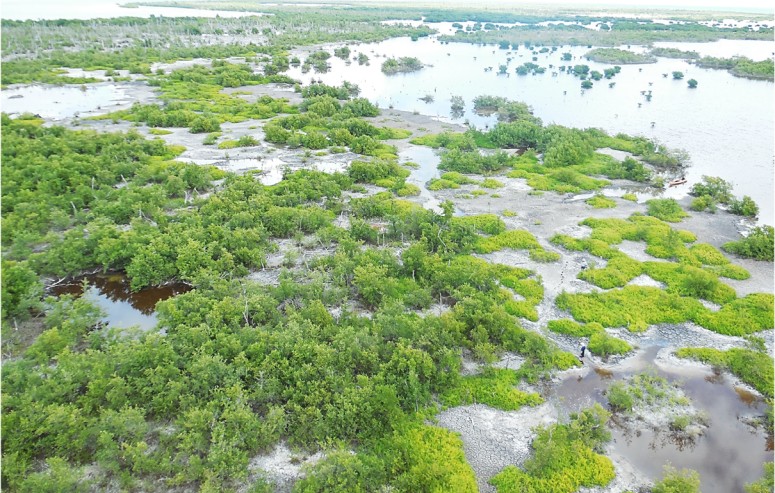

**Figure 11.** Cayo Centro (Banco Chinchorro) is a low island in the atoll reef of Banco Chinchorro, composed almost exclusively of shallow marshes and interior lagoons. The predomintant vegetation

are four species of mangrove: red (*Rhizophora mangle*), white (*Laguncularia racemosa*), black (*Avicennia germinans*), and buttonwood (*Conocarpus erectus*), along with large patches of "chit" (*Thrinax radiata*). The turtle seagrass (*Thalassia testudinum*) is abundant in shallow waters. With his 43 km long × 28 km wide, Banco Chinchorro is the largest coralline atoll in Mexico, and the second largest in the world. Photo by Lizbeth E. Lara-Sánchez.

*2.6. Comments on the Species List*

Comments on several taxa of pertinence are necessary, as follows (parenthesis means that note refers to a specific island popuation):

*Aspidoscelis tigris* (Roqueta). Niño-Gutierrez (2010) listed *A. tigris* for La Roqueta, but the slope of Guerrero is not inside the Tiger Whiptail distribution range [63]. The whiptail lizard from La Roqueta might be *A. deppii*, which is common near the port of Acapulco. We opt to list provisionally the population of La Roqueta as *A. deppii* until clarification is provided.

*Aspidoscelis tigris*. Chafin et al. [64] points out that the Tiger Whiptail species group is a major systematic challenge, "as it comprises a widely distributed complex encompassing a broad spectrum of biogeographic scenarios within desert regions of southwestern North America, northern México (to include Baja California), as well as numerous islands east and west of the peninsula". They found that many *A. tigris* lineages seem to have diverged much earlier than previously suspected but considered the current evidence insufficient to elevate those lineages to the species level. We follow this criterion and accept the current status of the Tiger Whiptails in Mexico as a polytypic complex, but with the warning that this complex is under revision and many mainland, and insular lineages, may be considered distinct species in the near future.

*Boa imperator* (Cozumel). The Central American Boa Constrictor was unknown on Cozumel until 1971, when, according to local independent informants, cinematographers filming the movie "*El Jardín de la Tía Isabel*" released several boas of various sizes in order to create a more "exotic" atmosphere [65]. The population of Cayo Centro (Banco Chinchorro) may be from alien origin too [66]. But Chinchorro has not had a long history of herpetofaunistical inventories as Cozumel has, thus there are not many arguments to state that *B. imperator* was previously absent.

*Crocodylus moreletii* (Contoy). The swamp crocodile is alien to the Mexican Yucatan Peninsula [57,67], where it has become well established in the mainland [65]. Lazcano-Barrero [68] acknowledged four intentional releases of *C. moreletii* on Isla Contoy from 1981 to 1991. The individuals came from zoos and from seizures at regional fairs. However, those crocodiles emigrated or failed to establish themselves around the area, since there is not any known record of Morelet's crocodiles since then [65]. Consequently, we do not enlist this species as part of the herpetofauna in Contoy.

*Crotalus caliginis*. Klauber [69] described the population of rattlesnakes from South Coronado island as a subspecies of *Crotalus viridis*, citing morphological differences from continental *C. viridis*, such as a smaller body, adulthood reached at a smaller body length, and differences in proportions of head and rattle. Grismer [70] considered these morphological differences enough to justify the status of *C. caliginis* as a full species, as did Grismer [71], Wallach et al. [72], and Samaniego-Herrera et al. [45] Ashton and Queiroz [73], however, split the *C. viridis* complex into two species, *C. viridis* and *C. oreganus*, placing the Coronado rattlesnakes as a subspecies of *C. oreganus*. Campbell and Lamar [74]**,** as well as Heimes [75] accepted this classification; this is also the name recognized by the SEMARNAT [76] in its species conservation plan. On the other hand, Pook et al. [77] found insufficient genetic divergence between *C. v. helleri* and *C. v. caliginis,* and indicated that the colonization of Coronado Sur must be a recent event. This put into question the validity of *C. v. caliginis* as a subspecies and stated that the Coronado rattlesnake was just an insular population of the close continental relative *C. helleri*. Davis et al. [78] found important morphological differences between continental *C. helleri* and *C. caliginis* but con-

sidered there were insufficient differences at the molecular level and continued to recognize the Coronado rattlesnake as subsumed within *C. helleri*. This population appears in recent papers named as a subspecies of *C. helleri* [79], *C. viridis* [80], or *C. oreganus* [81]. And, taking into account we do not consider subspecies in this paper, we opt to keep the Coronado Islands rattlesnake as *C. caliginis*, keeping in mind that it might be arranged in one of these species in the near future. (Figure 12)

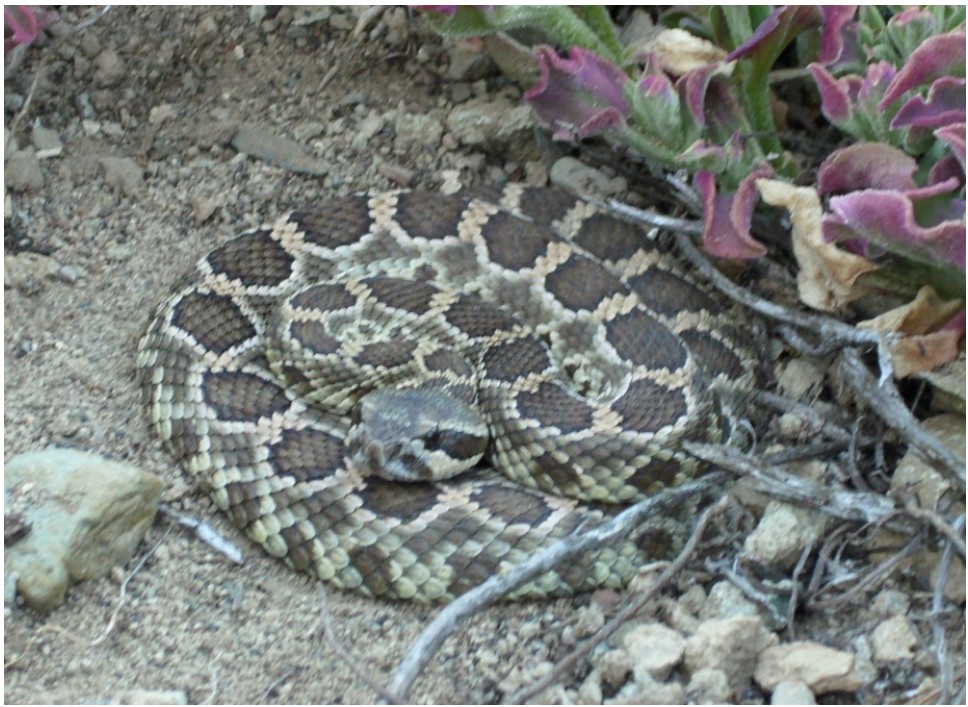

**Figure 12.** Coronado Island Rattlesnake (*Crotalus caliginis*). An endemic rattlesnake from Coronado, Sur Island, in the Coronado Archipielago near the Pacific US–Mexican border. Probably an isolated population of *C. helleri*, *C. viridis*, or (most likely) *C. oreganus*. Perhaps a relictual population from when these islands were joined to the mainland. EVS = H(19), IUCN = CT, NOM-059 = A. Photo by Victor Hugo Gonzalez-Sanchez.

*Crotalus lorenzoensis* originally was described as a subspecies of *C. ruber* by Radcliffe and Maslin [82]. Grismer [83] stated that the differences in the rattle morphology justified its recognition as a distinct species. Instead, Wallach et al. [72] considered *C. lorenzoensis* as a synonym of *C. ruber*. Johnson et al. [84] did not recognize the subspecies status and listed this snake as *C. lorenzoensis*. We follow this decision.

*Ctenosaura nolascensis*. The Nolasco spiny-tailed iguana was originally considered as a subspecies of *C. hemilopha* by Smith [85]. Later, Grismer [83] elevated this population to full species status, based on morphological evidence, although it was suggested that the San Pedro Nolasco iguana might have resulted from a deliberate introduction by the Seri people. Davy et al. [86] demonstrated that the divergence time preceded the human occupation of the area and also confirmed the specific status.

*Ctenosaura pectinata*. (Clarion) The Western Spiny-tailed Iguana occurs in low to intermediate elevations on the Pacific versant of Mexico from Sinaloa into Chiapas, including offshore islands. However, this iguana was introduced on Isla Clarion sometime in the mid-1990s [65] (Figure 13).

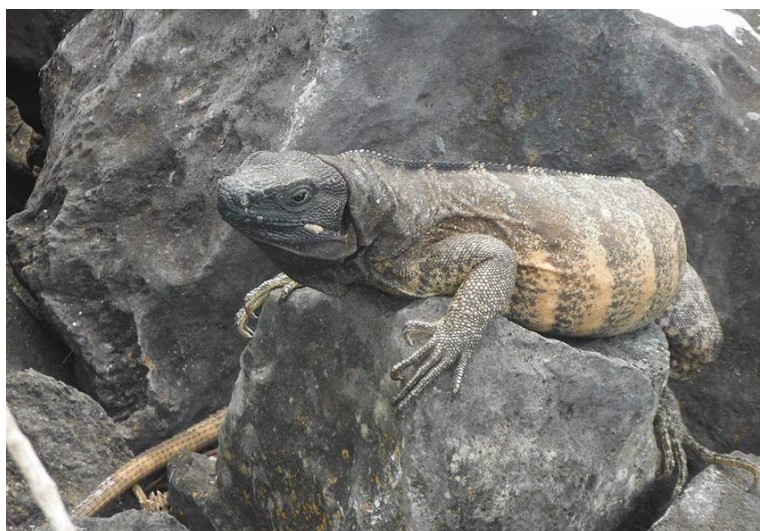

**Figure 13.** *Ctenosaura pectinata*. (Clarion) The Western Spiny-tailed Iguana is a common habitant on the Pacific versant of Mexico, including offshore islands. However, this iguana was introduced on Isla Clarion in the mid-1990s. Its impact on native biota is unknown. There are seven non-mexican native herpetofaunal species in Mexican islands, and another two, that, despite being native to Mexico, are alien in some islands, like *B. imperator* in Cozumel and *C. pectinata* in Clarion. EVS = H(15), IUCN = NE, NOM-059 = A. Photo by Humberto Almanza.

*Hypsiglena catalinae* was described originally as a subspecies of *H. torquata* by Tanner [87], recognized subsequently as a subspecies by Mulcahy [88] and later as synonym of *H. chlorophacea* by Wallach et al. [72]. Mulcahy et al. [89] recognized the Clarion nightsnake as a valid species (as *H. catalinae*) and Johnson et al. [84] followed this determination, a position which we accept here.

*Hypsiglena chlorophaea tiburonensis* initially was described as a subspecies of *H. torquata* by Tanner [90]. Grismer [71,91] indicated that the nightsnake populations from Tiburón and San Esteban should be allocated to *H. t. tiburonensis*. Mulcahy [88] reallocated this taxon as *H. chlorophacea tiburonensis*. Wallach et al. [72] listed the Tiburon and San Esteban Nightsnakes as *H. onchorhyncha*, but strangely, they also recognized *H. t. tiburonensis* as a synonym of *H. chlorophacea*. Given the proximity and shared geological history of Tiburón to the coast of Sonora, we agree that these nightsnakes should be allocated to *H. chlorophacea*, which is the nightsnake species that inhabits the shorelines of Sonora, Sinaloa, and the High Gulf of California [88,89,92]

*Hypsiglena (Eridiphas) marcosensis* was described first by Ottley and Tanner [93] as a subspecies of *Eridiphas slevini* from Isla San Marcos, where it is endemic. Grismer [83] recognized it as a full species (as *E. marcosensis*), although at that time *E. marcosensis* was known only from two specimens. Mulcahy and Archibald [94] revisited this classification but with more specimens available from Isla San Marcos, and found that overlap exists among the morphological characters used to differentiate these populations, and placed *E. marcosensis* as an insular population of *E. slevini*. Mulcahy [88] reiterated this decision and stated that San Marcos populations share ancestry with the Danzante populations, probably originating from mainland populations through land bridges existing in the past, but the evidence for that is lacking presently. Wallach et al. [72] listed the San Marcos Nightsnake among many other insular populations belonging to *H. slevini.*

*Hypsiglena ochrorhyncha unaocularis* Tanner [95] described the Clarion Nightsnake as a subspecies of *H. onchorhyncha* (as *H. o. unaocularus*), and it is treated as such by Wallach et al. [72]. Believed extinct for decades, its rediscovery allowed for molecular analysis, which concluded that it is a valid species (as *H. unaocularus*; Mulcahy et al. [89]). Johnson et al. [84] also listed it as *H. unaocularis*.

*Imantodes gemmistratus.* Casas-Andreu [96] listed the Central American Tree Snake as an unclear record from Isla María Magdalena. We opt to omit this snake for María Magdalena, since there are no other references in the literature and no record could be found in any electronic database nor the Coleccion Nacional de Anfibios y Reptiles (CNAR) from UNAM.

*Lampropeltis catalinensis.* The Santa Catalina Kingsnake is known only from the holotype described by Van Denburgh and Slevin [97](1921). Murphy and Ottley, as well as Wallach et al. (2014) listed *L. catalinensis* as a subspecies of *L. getula* and *L. californiae*, respectively, but Grismer [71,83] and Johnson et al. [84] regarded it as a full species. Since we do not recognize subspecies as a valid taxonomic category, we consider *L. catalinensis* as a valid species, until the phylogeny of this snake is clarified (Figure 14).

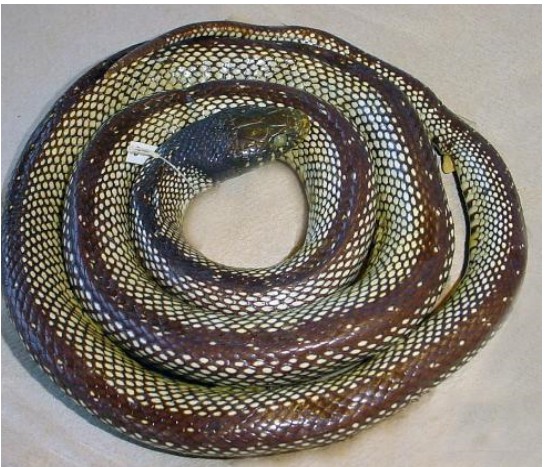 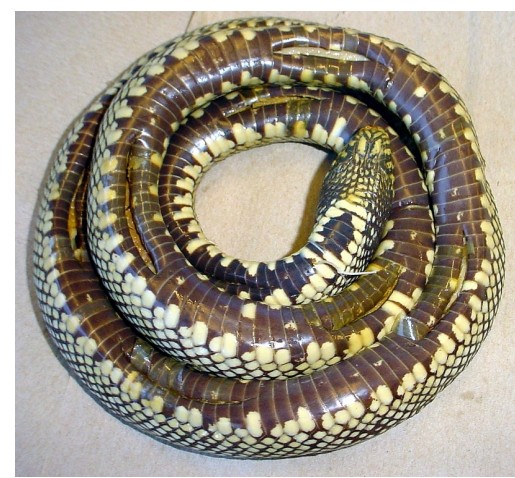

**Figure 14.** Holotype (CAS 50515) of *Lampopeltis catalinae*. The Isla Santa Catalina King Snake is known only from this single male specimen collected in 1920s. His ecology is unknown and since there is no other verifiable record in almost a century, we must assume it is extinct. Although survey efforts must be made in order to clarify if this assumption is correct. EVS = H(17), IUCN = DD, NOM-059 = NS. Photo courtesy of Brian Hubbs.

*Lampropeltis herrerae.* Described as a new kingsnake species on Todos Santos Sur [98], this taxon is frequently treated as a subspecies of *L. zonata* [72,99,100]. Rodríguez-Robles et al. [101] considered that *L. herrerae* might constitute a distinct lineage. Grismer (2001) pointed out several morphological differences between *L. herrerae* and continental *L. zonata*, a position which agrees with the analysis of Rodríguez-Robles et al. [101] and justifies the recognition of *L. herrerae* as a valid species. Grismer [71] and Johnson et al. [84] follow this nomenclature, as do we.

*Masticophis lineatus variolosus* was described as *Masticophis mentovarius variolosus* from María Magdalena Island (Tres Marías Archipelago) by Smith [102], but Smith and Taylor [103], as well as Zweifel [104] indicated that the Tres Marías subspecies should be assigned preferably to *M. lineatus* (considering *M. striolatus variolosus* as a synonym). O'Connell and Smith [105] stated that due to the difficulty of the reconstruction of the phylogeny of this species and the fact they lacked DNA samples from the Tres Marías whipsnake, a nomenclatural change is not supported presently; thus, they recommended the maintenance of *M.m.variolosus*. Until the phylogeny of this insular population is better understood, we apply the name *Masticophis lineatus*.

*Micrurus diastema*: The variable coral snake complex extends from the southern United States to Honduras [74]. In previous work for the Yucatán Peninsula in Mexico [35,36,57], this coral snake was known as *M. diastema*. But, recently, Reyes-Velasco et al. [106] stated that all populations east of the Tehuantepec Ishmus, previously referred to as *M. diastema,* comprise a distinct clade, and suggest the name *M. apiatus* for these populations. Diaz-Gamboa et al. [67] follow that name, as so do we.

*Norops microlepidotus*. Castro-Franco and Gaviño-De la Torre [107] listed *N. microlepidotus* for Isla Peña, but Ramírez-Reyes et al. [108] noted that the closest *N. microlepidotus* record is at least 700 km from Isla Peña. They reviewed the specimens collected by Castro-Franco and Gaviño-De la Torre [107] and concluded that this anole is actually *N. nebulosus*, which is a common lizard on Isla Peña.

*Norops sericeus*. Lara-Tufiño et al. [109] resurrected the name *N. ustus* for the former *N. sericeus* in the Mexican Yucatán Peninsula plus Belize, restricting *N. sericeus* to the Gulf of Mexico versant from Tamaulipas to western Tabasco. They indicated, however, that it is not clear where the area of contact is between *N. sericeus* and *N. ustus*, or if there is genetic flow between these populations. They specified that *N. sericeus* group species show important variations in morphological traits, which makes identification difficult. González-Sánchez et al. [57] assigned the *N. sericeus* populations in the Yucatán Península to *N. ustus*. We accept this position, so the insular populations of the Yucatán Península, previously indicated as *N. sericeus* are assigned to *N. ustus*.

*Phyllodactylus* spp. We accept six *Phyllodactylus* species as occurring in the Mexican insular systems: *P. bugastrolepis*, *P. homolepidurus*, *P. nocticolus*, *P. partidus*, *P. unctus*, and *P. xanti*. Blair et al. [110] stated that most of the *P. xanti* on the Baja California peninsula are referrable to *P. nocticolus* and that *P. xanti* is restricted to the Los Cabos region, where it overlaps the range of *P. unctus*. In their study, however, they did not include insular specimens; therefore, the identity of many insular *Phyllodactylus* is unclear. Additionally, it is unclear whether some insular populations originated by introduction [111,112] or by rafting [113]. Therefore, in the majority of the cases, we follow the insular distribution as understood by Grismer [71] and Blair et al. [110]. Thus, we restrict *P. xanti* to the continental region of Los Cabos and assign most of the insular populations to *P. nocticolus*, until more evidence is provided. We do not overlook, however, the possibility that the insular populations of *Phyllodactylus* close to Espiritu Santo or Los Cabos might be more closely related to *P. xanti*. (Figure 15).

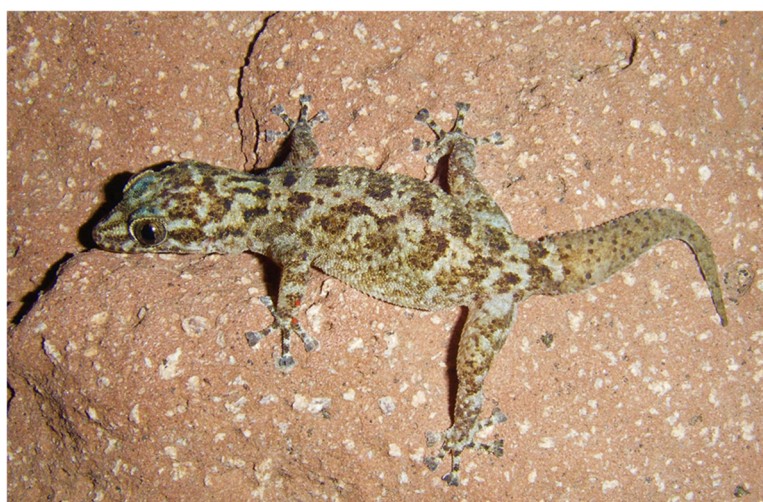

**Figure 15.** Peninsular Leaf-toed Gecko (*Phyllodactylus nocticolus*) in the Coronados island (Gulf of California). The leaf-toed geckos from the Mexican Pacific versant are a intriguin and complicted species complex. Currently several studies are being carried out to solve the *Phyllodactylus* taxonomic puzzle. For simplification, we assigned most of the insular populations as *P. nocticolus*, as was completed by Grismer [71] and Blair et al. [110]. But, as a reminder to the reader, this is a provisional statement until further taxonomic clarification, since many insular populations may constitute species on their own (as occurred with another insular *Phyllodactylus* recently described such: *P. angelensis*, *P. apricus*, *P. coronatus*, *P. cleofasensis*, etc.) or may be more related to *P. xanti*. Distribution status = NE, EVS = M(10), IUCN = NE, NOM-059 = NS. Photo by Ruben Alonso Carbajal-Márquez.

*Phyllodactylus tuberculosus*. Dixon [113] mentioned that the *Phyllodactylus* population on Farallon Island (Farallón de San Ignacio) was known (at that time) from only one specimen that "appears to be closely related to *P. tuberculosus* of the mainland." Later, Murphy and Ottley [114] as well as Murphy and Aguirre-León listed *P. tuberculosus* for that island, although Grismer [71,91] did not mention it. Peralta et al. [115], however, reported *P. homolepidurus* as a new record for Farallón de San Ignacio. We consider that this is the correct identity for this population.

*Rena boettgeri*. Werner [116] described a species of wormsnake (as *Glauconia boettgeri*) from an undetermined location. Many years later, Smith and Larsen [117] revisited the holotype and concluded that the Baja California Cape wormsnake is a subspecies of *Leptotyphlops humilis* and stated its distribution to be confined to the southern tip of the peninsula. Grismer [71,83] listed *Leptotyphlops humilis* as the only wormsnake present on the Baja California Peninsula. Adalsteinsson et al. [118] recognized *R. boettgeri* as a full species in southern Baja California Sur, while maintaining *R. humilis* as the name for the populations in the rest of the peninsula, but they did not indicate which insular populations would correspond to *R. boettgeri* or *R. humilis*. Wallach et al. [72] treated *R. boettgeri* (as *G. boettgeri*) as a synonym of *R. humilis*. As we wait for more information concerning insular populations, we follow this position provisionally.

*Sauromalus ater*. Grismer [71,91] and Hollingsworth [119] listed the Chuckwalla from Danzante as *S. ater/obesus*. Previously, however, Petren and Case [120] found genetic evidence that determined that this population belongs to *S. a. slevini*. Murphy and Aguirre-León [121] recognized this population as *S. slevini*. Finally, Montanucci [122] reviewed the morphological evidence and concluded that, indeed, the Chuckwallas from Danzante resemble *S. slevini*. We accept that conclusion.

*Sauromalus sp.* (Alcatraz Island). The Spiny Chuckwalla from Alcatraz Island appears to be a lineage of hybrid origin from *S. hispidus*, *S. varius*, and *S. obesus* [71,123], thus, it cannot be given a specific denomination. Krauss [124] and González-Sánchez et al. [65] discussed the probable alien origin for this population.

*Trachycephalus typhonius*. The taxonomy of this milk frog is one of the most controversial disputes in herpetology going back to site of collection of the holotype and its description by Linnaeus more than 250 years ago [125]. The controversy reaches the Central American milk frog we know in the Yucatan Peninsula as *T. typhonius* as listed by González-Sánchez et al. [57], *Trachycephalus venulosus* [126] or *Phrynohyas venulose* [34,36]. Recently, Ron et al. [127] intended to break up the *T. typhonius* complex, and restricted this name for the populations in Guyana and Surinam. While the populations from other parts of South America, Central America, and Mexico, may be independent species by their own. Although the efforts to solve this taxonomic puzzle are still going on [128], the populations from Mexico and Central America do not have a specific denomination yet. Thus, we follow the recommendation of Frost (amphibian species of the world) and consider this frog as *T. "vermiculatus"* to distinguish this populations from other *T. typhonius* (sensu strictu) populations, until clarification.

*Uta antiqua* was described as a new species (as *U. antiquus*) from Isla Salsipuedes, San Lorenzo Norte, and Sur [129], and recognized as valid by Liner [130,131] as well as Murphy and Aguirre-León [121,132]. Grismer [83] synonymized *U. antiqua* with *U. stansburiana*. Flores-Villela and Canseco-Márquez [133] and Wilson et al. [25] accepted this disposition.

*Uta stansburiana*. The Side-blotched Lizard is one of the most studied reptile species in Mexico, thus a taxonomic revision of this group exists [134], but there is still a great deal of uncertainty regarding the identity of several lineages and about the validity and status of many subspecific denominations [134]. Furthermore, introgression and interbreeding among different populations has occurred often. Also, many insular populations might be translocated and/or introduced [121]. Thus, determining how many insular lineages constitute undescribed species remains challenging.

*Uta stellata* is an endemic species from San Benito Island [sic] (i.e., San Benito Archipelago), distinguishable from *U. stansburiana* by scutellational differences, according to Van Denburgh [135]. This taxon was listed by Liner [130,131] as a valid species, but Grismer [83] synonymized *U. stellata* with *U. stansburiana*, which is a position we accept.

*2.7. Comments on Island Toponomy*

Any biologist, geographer, or geologist who had worked on the Mexican islands is aware of the complexity of island toponyms, in many cases a same island may be known by several synonyms, or the same name is used for different islands. Thus, tracing the biogeographical history of Mexican insular systems among many different authors and times, is a very confusing task. Below, we present the key issues we identified during the compilation of this list, aiming to aid the reader in tracking the identity of the most problematic islands more effectively.

Catalana (Santa Catalina) is likely the original name, since old maps depict this island as "Catalán" or "Catalana". It is believed that in the mid-1850s when the U.S. Navy charted the Gulf the mapmakers made a mistake and erroneously change the name to Isla Santa Catalina. We kept this latter denomination since it is the official one in the catalogue of INEGI [14] (num. 333). And it is, by far, more frequently used in the herpetological literature. But we point out that "Catalana" may appear from time to time in some papers or books.

Cardonosa ("a place of *cardones*", Spanish denomination for arborescent cactus); this name is a common synonym for Partida Norte, but in our text, Roca Cardonosa (number 39 in the Región Marina Golfo de California Section) refers to what Grismer [71,83] listed as Cardonosa Este.

Cerros (Cedros) means mountains in Spanish, frequent in old cartography [39], and is often used by people who are not from the island. The correct denomination is "Cedros"(cedars), given by the Spaniards due to the forest vegetation in several parts of the islands (although the trees are, in fact, pines) [40].

Coloradito (Lobos) is the island between El Muerto and La Encantada. We chose to make an exception to the nomenclature of INEGI [14], since "Lobos" is a very common name for many insular bodies. We chose to keep the name "Coloradito" to avoid confusion.

Coral or del Coral (La Peña) is a frequent synonym for La Peña. We chose this last denomination since it is the used by INEGI [14] (insular element 168 in the Region Marina Golfo de California section). Isla del Coral is the name that the tourist agencies use to promote the island.

Coronados Archipielago (The Coronado islands on the Pacific) is very often written as "Coronados." We use the name in its singular form, as indicated by INEGI [14], whereas the plural denomination corresponds to the island in the Gulf of California in the Loreto Bay region.

Danzantes is the name for Isla Danzante and the surrounding islets.

Grande, or Isla Grande, a synonym for Isla Ixtapa, which is near the port of Zihuatanejo.

Flecha (within Bahía de los Angeles) is a synonym of El Borrego (number 90 in the Región Marina Golfo de California Section).

"Isabela" is a frequent synonym for Isabel; sometimes both names are used even in the same document, for example in Woolrich-Piña et al. [136].

Jacques Cousteau (Cerralvo): in 2009 the Mexican government renamed the Cerralvo island in honor to the French explorer [137], and it is listed as such by INEGI [14] (151, in Region Marina Golfo de California). This denomination, however, is controversial, since the local people were not consulted for their approval of the name change [40]. Moreover, the name Cerralvo is strongly associated with the history of the exploration of the Gulf of California [138]. The name honors the regent viceroy of New Spain who authorized and founded one of the first and most important expeditions in the Gulf of California, which

discovered and named several islands in the region of La Paz. The Spanish navigators gave the name Cerralvo to the largest island they found on one of those trips. We agree in respecting the historical value of the toponomies and choose to maintain the name Cerralvo.

Partida Norte and Partida Sur are names referenced by INEGI [14] as Partida and La Partida for the islands of Archipielago San Lorenzo and Archipielago Espiritu Santo, respectively. We choose to keep the distinction words Norte and Sur, to avoid confusion.

Pond (Estanque): the English word "Pond" is more frequent in the scientific literature than its Spanish counterpart, "Estanque." We believe that the Spanish denomination should prevail since "Pond" is not a personal name (as occurs with Smith Island). Also, INEGI [14] uses "Estanque" (Insular element number 131 in the Región Marina Golfo de California Section).

Raza or La Raza (Rasa) alludes to "race" or "caste," probably confused with the homophonous word "rasa" = flat, which alludes to the flat and shallow topography of the island, which is a result of guano mining, principally.

Smith (within Bahía de los Angeles) is a frequent denomination for the island that INEGI [14] lists as Coronado (number 322 in the Región Marina Golfo de California Section). Although we discourage the use of foreign words in toponyms, in this case we employ the Spanish name as a secondary synonym, since there is a Coronado Archipielago in the Pacific, and a Coronados Island in Loreto Bay.

Turners (Datil). INEGI [14] recognized the English denomination (number 993 in the Región Marina Golfo de California Section); however, we place it as a secondary synonym, in order to give preference to the Spanish nomenclature.

Willard (San Luis Gonzaga) is listed as such by Grismer [71,91] for a rocky island within San Luis Gonzaga Bay. INEGI [14] used the bay as eponymous for that island (number 322 in the Región Marina Golfo de California Section).

## 3. Results

### *3.1. Composition of the Herpetofauna*

#### 3.1.1. Families

The herpetofauna of the insular systems of Mexico comprises 40 families (Table 1). Seven of these families contain amphibians (six anurans and one salamander). The reptiles comprise 33 families, including one crocodylian, 24 squamate, and 8 turtle families. The anuran families Bufonidae and Hylidae (Table 2) collectively contain slightly more than half of all the amphibian species (4 and 5, respectively, of a total of 16). The most speciose reptile families (Table 2) are the Phrynosomatidae (31), Teiidae (22), Colubridae (36), Dipsadidae (19), and Viperidae (18), comprising 66.5% of the squamates and 61.5% of all reptiles.

**Table 1.** Composition of the native and non-native herpetofauna of the insular systems of Mexico.

| Orders | Families | Genera | Species |
|---|---|---|---|
| Anura | 6 | 12 | 14 |
| Caudata | 1 | 2 | 2 |
| Subtotals | 7 | 14 | 16 |
| Crocodylia | 1 | 1 | 2 |
| Squamata | 24 | 69 | 195 |
| Testudines | 8 | 11 | 13 |
| Subtotals | 33 | 81 | 210 |
| Totals | 40 | 95 | 226 |

**Table 2.** Distribution of the insular herpetofauna of Mexico by physiograpc region. * = country endemic; ** insular endemic; ‡ = non-native.

| Taxa | Physiographic Regions | | | | | |
|---|---|---|---|---|---|---|
| | Islands of Gulf of California | Islands of Pacific Baja California | Islands of Tropical Pacific | Islands of Gulf of Mexico | Islands of Mexican Caribbean | Total Number of Regions |
| **Amphibia (16 species)** | | | | | | |
| **Anura (14 species)** | | | | | | |
| **Bufonidae (4 species)** | | | | | | |
| *Anaxyrus punctatus* | + | + | | | | 2 |
| *Incilius mazatlanensis* * | | | + | | | 1 |
| *Incilius valliceps* | | | | + | + | 2 |
| *Rhinella horribilis* | | | + | + | + | 3 |
| **Eleutherodactylidae (2 species)** | | | | | | |
| *Eleutherodactylus pallidus* * | | | + | | | 1 |
| *Eleutherodactylus planirostris* ‡ | | | | | + | 1 |
| **Hylidae (5 species)** | | | | | | |
| *Dendrosophus microcephalus* | | | | | + | 1 |
| *Hyliola regilla* | | + | | | | 1 |
| *Scinax staufferi* | | | | + | + | 2 |
| *Smilisca baudini* | | | + | + | + | 3 |
| *Trachycephalus vermiculatus* | | | | | + | 1 |
| **Leptodactylidae (1 species)** | | | | | | |
| *Leptodactylus fragilis* | | | | + | + | 2 |
| **Microhylidae (1 species)** | | | | | | |
| *Hypopachus variolosus* | | | + | + | | 2 |
| **Scaphiopodidae** | | | | | | |
| *Scaphiopus couchii* | + | | | | | 1 |
| **Caudata (2 species)** | | | | | | |
| **Plethodontidae (2 species)** | | | | | | |
| *Aneides lugubris* | | + | | | | 1 |
| *Batrachoseps major* | | + | | | | 1 |
| **Reptilia (2 species)** | | | | | | |
| **Crocodylia (2 species)** | | | | | | |
| **Crocodylidae (2 species)** | | | | | | |
| *Crocodylus acutus* | | | + | | + | 2 |
| *Crocodylus moreletii* | | | | + | | 1 |
| **Squamata (185 species)** | | | | | | |
| **Anguidae (3 species)** | | | | | | |
| *Elgaria cedrosensis* * | | + | | | | 1 |
| *Elgaria multicarinata* | | + | | | | 1 |
| *Elgaria nana* ** | | + | | | | 1 |
| **Anniellidae (2 species)** | | | | | | |
| *Anniella geronimensis* * | | + | | | | 1 |
| *Anniella pulchra* | | + | | | | 1 |
| **Bipedidae (1 species)** | | | | | | |
| *Bipes biporus* * | | + | | | | 1 |
| **Corytophanidae (1 species)** | | | | | | |
| *Basiliscus vittatus* | | | | + | + | 2 |

| Species | | | | | | |
|---|---|---|---|---|---|---|
| **Crotaphytidae (4 species)** | | | | | | |
| *Crotaphytus dickersonae* ** | + | | | | | 1 |
| *Crotaphytus insularis* ** | + | | | | | 1 |
| *Gambelia copeii* | | + | | | | 1 |
| *Gambelia wislizenii* | + | | | | | 1 |
| **Dactyloidae (6 species)** | | | | | | |
| *Anolis allisoni* ‡ | | | | | + | 1 |
| *Norops lemurinus* | | | | | + | 1 |
| *Norops nebulosus* * | | | + | | | 1 |
| *Norops rodriguezii* | | | | + | + | 2 |
| *Norops sagrei* ‡ | | | | + | + | 2 |
| *Norops ustus* | | | | + | + | 2 |
| **Eublepharidae (3 species)** | | | | | | |
| *Coleonyx elegans* | | | | | + | 1 |
| *Coleonyx gypsicolus* ** | + | | | | | 1 |
| *Coleonyx variegatus* | + | + | | | | 2 |
| **Gekkonidae (3 species)** | | | | | | |
| *Gehyra mutilata* ‡ | | | + | | | 1 |
| *Hemidactylus frenatus* ‡ | | | + | + | + | 3 |
| *Hemidactylus turcicus* ‡ | | | | + | + | 2 |
| **Iguanidae (13 species)** | | | | | | |
| *Ctenosaura conspicuosa* ** | + | | | | | 1 |
| *Ctenosaura hemilopha* * | + | | | | | 1 |
| *Ctenosaura nolascensis* ** | + | | | | | 1 |
| *Ctenosaura pectinata* | | | + | | | 1 |
| *Ctenosaura similis* | | | | + | + | 2 |
| *Dipsosaurus catalinensis* ** | + | | | | | 1 |
| *Dipsosaurus dorsalis* | + | + | | | | 2 |
| *Iguana iguana* | | | + | + | + | 3 |
| *Sauromalus ater* | + | | | | | 1 |
| *Sauromalus hispidus* * | + | | | | | 1 |
| *Sauromalus klauberi* ** | + | | | | | 1 |
| *Sauromalus slevini* * | + | | | | | 1 |
| *Sauromalus varius* * | + | | | | | 1 |
| **Mabuyidae (1 species)** | | | | | | |
| *Marisora aquilonaria* * | | | + | | | 1 |
| *Marisora lineola* | | | | + | + | 2 |
| **Phrynosomatidae (31 species)** | | | | | | |
| *Callisaurus draconoides* | + | + | | | | 2 |
| *Petrosaurus mearnsi* | + | | | | | 1 |
| *Petrosaurus repens* * | + | | | | | 1 |
| *Petrosaurus slevini* ** | + | | | | | 1 |
| *Petrosaurus thalassinus* * | + | | | | | 1 |
| *Phrynosoma cerroense* * | | + | | | | 1 |
| *Phrynosoma solare* | + | | | | | 1 |
| *Sceloporus angustus* ** | + | | | | | 1 |
| *Sceloporus chrysostictus* | | | | + | + | 2 |
| *Sceloporus clarkii* | + | + | + | | | 3 |
| *Sceloporus cozumelae* * | | | | | + | 1 |

| | | | | | | |
|---|:---:|:---:|:---:|:---:|:---:|:---:|
| *Sceloporus grandaevus* ** | + | | | | | 1 |
| *Sceloporus hunsakeri* * | + | | | | | 1 |
| *Sceloporus lineatulus* ** | + | | | | | 1 |
| *Sceloporus magister* | + | | | | | 1 |
| *Sceloporus occidentalis* | | + | | | | 1 |
| *Sceloporus orcutti* | + | | | | | 1 |
| *Sceloporus variabilis* | | | | + | | 1 |
| *Sceloporus zosteromus* * | + | + | | | | 2 |
| *Urosaurus auriculatus* ** | | | + | | | 1 |
| *Urosaurus bicarinatus* * | | | + | | | 1 |
| *Urosaurus clarionensis* ** | | | + | | | 1 |
| *Urosaurus nigricaudus* * | + | + | | | | 2 |
| *Urosaurus ornatus* | + | | + | | | 2 |
| *Uta encantadae* ** | + | | | | | 1 |
| *Uta lowei* ** | + | | | | | 1 |
| *Uta nolascensis* ** | + | | | | | 1 |
| *Uta palmeri* ** | + | | | | | 1 |
| *Uta squamata* | + | | | | | 1 |
| *Uta stansburiana* | + | + | | | | 2 |
| *Uta tumidarostra* ** | + | | | | | 1 |
| **Phyllodactylidae (9 species)** | | | | | | |
| *Phyllodactylus angelensis* ** | + | | | | | 1 |
| *Phyllodactylus apricus* ** | + | | | | | 1 |
| *Phyllodactylus benedetii* * | | | + | | | 1 |
| *Phyllodactylus bugastrolepis* ** | + | | | | | 1 |
| *Phyllodactylus cleofasensis* ** | | | + | | | 1 |
| *Phyllodactylus coronatus* ** | + | | | | | 1 |
| *Phyllodactylus homolepidurus* * | + | | | | | 1 |
| *Phyllodactylus isabelae* ** | | | + | | | 1 |
| *Phyllodactylus lanei* * | | | + | | | 1 |
| *Phyllodactylus lupitae* ** | | | + | | | 1 |
| *Phyllodactylus nocticolus* | + | + | | | | 2 |
| *Phyllodactylus partidus* ** | + | | | | | 1 |
| *Phyllodactylus tuberculosus* | | | + | | | 1 |
| *Phyllodactylus santacruzensis* ** | + | | | | | 1 |
| *Phyllodactylus unctus* * | + | | | | | 1 |
| **Scincidae (2 species)** | | | | | | |
| *Mesoscincus schwartzei* | | | | | + | 1 |
| *Plestiodon skiltonianus* | | + | | | | 1 |
| **Sphaerodactylidae (3 species)** | | | | | | |
| *Aristelliger georgeensis* | | | | | + | 1 |
| *Sphaerodactylus continentalis* | | | | | + | 1 |
| *Sphaerodactylus glaucus* | | | | + | + | 2 |
| **Teiidae (21 species)** | | | | | | |
| *Aspidoscelis bacata* ** | + | | | | | 1 |
| *Aspidoscelis cana* ** | + | | | | | 1 |
| *Aspidoscelis carmenensis* ** | + | | | | | 1 |
| *Aspidoscelis catalinensis* ** | + | | | | | 1 |
| *Aspidoscelis celeripes* ** | + | | | | | 1 |

| Species | C1 | C2 | C3 | C4 | C5 | N |
|---|---|---|---|---|---|---|
| *Aspidoscelis ceralbelsis* ** | + | | | | | 1 |
| *Aspidoscelis communis* * | | | + | | | 1 |
| *Aspidoscelis costata* * | | | + | | | 1 |
| *Aspidoscelis cozumela* * | | | | + | + | 2 |
| *Aspidoscelis danheimae* ** | + | | | | | 1 |
| *Aspidoscelis deppii* | | | + | + | + | 3 |
| *Aspidoscelis espiritensis* ** | + | | | | | 1 |
| *Aspidoscelis franciscensis* ** | + | | | | | 1 |
| *Aspidoscelis guttatus* * | | | + | | | 1 |
| *Aspidoscelis hyperythrus* | + | + | | | | 2 |
| *Aspidoscelis lineatissima* * | | | + | | | 1 |
| *Aspidoscelis martyris* ** | + | | | | | 1 |
| *Aspidoscelis maslini* | | | | + | + | 2 |
| *Aspidoscelis pictus* ** | + | | | | | 1 |
| *Aspidoscelis rodecki* * | | | | | + | 1 |
| *Aspidoscelis tigris* | + | + | | | | 2 |
| *Holcosus gaigeae* * | | | | + | + | 2 |
| **Boidae (2 species)** | | | | | | |
| *Boa sigma* * | | | + | | | 1 |
| *Boa imperator* | | | | + | + | 2 |
| **Charinidae (1 species)** | | | | | | |
| *Lichanura trivirgata* | + | + | | | | 2 |
| **Colubridae (36)** | | | | | | |
| *Bogertophis rosaliae* | + | | | | | 1 |
| *Drymarchon melanurus* | | | + | | | 1 |
| *Drymobius margaritiferus* | | | | + | | 1 |
| *Lampropeltis abnorma* | | | | + | | 1 |
| *Lampropeltis californiae* | + | | | | | 1 |
| *Lampropeltis catalinensis* ** | + | | | | | 1 |
| *Lampropeltis herrerae* ** | | + | | | | 1 |
| *Lampropeltis polyzona* * | | | + | | | 1 |
| *Leptophis diplotropis* * | | | + | | | 1 |
| *Leptophis mexicanus* | | | | | + | 1 |
| *Masticophis anthonyi* ** | | | + | | | 1 |
| *Masticophis barbouri* | + | | | | | 1 |
| *Masticophis bilineatus* | + | | | | | 1 |
| *Masticophis fuliginosus* | + | + | | | | 2 |
| *Masticophis mentovarius* | | | + | | + | 2 |
| *Masticophis slevini* ** | + | | | | | 1 |
| *Mastigodryas melanolomus* | | | + | | + | 2 |
| *Oxybelis aeneus* | | | | + | + | 2 |
| *Oxybelis fulgidus* | | | | | + | 1 |
| *Oxybelis microphtalmus* * | | | + | | | 1 |
| *Phyllorhynchus decurtatus* | + | | | | | 1 |
| *Pituophis catenifer* | + | + | | | | 2 |
| *Pituophis insularis* ** | | + | | | | 1 |
| *Pituophis vertebralis* * | + | + | | | | 2 |
| *Pseudelaphe flavirufa* | | | | + | | 1 |
| *Rhinocheilus etheridgei* ** | + | | | | | 1 |

| | | | | | | |
|---|---|---|---|---|---|---|
| *Salvadora hexalepis* | + | + | | | | 2 |
| *Sonora savagei* * | + | | | | | 1 |
| *Sonora semiannulata* | + | | | | | 1 |
| *Sonora straminea* | + | + | | | | 2 |
| *Spilotes pullatus* | | | | + | | 1 |
| *Tantilla bocourti* * | | | + | | | 1 |
| *Tantilla calamarina* * | | | + | | | 1 |
| *Tantilla moesta* | | | | | + | 1 |
| *Tantilla planiceps* | + | | | | | 1 |
| *Trimorphodon lyrophanes* | + | | | | | 1 |
| **Dipsadidae (19 species)** | | | | | | |
| *Coniophanes imperialis* | | | | + | | 1 |
| *Conophis lineatus* | | | | + | + | 2 |
| *Conophis vittatus* | | | + | | | 1 |
| *Diadophis punctatus* | | + | | | | 1 |
| *Dipsas brevifacies* | | | | + | | 1 |
| *Geophis annuliferus* * | | | + | | | 1 |
| *Hypsiglena catalinae* ** | + | | | | | 1 |
| *Hypsiglena chlorophaea* | + | | | | | 1 |
| *Hypsiglena ochrorhynchus* | + | + | | | | 2 |
| *Hypsiglena slevini* * | + | + | + | | | 3 |
| *Hypsiglena torquata* | | | + | | | 1 |
| *Hypsiglena unaocularis* ** | | | + | | | 1 |
| *Imantodes cenchoa* | | | | + | | 1 |
| *Imantodes gemmistratus* | | | + | | | 1 |
| *Leptodeira frenata* | | | | + | + | 2 |
| *Ninia sebae* | | | | + | | 1 |
| *Rhadinaea hesperia* * | | | + | | | 1 |
| *Sibon nebulatus* | | | + | + | | 2 |
| *Tropidodipsas sartorii* | | | | + | | 1 |
| **Elapidae (3 species)** | | | | | | |
| *Hydrophis platurus* | + | | + | | | 2 |
| *Micruroides euryxanthus* | + | | | | | 1 |
| *Micrurus apiatus* | | | | + | + | 2 |
| **Leptotyphlopidae (3 species)** | | | | | | |
| *Epictia bakewelli* * | | | + | | | 1 |
| *Epictia magnamaculata* | | | | | + | 1 |
| *Rena humilis* | + | + | + | | | 3 |
| **Natricidae (1 species)** | | | | | | |
| *Thamnophis proximus* | | | | | + | 1 |
| **Typhlopidae (1 species)** | | | | | | |
| *Indotyphlops braminus* ‡ | | | + | + | + | 3 |
| **Viperidae (18 species)** | | | | | | |
| *Agkistrodon bilineatus* | | | + | | | 1 |
| *Agkistrodon russeolus* * | | | | | + | 1 |
| *Crotalus angelensis* ** | + | | | | | 1 |
| *Crotalus atrox* | + | | | | | 1 |
| *Crotalus caliginis* ** | | + | | | | 1 |
| *Crotalus catalinensis* | + | | | | | 1 |

| | | | | | | |
|---|---|---|---|---|---|---|
| *Crotalus cerastes* | + | | | | | 1 |
| *Crotalus enyo* * | + | + | | | | 2 |
| *Crotalus estebanensis* ** | + | | | | | 1 |
| *Crotalus lorenzoensis* ** | + | | | | | 1 |
| *Crotalus mitchellii* | + | + | | | | 2 |
| *Crotalus molossus* | + | | | | | 1 |
| *Crotalus polisi* ** | + | | | | | 1 |
| *Crotalus pyrrhus* | + | | | | | 1 |
| *Crotalus ruber* | + | + | | | | 2 |
| *Crotalus thalassoporus* ** | + | | | | | 1 |
| *Crotalus tigris* | + | | | | | 1 |
| *Crotalus tortuguensis* ** | + | | | | | 1 |
| **Testudines (13 species)** | | | | | | |
| **Chelonidae (5 species)** | | | | | | |
| *Caretta caretta* | + | | + | + | + | 4 |
| *Chelonia mydas* | + | | + | + | + | 4 |
| *Eretmochelys imbricata* | + | | + | + | + | 4 |
| *Lepidochelys kempii* | | | | + | + | 2 |
| *Lepidochelys olivacea* | + | | + | | | 2 |
| **Dermatemydidae (1 species)** | | | | | | |
| *Dermatemys mawii* | | | | + | | 1 |
| **Dermochelyidae (1 species)** | | | | | | |
| *Dermochelys coriacea* | + | | + | | | 2 |
| **Emydidae (1 species)** | | | | | | |
| *Trachemys venusta* | | | | | + | 1 |
| **Geoemydidae (1 species)** | | | | | | |
| *Rhinoclemmys areolata* | | | | | + | 1 |
| **Kinosternidae (2 species)** | | | | | | |
| *Kinosternon integrum* * | | | + | | | 1 |
| *Kinosternon scorpioides* | | | | + | + | 2 |
| **Staurotypidae (1 species)** | | | | | | |
| *Staurotypus triporcatus* | | | | + | | 1 |
| **Testudinidae (1 species)** | | | | | | |
| *Gopherus morafkai* | + | | | | | 1 |
| **Totals** | **108** | **40** | **57** | **46** | **53** | — |

3.1.2. Genera

Ninety-five genera are represented within the herpetofauna of the insular systems of Mexico (Table 1). Fourteen genera represent amphibians, including two salamander and twelve anuran genera (Table 1). Except for *Incilius* and *Eleutherodactylus*, with two species each, all the other amphibian genera are monospecific for the islands of Mexico (however, one of the two species of *Eleutherodactylus*, *E. planirostris*, is an introduced species). The reptiles are arranged among 81 genera, including 1 of crocodiles, 11 of turtles, and 69 of squamates (Table 1). Ranked among the most speciose reptile genera are *Aspidoscelis* (21 species), *Crotalus* (16), *Sceloporus* (12), *Phyllodactylus* (15), *Uta* (7), *Masticophis* (6), and *Lampropeltis* (5), all comprising squamates. In the Testudines there are 11 genera (Table 1), with 13 species; *Kinosternon* and *Lepidochelys* have 2 species each (Table 2).

3.1.3. Species

Despite the insular territories of Mexico representing only 0.26% of the country's emerged surface, they host an outstanding total amount of 226 herpetofaunal species (Table 1). This number is 16.1% of the 1405 species known for Mexico [26]. Of these 226 species, 16 are amphibians (7.4%) and 210 are reptiles (92.9%). This number of reptile species is higher than that reported for some large Mexican states, such as Chihuahua (155), Durango (138), Sonora (169), Tamaulipas (160), Jalisco (194), Puebla (184) [139], and 120 species reported within the three states that comprise the Mexican Yucatan Peninsula [57,67]. If the insular territories of Mexico constituted a state, the reptile fauna would rank fourth in size, only below that of Oaxaca, with 328, Chiapas with 254, and Veracruz with 244, and only slightly higher than Guerrero with 200, and Jalisco with 194 [139].

On the other hand, the amphibians have a low specific richness, with 16 species, 15 of which are native species. These 15 species comprise 3.5% of the 430 native amphibian species in Mexico [26]. The low specific richness of amphibians on islands is not surprising; the physiological traits of amphibians, such as thin moist skin, soft eggs, and life cycles with an aquatic phase, are not suitable for the dry conditions often found in Mexican insular ecosystems. These habitats are generally scarce also in resources, and on the great majority of the Mexican islands there are no permanent sources of freshwater available. Furthermore, the probability of amphibian colonization in such environments through rafting is low, as only a limited number of amphibians can endure prolonged dehydration and exposure to a saline environmentThis contrast in low amphibian and high reptile richness on islands is very evident if we compare a Mexican state similar in size to the area of the Mexican insular territories, such as Aguascalientes, which has an area of 5680.3 km$^2$, which represents 0.3% of the country's land surface. This state has a herpetofaunal diversity of 20 amphibian species and 78 reptiles [26] in contrast to the 16 amphibian species (80% of the diversity in Aguascalientes) and 210 reptile species (269.2% of the diversity in Aguascalientes) on islands. Another example is Morelos (4878.9 km$^2$, 0.25% of the Mexican territory), with 43 amphibian species and 105 reptile species [26]. Comparatively, the insular amphibians and reptiles are the 37.2 and 205.7%, respectively, of the herpetofauna of Morelos. Of particular interest is the high specific richness of insular rattlesnakes (16 spp.). Forty-five species in the genus *Crotalus* (counting *C. caliginis*) inhabit Mexico [26]. The 16 of these rattlesnake taxa (35.6%) that occur in the Mexican insular systems are restricted to the two physiographic regions associated with the peninsula of Baja California (the Gulf of California islands and the Pacific islands of Baja California). Interestingly, there are no rattlesnakes recorded from any of the other three Mexican insular systems physiographic regions. Of the sixteen insular species, eight (50.0%) are insular endemics (*Crotalus angelensis*, *C. caliginis*, *C. catalinensis* (Figure 16), *C. estebanensis*, *C. lorenzoensis*, *C. polisi*, *C. thalassoporus*, and *C. tortuguensis*), one (*C. enyo*) is a country endemic (6.3%), and the remaining seven species (43.8%) are Mexican–US species (*C. atrox*, *C. cerastes*, *C. mitchellii*, *C. molossus*, *C. pyrrhus*, *C. ruber*, and *C. tigris*). Only 3 of the 16 species (*C. enyo*, *C. mitchelli*, and *C. ruber*) inhabit islands on both sides of the Baja California peninsula.

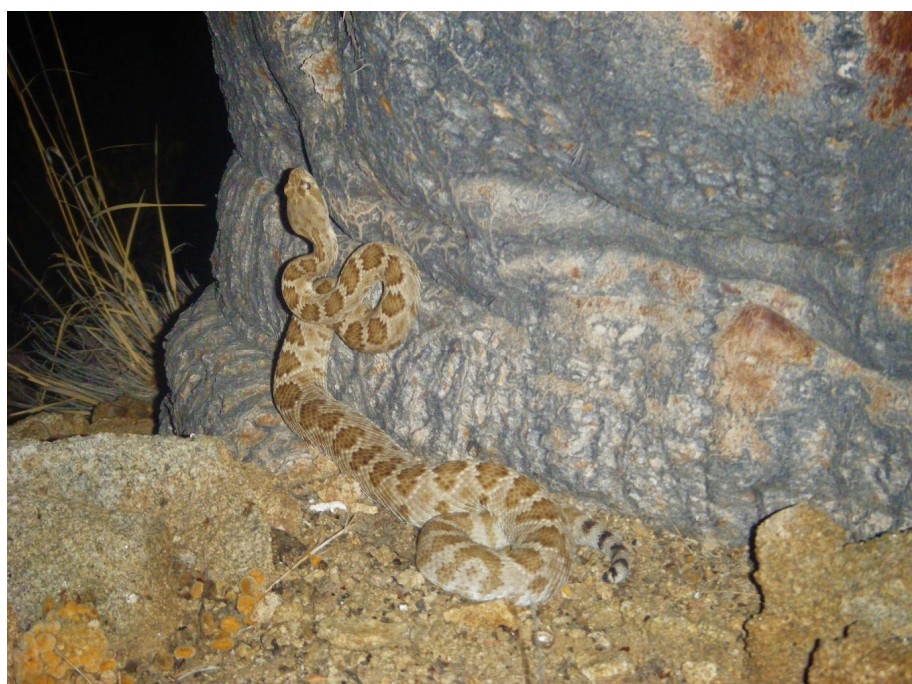

**Figure 16.** Catalina Island Rattlesnake (*Crotalus catalinensis*) searching for prey. A distinctive characteristic of this insular rattlesnake is the absence of rattle, which is reduced only to a button on the tip of the tail. Perhaps an adaptation for arboreality and/or simply a loss of this structure due to the lack of large mammals in the island, which would make the rattle useless as a warning mechanism. EVS = H(19), IUCN = CT, NOM-059 = A. Photo by Ruben Alonso Carbajal-Márquez.

### 3.2. *Patterns of Physiographic Distribution*

We utilized a system of five physiographic regions to examine the distribution of the herpetofauna of the insular systems of Mexico and documented the distribution of these species in Table 2. Finally, we present a summary of these data in Table 3.

The total number of species in the different biogeographic regions ranges from 40 to 108, ranked in descending order, as follows: Islands of the Gulf of California (108), Tropical islands of the Pacific (57), Mexican Caribbean islands (52), Islands of the Gulf of Mexico (46), and the Islands of the Pacific of Baja California [40]. Notably, the islands of the Gulf of Baja California are far richer in species than any the other physiographic regions, which have only 52.8% (Tropical Pacific), 48.2% (Caribbean), 42.6% (Golfo de México), and 37.0% (Pacific of Baja California) of the richness of the islands of the Sea of Cortes (Table 3).

**Table 3.** Summary of the distributional occurrence of herpetofaunal families in the Mexican insular systems by physiographic region. Shaded lines indicates totals/subtotals.

| Families | Number of Species | Distributional Occurrence | | | | |
|---|---|---|---|---|---|---|
| | | Gulf of California | Pacific Baja California | Tropical Pacífic | Gulf of Mexico | Caribbean Islands |
| Bufonidae | 4 | 1 | 1 | 2 | 2 | 2 |
| Eleutherodactylidae | 2 | — | — | 1 | — | 1 |
| Hylidae | 5 | — | 1 | 1 | 2 | 4 |
| Leptodactylidae | 1 | — | — | — | 1 | 1 |
| Microhylidae | 1 | — | — | 1 | 1 | — |
| Scaphiopodidae | 1 | 1 | — | — | — | — |
| **Subtotals** | **14** | **2** | **2** | **5** | **6** | **8** |
| Plethodontidae | 2 | — | 2 | — | — | — |
| **Subtotals** | **2** | **—** | **2** | **—** | **—** | **—** |

| Totals | **16** | **2** | **4** | **5** | **6** | **8** |
|---|---|---|---|---|---|---|
| Crocodylidae | 2 | — | — | 1 | 1 | 1 |
| **Subtotals** | **2** | **—** | **—** | **1** | **1** | **1** |
| Anguidae | 3 | — | 3 | — | — | — |
| Anniellidae | 2 | — | 2 | — | — | — |
| Bipedidae | 1 | — | 1 | — | — | — |
| Corytophanidae | 1 | — | — | — | 1 | 1 |
| Crotaphytidae | 4 | 3 | 1 | — | — | — |
| Dactyloidae | 6 | — | — | 1 | 3 | 5 |
| Eublepharidae | 3 | 2 | 1 | — | — | 1 |
| Gekkonidae | 3 | — | — | 2 | 2 | 2 |
| Iguanidae | 13 | 10 | 1 | 2 | 2 | 2 |
| Mabuyidae | 2 | — | — | 1 | 1 | 1 |
| Phrynosomatidae | 31 | 23 | 7 | 5 | 2 | 2 |
| Phyllodactylidae | 15 | 9 | 1 | 6 | — | — |
| Scincidae | 2 | — | 1 | — | — | 1 |
| Sphaerodactylidae | 3 | — | — | — | 1 | 3 |
| Teiidae | 23 | 13 | 2 | 5 | 5 | 5 |
| **Subtotals** | **111** | **60** | **20** | **22** | **16** | **23** |
| Boidae | 2 | — | — | 1 | 1 | 1 |
| Charinidae | 1 | 1 | 1 | — | — | — |
| Colubridae | 36 | 17 | 7 | 9 | 5 | 6 |
| Dipsadidae | 19 | 4 | 3 | 8 | 8 | 2 |
| Elapidae | 3 | 2 | — | 1 | 1 | 1 |
| Leptotyphlopidae | 3 | 1 | 1 | 2 | — | 1 |
| Natricidae | 1 | — | — | — | — | 1 |
| Typhlopidae | 1 | — | — | 1 | 1 | 1 |
| Viperidae | 18 | 15 | 4 | 1 | — | 1 |
| **Subtotals** | **84** | **40** | **16** | **23** | **16** | **14** |
| Chelonidae | 5 | 4 | — | 4 | 4 | 4 |
| Dermatemydidae | 1 | — | — | — | 1 | — |
| Dermochelyidae | 1 | 1 | — | 1 | — | — |
| Emydidae | 1 | — | — | — | — | 1 |
| Geoemydidae | 1 | — | — | — | — | 1 |
| Kinosternidae | 2 | — | — | 1 | 1 | 1 |
| Staurotypidae | 1 | — | — | — | 1 | — |
| Testudinidae | 1 | 1 | — | — | — | — |
| **Subtotals** | **13** | **6** | **—** | **6** | **7** | **7** |
| **Totals** | **210** | **106** | **36** | **52** | **40** | **44** |
| **Sum Totals** | **226** | **108** | **40** | **57** | **46** | **53** |

Since the insular elements of Mexico are scattered in both oceans, it is to be expected that no insular reptile or amphibian species would be distributed among all of the five physiographic regions we recognize. In addition, only three species occupy four of the five regions, all sea turtles (*Caretta caretta*, *Chelonia mydas* and *Eretmochelys imbricata*). Only 9 species of the total 226 (~4%) occupy three regions, including two anurans (*Rhinella horribilis* and *Smilisca baudini*), three lizards (*Iguana iguana*, *Marisora brachypoda*, *Sceloporus clarkii*, *Aspidoscelis deppii*, and the non-native gecko *Hemidactylus frenatus*), three snakes (*Hypsiglena slevini*, *Rena umilis* and the non-native blindsnake *Indotyphlops braminus*). In

addition, among these six terrestrial species, five species are restricted to the tropical regions (Tropical Pacific, Gulf of Mexico, and the Mexican Caribbean), and the sixth one, *Hypsiglena slevini*, occurs along the regions of the Pacific (Pacific of Baja California, Tropical Pacific, and Tropical islands of the Gulf).

The rest of the species are distributed in either two (51 species or 22.5%) or one (163 species or 72.1%) physiographic region(s). The mean value for insular distribution is 93.3. Thus, almost three-quarters of the 226 insular species are limited to single-island regions.

Based on the data in Table 2, there are 78 single-region species limited to the islands of the Gulf of California (as indicated below). Of these 78 species, 44 (56.4%) are insular endemics, 10 (12.8%) are country endemics, and 24 (30.8%) are non-endemics. As perhaps expected, almost half (47.8%) of the single-region species are found in the Gulf of California region. Remarkable examples of endemicity are the species of the genus *Crotalus* and *Uta*. (Figures 17 and 18) (No asterisk = Non-endemic; * = country endemic; ** = regional endemic; and ‡ = non-native).

| | |
|---|---|
| *Scaphiopus couchii* | *Aspidoscelis bacata* ** |
| *Crotaphytus dickersonae* ** | *Aspidoscelis cana* ** |
| *Crotaphytus insularis* ** | *Aspidoscelis carmenensis* ** |
| *Gambelia wislizenii* | *Aspidoscelis catalinensis* ** |
| *Coleonyx gypsicolus* ** | *Aspidoscelis celeripes* ** |
| *Ctenosaura conspicuosa* ** | *Aspidoscelis ceralbelsis* ** |
| *Ctenosaura hemilopha* * | *Aspidoscelis danheimae* ** |
| *Ctenosaura nolascensis* ** | *Aspidoscelis espiritensis* ** |
| *Dipsosaurus catalinensis* ** | *Aspidoscelis franciscensis* ** |
| *Sauromalus ater* | *Aspidoscelis martyris* ** |
| *Sauromalus hispidus* * | *Aspidoscelis pictus* ** |
| *Sauromalus klauberi* ** | *Bogertophis rosaliae* |
| *Sauromalus slevini* * | *Lampropeltis californiae* |
| *Sauromalus varius* * | *Lampropeltis catalinensis* ** |
| *Petrosaurus mearnsi* | *Masticophis barbouri* |
| *Petrosaurus repens* * | *Masticophis bilineatus* |
| *Petrosaurus slevini* ** | *Masticophis slevini* ** |
| *Petrosaurus thalassinus* * | *Phyllorhynchus decurtatus* |
| *Phrynosoma solare* | *Rhinocheilus etheridgei* ** |
| *Sceloporus angustus* ** | *Sonora savagei* * |
| *Sceloporus grandaevus* ** | *Sonora semiannulata* |
| *Sceloporus hunsakeri* * | *Tantilla planiceps* |
| *Sceloporus lineatulus* ** | *Trimorphodon lyrophanes* |
| *Sceloporus magister* | *Hypsiglena catalinae* ** |
| *Sceloporus orcutti* | *Hypsiglena chlorophaea* |
| *Uta encantadae* ** | *Micruroides euryxanthus* |
| *Uta lowei* ** | *Crotalus angelensis* ** |
| *Uta nolascensis* ** | *Crotalus atrox* |
| *Uta palmeri* ** | *Crotalus catalinensis* |
| *Uta squamata* | *Crotalus cerastes* |

*Uta tumidarostra* **

*Phyllodactylus angelensis* **

*Phyllodactylus apricus* **

*Phyllodactylus bugastrolepis* **

*Phyllodactylus coronatus* **

*Phyllodactylus homolepidurus* *

*Phyllodactylus partidus* **

*Phyllodactylus santacruzensis* **

*Phyllodactylus unctus* *

*Crotalus estebanensis* **

*Crotalus lorenzoensis* **

*Crotalus molossus*

*Crotalus polisi* **

*Crotalus pyrrhus*

*Crotalus thalassoporus* **

*Crotalus tigris*

*Crotalus tortuguensis* **

*Gopherus morafkai*

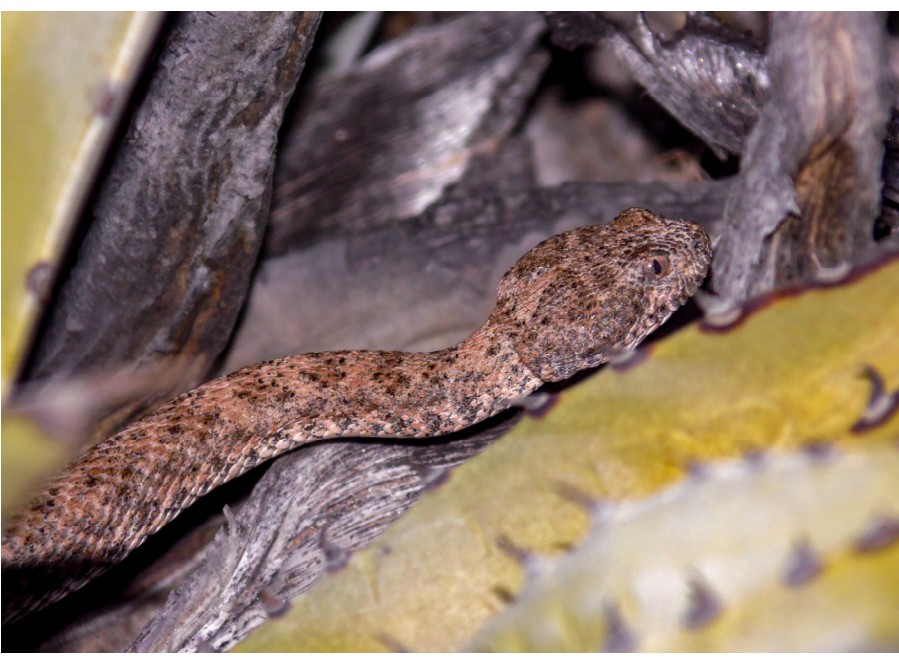

**Figure 17.** Horsehead Island Speckled Rattlesnake (*Crotalus polisi*), only known from Isla Cabeza de Caballo (Gulf of California). The islands of the Sea of Cortes are home to fifteen *Crotalus* species, this number alone is superior to the *Crotalus* species diversity of many mainland Mexican states and represents the 32.6% of all *Crotalus* species in Mexico (46). Five of them are insular endemics (11% of Mexican *Crotalus*) and, consequently only known in their type locality. Distributional status = IE, EVS = H (18), IUCN = NE, NOM-059 = NS. Photo by Tania Pérez-Fiol.

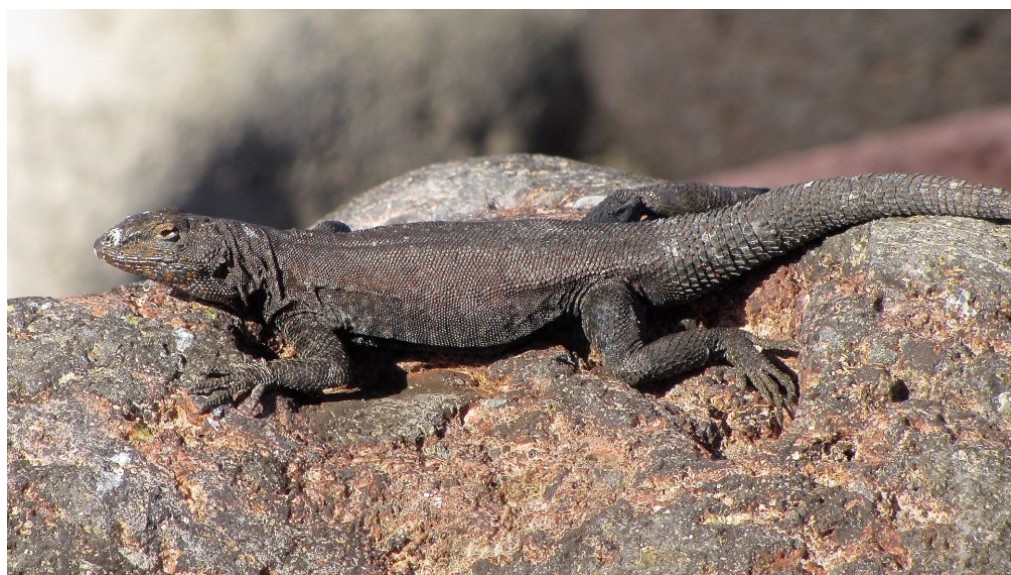

**Figure 18.** Enchanted Side-blotched Lizard (*Uta Encantadae*) from Encantada. Phrynosomatid lizards are diverse in the islas of the Sea of Cortes with 23 species. Interestingly, five of the six *Uta* species in these islands are insular endemics. Distributional status = IE, EVS = H (17), IUCN = VU, NOM-059 = NS. Photo by Jorge H Valdez.

The number of single-region species in the tropical Pacific islands is 36 (22.1%). Eight of these 36 species (22.2%) are insular endemics, 20 (55.5%) are country endemics, 7 (19.5.3%) are non-endemics, and one (2.8%) is a non-native species (See below). *Lampropeltis* and *Phyllodactylus* are typical examples of herpetofaunistical diversity within the Tres Marias Archipielago (Figure 19). (No asterisk = Non-endemic; * = country endemic; ** = insular endemic; and ‡ = non-native):

| | |
|---|---|
| *Incilius mazatlanensis* * | *Aspidoscelis lineatissima* * |
| *Eleutherodactylus pallidus* * | *Boa sigma* * |
| *Norops nebulosus* * | *Drymarchon melanurus* |
| *Gehyra mutilata* ‡ | *Lampropeltis polyzona* * |
| *Ctenosaura pectinata* | *Leptophis diplotropis* * |
| *Marisora aquilonaria* * | *Masticophis anthonyi* ** |
| *Urosaurus auriculatus* ** | *Oxybelis microphtalmus* * |
| *Urosaurus bicarinatus* * | *Tantilla bocourti* * |
| *Urosaurus clarionensis* ** | *Tantilla calamarina* * |
| *Phyllodactylus benedetii* * | *Conophis vittatus* |
| *Phyllodactylus cloefasensis* ** | *Geophis annuliferus* * |
| *Phyllodactylus isabelae* ** | *Hypsiglena torquata* |
| *Phyllodactylus lanei* * | *Hypsiglena unaocularis* ** |
| *Phyllodactylus lupitae* ** | *Imantodes gemmistratus* |
| *Phyllodactylus tuberculosus* | *Rhadinaea hesperia* * |
| *Aspidoscelis communis* * | *Epictia bakewelli* * |
| *Aspidoscelis costata* * | *Agkistrodon bilineatus* |
| *Aspidoscelis guttatus* * | *Kinosternon integrum* * |

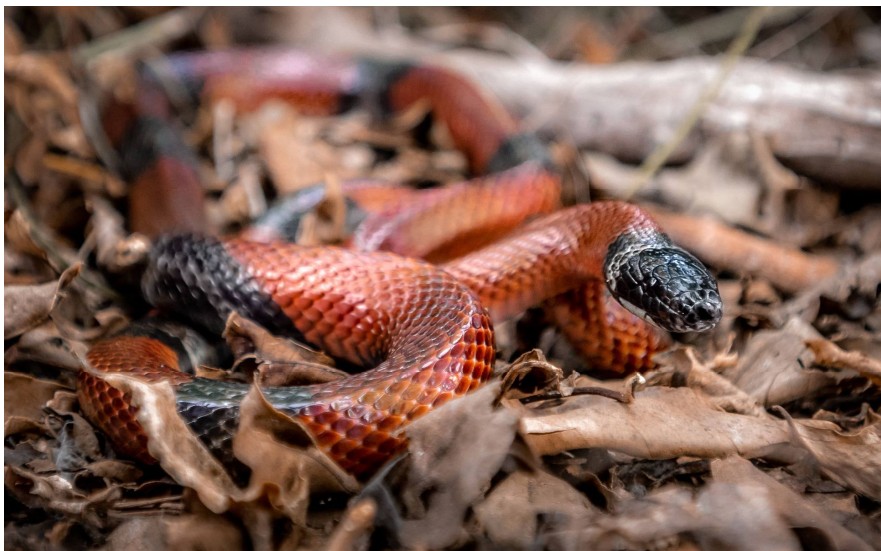

**Figure 19.** West Mexican Milksnake (*Lampropeltis polyzona*) from Isla Isabel, Tropical Pacific. Milksnakes are especially vulnerable to depredation from introduced mammals, such as the black rat (*Rattus rattus*) and feral cats (*Felis silvestris catus*). Thus, the population's number of milksnakes can constitute a reference indicator of impact from introduced mammals in islands. Distributional status = CE, EVS = M (11), IUCN = NE, NOM-059 = A. Photo by Edgar Alvarado-Rodríguez.

The number of single-region species in the Caribbean islands is 19 (11.7%). Three of these species (15.8%) are country endemics, fourteen species (73.7%) are non-endemics, and the remaining two (10.5%) are non-natives (see below). As expected, the *Gekkonidae* are conspicuous in most of these islands (Figure 20) (No asterisk = Non-endemic; * = country endemic; ** = insular endemic; and ‡ = non-native).

*Eleutherodactylus planirostris* ‡

*Dendrosophus microcephalus*

*Trachycephalus vermiculatus*

*Anolis allisoni* ‡

*Norops lemurinus*

*Coleonyx elegans*

*Sceloporus cozumelae* *

*Mesoscincus schwartzei*

*Aristelliger georgeensis*

*Sphaerodactylus continentalis*

*Aspidoscelis rodecki* *

*Leptophis mexicanus*

*Oxybelis fulgidus*

*Tantilla moesta*

*Epictia magnamaculata*

*Thamnophis proximus*

*Agkistrodon russeolus* *

*Trachemys venusta*

*Rhinoclemmys areolata*

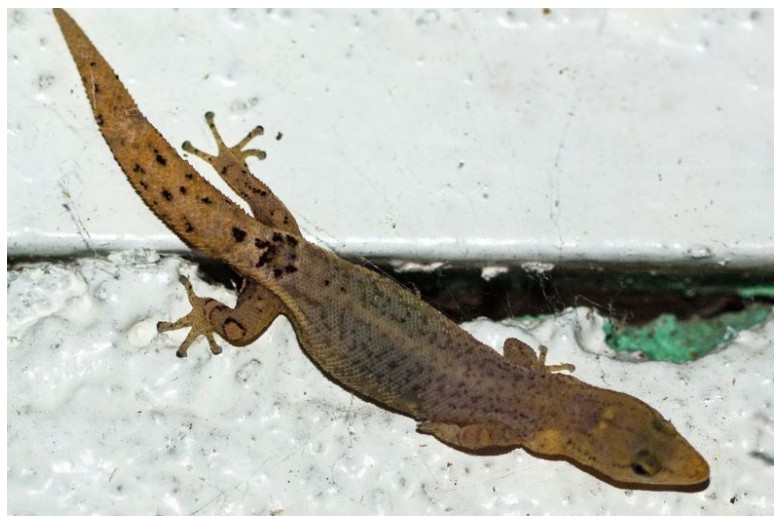

**Figure 20.** Spotted least gecko (*Sphaerodactylus continentalis*) from Cozumel. Little is known from this gecko in the Mexican Yucatan Peninsula with Cozumel being the location of its northernmost population. Distributional status = NE, EVS= M (10), IUCN = NE, NOM-059 = NS. Photo by Luis Díaz-Gamboa.

In the Pacific islands of Baja California, there are seventeen (10.4%) single-region species, of these seventeen species, four (23.5%) are insular endemics, another four (23.5%) are country endemics, and nine (52.9%) are non-endemics (See below). This is the only insular region in which worm lizards can be found (Figure 21) (No asterisk = Non-endemic; * = country endemic; ** = insular endemic; and ‡ = non-native).

*Hyliola rejilla*

*Aneides lugubris*

*Batrachoseps major*

*Elgaria cedrosensis* *

*Elgaria multicarinata*

*Elgaria nana* **

*Aniella geronimensis* *

*Aniella pulchra*

*Bipes biporus* *

*Gambelia copeii*

*Phrynosoma cerroense* *

*Sceloporus occidentalis*

*Plestiodon skiltonianus*

*Lampropeltis herrerae* **

*Pituophis insularis* **

*Diadophis punctatus*

*Crotalus caliginis* **

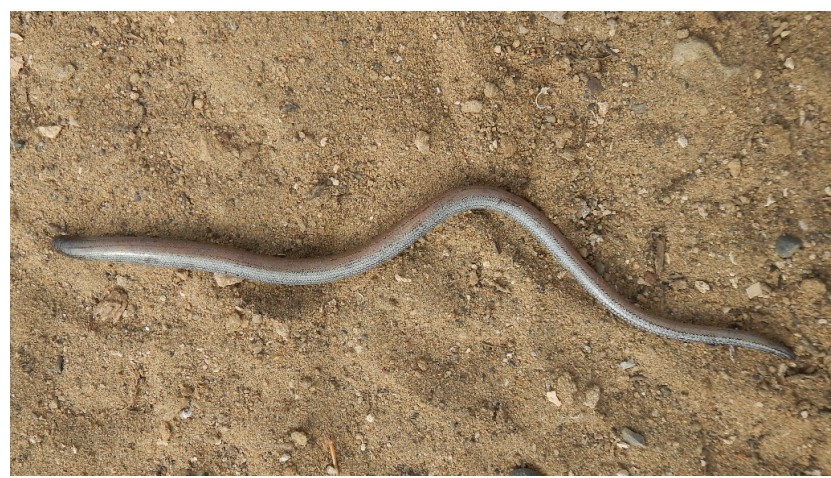

**Figure 21.** The Baja California Legless lizard (*Aniella geronimensis*) is one of the only two species of legless lizards inhabiting Mexican insular systems. Its range goes along the Pacific coast of the Mexican state of Baja California, and the islands San Martin and San Jerónimo. Although it is of secretive habits, it is common to find between the roots of the introduced iceplant (*Mesembrianthemum cristallynum*) that overpopulates the sandunes of that island (VHGS pers. Observ.) EVS = M (13), IUCN = EN, NOM-059 = Pr.

Finally, in the islands of the Gulf of Mexico, there are 13 (~8%) single-region species and all 13 of these species are non-endemics, as are enlisted below (No asterisk = Non-endemic; * = country endemic; ** = insular endemic; and ‡ = non-native):

*Crocodylus moreletii*  *Dipsas brevifacies*

*Sceloporus variabilis*  *Imantodes cenchoa*

*Drymobius margaritiferus*  *Ninia sebae*

*Lampropeltis abnormal*  *Tropidodipsas sartorii*

*Pseudelaphe flavirufa*  *Dermatemys mawii*

*Spilotes pullatus*  *Staurotypus triporcatus*

*Coniophanes imperialis*

In summary, of the 163 single-region species documented in the insular systems herpetofauna of Mexico, 56 (34.3%) are insular endemics, 37 (22.7%) are country endemics, 67 (41.1%) are non-endemics, and 3 (1.8%) are non-natives. Insular endemic species constitute the largest proportion of these species' categories only in the Gulf of California physiographic region (44 of 78 species, or 56.4%). Country endemics comprise the greatest proportion only in the Tropical Pacific Islands physiographic region (20 of 36 species or 55.5%). Non-endemic species make up the highest proportion in the Baja California Pacific Islands (9 of 17 species or 52.9%), Gulf of Mexico Islands (all 13 of 13 species or 100%), and Caribbean Islands physiographic regions (14 of 19 species or 73.7%). Non-native species are in the Tropical Pacific Islands, Gulf of Mexico, and Mexican Caribbean Islands physiographic regions (3, 4, and 6, respectively). *Hemidactylus frenatus* and *Indotyphlops braminus* are within these three regions.

In order to analyze the herpetofaunal similarity relationships among the five insular physiographic regions, we constructed a Coefficient of Biogeographic Resemblance (CBR) matrix using the algorithm of Duellman [140]. As mentioned above, the greatest species richness of 108 species is found in the Gulf of California region and the least of 40 species in the Pacific islands of Baja California. The mean species richness for the five regions is 60.6. The number of shared species between all the regional pairs ranges from 0 in two instances to 23 between the Gulf of California islands and those in the Pacific region of Baja California, and the same value between the islands of the Gulf of Mexico and those of the Caribbean Sea (Table 4). The mean value of shared species among all five regions is 7.0. As expected, the greatest similarity exists between those regions in closest proximity to one another, i.e., between the islands of the Gulf of Mexico and those of the Caribbean (also 27 species), closely followed by the shared species between the islands of the Pacific regions of Baja California and those of the Gulf of California (23 species) and, also expected, and here demonstrated, was the complete lack of species shared between the two regions associated with Baja California and the two on the eastern coast of Mexico (Figure 22).

**Table 4.** Pair-wise comparison matrix of Coefficient of Biogeographic Resemblance (CBR). Bold/Underlined values = number of species in each region; upper triangular matrix values = species in common between two regions; and lower triangular matrix values = CBR values. The formula for this algorithm is CBR = $2C/(N_1 + N_2)$, where C is the number of species in common to both regions, $N_1$ is the number of species in the first region, and $N_2$ is the number of species in the second region.

| | Gulf of California | Pacific Baja California | Tropical Pacific | Gulf of Mexico | Mexican Caribbean |
|---|---|---|---|---|---|
| **Gulf of California** | **108** | 23 | 10 | 3 | 3 |
| **Pacific Baja California** | 0.31 | **40** | 3 | 0 | 0 |
| **Tropical Pacific** | 0.123 | 0.064 | **54** | 9 | 10 |
| **Gulf of Mexico** | 0.04 | 0 | 0.187 | **42** | 27 |
| **Mexican Caribbean** | 0.0387 | 0 | 0.198 | 0.607 | **47** |

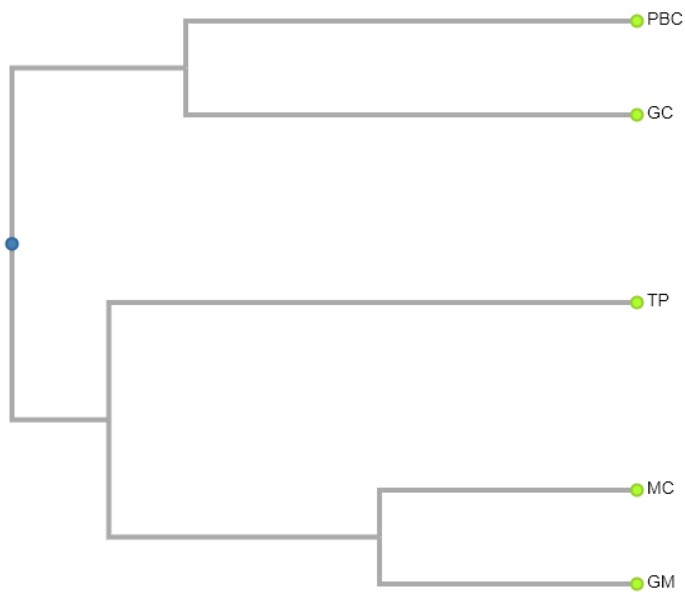

**Figure 22.** UPGMA-generated dendrogram illustrating the similarity relationships of species richness among the herpetofauna in the five physiographic regions of the Mexican Insular Systems (based on the data in Table 4). We calculated the similarity values using Duellman's (1990) Coefficient of Biogeographic Resemblance (CBR). PBC = Pacific of Baja California; GC = Gulf of California (Sea of Cortés), TP = Tropical Pacific, GM = Gulf of Mexico, MC= Mexican Caribbean ((GC:0.345, PBC:0.345):0.133,(TP:0.404,(GM:0.197,MC:0.197):0.207):0.074).

The herpetofauna of the tropical Pacific islands exhibits a limited overlap of species (ranging from three to ten) with both the Baja California regions and the two east coast Mexican regions. The species that are part of the Pacific Tropical Islands herpetofauna shared with those in the Gulf of California are as follows: *Sceloporus clarkia*; *Urosaurus ornatus*; *Hypsiglena slevini*; *Hydrophis platurus*; *Rena humilis*; *Caretta caretta*; *Chelonia mydas*; *Eretmochelys imbricata*; *Lepidochelys olivacea*; and *Dermochelys coriacea*. Of these ten species, only four are terrestrial, the first two listed are lizards and the other two are snakes; the remainder are the six marine species occurring along the Pacific shores of Mexico, one a snake and five turtles (Figure 23). The species that are part of the Pacific tropical islands herpetofauna and also those of the islands of the Gulf of Mexico (eleven species) and the islands of the Caribbean (twelve species) are as follows: *Rhinella horribilis*; *Smilisca baudinii*; *Hypopachus variolosus* (only the islands of the Gulf of Mexico); *Crocodylus acutus* (only the Caribbean islands); *Hemidactylus frenatus*; *Iguana iguana*; *Aspidoscelis deppii*; *Mastigodryas melanolomus* (only the Caribbean islands); *Oxybelis aeneus* (only the Caribbean islands);

*Sibon nebulatus* (only the islands of the Gulf of Mexico); *Indotyphlops braminus* and three sea turtles (*Caretta caretta*, *Chelonia mydas*, and *Eretmochelys imbricata*). The pattern on the eastern coast of Mexico is distinct from that seen on the Pacific coast. Taking apart the sea turtles, the species involved are all terrestrial, except for *C. acutus*, which can occupy both fresh and saltwater habitats, in addition to occurring on land. Most of the species are native to the area they occupy in the insular systems, except for two non-native species, *H. frenatus* and *I. braminus*. Three of the species are anurans, one is a crocodilian, four are lizards (including the non-native gecko *H. frenatus*), and four are snakes (including the non-native blindsnake *I. braminus*). The terrestrial species involved are among some of the most widespread species both inside and outside Mexico.

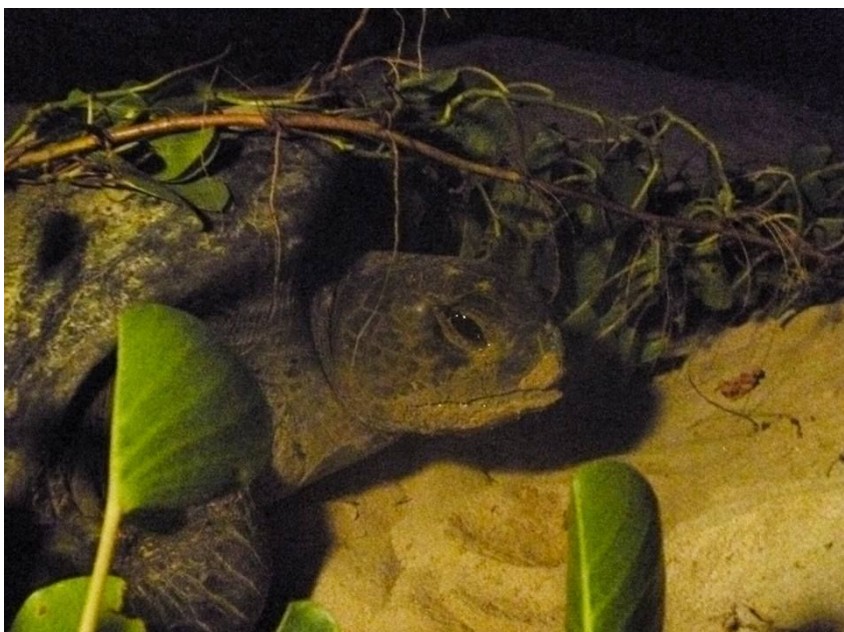

**Figure 23.** Green sea turtle (*Chelonia agassizi*) Isla Clarión (Tropical Pacific). As expected, the different physiographic regions used in this paper show little biogeographic resemblance among them. This is because of the characteristic isolation of insular systems along with the almost inexistent species turnover. It is not surprising that marine reptiles are the only species shared among several different physiographic insular regions. The sea turtles occur in the surrounding waters of many Mexican islands, but only spawn or nest in few of them. Distributional status = NE, IUCN = EN, NOM-059 = P (EVS do not apply for marine species). Photo by Humberto Almanza.

*3.3. Patterns of Distribution within Physiograph Regions*

We documented the distribution of the insular herpetofauna within each of the five insular physiographic regions recognized in a series of Tables 5–9, which are discussed below.

In Table 5, we list 17 islands that are part of the Pacific Baja California physiographic region. These islands are mapped in Figure 2. The numbers of the 40 herpetofaunal species (two anurans, two salamanders, and thirty-five squamates) occurring in these islands range from 1 to 19 (mean, 5.6). The largest number of species (19) is found on Isla Santa Margarita, one of two sandy barrier islands enclosing Bahía Magdalena (wikipedia.com; accessed 3 March 2020). The herpetofauna of this island consists of 1 anuran, 10 lizards, and 8 snakes. The other of the two barrier islands is Isla Magdalena, the herpetofauna of which contains the third largest number of species (14), including 10 lizards and 4 snakes. The second largest herpetofauna is found on Isla Cedros, situated approximately 100 km off the westernmost point of the mainland of Baja California (Punta Eugenia), at approximately the same latitude as the border between Baja California and Baja California Sur.

Cedros is the fourth largest island in Mexico, after Isla Tiburón, Isla Ángel de la Guarda, and Cozumel (wikipedia.com; accessed on 3 March 2020). The herpetofauna of this island comprises 15 species, including one anuran, eight lizards (Including the horned lizard *Phrynosoma cerroense*, Figure 24), and six snakes. Six of the seventeen islands support only one recorded species, which is the same species in all cases (*Uta stansburiana*). Not surprisingly, this phrynosomatid lizard is found on all 17 islands in this region. The remainder of the 39 species in this region occur in from one to six islands. The next most widely distributed species is *Aspidoscelis tigris*, which occupies six islands in this region. The rest of the 39 species occur in these islands as follows (Table 5): one island (13 species); two islands (17 species); three islands (three species); four islands (three species); and five islands (one species).

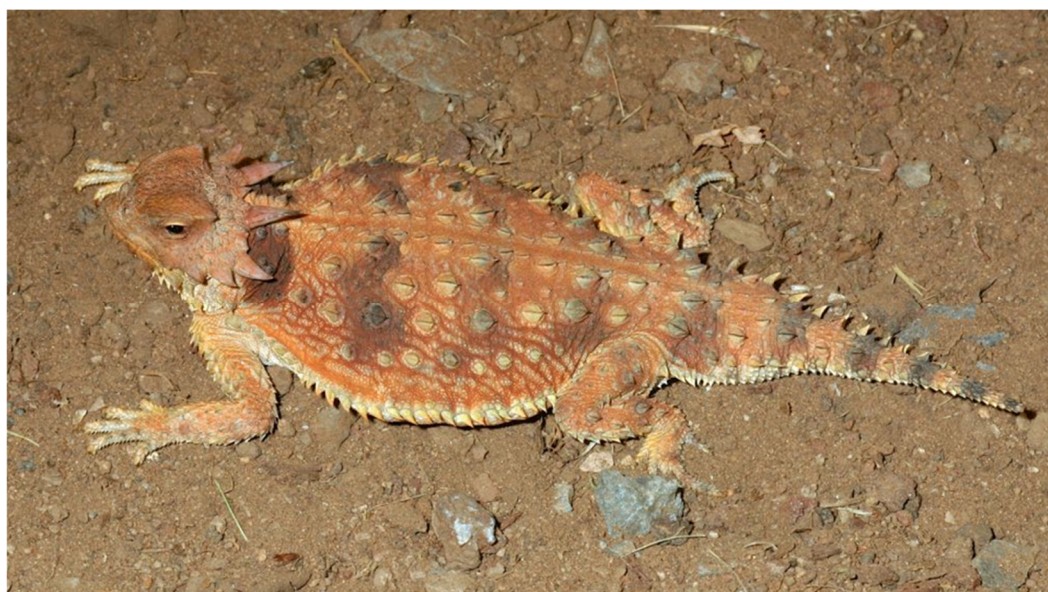

**Figure 24.** Cedros island horned lizard (*Phrynosoma cerroense*). The only horned lizard on the Mexican islands, despite its common name, it is not exclusive of Cedros Island, but is endemic from the Baja California Peninsula (as is this individual photographed in San Fernando, in the Vizcaino desert in mainland Baja California). Distributional Status = CE, EVS = H (16), IUCN = NE, NOM-059 = A. Photo from the Amphibian and Reptile Atlas of Peninsular California ((herpatlas.sdnhm.org) San Diego Natural History Museum), courtesy of Bradford Hollingsworth.

**Table 5.** Distribution of the insular herpetofauna in the Pacific Baja California physiogaphic region. Notes: Islas San Benito comprises an archipielago of three islands: San Benito Oeste, San Benito Medio, and San Benito Este, *Uta stansburiana* is present in all three islands (VHGS pers. Observ.). No asterisk = Non-endemic; * = country endemic; ** = insular endemic.

| Taxa | Asunción | Cedros | Coronado Sur | Coronado medio | Coronado norte | Magdalena | Natividad | Islas San Benito | Las Brosas | Las Piedras | Pata | San Martín | San Jerónimo | San Roque | Santa Margarita | Todos Santos Sur | Todos Santos Norte | Total Number of Islands |
|---|---|---|---|---|---|---|---|---|---|---|---|---|---|---|---|---|---|---|
| | 1 | 2 | 3 | 4 | 5 | 6 | 7 | 8 | 9 | 10 | 11 | 12 | 13 | 14 | 15 | 16 | 17 | |
| **Amphibians** | | | | | | | | | | | | | | | | | | |
| **Anura** | | | | | | | | | | | | | | | | | | |
| **Bufonidae** | | | | | | | | | | | | | | | | | | |
| *Anaxyrus punctatus* | | | | | | | | | | | | | | | + | | | 1 |
| **Hylidae** | | | | | | | | | | | | | | | | | | |
| *Hyliola regilla* | | + | | | | | | | | | | | | | | | | 1 |
| **Caudata** | | | | | | | | | | | | | | | | | | |
| **Plethodontidae** | | | | | | | | | | | | | | | | | | |
| *Aneides lugubris* | | | | | + | | | | | | | | | | | | | 1 |
| *Batrachoseps major* | | | + | + | + | | | | | | | | | | | + | | 4 |
| **Reptiles** | | | | | | | | | | | | | | | | | | |
| **Squamata** | | | | | | | | | | | | | | | | | | |
| **Anguidae** | | | | | | | | | | | | | | | | | | |
| *Elgaria cedrosensis* * | | + | | | | | | | | | | | | | | | | 1 |
| *Elgaria multicarinata* | | | | | | | | | | | | + | | | | | | 1 |
| *Elgaria nana* ** | | | + | | + | | | | | | | | | | | | | 2 |
| **Anniellidae** | | | | | | | | | | | | | | | | | | |
| *Anniella geronimensis* * | | | | | | | | | | | | + | + | | | | | 2 |
| *Anniella pulchra* | | | + | | + | | | | | | | | | | | + | + | 4 |
| **Bipedidae** | | | | | | | | | | | | | | | | | | |
| *Bipes biporus* * | | | | | | + | | | | | | | | | | | | 1 |
| **Crotaphytidae** | | | | | | | | | | | | | | | | | | |

| Species | 1 | 2 | 3 | 4 | 5 | 6 | 7 | 8 | 9 | 10 | 11 | 12 | 13 | 14 | 15 | 16 | 17 | Total |
|---|---|---|---|---|---|---|---|---|---|---|---|---|---|---|---|---|---|---|
| *Gambelia copeii* | | + | | | | + | | | | | | | | | + | | | 3 |
| **Eublepharidae** | | | | | | | | | | | | | | | | | | |
| *Coleonyx variegatus* | | + | | | | | | | | | | | | | + | | | 2 |
| **Iguanidae** | | | | | | | | | | | | | | | | | | |
| *Dipsosaurus dorsalis* | | | | | | + | | | | | | | | | + | | | 2 |
| **Phrynosomatidae** | | | | | | | | | | | | | | | | | | |
| *Callisaurus draconoides* | | | | | | + | | | | | | | | | + | | | 2 |
| *Phrynosoma cerroense* * | | + | | | | | | | | | | | | | | | | 1 |
| *Sceloporus occidentalis* | | + | | | | | | | | | | | | | | + | | 2 |
| *Sceloporus zosteromus* * | | + | | | | + | | | | | | | | | + | | | 3 |
| *Urosaurus nigricaudus* * | | | | | | + | | | | | | | | | + | | | 2 |
| *Uta stansburiana* | + | + | + | + | + | + | + | + | + | + | + | + | + | + | + | + | + | 17 |
| **Phyllodactylidae** | | | | | | | | | | | | | | | | | | |
| *Phyllodactillus nocticolus* | | | | | | + | | | | | | | | | + | | | 2 |
| **Scincidae** | | | | | | | | | | | | | | | | | | |
| *Plestiodon skiltonianus* | | | + | | + | | | | | | | | | | + | + | | 4 |
| **Teiidae** | | | | | | | | | | | | | | | | | | |
| *Aspidoscelis hyperythrus* | | | | | | + | | | | | | | | | + | | | 2 |
| *Aspidoscelis tigris* | | + | + | | + | + | + | | | | | | | | + | | | 6 |
| **Charinidae** | | | | | | | | | | | | | | | | | | |
| *Lichanura trivirgata* | | + | | | | | | | | | | | | | + | | | 2 |
| **Colubridae** | | | | | | | | | | | | | | | | | | |
| *Lampropeltis herrerae* ** | | | | | | | | | | | | | | | | + | | 1 |
| *Masticophis fuliginosus* | | | | | | + | | | | | | | | | + | | | 2 |
| *Pituophis catenifer* | | | + | | | | | | | | | + | | | | | | 2 |
| *Pituophis insulanus* ** | | + | | | | | | | | | | | | | | | | 1 |
| *Pituophis vertebralis* * | | | | | | + | | | | | | | | | + | | | 2 |
| *Salvadora hexalepis* | | | | | | | | | | | | | + | | | + | | 2 |
| *Sonora straminea* | | + | | | | + | | | | | | | | | + | | | 3 |
| **Dipsadidae** | | | | | | | | | | | | | | | | | | |
| *Diadophis punctatus* | | | | | | | | | | | | + | | | | + | | 2 |
| *Hypsiglena ochrorhyncus* | | + | + | + | + | | | | | | | + | | | | | | 5 |
| *Hypsiglena slevini* * | | | | | | | | | | | | | | | + | | | 1 |
| **Leptotyphlopidae** | | | | | | | | | | | | | | | | | | |
| *Rena humilis* | | + | | | | | | | | | | | | | | | | 1 |

| | 1 | 15 | 9 | 3 | 8 | 14 | 2 | 1 | 1 | 1 | 1 | 6 | 3 | 1 | 19 | 8 | 3 | |
|---|---|---|---|---|---|---|---|---|---|---|---|---|---|---|---|---|---|---|
| **Viperidae** | | | | | | | | | | | | | | | | | | |
| *Crotalus caliginis* ** | | | + | | | | | | | | | | | | | | | 1 |
| *Crotalus enyo* * | | | | | | + | | | | | | | | | + | | | 2 |
| *Crotalus mitchellii* | | | | | | | | | | | | | | | + | | | 1 |
| *Crotalus ruber* | | + | | | | | | | | | | | | | + | | | 2 |
| **Totals (39 species)** | 1 | 15 | 9 | 3 | 8 | 14 | 2 | 1 | 1 | 1 | 1 | 6 | 3 | 1 | 19 | 8 | 3 | — |

**Documentation:** 1, 4–5, 7–8, 12–14, 16 ([45,71] Samaniego et al., 2007; Grismer 2002); 2–3 ([45,71,72] Samaniego et al., 2007; Grismer 2002; Wallach et al., 2014); 6, 17 ([45,71,115] Samaniego et al., 2007; Grismer 2002; Peralta 2007); 9–11 ([71] Grismer 2002); 15 ([45,71,72,115,141] Samaniego et al., 2007; Grismer 2002; Meik et al., 2015; Wallach et al., 2014; Peralta 2007).

In Table 6, we have placed the 108 species distributed on the islands of the Gulf of California. Some of these islands are illustrated in Figure 4. In this large table we have included data on 70 islands in this physiographic region, which makes it the most substantial compendium in this paper. The numbers of the 108 herpetofaunal species (2 anurans, 96 squamates, and 1 turtle, minus the marine species) found on these islands range from an occupancy of 1 to 43 (mean, 5.9). The most widely distributed species on these islands is the lizard *Uta stansburiana*, which is the same species as is most widely distributed on the Pacific islands of Baja California. The greatest number of species (28) occurs on Isla Tiburón, the large island lying off the coast of Sonora. The herpetofauna of this island comprises 2 anurans, 12 lizards, 13 snakes, and 1 turtle. The second largest herpetofauna (23 species) is found on Isla San Marcos and consists of 11 lizards and 12 snakes. Seventeen of the seventy islands support but a single recorded species. The remainder of the species in these regions occupy the following numbers (Table 6): two islands (11 species); three islands (10); four (4); five (6); six (3); seven (3); eight (1); nine (1); ten (2); thirteen (1); fifteen (1); sixteen (1); seventeen (2); twenty (2); twenty-one (2); twenty-three (1); and twenty-eight (1).

Table 7 documents the herpetofauna of the 16 islands in the Tropical Pacific region. Several of these islands are illustrated in Figure 6. The total number of islands occupied by the 54 resident species (five anurans, one crocodilian, forty-one squamates, and six turtles) vary from 1 to 13 (mean, 2.9). The greatest number of species (22) is recorded from Isla María Madre, the largest of the four principal islands making up the Islas Marías (wikipedia.com; accessed 4 March 2020). The second and third next-largest herpetofaunal segments are found on the second and third largest islands in the archipelago, i.e., Isla María Magdalena (18 species) and Isla María Cleofas (14 species). The remaining island in this archipelago is Isla San Juanito, with seven species. Of the total number of species (30, including *Dermochelys coriacea,* recorded only from the archipelago in general without documentation on a specific island; see Table 7) inhabiting the Islas Marías archipelago, including four anurans, one crocodylian, twenty squamates, and four turtles, none are regional endemics, whereas eleven are country endemics (with *P. cleofasensis* as the most reciente description of an insular endemic species, Figure 25); no non-native species are recorded and nineteen are non-endemics. Species found on all four of the major islands in this archipelago are *Ctenosaura pectinata* and *Aspidoscelis communis*; otherwise, there are 11 species inhabiting three of the four islands, 3 on two, and 13 on a single island (*D. coriacea* has to be excused from this accounting). Lying between the Islas Marías archipelago and the mainland of Nayarit is Isla Isabela, which is designated as a national park. Eleven species are known to occur on this small island, including one anuran and ten squamates (Table 7). Of these eleven species, three are the non-native species *Gehyra mutilata*, *Hemidactylus frenatus*, and *Indotyphlops braminus*, five are non-endemics, two are country endemics, and one is a regional endemic. In summary, the numbers of species found on the 17 islands in this region range from 4 to 22. The most broadly distributed species in this region is *Ctenosaura pectinata*, which occupies 13 islands. The next most widely ranging species is *Norops nebulosus* on 10 islands. The remainder of the 53 species occur on the islands of this region as follows (Table 7): one island (22 species); two islands (5 species); three islands (12 species); four islands (6 species); five islands (1 species); six islands (2 species); and seven islands (3 species). Another group of notable islands in this physiographic region is the Las Marietas, which are uninhabited and located a few kilometers off Punta Mita in extreme southwestern Nayarit. The known herpetofauna of these islands comprises ten species, including eight squamates and two sea turtles. These ten species include nine non-endemics and one country endemic.

**Table 6.** (Part 1). Distribution of the insular herpetofauna of the region of the Gulf of California. Numbers below island names refer to numbered references at the bottom of the table providing documentation for data in body of table. (Part 2). Distribution of the insular herpetofauna of the region of the Gulf of California. Numbers below island names refer to numbered references at the bottom of the table providing documentation for data in body of table. No asterisk = Non-endemic; * = country endemic; ** = insular endemic; and ‡ = non-native. Shaded grey cells only to facilitate reading.

**(Part 1)**

**Islands of the Gulf of California**

| Taxa | Alcatraz (Pelícano) | Ángel de la Guarda | Isla Ballena | Isla Bota | Cabeza de caballo | Cardonosa Este | Carmen | Cayo | Cerraja | Isla Cerralvo | Cholludo | Coloradito (Lobos) | Coronados | Danzante | Dátil (Turners) | El Coyote | El Muerto | El Pardito | El requesón | Coloradita | Encantada | Espíritu Santo | Isla flecha | Farallon de San Ignacio | Gallina | Gallo | Gaviota | Granito | Islitas | Lagartija | La ventana | Las ánimas | Las galeras | Mejía | Mitlán | Total number of islands |
|---|---|---|---|---|---|---|---|---|---|---|---|---|---|---|---|---|---|---|---|---|---|---|---|---|---|---|---|---|---|---|---|---|---|---|---|---|
| | 1 | 2 | 3 | 4 | 5 | 6 | 7 | 8 | 9 | 10 | 11 | 12 | 13 | 14 | 15 | 16 | 17 | 18 | 19 | 20 | 21 | 22 | 23 | 24 | 25 | 26 | 27 | 28 | 29 | 30 | 31 | 32 | 33 | 34 | 35 | |
| **Amphibia** | | | | | | | | | | | | | | | | | | | | | | | | | | | | | | | | | | | | |
| **Anura** | | | | | | | | | | | | | | | | | | | | | | | | | | | | | | | | | | | | |
| **Bufonidae** | | | | | | | | | | | | | | | | | | | | | | | | | | | | | | | | | | | | |
| *Anaxyrus punctatus* | | | | | | | | | | + | | | | | | | | | | | | + | | | | | | | | | | | | | | 2 |
| **Scaphiopodidae** | | | | | | | | | | | | | | | | | | | | | | | | | | | | | | | | | | | | |
| *Scaphiopus couchii* | | | | | | | | | | + | | | | | | | | | | | | + | | | | | | | | | | | | | | 2 |
| **Reptiles** | | | | | | | | | | | | | | | | | | | | | | | | | | | | | | | | | | | | |
| **Squamata** | | | | | | | | | | | | | | | | | | | | | | | | | | | | | | | | | | | | |
| **Crotaphytidae** | | | | | | | | | | | | | | | | | | | | | | | | | | | | | | | | | | | | |
| *Crotaphytus dickersonae* ** | | | | | | | | | | | | | | | | | | | | | | | | | | | | | | | | | | | | |
| *Crotaphytus insularis* ** | | + | | | | | | | | | | | | | | | | | | | | | | | | | | | | | | | | | | 1 |
| *Gambelia wislizenii* | | | | | | | | | | | | | | | | | | | | | | | | | | | | | | | | | | | | |
| **Eublepharidae** | | | | | | | | | | | | | | | | | | | | | | | | | | | | | | | | | | | | |
| *Coleonyx gypsicolus* ** | | | | | | | | | | | | | | | | | | | | | | | | | | | | | | | | | | | | |
| *Coleonyx variegatus* | | + | | | | | | | | | | | + | + | | | | | | | | + | | | | | | | | | | | | | | 4 |
| **Gekkonidae** | | | | | | | | | | | | | | | | | | | | | | | | | | | | | | | | | | | | |
| *Hemidactylus frenatus* ‡ | | | | | | | | | | | | | | | | | + | | | | | | | | | | | | | | | | | | | |
| **Iguanidae** | | | | | | | | | | | | | | | | | | | | | | | | | | | | | | | | | | | | |
| *Ctenosaura conspicuosa* ** | | | | | | | | | | + | | | | | | | | | | | | | | | | | | | | | | | | | | 1 |
| *Ctenosaura hemilopha* * | | | | | | | | | | + | | | | | | | | | | | | | | | | | | | | | | | | | | 1 |
| *Ctenosaura nolascensis* ** | | | | | | | | | | | | | | | | | | | | | | | | | | | | | | | | | | | | |
| *Dipsosaurus catalinensis* ** | | | | | | | | | | | | | | | | | | | | | | | | | | | | | | | | | | | | |
| *Dipsosaurus dorsalis* | | + | | | | | + | | | + | | | + | | | | | + | | | | + | | | | | | | | | | | | | | 6 |
| *Sauromalus ater* | | | + | | | | | | | | | | | | | + | | | | | | + | | | | | + | | | | | | | | | 4 |
| *Sauromalus hispidus* * | | + | | | + | | | | | | | | | | | | | | | | | | | | | | | | | | + | | + | + | + | 6 |
| *Sauromalus klauberi* ** | | | | | | | | | | | | | | | | | | | | | | | | | | | | | | | | | | | | |

| Species | Count |
|---|---|
| *Sauromalus slevini* * | 3 |
| *Sauromalus* spp. | 1 |
| *Sauromalus varius* * | |
| **Phrynosomatidae** | |
| *Callisaurus draconoides* | 6 |
| *Petrosaurus mearnsi* | 1 |
| *Petrosaurus repens* * | 1 |
| *Petrosaurus slevini* ** | 2 |
| *Petrosaurus thalassinus* * | 1 |
| *Phrynosoma solare* | |
| *Sceloporus angustus* ** | |
| *Sceloporus clarkia* | |
| *Sceloporus grandaevus* ** | 1 |
| *Sceloporus hunsakeri* * | 3 |
| *Sceloporus lineatulus* ** | |
| *Sceloporus magister* | |
| *Sceloporus orcutti* | 2 |
| *Sceloporus zosteromus* * | 3 |
| *Urosaurus nigricaudus* * | 13 |
| *Urosaurus ornatus* | 1 |
| *Uta encantadae* ** | 1 |
| *Uta lowei* ** | 1 |
| *Uta nolascensis* ** | |
| *Uta palmeri* ** | |
| *Uta squamata* | |
| *Uta stansburiana* | 21 |
| *Uta tumidarostra* ** | 2 |
| **Phyllodactylidae** | |
| *Phyllodactylus angelensis* ** | |
| *Phyllodactylus apricus* ** | |
| *Phyllodactylus bugastrolepis* ** | |
| *Phyllodactylus coronatus* ** | |
| *Phyllodactylus homolepidurus* * | 1 |
| *Phyllodactylus nocticolus* | 14 |
| *Phyllodactylus partidus* ** | 1 |
| *Phyllodactylus santacruzensis* ** | |
| *Phyllodactylus unctus* * | 5 |
| **Teiidae** | |
| *Aspidoscelis bacata* ** | |
| *Aspidoscelis cana* ** | |
| *Aspidoscelis carmenensis* ** | 1 |
| *Aspidoscelis catalinensis* ** | |
| *Aspidoscelis celeripes* ** | |
| *Aspidoscelis ceralbelsis* ** | 1 |
| *Aspidoscelis danheimae* ** | |

| Species | | | | | | | | | | | | n |
|---|---|---|---|---|---|---|---|---|---|---|---|---|
| *Aspidoscelis espiritensis* ** | | | | | | | | | + | | | 1 |
| *Aspidoscelis franciscensis* ** | | | | | | | | | | | | |
| *Aspidoscelis hyperythrus* | | | | | + | | | | | | | 1 |
| *Aspidoscelis martyris* ** | | | | | | | | | | | | |
| *Aspidoscelis pictus* ** | | | | | | | | | | | | |
| *Aspidoscelis tigris* | + | + | + | | + | + | | | + | + | | 7 |
| **Charinidae** | | | | | | | | | | | | |
| *Lichanura trivirgata* | + | | + | + | + | | | | + | | | 5 |
| **Colubridae** | | | | | | | | | | | | + |
| *Bogertophis rosaliae* | | | | | | + | | | | | | 1 |
| *Lampropeltis californiae* | + | | | + | | | | | | | | 2 |
| *Lampropeltis catalinensis* ** | | | | | | | | | | | | |
| *Masticophis barbouri*** | | | | | | | | | + | | | 1 |
| *Masticophis bilineatus* | | | | | | | | | | | | |
| *Masticophis fuliginosus* | | | + | + | + | + | + | | + | | | 6 |
| *Masticophis slevini* ** | | | | | | | | | | | | |
| *Phyllorhynchus decurtatus* | + | | | + | | | | | | | | 2 |
| *Pituophis catenifer* | | | | | | | | | | | | |
| *Pituophis vertebralis* * | | | | | | | | | | | | |
| *Rhinocheilus etheridgei* ** | | | | + | | | | | | | | 1 |
| *Salvadora hexalepis,* | | | + | | | | | | + | | | 1 |
| *Sonora savagei* * | | | | + | | | | | | | | 1 |
| *Sonora semiannulata* | | | | | | | | | | | | |
| *Sonora straminea* | | | | | | + | | | + | | | 2 |
| *Tantilla planiceps* | | | + | | | | | | | | | 1 |
| *Trimorphodon lyrophanes* | | | | + | | + | | + | | | | 3 |
| **Dipsadidae** | | | | | | | | | | | | |
| *Hypsiglena catalinae* ** | | | | | | | | | | | | |
| *Hypsiglena chlorophaea* | | | | | | | | | | | | |
| *Hypsiglena ochrorhynchus* | + | | + | + | + | + | | + | | | | 6 |
| *Hypsiglena slevini* * | | | | + | + | + | | | | | + | 4 |
| **Elapidae** | | | | | | | | | | | | |
| *Micruroides euryxanthus* | | | | | | | | | | | | |
| **Leptotyphlopidae** | | | | | | | | | | | | |
| *Rena humilis* | | | + | + | | + | | + | | | | 4 |
| **Viperidae** | | | | | | | | | | | | |
| *Crotalus angelensis* ** | | + | | | | | | | | | | 1 |
| *Crotalus atrox* | | | | | | | + | | | | | 1 |
| *Crotalus catalinensis* ** | | | | | | | | | | | | |
| *Crotalus cerastes* | | | | | | | | | | | | |
| *Crotalus enyo* * | | | + | + | + | | | | + | | | 4 |
| *Crotalus estebanensis* ** | | | | | | | | | | | | |
| *Crotalus lorenzoensis* ** | | | | | | | | | | | | |
| *Crotalus mitchellii* | | | + | + | | | | | + | | | 3 |
| *Crotalus molossus* | | | | | | | | | | | | |

| Taxa | | Total |
|---|---|---|
| *Crotalus polisi* ** | + | 1 |
| *Crotalus pyrrhus* | + | 1 |
| *Crotalus ruber* | + ... + + | 3 |
| *Crotalus thalassoporus* ** | | |
| *Crotalus tigris* | | |
| *Crotalus tortuguensis* ** | | |
| **Testudines** | | |
| **Testudinidae** | | |
| *Gopherus morafkai* | + | 1 |
| **Totals** | 3 15 5 1 3 3 18 2 1 20 2 1 17 16 5 3 8 3 1 1 1 21 1 3 2 5 1 2 1 1 3 2 1 6 2 | — |

**(Part 2).**

**Islands of the Gulf of California**

| Taxa | Monserrate | Moscas | Pardo | Partida Norte | Partida Sur | Pata | Patos | Piojo | Pond (estanque) | Rasa | Roca Lobos | Salsipuedes | San Cosme | San Damián | San Diego | San Esteban | San Francisco | San Ildefonso | San José | San Lorenzo Norte | San Lorenzo Sur | San Luis | San Marcos | San Pedro Mártir | San Pedro Nolasco | Santa Catalina | Santa Cruz | Santiago | Smith | Tiburón | Tijeras | Tortuga | San Luis Gonzaga | Islas Santa Inez | Islotes Blancos | Total number of islands |
|---|---|---|---|---|---|---|---|---|---|---|---|---|---|---|---|---|---|---|---|---|---|---|---|---|---|---|---|---|---|---|---|---|---|---|---|---|
| | 36 | 37 | 38 | 39 | 40 | 41 | 42 | 43 | 44 | 45 | 46 | 47 | 48 | 49 | 50 | 51 | 52 | 53 | 54 | 55 | 56 | 57 | 58 | 59 | 60 | 61 | 62 | 63 | 64 | 65 | 66 | 67 | 68 | 69 | 70 | |
| **Amphibia** | | | | | | | | | | | | | | | | | | | | | | | | | | | | | | | | | | | | |
| **Anura** | | | | | | | | | | | | | | | | | | | | | | | | | | | | | | | | | | | | |
| **Bufonidae** | | | | | | | | | | | | | | | | | | | | | | | | | | | | | | | | | | | | |
| *Anaxyrus punctatus* | | | | | + | | | | | | | | | | | | | | | | | | | | | | | | | + | | | | | | 4 |
| **Scaphiopodidae** | | | | | | | | | | | | | | | | | | | | | | | | | | | | | | | | | | | | |
| *Scaphiopus couchii* | | | | | + | | | | | | | | | | | | | | | | | | | | | | | | | + | | | | | | 4 |
| **Reptiles** | | | | | | | | | | | | | | | | | | | | | | | | | | | | | | | | | | | | |
| **Squamata** | | | | | | | | | | | | | | | | | | | | | | | | | | | | | | | | | | | | |
| **Crotaphytidae** | | | | | | | | | | | | | | | | | | | | | | | | | | | | | | | | | | | | |
| *Crotaphytus dickersonae* | | | | | | | | | | | | | | | | | | | | | | | | | | | | | | + | | | | | | 1 |
| *Crotaphytus insularis* ** | | | | | | | | | | | | | | | | | | | | | | | | | | | | | | | | | | | | 1 |
| *Gambelia wislizenii* | | | | | | | | | | | | | | | | | | | | | | | | | | | | | | + | | | | | | 1 |
| **Eublepharidae** | | | | | | | | | | | | | | | | | | | | | | | | | | | | | | | | | | | | |
| *Coleonyx gypsicolus* | | | | | | | | | | | | | | | | | | | | | | | + | | | | | | | | | | | | | 1 |
| *Coleonyx variegatus* | | | | | + | | | | | | | | | | | | | | + | | | | + | | | | | | | + | | | | + | | 9 |
| **Iguanidae** | | | | | | | | | | | | | | | | | | | | | | | | | | | | | | | | | | | | |
| *Ctenosaura conspicuosa* | | | | | | | | | | | | | | | | + | | | | | | | | | | | | | | | | | | | | 2 |
| *Ctenosaura hemilopha* | | | | | | | | | | | | | | | | | | | | | | | | | | | | | | | | | | | | 1 |
| *Ctenosaura nolascens* | | | | | | | | | | | | | | | | | | | | | | | | | + | | | | | | | | | | | 1 |
| *Dipsosaurus catalinensis* | | | | | | | | | | | | | | | | | | | | | | | | | | + | | | | | | | | | | 1 |
| *Dipsosaurus dorsalis* | + | | | | + | | | | | | | | | | | | | | + | + | + | | | | | | | + | | | | | | | | 12 |
| *Sauromalus ater* | | + | | | + | | | | | | | | + | + | + | | | | | | | | + | | | + | | | | + | | | | + | | 13 |

| Species | Count |
|---|---|
| *Sauromalus hispidus* ** | 12 |
| *Sauromalus klauberi* | 1 |
| *Sauromalus slevini* | 4 |
| *Sauromalus spp.* | 1 |
| *Sauromalus varius* | 2 |
| **Phrynosomatidae** | |
| *Callisaurus draconoides* | 15 |
| *Petrosaurus mearnsi* | 1 |
| *Petrosaurus repens* | 1 |
| *Petrosaurus slevini* ** | 2 |
| *Petrosaurus thalassinus* | 2 |
| *Phrynosoma solare* | 1 |
| *Sceloporus angustus* | 2 |
| *Sceloporus clarkii* | 2 |
| *Sceloporus grandaevus* | 1 |
| *Sceloporus hunsakeri* | 4 |
| *Sceloporus lineatulus* | 1 |
| *Sceloporus magister* | 1 |
| *Sceloporus orcutti* | 7 |
| *Sceloporus zosteromus* | 6 |
| *Urosaurus nigricaudus* | 21 |
| *Urosaurus ornatus* | 2 |
| *Uta encantadae* | 2 |
| *Uta lowei* | 1 |
| *Uta nolascensis* | 1 |
| *Uta palmeri* | 1 |
| *Uta squamata* | 1 |
| *Uta stansburiana* | 43 |
| *Uta tumidarostra* | 2 |
| **Phyllodactylidae** | |
| *Phyllodactylus angelensis* ** | 6 |
| *Phyllodactylus apricus* ** | |
| *Phyllodactylus bugastrolepis* | 1 |
| *Phyllodactylus homolepidurus* | 2 |
| *Phyllodactylus nocticolus* | 32 |
| *Phyllodactylus partidus* ** | 2 |
| *Phyllodactylus santacruzensis* ** | |
| *Phyllodactylus unctus* | 6 |
| **Teiidae** | |
| *Aspidoscelis bacata* | 1 |
| *Aspidoscelis cana* | 3 |
| *Aspidoscelis carmenensis* | 1 |
| *Aspidoscelis catalinensis* | 1 |
| *Aspidoscelis celeripes* | 2 |
| *Aspidoscelis ceralbelsis* | 1 |

| Species | 1 | 2 | 3 | 4 | 5 | 6 | 7 | 8 | 9 | 10 | 11 | 12 | 13 | 14 | 15 | 16 | 17 | 18 | 19 | 20 | 21 | 22 | Count |
|---|---|---|---|---|---|---|---|---|---|---|---|---|---|---|---|---|---|---|---|---|---|---|---|
| *Aspidoscelis danheimae* | | | | | | | | | | | + | | | | | | | | | | | | 1 |
| *Aspidoscelis espiritensis* | | | | + | | | | | | | | | | | | | | | | | | | 2 |
| *Aspidoscelis franciscensis* | | | | | | | | | + | | | | | | | | | | | | | | 1 |
| *Aspidoscelis hyperythrus* | | | | | | | | | | | | | | + | | | | | | | | | 2 |
| *Aspidoscelis martyris* | | | | | | | | | | | | | | | + | | | | | | | | 1 |
| *Aspidoscelis pictus* | + | | | | | | | | | | | | | | | | | | | | | | 1 |
| *Aspidoscelis tigris* | | | + | + | + | | + | + | | | | | | + | | | | | | + | + | | 15 |
| **Serpentes** | | | | | | | | | | | | | | | | | | | | | | | |
| **Charinidae** | | | | | | | | | | | | | | | | | | | | | | | |
| *Lichanura trivirgata* | | | | | | | | | | | | | | + | | | | | | | + | | 8 |
| **Colubridae** | | | | | | | | | | | | | | | | | | | | | | | |
| *Bogertophis rosaliae* | | | | | | | | | | | | | | + | | | | | | | | | 2 |
| *Lampropeltis californiae* | + | | | | | + | | + | | | | + | + | | | + | + | + | | | | + | 11 |
| *Lampropeltis catalinensis* | | | | | | | | | | | | | | | | | | + | | | | | 1 |
| *Masticophis barbouri* | | | | + | | | | | | | | | | | | | | | | | | | 2 |
| *Masticophis bilineatus* | | | | | | | | | | | | | | | | | | | | | + | | 1 |
| *Masticophis fuliginosus* | + | | | + | | | | | | + | + | | | + | | | | | | | + | | 12 |
| *Masticophis slevini* | | | | | | | | + | | | | | | | | | | | | | | | 1 |
| *Phyllorhynchus decurtatus* | + | | | | | | | | | | + | | | + | | | | | | | | | 5 |
| *Pituophis catenifer* | | | | | | | | | | | | | | | | | | | | | + | | 1 |
| *Pituophis vertebralis* | | | | | | | | | | | + | | | | | | | | | | | | 1 |
| *Rhinocheilus etheridgei* ** | | | | | | | | | | | | | | | | | | | | | | | 1 ** |
| *Salvadora hexalepis,* | | | | | | | | | | | + | | | | | | | | | | + | | 3 |
| *Sonora savagei* * | | | | | | | | | | | | | | | | | | | | | | | 1 |
| *Sonora semiannulata* | | | | | | | | | | | + | | | + | | | | | | | | | 2 |
| *Sonora straminea* | + | | | + | | | | | | | + | | | + | | | | | | | + | | 7 |
| *Tantilla planiceps* | | | | | | | | | | | | | | + | | | | | | | | | 2 |
| *Trimorphodon lyrophanes* | | | | | | | | | | | + | | | + | | | | | | | + | | 6 |
| **Dipsadidae** | | | | | | | | | | | | | | | | | | | | | | | |
| *Hypsiglena catalinae* | | | | | | | | | | | | | + | | | | | | | | | | 1 |
| *Hypsiglena chlorophaea* | | | | | | | | + | | | | | | | | | | | | | + | | 2 |
| *Hypsiglena ochrorhynchus* | + | | + | + | | + | | | + | | + | | + | + | | | | + | | + | | + | 18 |
| *Hypsiglena slevini* | | | | | | | | | | | | | | + | | | | | | | | | 4 |
| **Elapidae** | | | | | | | | | | | | | | | | | | | | | | | |
| *Micruroides euryxanthus* | | | | | | | | | | | | | | | | | | | | | + | | 1 |
| **Leptotyphlopidae** | | | | | | | | | | | | | | | | | | | | | | | |
| *Rena humilis* | | | | | | | | | | | | | | + | | | | + | + | | | | 7 |
| **Viperidae** | | | | | | | | | | | | | | | | | | | | | | | |
| *Crotalus angelensis* | | | | | | | | | | | | | | | | | | | | | | | 1 |
| *Crotalus atrox* | | | | | | | | | | | | | | | | + | | + | | | + | | 4 |
| *Crotalus catalinensis* | | | | | | | | | | | | | | | | | + | | | | | | 1 |
| *Crotalus cerastes* | | | | | | | | | | | | | | | | | | | | | + | | 1 |
| *Crotalus enyo* | | + | | + | | | | | + | | + | | | + | | | | | | | | | 9 |
| *Crotalus estebanensis* | | | | | | | | + | | | | | | | | | | | | | | | 1 |
| *Crotalus lorenzoensis* | | | | | | | | | | | | + | | | | | | | | | | | 1 |

| Taxa | | | | | | | | | | | | | | | | | | | | | | | | | | | | | | | | | | | | Total |
|---|---|---|---|---|---|---|---|---|---|---|---|---|---|---|---|---|---|---|---|---|---|---|---|---|---|---|---|---|---|---|---|---|---|---|---|---|
| *Crotalus mitchellii* | + | | | + | | | | + | | | | | | | | | | | + | | | | | | | | | | | | | | + | | | | 7 |
| *Crotalus molossus* | | | | | | | | | | | | | | | | | | | | | | | | | | | | | | | | + | | | | | 1 |
| *Crotalus polisi* | | | | | | | | | | | | | | | | | | | | | | | | | | | | | | | | | | | | | 1 |
| *Crotalus pyrrhus* | | | | | | | | | | | | | | | | | | | | | | | | | | | | | | | + | | | | | | 2 |
| *Crotalus ruber* | + | | | | | + | | | | | | | | | | | + | | + | | | | | | | | | | | | | | | | | | 7 |
| *Crotalus thalassoporus* | | | | + | | | | | | | | | | | | | | | | | | | | | | | | | | | | | | | | | 1 |
| *Crotalus tigris* | | | | | | | | | | | | | | | | | | | | | | | | | | | | | | | | + | | | | | 1 |
| *Crotalus tortuguensis* | | | | | | | | | | | | | | | | | | | | | | | | | | | | | | | | | | + | | | 1 |
| **Testudines** | | | | | | | | | | | | | | | | | | | | | | | | | | | | | | | | | | | | | |
| **Testudinidae** | | | | | | | | | | | | | | | | | | | | | | | | | | | | | | | | | | | | | |
| *Gopherus morafkai* | | | | | | | | | | | | | | | | | | | | | | | | | | | | | | | | + | | | | | 2 |
| **Totals** | 13 | 1 | 4 | 4 | 20 | 1 | 2 | 4 | 5 | 3 | 2 | 6 | 3 | 1 | 3 | 9 | 10 | 4 | 21 | 5 | 7 | 3 | 24 | 4 | 6 | 10 | 7 | 1 | 7 | 28 | 1 | 5 | 2 | 2 | 1 | — |

**Documentation:** 1–4, 6, 8–9, 11–12, 15–16, 18–21, 23, 25–35, 36–42, 44–53, 55–57, 59–60, 63–64, 66–70 ([71,91]); 5 ([71,91,142]); 7 (([71,91];[143] [*Dipsosaurus dorsalis*]; [72] [*Rena humilis*]; [144][*Lichanura trivirgata*], [145][*Salvadora hexalepis*]); 10 (([71,91]; [72,146] [*T. lyrophanes*]); 13 ([71,91,147,148]); 14 ([71,91]; [72,146] [*T. lyrophanes*]); 17 ([71,91,149]); 18 ([71,91]; [150][*H. frenatus*] 22 ([71,91,144]); 24 ([71,91,115]; [151] [*Aspidoscelis tigris*]); 43 ([71,91,142]); 54 ([71,91,146]), 58 ([71,91,146]; [152] [*Tantilla planiceps*]); 61 ([71,89,91]); 62 ([71,91,132]); and 65 ([71,91,146,153]).

**Table 7.** Distribution of the insular herpetofauna of the region of Tropical Pacific. Note: *Dermochelys coriacea* is recorded from the "Islas Marías" but without reference to a particular island in the archipelago. A column on the collective herpetofauna of the Islas Bahía de Chamela is provided in order to include data on species known only from the island group without reference to specific islands in the group. No asterisk = Non-endemic; * = country endemic; ** = insular endemic; and ‡ = non-native.

| | **Islands of Tropical Pacific** | | | | | | | | | | | | | | | | |
|---|---|---|---|---|---|---|---|---|---|---|---|---|---|---|---|---|---|
| | | | **Bahia de Chamela Islands** | | | **Islas Marietas Archipielago** | | | **Tres Marias Archipielago** | | | | **Revillagigedo Archipielago** | | | | |
| **Taxa** | **Isabel** | **La Peña (Coral)** | **San Pancho** | **Cocinas** | **Unspecified island** | **Isla Larga** | **Isla Redonda** | **Unspecified island** | **San Juanito** | **María Madre** | **María Magdalena** | **Maria Cleofas** | **Clarion** | **Socorro** | **Isla Ixtapa** | **La Roqueta** | **Total Number of Islands** |
| | **1** | **2** | **3** | **4** | **5** | **6** | **7** | **8** | **9** | **10** | **11** | **12** | **13** | **14** | **15** | **16** | |
| **Amphibians** | | | | | | | | | | | | | | | | | |
| **Anura** | | | | | | | | | | | | | | | | | |
| **Bufonidae** | | | | | | | | | | | | | | | | | |
| *Incilius mazatlanensis* * | + | | | | | | | | | + | | | | | | | 2 |
| *Rhinella horribilis* | | + | | | | | | | | | | | | | | | 1 |

| | 1 | 2 | 3 | 4 | 5 | 6 | 7 | 8 | 9 | 10 | 11 | 12 | 13 | 14 | 15 | |
|---|---|---|---|---|---|---|---|---|---|---|---|---|---|---|---|---|
| **Eleutherodactylidae** | | | | | | | | | | | | | | | | |
| *Eleutherodactylus pallidus* * | | | | | | | | | | + | + | + | | | | 3 |
| **Hylidae** | | | | | | | | | | | | | | | | |
| *Smilisca baudinii* | | | | | | | | | | + | | + | | | | 2 |
| **Microhylidae** | | | | | | | | | | | | | | | | |
| *Hypopachus variolosus* | | | | | | | | | | + | | | | | | 1 |
| **Reptiles** | | | | | | | | | | | | | | | | |
| **Crocodylia** | | | | | | | | | | | | | | | | |
| **Crocodylidae** | | | | | | | | | | | | | | | | |
| *Crocodylus acutus* | | | | | | | | | | | + | | | | | 1 |
| **Squamata** | | | | | | | | | | | | | | | | |
| **Dactyloidae** | | | | | | | | | | | | | | | | |
| *Norops nebulosus* * | | + | + | | + | | + | + | * | + | + | + | | + | + | 10 |
| **Gekkonidae** | | | | | | | | | | | | | | | | |
| *Gehyra mutilata* ‡ | + | | | | | | | | | | | | | | | 1 |
| *Hemidactylus frenatus* ‡ | + | + | + | + | + | | | * | | | | | + | + | + | 7 |
| **Iguanidae** | | | | | | | | | | | | | | | | |
| *Ctenosaura pectinata* | + | + | | + | | + | + | + | + | + | + | + | + | + | + | 13 |
| *Iguana iguana* | + | | | + | + | | + | + | | | + | | | | + | 7 |
| **Mabuyidae** | | | | | | | | | | | | | | | | |
| *Marisora aquilonaria* * | | | | | | + | + | + | | | | | | | | 3 |
| **Phrynosomatidae** | | | | | | | | | | | | | | | | |
| *Sceloporus clarkii* | + | | | | | | | | | | | | | | | 1 |
| *Urosaurus auriculatus* ** | | | | | | | | | | | | | + | | | 1 |
| *Urosaurus bicarinatus* * | | + | + | + | + | | | | | | | | | + | | 4 |
| *Urosaurus clarionensis* ** | | | | | | | | | | | | + | | | | 1 |
| *Urosaurus ornatus* | | | | | | | | | + | + | + | | | | | 3 |
| **Phyllodactylidae** | | | | | | | | | | | | | | | | |
| *Phyllodactylus isabelae* ** | + | | | | | + | + | | | | | | | | | 3 |
| *Phyllodactylus benedetii* * | | | + | | + | | | | | | | | | | | 2 |

| | | | | | | | | | | | | | | | | |
|---|---|---|---|---|---|---|---|---|---|---|---|---|---|---|---|---|
| *Phyllodactylus lanei* * | | | | | | | | | | | | | | + | | 1 |
| *Phyllodactylus lupitae* ** | | + | | | | | | | | | | | | | | 1 |
| *Phyllodactylus tuberculosus* | + | | | | | | | | | | | | | | | 1 |
| *Phyllodactylus cleofanensis* ** | | | | | | | | | | + | + | + | | | | 3 |
| **Teiidae** | | | | | | | | | | | | | | | | |
| *Aspidoscelis communis* * | | | | | | | | | + | + | + | + | | | | 4 |
| *Aspidoscelis costata* * | + | | | | | | | | | | | | | | | 1 |
| *Aspidoscelis deppii* | | | | | | | | | | | | | | | + | 1 |
| *Aspidoscelis guttatus* * | | | | | | | | | | | | | | + | | 1 |
| *Aspidoscelis lineattissima* * | | + | + | + | + | + | + | + | | | | | | | | 7 |
| **Boidae** | | | | | | | | | | | | | | | | |
| *Boa sigma* * | | | + | | + | | | | | + | + | + | | | | 5 |
| **Colubridae** | | | | | | | | | | | | | | | | |
| *Drymarchon melanurus* | | | | | | | | | | + | + | + | | | | 3 |
| *Lampropeltis polyzona* * | + | | | | | | | | | + | | * | | | | 2 |
| *Leptophis diplotropis* * | | | | | | | | | | + | + | + | | | | 3 |
| *Masticophis anthonyi* ** | | | | | | | | | | | | | + | | | 1 |
| *Masticophis mentovarius* | | + | | | | + | | + | + | + | + | | | | | 6 |
| *Mastigodryas melanolomus* | | | | | | | | | | + | + | + | | | | 3 |
| *Oxybelis microphtalmus* * | | | | | | | | | | + | + | + | | | + | 4 |
| *Tantilla bocourti* * | | | | | | | | | | | | + | | | | 1 |
| *Tantilla calamarina* * | | | | | | | | | | + | | * | | | | 1 |
| **Dipsadidae** | | | | | | | | | | | | | | | | |
| *Conophis vittatus* | | | | | + | | | | | | | | | | | 1 |
| *Hypsiglena slevini* * | | + | | | | | | | | | | | | | | 1 |
| *Hypsiglena torquata* | | | | | | + | | + | | | + | | | | | 3 |
| *Hypsiglena unaocularis* ** | | | | | | | | | | | | | + | | | 1 |
| *Imantodes gemmistratus* | | | | | | | | | | + | | + | | | | 2 |
| *Rhadinaea hesperia* * | | | | | | | | | | | | + | | | | 1 |
| *Geophis annuliferus* * | | | | | | | | | | * | | | | | | 1 |

| | 1 | 2 | 3 | 4 | 5 | 6 | 7 | 8 | 9 | 10 | 11 | 12 | 13 | 14 | 15 | 16 | Total |
|---|---|---|---|---|---|---|---|---|---|---|---|---|---|---|---|---|---|
| *Sibon nebulatus* | | | | | | | | | | + | | | | | | | 1 |
| **Elapidae** | | | | | | | | | | | | | | | | | |
| *Hydrophis platurus* | | | | | + | | | + | | | + | | | | | | 3 |
| **Leptotyphlopidae** | | | | | | | | | | | | | | | | | |
| ***Rhena humilis*** | | | | | | | | | | + | | | | | | | |
| *Epictia bakewelli* | | | | | | | | | | | | | | + | | | 1 |
| **Typhlopidae** | | | | | | | | | | | | | | | | | |
| *Indotyphlops braminus* ‡ | + | | | | | | | | | | | | | | | | 1 |
| **Viperidae** | | | | | | | | | | | | | | | | | |
| *Agkistrodon bilineatus* | | | | | | | | | + | + | + | | | | | | 3 |
| **Testudines** | | | | | | | | | | | | | | | | | |
| **Chelonidae** | | | | | | | | | | | | | | | | | |
| *Caretta caretta* | | | | | | | | | | | | | + | | | | 1 |
| *Chelonia mydas* | | | | | + | | | + | | | | | + | + | | | 4 |
| *Eretmochelys imbricata* | | | | | + | | | + | | + | + | | + | + | | | 6 |
| *Lepidochelys olivacea* | | | | | + | | | + | | | | | + | + | | | 4 |
| **Dermochelydae** | | | | | | | | | | | | | | | | | |
| *Dermochelys coriacea* | | | | | + | | | | | | | | + | + | | | 3 |
| **Kinosternidae** | | | | | | | | | | | | | | | | | |
| *Kinosternon integrum* * | | | | | | | | | | + | | | | | | | 1 |
| **Totals** | 11 | 9 | 6 | 4 | 14 | 6 | 6 | 10 | 7 | 22 | 18 | 14 | 9 | 6 | 7 | 6 | — |

**Documentation**: 1 ([136,154,155]); 2 ([107,108]); 3,4 ([156]); 4 ([156,157]; 5 ([158]); 6–7, 11 ([96,154,159]); 8 ([96,160]); 9 ([96,136,161]); 10 ([96,136,161,162]); 11 ([161]); 12 ([96,136,161,163–165]); 13 ([89,166–168]; Inaturalist); 14 ([166,167,169]); 15 ([170]; IBUNAM: CNAR:10589); and 16 ([171]; IBUNAM; CNAR).

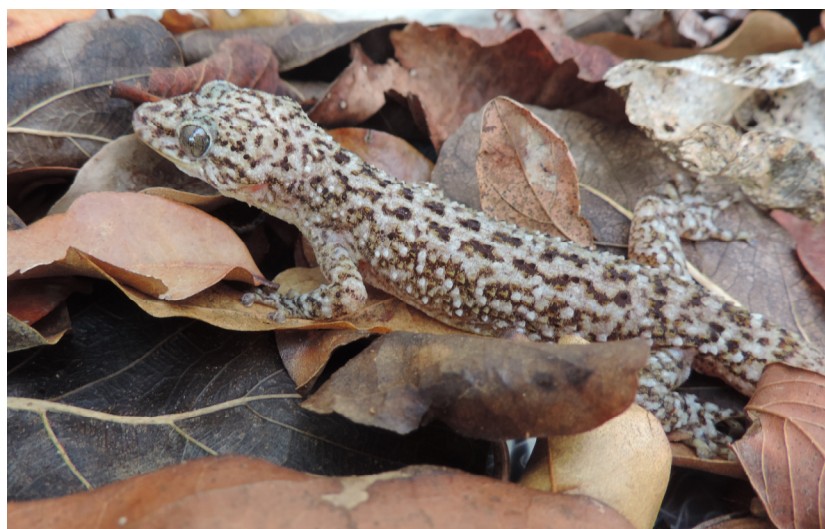

**Figure 25.** María Cleofas leaf-toed Gecko (Phyllodactyllys cleofasensis) from Isla María Cleofas (Tres Marías Archipielago), Tropical Pacific. In the past it was assumed that insular amphibians and reptiles from the Western Mexican Pacific were representatives of taxa found on the adjacent mainland. However, recent molecular studies had revealed that many Phylldactylus geckos dwelling within the Pacific Insular Systems are, in fact, distinct and independent species. One notable example is Phyllodactylus cleofasensis, marking the identification of the third gecko species residing on the islands off the coast of Nayarit. This finding is accompanied by the recognition of *P. isabelae* as an endemic species to the Marietas Islands and *P. lupitae* as an endemic species to El Coral Island. Photo by José Rafael Nolasco-Luna**.**

The herpetofauna of the seven islands in the Gulf of Mexico region is detailed in Table 8. Several of these islands are mapped in Figure 8. The total number of islands occupied by the 46 resident species (six anurans, one crocodylian, thirty-two squamates, and eight turtles).

**Table 8.** Distribution of the insular herpetofauna for the region of the islands of the Gulf of Mexico. No asterisk = Non-endemic; * = country endemic; and ‡ = non-native.

| | Islands of the Gulf of Mexico | | | | | | | |
|---|---|---|---|---|---|---|---|---|
| **Taxa** | **Cayo Arcas** | **Islas de Barrera de Laguna Madre** | **Islas de Laguna de Términos** | **Isla Lobos** | **Sistema Arrecifal Veracruzano** | | **Arrecife Alacranes** | **Total Number of Islands** |
| | | | | | **Isla de Sacrificios** | **Unspecified islands** | | |
| | **1** | **2** | **3** | **4** | **5** | **6** | **7** | |
| **Anura** | | | | | | | | |
| **Bufonidae** | | | | | | | | |
| *Incilius valliceps* | | | + | | | | | 1 |
| *Rhinella horribilis* | | | + | | | | | 1 |
| **Eleutherodactylidae** | | | | | | | | |
| **Hylidae** | | | | | | | | |
| *Scinax staufferi* | | | + | | | | | 1 |
| *Smilisca baudinii* | | | + | | | | | 1 |
| **Leptodactylidae** | | | | | | | | |
| *Leptodactylus fragilis* | | | + | | | | | 1 |
| **Microhylidae** | | | | | | | | |

| | | | | | | |
|---|:---:|:---:|:---:|:---:|:---:|:---:|
| *Hypopachus variolosus* | | + | | | | 1 |
| **Crocodylia** | | | | | | |
| **Crocodylidae** | | | | | | |
| *Crocodylus moreletii* | + | + | | | | 2 |
| **Squamata** | | | | | | |
| **Corytophanidae** | | | | | | |
| *Basiliscus vittatus* | | + | | | | 1 |
| **Dactyloidae** | | | | | | |
| *Norops rodriguezii* | | + | | | | 1 |
| *Norops sagrei* ‡ | | + | | | + | 2 |
| *Norops ustus* | | + | | | | 1 |
| *Norops sp.* | | | + | + | | 2 |
| **Gekkonidae** | | | | | | |
| *Hemidactylus frenatus* ‡ | | + | | | | 1 |
| *Hemidactylus turcicus* ‡ | | + | | | | 1 |
| **Iguanidae** | | | | | | |
| *Ctenosaura similis* | | + | + | + | | 3 |
| *Iguana iguana* | | + | + | + | | 3 |
| **Mabuyidae** | | | | | | |
| *Marisora lineola* | | + | | | + | 2 |
| **Phrynosomatidae** | | | | | | |
| *Sceloporus chrysostictus* | | + | | | | 1 |
| *Sceloporus variabilis* | | | | + | | 1 |
| **Sphaerodactylidae** | | | | | | |
| *Sphaerodactylus glaucus* | | + | | | | 1 |
| **Teiidae** | | | | | | |
| *Aspidoscelis cozumelae* | | + | | | | |
| *Aspidoscelis deppii* | | + | | | | 1 |
| *Aspidoscelis maslini* | | + | | | | 1 |
| *Holcosus gaigeae* * | | + | | | | 1 |
| **Boidae** | | | | | | |
| *Boa imperator* | | + | + | + | | 3 |
| **Colubridae** | | | | | | |
| *Drymobius margaritiferus* | | + | | | | 1 |
| *Lampropeltis abnorma* | | + | | | | 1 |
| *Pseudelaphe flavirufa* | | + | | | | 1 |
| *Spilotes pullatus* | | + | | | | 1 |
| *Oxybelis aeneus* | | + | | | | |
| **Dipsadidae** | | | | | | |
| *Coniophanes imperialis* | | + | | | | 1 |
| *Conophis lineatus* | | + | | | | 1 |
| *Dipsas brevifacies* | | + | | | | 1 |
| *Imantodes cenchoa* | | + | | | | 1 |
| *Leptodeira frenata* | | + | | | | 1 |
| *Ninia sebae* | | + | | | | 1 |
| *Sibon nebulatus* | | + | | | | 1 |
| *Tropidodipsas sartorii* | | + | | | | 1 |
| **Elapidae** | | | | | | |
| *Micrurus apiatus* | | + | | | | 1 |
| **Typhlopidae** | | | | | | |
| *Indotypholps braminus* | | + | | | | |

| Taxa | 1 | 2 | 3 | 4 | 5 | 6 | 7 | Total |
|---|---|---|---|---|---|---|---|---|
| **Testudines** | | | | | | | | |
| **Cheloniidae** | | | | | | | | |
| *Caretta caretta* | | + | + | + | + | + | + | 6 |
| *Chelonia mydas* | | + | + | + | + | + | + | 6 |
| *Eretmochelys imbricata* | | | + | | + | + | + | 4 |
| *Lepidochelys kempii* | | + | + | + | + | + | | 5 |
| **Dermatemydidae** | | | | | | | | |
| *Dermatemys mawii* | | | + | | | | | 1 |
| **Dermochelyidae** | | | | | | | | |
| *Dermochelys coriacea* | + | | | + | + | + | + | 5 |
| **Kinosternidae** | | | | | | | | |
| *Kinosternon scorpioides* | | | + | | | | | 1 |
| **Staurotypidae** | | | | | | | | |
| *Staurotypus triporcatus* | | | + | | | | | 1 |
| **Totals** | 1 | 4 | 42 | 4 | 9 | 10 | 6 | |

**Documentation:** 1 ([36]); 2 ([172]); 3 ([57,67]); 4 ([173]); 5,6 ([61]); 7 ([57,67]).

The herpetofauna of the ten islands of the Mexican Caribbean region consists of eight anurans, one crocodylian, thirty-eight squamates, and eight turtles. This herpetofauna is documented in Table 9 and the islands are illustrated in Figure 10.

**Table 9.** Distribution of the insular herpetofauna for the region of the Mexican Caribbean. No asterisk = Non-endemic; * = country endemic;; and ‡ = non-native. Notes: Braga (2000) enlists several additional species at a genus level, not enlisted here. The ? Sign indicates that those species may occur incidentally or that the record is doubtful.

| | Islands of the Mexican Caribbean | | | | | | | | | | | |
| | | | | | Banco Chinchorro Archipielago | | | | Sian Ka'an islands | | | |
| Taxa | Isla Contoy | Isla Holbox | Isla Mujeres | Cozumel | Cayo Centro | Cayo Norte Mayor | Cayo Norte Menor | Cayo Lobos | Cayo Culebras | Cayo Iguanas | Tamalcab | Total Number of Islands |
| | 1 | 2 | 3 | 4 | 5 | 6 | 7 | 8 | 9 | 10 | 11 | |
| **Amphibia** | | | | | | | | | | | | |
| **Anura** | | | | | | | | | | | | |
| **Bufonidae** | | | | | | | | | | | | |
| *Incilius valliceps* | | + | + | + | | | | | | | | 3 |
| *Rhinella horribilis* | | | | + | | | | | | | | 1 |
| **Eleutherodactylidae** | | | | | | | | | | | | |
| *Eleutherodactylus planirostris* ‡ | | | | + | | | | | | | | 1 |
| **Hylidae** | | | | | | | | | | | | |
| *Dendrosophus microcephalus* | | | | + | | | | | | | | 1 |
| *Scinax staufferi* | | | | + | | | | | | | | 1 |
| *Smilisca baudinii* | | + | | + | | | | | | | | 1 |
| *Trachycephalus vermiculatus* | | + | | + | | | | | | | | 1 |
| **Leptodactylidae** | | | | | | | | | | | | |
| *Leptodactylus fragilis* | | | | + | | | | | | | | 1 |
| **Reptiles** | | | | | | | | | | | | |
| **Crocodylia** | | | | | | | | | | | | |
| **Crocodylidae** | | | | | | | | | | | | |
| *Crocodylus acutus* | + | + | + | + | + | + | + | | | | | 6 |
| *Crocodylus moreletti* | | ? | | | | | | | | | | |
| **Squamata** | | | | | | | | | | | | |

| | | | | | | | | | | | |
|---|---|---|---|---|---|---|---|---|---|---|---|
| **Corytophanidae** | | | | | | | | | | | |
| *Basiliscus vittatus* | | | | + | | | | | | + | 2 |
| **Dactyloidae** | | | | | | | | | | | |
| *Anolis allisoni* ‡ | | | | | + | + | + | | | | 3 |
| *Norops lemurinus* | | | + | | | | | | | + | 2 |
| *Norops rodriguezii* | | + | + | + | | | | | | + | 4 |
| *Norops sagrei* ‡ | + | + | + | + | + | + | + | + | | | 8 |
| *Norops ustus* | | + | + | | | | | | | + | 2 |
| **Eublepharidae** | | | | | | | | | | | |
| *Coleonyx elegans* | | | + | | | | | | | | 1 |
| **Gekkonidae** | | | | | | | | | | | |
| *Hemidactylus frenatus* ‡ | + | + | | + | + | + | | | | + | 6 |
| *Hemidactylus turcicus* ‡ | | | | + | | | | | | | 1 |
| **Iguanidae** | | | | | | | | | | | |
| *Ctenosaura similis* | + | + | + | + | + | + | | | + | + | 8 |
| *Iguana iguana* | | | | + | + | + | + | | | | 4 |
| **Mabuyidae** | | | | | | | | | | | |
| *Marisora lineola* | + | + | + | + | | | | | | | 4 |
| **Phrynosomatidae** | | | | | | | | | | | |
| *Sceloporus chrysostictus* | | + | + | + | | | | | | | 2 |
| *Sceloporus cozumelae* * | + | + | + | + | | | | | | | 4 |
| **Scincidae** | | | | | | | | | | | |
| *Mesoscincus schwartzei* | | | | + | | | | | | | 1 |
| **Sphaerodactylidae** | | | | | | | | | | | |
| *Aristelliger georgeensis* | + | | + | + | + | + | + | | | | 6 |
| *Sphaerodactylus continentalis* | | | | + | | | | | | | 1 |
| *Sphaerodactylus glaucus* | | | | + | | | | | | + | 2 |
| **Teiidae** | | | | | | | | | | | |
| *Aspidoscelis angusticeps* | | ? | | | | | | | | | 1 |
| *Aspidoscelis cozumela* * | | | | + | | | | | | | 1 |
| *Aspidoscelis depii* | | | | + | | | | | | | |
| *Aspidoscelis maslini* | | | | | + | + | + | | | | 3 |
| *Aspidoscelis rodecki* * | + | | + | | | | | | | | 2 |
| *Holcosus gaigeae* * | | | + | | | | | | | | 1 |
| **Boidae** | | | | | | | | | | | |
| *Boa imperator* | + | | + | + | + | | | | | | 4 |
| **Colubridae** | | | | | | | | | | | |
| *Leptophis mexicanus* | | + | + | | | | | | | + | 2 |
| *Masticophis mentovarius* | | + | | | | | | | | | |
| *Mastigodryas melanolomus* | | | | + | | | | | | + | 2 |
| *Oxybelis aeneus* | + | | + | | | | | | | | 2 |
| *Oxybelis fulgidus* | | + | + | + | | | | | | + | 4 |
| *Tantilla moesta* | | | | + | | | | | | | 1 |
| **Dipsadidae** | | | | | | | | | | | |
| *Coniophanes meridanus* * | | + | | | | | | | | | 1 |
| *Conophis lineatus* | + | + | | | | | | | | | 2 |
| *Leptodeira frenata* | | | | + | | | | | | | 1 |
| **Elapidae** | | | | | | | | | | | |
| *Micrurus apiatus* | | | | | | | | | | + | 1 |
| **Leptotyphlopidae** | | | | | | | | | | | |
| *Epictia magnamaculata* | | | | + | | | | | | | 1 |
| **Natricidae** | | | | | | | | | | | |
| *Thamnophis proximus* | | | | + | | | | | | | 1 |
| **Typhlopidae** | | | | | | | | | | | |
| *Indotyphlops braminus* ‡ | | | | + | | | | | | | 1 |

| Taxa | | | | | | | | | | | | |
|---|---|---|---|---|---|---|---|---|---|---|---|---|
| **Viperidae** | | | | | | | | | | | | |
| *Agkistrodon russeolus* * | | + | | | | | | | | | | 1 |
| **Testudines** | | | | | | | | | | | | |
| **Cheloniidae** | | | | | | | | | | | | |
| *Caretta caretta* | + | + | + | + | + | + | ? | + | | | | 7 |
| *Chelonia mydas* | + | + | + | + | + | + | ? | + | | | | 7 |
| *Eretmochelys imbricata* | + | + | + | + | + | + | ? | + | | | | 7 |
| *Lepidochelys kempii* | | | | + | | | | | | | | |
| **Dermochelyidae** | | | | | | | | | | | | |
| *Dermochelys coriacea* | + | | + | + | | + | | + | | | | 5 |
| **Emydidae** | | | | | | | | | | | | |
| *Trachemys venusta* | | | | + | | | | | | | | 1 |
| **Geoemydidae** | | | | | | | | | | | | |
| *Rhinoclemmys areolata* | | + | + | + | | | | | | | | 2 |
| **Kinosternidae** | | | | | | | | | | | | |
| *Kinosternon scorpioides* | | | | + | | | | | | | | 1 |
| **Totals** | 15 | 13 | 23 | 39 | 12 | 12 | 6 | 4 | 1 | 1 | 11 | — |

**Documentation:** 1–4 ([36,57,67,174]); 2,4 (iNaturalist, [67]); 5–7 ([66]; Charruau and González-Sánchez, pers. Observ.); 8 ([175]); 9 ([36]); 10 ([176]); and 11 ([177]).

### 3.4. Patterns of Distribution within Physiograph Regions

We organized the insular species of Mexico into four distributional categories, including non-endemic species, country endemic species, insular (regional) endemic species, and non-native species. The status of the 226 species is indicated in Table 10. The data on distributional status are summarized in Table 11.

**Table 10.** Distributional and conservation status measures for members of the herpetofauna of the Mexican insular systems. Distributional Status: IE = endemic to Mexican Insular Systems; CE = endemic to country of Mexico; NE = not endemic to insular systems or country; and NN = non-native. Environmental Vulnerability Score (taken from Wilson et al., 2013a,b): low (L) vulnerability species (EVS of 3–9); medium (M) vulnerability species (EVS of 10–13); and high (H) vulnerability species (EVS of 14–20). IUCN Categorization: CR = Critically Endangered; EN = Endangered; VU = Vulnerable; NT = Near Threatened; LC = Least Concern; NE = Not Evaluated (no DD species are identified). SEMARNAT Status: A = Threatened; P = Endangered; Pr = Special Protection; and NS = No Status.

| Taxa | Distributional Status | Environmental Vulnerability Category (Score) | IUCN Categorization | SEMARNAT Status |
|---|---|---|---|---|
| **Anura** | | | | |
| **Bufonidae** | | | | |
| *Anaxyrus punctatus* | NE | L (5) | LC | NS |
| *Incilius mazatlanensis* | CE | M (12) | LC | NS |
| *Incilius valliceps* | NE | L (6) | LC | NS |
| *Rhinella horribilis* | NE | L (3) | LC | NS |
| **Eleutherodactylidae** | | | | |
| *Eleutherodactylus pallidus* | CE | H (17) | DD | Pr |
| *Eleutherodactylus planirostris* | NN | - | LC | - |
| **Hylidae** | | | | |
| *Dendropsophus microcephalus* | NE | L (7) | LC | NS |
| *Hyliola regilla* | NE | L (3) | LC | NS |
| *Scinax staufferi* | NE | L (4) | LC | NS |
| *Smilisca baudinii* | NE | L (3) | LC | NS |
| *Trachycephalus vermiculatus* | NE | L (4) | LC | NS |
| **Leptodactylidae** | | | | |

| | | | | |
|---|---|---|---|---|
| *Leptodactylus fragilis* | NE | L (5) | LC | NS |
| **Microhylidae** | | | | |
| *Hypopachus variolosus* | NE | L (4) | LC | NS |
| **Scaphiopodidae** | | | | |
| *Scaphiopus couchii* | NE | L (3) | LC | NS |
| **Caudata** | | | | |
| **Plethodontidae** | | | | |
| *Aneides lugubris* | NE | H (14) | LC | Pr |
| *Batrachoseps major* | NE | H (14) | LC | NS |
| **Reptiles** | | | | |
| **Crocodylia** | | | | |
| **Crocodylidae** | | | | |
| *Crocodylus acutus* | NE | H (14) | VU | Pr |
| *Crocodylus moreletii* | NE | M (13) | LC | Pr |
| **Squamata** | | | | |
| **Anguidae** | | | | |
| *Elgaria cedrosensis* | CE | H (16) | NE | NS |
| *Elgaria multicarinata* | NE | M (10) | LC | Pr |
| *Elgaria nana* | IE | H (16) | LC | Pr |
| **Anniellidae** | | | | |
| *Anniella geronimensis* | CE | M (13) | EN | Pr |
| *Anniella pulchra* | NE | M (12) | LC | Pr |
| **Bipedidae** | | | | |
| *Bipes biporus* | CE | H (14) | LC | Pr |
| **Corytophanidae** | | | | |
| *Basiliscus vittatus* | NE | L (7) | LC | NS |
| **Crotaphytidae** | | | | |
| *Crotaphytus dickersonae* | IE | H (16) | LC | NS |
| *Crotaphytus insularis* | IE | H (16) | LC | NS |
| *Gambelia copeii* | NE | M (11) | LC | NS |
| *Gambelia wislizenii* | NE | M (13) | LC | Pr |
| **Dactyloidae** | | | | |
| *Anolis allisoni* | NN | - | NE | - |
| *Norops lemurinus* | NE | L (8) | NE | NS |
| *Norops nebulosus* | CE | M (13) | LC | NS |
| *Norops rodriguezii* | NE | M (10) | NE | NS |
| *Norops sagrei* | NN | - | - | - |
| *Norops ustus* | NE | L (8) | NE | NS |
| **Eublepharidae** | | | | |
| *Coleonyx elegans* | NE | L (9) | LC | A |
| *Coleonyx gypsicolus* | IE | H (18) | LC | NS |
| *Coleonyx variegatus* | NE | M (11) | LC | Pr |
| **Gekkonidae** | | | | |
| *Gehyra mutilata* | NN | - | - | - |
| *Hemidactylus frenatus* | NN | - | - | - |
| *Hemidactylus turcicus* | NN | - | - | - |
| **Iguanidae** | | | | |
| *Ctenosaura conspicuosa* | IE | H (16) | NE | Ns |
| *Ctenosaura hemilopha* | CE | H (18) | NE | Pr |

| *Ctenosaura nolascensis* | IE | H (17) | VU | Ns |
|---|---|---|---|---|
| *Ctenosaura pectinata* | NE | H (15) | NE | A |
| *Ctenosaura similis* | NE | L (8) | LC | A |
| *Dipsosaurus catalinensis* | IE | H (17) | NE | Ns |
| *Dipsosaurus dorsalis* | NE | M (11) | LC | Ns |
| *Iguana iguana* | NE | M (12) | LC | Pr |
| *Sauromalus ater* | NE | M (13) | LC | Pr |
| *Sauromalus hispidus* | CE | H (14) | NT | A |
| *Sauromalus klauberi* | IE | H (16) | NE | A |
| *Sauromalus slevini* | CE | H (16) | NE | A |
| *Sauromalus varius* | CE | H (16) | NE | A |
| **Mabuyidae** | | | | |
| *Marisora aquilonaria* | CE | M (13) | NE | Ns |
| *Marisora lineola* | NE | M (10) | LC | Ns |
| **Phrynosomatidae** | | | | |
| *Callisaurus draconoides* | NE | M (12) | LC | A |
| *Petrosaurus mearnsi* | NE | M (12) | LC | Pr |
| *Petrosaurus repens* | CE | M (13) | LC | Ns |
| *Petrosaurus slevini* | IE | H (16) | LC | Ns |
| *Petrosaurus thalassinus* | CE | M (13) | LC | Pr |
| *Phrynosoma cerroense* | CE | H (16) | NE | A |
| *Phrynosoma solare* | NE | H (14) | LC | NS |
| *Sceloporus angustus* | IE | H (16) | LC | A |
| *Sceloporus chrysostictus* | NE | M (13) | LC | NS |
| *Sceloporus clarkii* | NE | M (10) | LC | NS |
| *Sceloporus cozumelae* | CE | H (15) | LC | Pr |
| *Sceloporus grandaevus* | IE | H (16) | LC | A |
| *Sceloporus hunsakeri* | CE | H (14) | LC | Pr |
| *Sceloporus lineatulus* | IE | H (17) | LC | A |
| *Sceloporus magister* | NE | L (9) | LC | NS |
| *Sceloporus occidentalis* | NE | M (12) | LC | NS |
| *Sceloporus orcutti* | NE | L (7) | LC | NS |
| *Sceloporus variabilis* | NE | L (5) | LC | NS |
| *Sceloporus zosteromus* | CE | M (12) | LC | Pr |
| *Urosaurus auriculatus* | IE | H (16) | EN | Ns |
| *Urosaurus bicarinatus* | CE | M (12) | LC | NS |
| *Urosaurus clarionensis* | IE | H (17) | VU | NS |
| *Urosaurus nigricaudus* | CE | L (8) | LC | A |
| *Urosaurus ornatus* | NE | M (10) | LC | NS |
| *Uta encantadae* | IE | H (17) | VU | NS |
| *Uta lowei* | IE | H (17) | VU | NS |
| *Uta nolascensis* | IE | H (17) | LC | A |
| *Uta palmeri* | IE | H (17) | VU | A |
| *Uta squamata* | NE | H (17) | LC | A |
| *Uta stansburiana* | NE | L (7) | LC | A |
| *Uta tumidarostra* | IE | H (17) | VU | NS |
| **Phyllodactylidae** | | | | |
| *Phyllodactylus angelensis* | IE | H (16) | NE | NS |
| *Phyllodactylus apricus* | IE | H (17) | NE | NS |

| | | | | |
|---|---|---|---|---|
| *Phyllodactylus benedetii* | CE | H (15) | NE | NS |
| *Phyllodactylus bugastrolepis* | IE | H (17) | LC | A |
| *Phyllodactylus cleofasensis* | IE | H (16) | NE | NS |
| *Phyllodactylus coronatus* | IE | H (16) | NE | NS |
| *Phyllodactylus homolepidurus* | CE | H (15) | LC | Pr |
| *Phyllodactylus isabelae* | IE | H (16) | NE | NS |
| *Phyllodactylus lanei* | CE | H (15) | LC | NS |
| *Phyllodactylus lupitae* | IE | H (16) | LC | NS |
| *Phyllodactylus nocticolus* | NE | M (10) | NE | NS |
| *Phyllodactylus partidus* | IE | H (16) | LC | Pr |
| *Phyllodactylus santacruzensis* | IE | H (17) | NE | NS |
| *Phyllodactylus tuberculosus* | NE | L (8) | NE | NS |
| *Phyllodactylus unctus* | CE | H (15) | LC | Pr |
| **Scincidae** | | | | |
| *Mesoscincus schwartzei* | NE | M (11) | LC | NS |
| *Plestiodon skiltonianus* | NE | M (11) | LC | NS |
| **Sphaerodactylidae** | | | | |
| *Aristelliger georgeensis* | NE | M (13) | LC | Pr |
| *Sphaerodactylus continentalis* | NE | M (10) | NE | NS |
| *Sphaerodactylus glaucus* | NE | M (12) | LC | Pr |
| **Teiidae** | | | | |
| *Aspidoscelis bacata* | IE | H (17) | LC | Pr |
| *Aspidoscelis cana* | IE | H (16) | LC | A |
| *Aspidoscelis carmenensis* | IE | H (17) | LC | NS |
| *Aspidoscelis catalinensis* | IE | H (17) | VU | Pr |
| *Aspidoscelis celeripes* | IE | H (15) | LC | Pr |
| *Aspidoscelis ceralbensis* | IE | H (17) | LC | Pr |
| *Aspidoscelis communis* | CE | H (14) | LC | Pr |
| *Aspidoscelis costata* | CE | M (11) | LC | NS |
| *Aspidoscelis cozumela* | CE | H (16) | LC | NS |
| *Aspidoscelis danheimae* | IE | H (16) | LC | A |
| *Aspidoscelis deppii* | NE | L (8) | LC | NS |
| *Aspidoscelis espiritensis* | IE | H (16) | LC | A |
| *Aspidoscelis franciscensis* | IE | H (17) | LC | NS |
| *Aspidoscelis guttatus* | CE | M (12) | LC | NS |
| *Aspidoscelis hyperythrus* | NE | M (10) | LC | A |
| *Aspidoscelis lineatissima* | CE | H (14) | LC | Pr |
| *Aspidoscelis martyris* | IE | H (17) | VU | Pr |
| *Aspidoscelis maslini* | NE | H (15) | LC | NS |
| *Aspidoscelis pictus* | IE | H (17) | LC | NS |
| *Aspidoscelis rodecki* | CE | H (16) | NT | P |
| *Aspidoscelis tigris* | NE | L (8) | LC | NS |
| *Holcosus gaigeae* | CE | H (15) | NE | NS |
| **Boidae** | | | | |
| *Boa imperator* | NE | M (10) | NE | A |
| *Boa sigma* | CE | H (15) | NE | A |
| **Charinidae** | | | | |
| *Lichanura trivirgata* | NE | M (10) | LC | A |
| **Colubridae** | | | | |

| | | | |
|---|---|---|---|
| *Bogertophis rosaliae* | NE | M (10) | LC | NS |
| *Drymarchon melanurus* | NE | L (6) | LC | NS |
| *Drymobius margaritiferus* | NE | L (6) | LC | NS |
| *Lampropeltis abnorma* | NE | L (9) | NE | A |
| *Lampropeltis californiae* | NE | M (10) | NE | A |
| *Lampropeltis catalinensis* | IE | H (17) | DD | NS |
| *Lampropeltis herrerae* | IE | H (17) | CR | A |
| *Lampropeltis polyzona* | CE | M (11) | NE | A |
| *Leptophis diplotropis* | CE | H (14) | LC | A |
| *Leptophis mexicanus* | NE | L (6) | LC | A |
| *Masticophis anthonyi* | IE | H (17) | CR | A |
| *Masticophis barbouri* | IE | H (17) | DD | A |
| *Masticophis bilineatus* | NE | M (11) | LC | NS |
| *Masticophis fuliginosus* | NE | L (9) | NE | NS |
| *Masticophis mentovarius* | NE | L (6) | LC | NS |
| *Masticophis slevini* | IE | H (17) | LC | NS |
| *Mastigodryas melanolomus* | NE | L (6) | LC | NS |
| *Oxybelis aeneus* | NE | L (5) | NE | NS |
| *Oxybelis fulgidus* | NE | L (9) | NE | NS |
| *Oxybelis microphtalmus* | CE | M (11) | NE | NS |
| *Phyllorhynchus decurtatus* | NE | M (11) | LC | NS |
| *Pituophis catenifer* | NE | L (9) | LC | NS |
| *Pituophis insulanus* | IE | H (16) | LC | NS |
| *Pituophis vertebralis* | CE | M (12) | LC | NS |
| *Pseudelaphe flavirufa* | NE | M (10) | LC | NS |
| *Rhinocheilus etheridgei* | IE | H (16) | DD | NS |
| *Salvadora hexalepis* | NE | M (10) | LC | NS |
| *Sonora savagei* | CE | H (15) | LC | Pr |
| *Sonora semiannulata* | NE | L (5) | LC | NS |
| *Sonora straminea* | NE | L (8) | LC | Pr |
| *Spilotes pullatus* | NE | L (6) | NE | NS |
| *Tantilla bocourti* | CE | L (9) | LC | NS |
| *Tantilla calamarina* | CE | M (12) | LC | Pr |
| *Tantilla moesta* | NE | M (13) | LC | NS |
| *Tantilla planiceps* | NE | L (9) | LC | NS |
| *Trimorphodon lyrophanes* | NE | M (10) | NE | NS |
| **Dipsadidae** | | | |
| *Coniophanes imperialis* | NE | L (8) | LC | NS |
| *Conophis lineatus* | NE | L (9) | LC | NS |
| *Conophis vittatus* | NE | M (11) | LC | NS |
| *Diadophis punctatus* | NE | L (4) | LC | NS |
| *Dipsas brevifacies* | NE | H (15) | LC | Pr |
| *Geophis annuliferus* | CE | M (13) | LC | Pr |
| *Hypsiglena catalinae* | IE | H (16) | NE | NS |
| *Hypsiglena chlorophaea* | NE | L (8) | NE | NS |
| *Hypsiglena ochrorhyncus* | NE | L (8) | NE | NS |
| *Hypsiglena slevini* | CE | M (11) | LC | A |
| *Hypsiglena torquata* | NE | L (8) | LC | Pr |
| *Hypsiglena unaocularis* | IE | H (16) | NE | NS |

| | | | | |
|---|---|---|---|---|
| *Imantodes cenchoa* | NE | L (6) | NE | Pr |
| *Imantodes gemmistratus* | NE | L (6) | LC | Pr |
| *Leptodeira frenata* | NE | M (13) | LC | NS |
| *Ninia sebae* | NE | L (5) | LC | NS |
| *Rhadinaea hesperia* | CE | M (10) | LC | Pr |
| *Sibon nebulatus* | NE | L (5) | NE | NS |
| *Tropidodipsas sartorii* | NE | L (9) | LC | Pr |
| **Elapidae** | | | | |
| *Hydrophis platurus* | NE | - | LC | NS |
| *Micruroides euryxanthus* | NE | H (15) | LC | A |
| *Micrurus apiatus* | NE | L(8) | NE | NS |
| **Leptotyphlopidae** | | | | |
| *Epictia bakewelli* | CE | M (12) | NE | NS |
| *Epictia magnamaculata* | NE | M (11) | NE | NS |
| *Rena humilis* | NE | L (8) | NE | NS |
| **Natricidae** | | | | |
| *Thamnophis proximus* | NE | L (7) | LC | A |
| **Typhlopidae** | | | | |
| *Indotyphlops braminus* | NN | - | NE | - |
| **Viperidae** | | | | |
| *Agkistrodon bilineatus* | NE | M (11) | NT | Pr |
| *Agkistrodon russeolus* | CE | H (15) | NE | NS |
| *Crotalus angelensis* | IE | H (18) | LC | NS |
| *Crotalus atrox* | NE | L (9) | LC | Pr |
| *Crotalus caliginis* | IE | H (18) | LC | NS |
| *Crotalus catalinensis* | IE | H (19) | CR | A |
| *Crotalus cerastes* | NE | H (16) | LC | Pr |
| *Crotalus enyo* | CE | M (13) | LC | A |
| *Crotalus estebanensis* | IE | H (19) | LC | NS |
| *Crotalus lorenzoensis* | IE | H (19) | LC | NS |
| *Crotalus mitchellii* | NE | M (12) | LC | Pr |
| *Crotalus molossus* | NE | L (8) | LC | Pr |
| *Crotalus polisi* | IE | H (18) | NE | NS |
| *Crotalus pyrrhus* | NE | M (12) | NE | NS |
| *Crotalus ruber* | NE | L (9) | LC | Pr |
| *Crotalus thalassoporus* | IE | H (18) | NE | NS |
| *Crotalus tigris* | NE | H (16) | LC | Pr |
| *Crotalus tortuguensis* | IE | H (18) | NE | Pr |
| **Testudines** | | | | |
| **Cheloniidae** | | | | |
| *Caretta caretta* | NE | - | VU | P |
| *Chelonia mydas* | NE | - | EN | P |
| *Eretmochelys imbricata* | NE | - | CR | P |
| *Lepidochelys kempii* | NE | - | CR | P |
| *Lepidochelys olivacea* | NE | - | VU | P |
| **Dermatemydidae** | | | | |
| *Dermatemys mawii* | NE | H (17) | CR | P |
| **Dermochelyidae** | | | | |
| *Dermochelys coriacea* | NE | - | VU | P |

| Emydidae | | | | |
|---|---|---|---|---|
| *Trachemys venusta* | NE | H (19) | NE | NS |
| **Geoemydidae** | | | | |
| *Rhinoclemmys areolata* | NE | M (13) | NT | A |
| **Kinosternidae** | | | | |
| *Kinosternon integrum* | CE | M (11) | LC | Pr |
| *Kinosternon scorpioides* | NE | M (10) | NT | Pr |
| **Staurotypidae** | | | | |
| *Staurotypus triporcatus* | NE | H (14) | NT | A |
| **Testudinidae** | | | | |
| *Geophis morafkai* | NE | H (15) | NE | A |

**Table 11.** Summary of the distributional status of herpetofaunal families in the Mexican insular systems.

| Families | Number of Species | Distributional Status | | | |
|---|---|---|---|---|---|
| | | Non-endemic (NE) | Country Endemic (CE) | Insular Endemic (IE) | Non-Native (NN) |
| Bufonidae | 4 | 3 | 1 | — | — |
| Eleutherodactylidae | 2 | — | 1 | — | 1 |
| Hylidae | 5 | 5 | — | — | — |
| Leptodactylidae | 1 | 1 | — | — | — |
| Microhylidae | 1 | 1 | — | — | — |
| Scaphiopodidae | 1 | 1 | — | — | — |
| **Subtotals** | **14** | **11** | **2** | **—** | **1** |
| Plethodontidae | 2 | 2 | — | — | — |
| **Subtotals** | **2** | **2** | **—** | **—** | **—** |
| **Totals** | **16** | **13** | **2** | **—** | **1** |
| Crocodylidae | 2 | 2 | — | — | — |
| **Subtotals** | **2** | **2** | **—** | **—** | **—** |
| Anguidae | 3 | 1 | 1 | 1 | — |
| Anniellidae | 2 | 1 | 1 | — | — |
| Bipedidae | 1 | — | 1 | — | — |
| Corytophanidae | 1 | 1 | — | — | — |
| Crotaphytidae | 4 | 2 | — | 2 | — |
| Dactyloidae | 6 | 3 | 1 | — | 2 |
| Eublepharidae | 3 | 2 | — | 1 | — |
| Gekkonidae | 3 | — | — | — | 3 |
| Iguanidae | 13 | 5 | 4 | 4 | — |
| Mabuyidae | 2 | 1 | 1 | — | — |
| Phrynosomatidae | 31 | 12 | 8 | 11 | — |
| Phyllodactylidae | 15 | 2 | 4 | 9 | — |
| Scincidae | 2 | 2 | — | — | — |
| Sphaerodactylidae | 3 | 3 | — | — | — |
| Teiidae | 22 | 4 | 7 | 11 | — |
| **Subtotals** | **111** | **39** | **28** | **39** | **5** |
| Boidae | 2 | 1 | 1 | — | — |
| Charinidae | 1 | 1 | — | — | — |
| Colubridae | 36 | 22 | 7 | 7 | — |
| Dipsadidae | 19 | 14 | 3 | 2 | — |

| | | | | |
|---|---|---|---|---|
| Elapidae | 3 | 3 | — | — | — |
| Leptotyphlopidae | 3 | 2 | 1 | — | — |
| Natricidae | 1 | 1 | — | — | — |
| Typhlopidae | 1 | — | — | — | 1 |
| Viperidae | 18 | 8 | 2 | 8 | — |
| **Subtotals** | **84** | **52** | **14** | **17** | **1** |
| Cheloniidae | 5 | 5 | — | — | — |
| Dermatemydidae | 1 | 1 | — | — | — |
| Dermochelyidae | 1 | 1 | — | — | — |
| Emydidae | 1 | 1 | — | — | — |
| Geoemydidae | 1 | 1 | — | — | — |
| Kinosternidae | 2 | 1 | 1 | — | — |
| Staurotypidae | 1 | 1 | — | — | — |
| Testudinidae | 1 | 1 | — | — | — |
| **Subtotals** | **13** | **12** | **1** | **—** | **—** |
| **Totals** | **210** | **105** | **43** | **56** | **6** |
| **Sum Totals** | **226** | **118** | **45** | **56** | **7** |

The largest number of species (118 or 52.2% of 226 species) is allocated to the non-endemic category. The ordinal proportion of non-endemic species is as follows: Anura (11 of 14 species or 78.6%); Caudata (2 of 2 species or 100%); Crocodylia (2 of 2 or 100%); Squamata (91 of 195 species or 46.7%); and Testudines (12 of 13 species or 92.3%).

The next largest group of species comprises the regional or insular endemics, amounting to 56 species (24.8% of 226 species). The ordinal proportions are as follows: Anura (0 of 14 species or 0%); Caudata (0 of 2 species or 0%); Crocodylia (0 or 2 species or 0%); Squamata (56 of 195 species or 28.7%); and Testudines (0 of 13 species or 0%).

The third largest group of species consists of the country endemics of which there are 43 species (19.0% of 226 species). The ordinal proportions are as follows: Anura (2 of 14 species or 14.3%); Caudata (0 of 2 species or 0%); Crocodylia (0 of 2 species or 0%); Squamata (42 of 195 species or 21.1%); and Testudines (1 of 13 species or 7.7%).

The total number of endemic species (country endemic plus regional or insular endemics) is 99 or 43.8% of the total insular herpetofauna of 226 species. This proportional endemicity is less than that reported for Mexican states such as Jalisco (64.6%;[139]); Michoacán (63.7%; [42]), Nayarit (57.1%; [136]), Oaxaca (58.1%; [178]), or Puebla [179], but higher than that documented for Chiapas (17.6%;[180]), Coahuila (28.0%; [44]), Nuevo León (28.1%; [43]), and Tamaulipas (32.1%; [181]). This dichotomy is principally due to the positioning of the state relative to either the Mexico–US border or the Mexico–Central American border, with states lying in the vicinity of one or the other of these borders having endemic species proportions lower than that of the insular regions and of those states lying relatively remotely from those two borders having higher proportions.

The smallest group of species constitutes the non-native species, of which there are seven species (3.2%). These seven species consist of one anuran (*Eleutherodactylus planirostris*) (Figure 26), five lizards (*Anolis allisoni*, *Norops sagrei*, *Gehyra mutilata*, *Hemidactylus frenatus*, and *H. turcicus*), and one snake (*Indotyphlops braminus*).

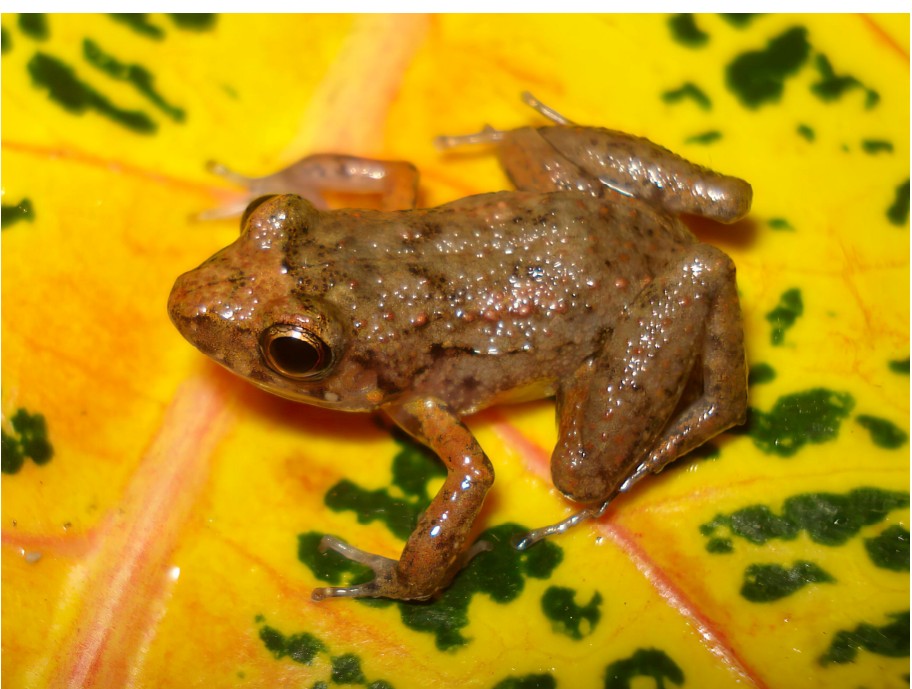

**Figure 26.** The Greenhouse Frog (*Eleutherodactylus planirostris*) from Cozumel (Mexican Caribbean) is extremely small-sized (adults < 30 mm in length). Native to Cuba, the Bahamas, and Cayman Islands. It is the only introduced amphibian inhabiting any Mexican insular system. Apparently, the populations colonizing the Mexican Yucatan Peninsula are related to the Greenhouse Frog's populations from Panama and/or the Philippines. Environmental impacts produced by the diminutive Greenhouse Frogs need to be determined, since there has been no direct evidence for it being particularly harmful [81]. Islands are hostile environments for most amphibians, so it is not rare that there are not other examples of introduced amphibians in Mexican islands. But Caribbean, and some Tropical Pacific islands (like Tres Marias Archipielago) may be humid enough to allow colonization by the Greenhouse Frog, Cane Toad (*Rhinella marina*), Bullfrog (*Lithobates catesbeianus*) or Cuban tree frog (*Osteopilus septentrionalis*). Distribution status = NN, IUCN = LC. Photo by Carlos Pavón-Vázquez**.**

None of the amphibians are exclusively distributed on any of the islands, and just two (12.5%) are country endemics, i.e., *Eleutherodactylus pallidus* and *Incilius mazatlanensis*. The remainder are either non-endemics (thirteen or 81.3%) or non-natives (one or 6.3%).

Only the squamates have taxa occupying all four of the distributional categories. Of the 195 squamate species, 91 (46.6%) are non-endemic species, 42 (21.54%) are country endemic species, 56 (28.7%) are regional or insular endemic species, and 6 (3%) are non-native species. Among the 84 snake species, 14 are country endemics (16.6%) and 17 are regional or insular endemics (20.2%); a single species is non-native (1.2%). Of the 17 regional or insular endemic snake species, eight or 47.1% are rattlesnakes. Among the 111 lizard species, 28 are country endemics (25.2%) and 39 are insular endemics (35.1%), and 5 species are non-natives (4.5%). Among the 39 insular endemic lizard species, 11 are phrynosomatids (28.2%), 11 are teiids (28.2%), and 17 belong to five other families. No turtle species is exclusive to the islands and only one (*Kinosternon integrum*) is a country endemic. Most likely, molecularly based taxonomic studies of insular populations of *Phyllodactylus*, *Sceloporus*, and *Uta* will lead to the uncovering of new species.

### 3.5. Comparison of Distributional Categorizations and Physiographic Regional Categorizations

In Table 12 we compared the distributional categorizations and the physiographic regional categorizations. The data in this table indicate that the largest proportion of species in each physiographic region consists of the non-endemic species, as follows: Gulf of California (49 of 108 species or 45.4%); Pacific of Baja California (27 of 40 species or 67.5%); Tropicales from the Pacific (25 of 57 species or 43.9%); Gulf of México (39 of 46 species or

84.8%); and Mexican Caribbean (39 of 46 species or 84.8%). A second conclusion is that the country endemic species are represented most evidently in the physiographic regions on the Pacific side of Mexico, as opposed to those on the Atlantic side. The proportions range from 13.9% (15 of 108 species) in the Gulf of California to 36.7% (21 of 57 species) in the Tropicals of the Pacific, respectively. On the Atlantic side the proportions are 6.5% (3 of 46 species) and 11.3% (6 of 53 species) in the Gulf of Mexico and Mexican Caribbean, respectively. The insular endemics are restricted in distribution to the physiographic regions on the Pacific side of Mexico and are most prominently a part of the herpetofauna of the Gulf of California, i.e., 44 of 108 species or 40.7%. The numbers of such species in the other two Pacific coastal regions are 8 of 57 species in the Tropical of the Pacific (14.0%) and 4 of 40 species in the Pacific of Baja California (10.0%). The non-native species are found only in three of the five regions, i.e., the Tropicals of the Pacific (3 of 57 species or 5.2%), the Gulf of Mexico (3 of 46 species or 6.5%), and the Mexican Caribbean (6 of 53 species or 11.3%). Interestingly, six of the seven non-native species in the insular regions occur in the Mexican Caribbean (all except for the gecko *Gehyra mutilata*), with three each in the Tropicals of the Pacific and four in the Gulf of Mexico.

**Table 12.** Comparison of distributional categorizations and physiographic regional categorizations for the members of the insular herpetofauna of Mexico.

| Distributional | Physiographic Regions | | | | |
|---|---|---|---|---|---|
| Categories | Gulf of California | Pacific of Baja California | Tropical Pacific | Gulf of Mexico | Mexican Caribbean |
| Non-endemic | 49 | 27 | 25 | 39 | 41 |
| Country endemic | 15 | 9 | 21 | 3 | 6 |
| Insuar endemic | 44 | 4 | 8 | — | — |
| Non-native | — | — | 3 | 4 | 6 |
| Totals | 108 | 40 | 57 | 46 | 53 |

*3.6. Principal Environmental Threats*

3.6.1. Deforestation, Agriculture, and Urban Development

Most Mexican islands are of a relatively small size, with most of them also uninhabited, but many are large enough to support temporary settlements of fishermen or navy bases, and all the larger islands have permanent towns, with the notable exception of Isla Tiburón, which is only occupied by a Mexican Navy outpost and the settlements of the Seri people [182,183], and Isla Ángel de la Guarda, which is uninhabited due to the lack of fresh water on the island [184]. Urban development is intense on the tourist islands of the Caribbean, such as Isla Mujeres and Cozumel, where the properties can have a very high economic value. Fortunately, the decree of several protected areas has limited considerably the space available for tourist development. Nevertheless, the demand for land continues on those islands to such a degree that the economic interest has involved even senior officials of the Mexican government in acts of land price speculation and land use changes. Cancún is the paradigmatic example of the almost complete destruction of an insular territory due to urban development. The process of urbanization and the transformation of Cancún into a peninsula is discussed by González-Sánchez et al. [57].

Another interesting case is that of María Madre Island, which supported a penitentiary colony since 1905. At its population peak (approximately 1986), it hosted nearly 5000 people. Environmental impact has occurred on the island due to the operation of the colony, such an open-air dump, the establishment of a sawmill, the introduction of cattle and other livestock, the proliferation of feral cats, and the furtive capture and consumption of fauna such as boas, iguanas, and sea turtles by the colonists. Even so, several conservation achievements exist, such as the decree of the Islas Marías Archipelago as a Protected Area and World Heritage site [185]. By federal decree, the penitentiary colony was closed in

2019 and, at the moment of this paper's redaction, these installations are under the transition to become a cultural center. This change opens the Tres Marias Archipielago to biodiversity conservation and ecological restoration programs, since in the past, due security restrictions, the Tres Marías islands were hardly accessible for conservationists. However, the access, management, and operation of this touristic and cultural center is under the control of the Mexican Navy, and this raises concerns about the accessibility these islands will have in the near future.

Also, Isla de Términos supports the important Ciudad del Carmen, which has been an important hub of the nationalized Mexican petroleum industry since 1971 [186,187] and has hosted a population of approximately 27% of the total population of the state of Campeche [187]. By far, it is the most highly populated island in Mexico [40]. The island used to be covered extensively by mangroves (*Rhizophora mangle*, *Avicennia germinans*, *Laguncularia racemosa*, and in a much lesser proportion *Conocarpus erecta*), with the shallow water on the lagunar side of the island being dominated by beds of seagrasses (*Thalassia testudinum*, *Halodule wrightii*, and *Siringodium filiforme* [188]). Most of this vegetation has been replaced by coconut and rice plantations and by the expansion of Ciudad del Carmen **[58]**, an urban area of approximately 2700 ha that supports 170,000 inhabitants [186]. Several changes in land use and population dynamics of Ciudad del Carmen have occurred in recent years, since the oil industry of the region has faced great uncertainty since the mid-2010s [187]. Even so, Isla del Carmen, along with the entire Laguna de Términos system, is one of the most important biodiversity areas in Mesoamerica **[58]**. The associated effects of urbanization on that island are summarized in the other portions of this section.

Other islands with important settlements and a growing population are Isla Cedros, Holbox, San Marcos, and Natividad [40]. Of special concern is the growing population in Holbox, where, in the last few years, uncontrolled development and a lack of a proper waste management had led to an environmental crisis [189,190] (Figure 27).

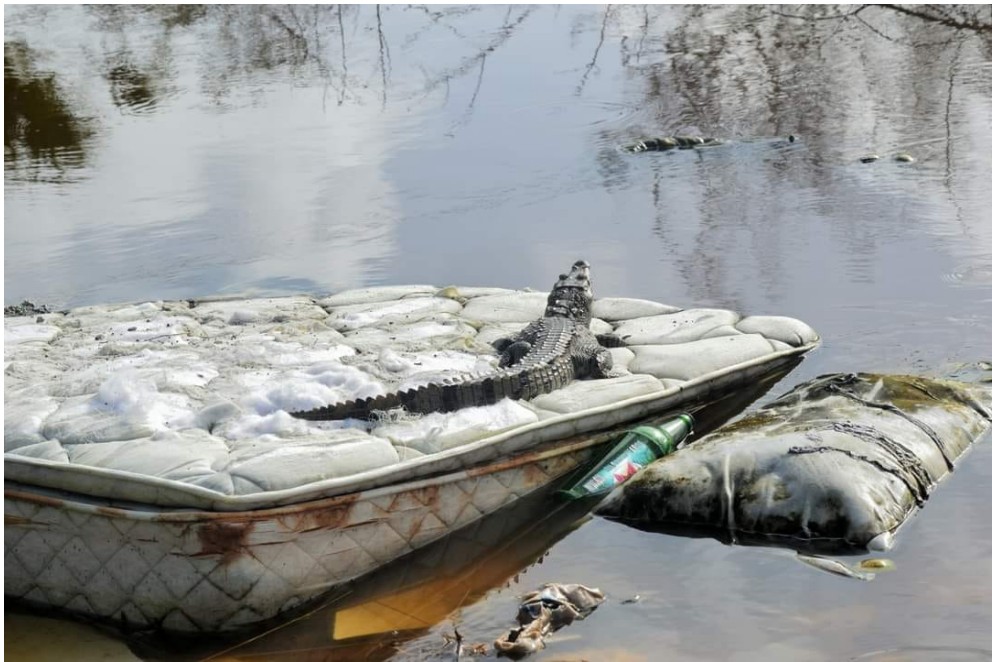

**Figure 27.** A Morelet's Crocodile (*C. moreletti*) taking a sunbath on a disposed spring in one of the many marshes on Holbox island. A major problem in populated islands is the waste disposal, which is a major logistical challenge for municipal authorities. This is more severe in those islands subject to rapid urban development due to tourism activities, such as in Holbox, Mujeres, and Cozumel. Photo courtesy of Eduardo Pacheco Cetina.

### 3.6.2. Agriculture and Cattle

The practice of agriculture is almost absent from the islands, but is extensive in Isla del Carmen, Campeche. In the past, the dominant vegetation consisted of mangroves and sea grasses, but urban development and fields of coconuts and rice [57] have replaced most of this vegetation.

The beginning of the introduction of cattle can be traced to the colonial times, when Spaniard navigators left on many islands several animals, such as goats, sheep, and pigs, with the aim of having food stocks on their navigation routes. Generally, however, this activity was limited only to the release of these animals with no particular interest in nurturing. The outcome of this activity was the establishment of feral populations. The problems associated with such populations are discussed in the section on invasive species [191] (Figure 28).

The existence of formal cattle raising as a productive economic activity Is present on few islands, most notoriously on Isla del Carmen, where extensive areas of cattle pasture exist. In the last few decades, the area devoted to agriculture and cattle raising on the islands has decreased [192] because of the increase of oil industry activities [193].

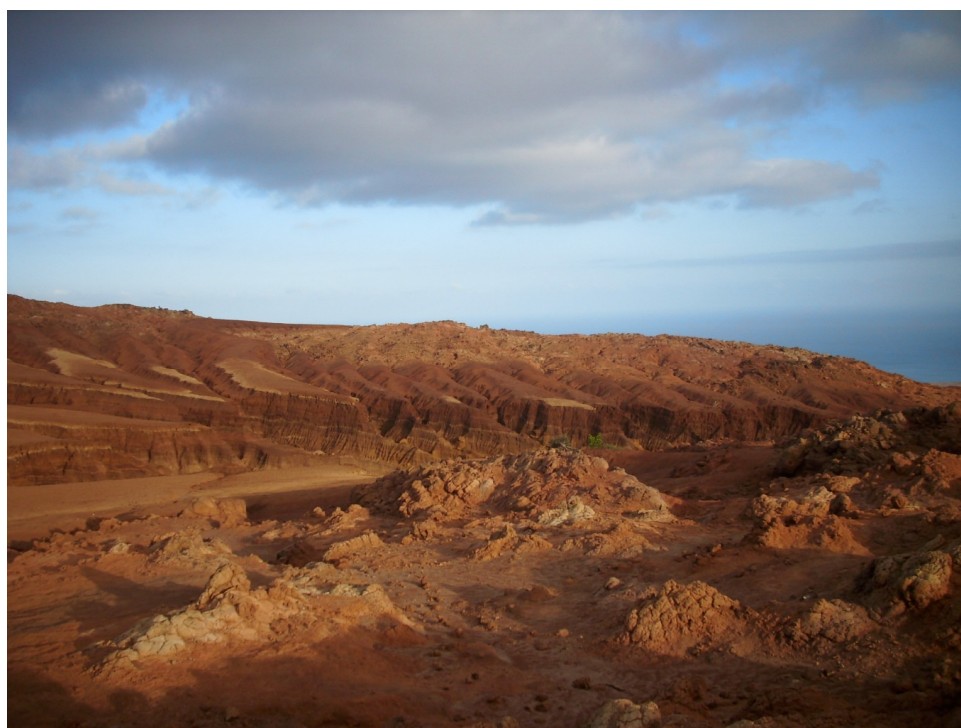

**Figure 28.** Eroded Landscape (18°45′06″ N y 110°58′07″ W) in Socorro Island (Revillagigedo Archipielago). In one of the few cases in which the introduction of an invasive species can be traced to one single event and/or specific date, approximately 100 sheep were introduced to Socorro in approximately 1869 with the intention of them becoming food sources for fishermen and boaters. They became feral and roamed free in the southern and eastern sides of the island and, by, 1989, they had an estimated population of 3000 individuals. They were eradicated completely by 2010. But by then, several portions of this island had been overgrazed. As is shown in this picture, the whole Horizon 0 and A are absent. The removal of the canopy and vegetation exposed the soil to the elements, since Socorro is from volcanic origin, several portions of the island are of pronounced slides, which aggravated the effects of hydric erosion, cleared all the superior Horizons of the soil, and resulted in this guilled landscape in which only parent material remains.

### 3.6.3. Hurricanes and Other Tropical Storms

Hurricanes and tropical storms are a major force that shape the structure of the insular and coastal ecosystems of Mexico, and are particularly important on the Mexican Yucatán Peninsula, where tropical storms occur with relatively high frequency. Hurricanes

can be highly destructive for those species who spawn in beaches or sand dunes, like crocodiles or sea turtles. For instance, based on internal reports from CONANP (accessed through our request to the Instituto Nacional de Acceso a la Información), Hurricane Delta flooded 40% of Chelonia mydas nests in Ixpalbarco (Cozumel) (7 October 2020). Almost one year later (19 August 2021), on that same beach, Hurricane Grace destroyed approximately 90% of the nests of Chelonia mydas and 100% of Caretta caretta nests (totaling ~400 nests). That year, approximately 80% of nests from APFF Isla de Cozumel and PN Arrecifes de Cozumel were lost (~1300 nests of both species) by flooding and/or erosion. This means that in 2021, a whole reproductive season of marine turtles from Cozumel was almost lost (inserter cita de reporte interno). However, recent studies on the effect of such intense climatic events on American crocodiles' population of Isla Cozumel and Banco Chinchorro Atoll are highlighting the negative impacts of hurricanes on the short-term but also positive impacts on the longer term on the ecology and health of these reptiles [194–196]. Then, the high frequency of tropical storms surely is a threat but also an important factor in the evolution of crocodiles and other reptiles in the Caribbean islands (Figure 29).

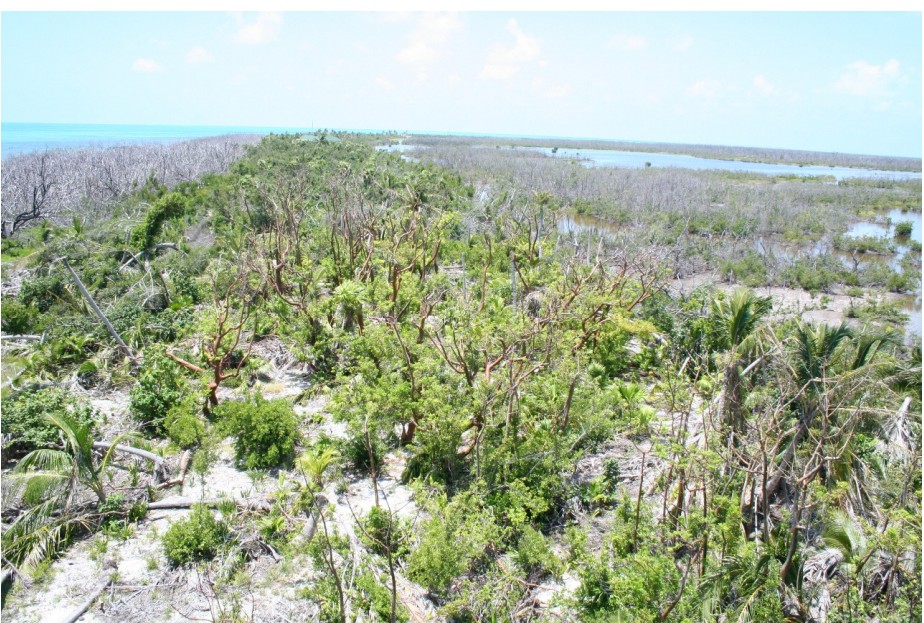

**Figure 29.** Severe devastation observed in the nesting grounds of the American crocodile (*C. acutus*) within Cayo Centro, Banco Chinchorro (Mexican Caribbean), following the impact of a hurricane. This nesting area is highly susceptible to erosion, flooding, and canopy loss (resulting in increased exposure to intense solar radiation). The escalating frequency of tropical storms, driven by the effects of climate change, intensifies the vulnerability of this location, and amplifies the profound risk of losing an entire cohort or even causing extinction of this isolated population. Photo by Pierre Charruau.

Those meteorological phenomena, however, are not exclusive to the Atlantic; in the tropical Pacific, we found that CONANP identified three hurricanes that impacted Isla Isabel and Isla Marietas, and although there is no information on the damage of turtles' nesting sites CONANP identified Hurricane Kena (5 in Saffir Simpson) as being severely damaging for birds and vegetation, with almost all trees of Isla Isabel being defoliated. It is important to notice that vegetation loss can be an important factor affecting the nesting success, since the canopy loss may result in major exposure to sunlight by the nesting sites, and thus changing the temperature of the nesting site.

In October 2015, Hurricane Patricia became the most intense (by central pressure) and powerful (by wind speed) tropical storm ever measured in history, and impacted the

coasts of Jalisco and Colima [197]. The trajectory of the storm passed near the islas Marietas and impacted the region of Chamela in Jalisco. Apparently, in the Marietas the damages were minimal (https://www.gob.mx/semarnat/prensa/reporta-sector-ambiental-danos-menores-en-zonas-costeras-por-paso-de-huracan-patricia accessed on 1 June 2020), but in the region of Chamela, the impacts on the vegetal structure and changes in the hydrological regimen were intense [198,199]. Unfortunately, post-storm surveys were made only on the mainland. So, what the effects were on the islands of Chamela Bay are unknown.

### 3.6.4. Wildfires

Wildfires are particularly catastrophic on islands since the area available for the fauna to escape is very reduced or even nonexistent (Figure 30). For example, in February of 1997, a wildfire consumed in less than 24 h the totality of the vegetation of Isla La Larga, one of the Marietas islands in Bahía de Banderas, probably caused by an uncontrolled bonfire set by tourists. That incident encouraged authorities to create stricter regulations for tourism on islands.

Given the isolation of the sites, the logistics for containing fires are extremely difficult in many cases. A clear example of the difficulty in attending to such contingencies is the event on Isla Guadalupe on the 15th of September 2008, when an uncontrolled burning of trash (made by workers of an ONG) near a biological station grew into a wildfire. The transport of personnel and equipment onto the island to attend to this contingency required the use of airplanes and ships and required the coordination of ~90 fieldworkers from several government institutions and ONGs, among other actors. It took approximately 80 h to be controlled and more than 300 h for its total extinction. This fire resulted in the damage of ~637 ha of vegetation (mostly grasslands) [200].

As it occurs in continental areas of Quintana Roo [57], the pressure for urbanization can provide a reason to clear the terrain and gain space for developing tourist infrastructure. This could be the reason behind the wildfire on Holbox Island between the 17th and 21st September 2016, during those days, a forest fire broke out east of the small island of Holbox, in the area known as "La Ensenada." The official report of the Federal Attorney for Environmental Protection (PROFEPA) determined that it was an arson attack, which affected a total area of 87.22 hectares of low deciduous forest and coastal scrub. Despite Holbox being near a mainland island, to extinguish the fire required an important inter-institutional effort which involved 85 people (39 brigades from CONAFOR, 7 from CONANP, 7 from SEMAR, 7 from SEDENA, 6 from PROFEPA, 6 from PRONATURA Yucatan Peninsula and 13 from the island of Holbox); 22 vehicles were also used, including trucks, boats, a helicopter, and a twin-engine plane. Despite the early warning and quick response, it took four days to extinguish the fire, and destroyed 87 has.

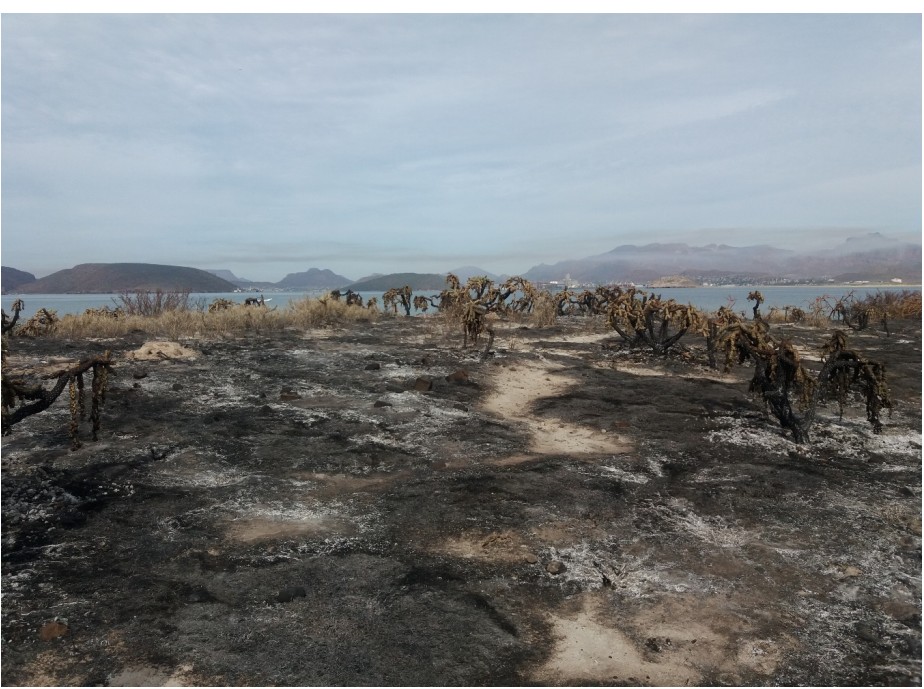

**Figure 30.** Vegetation damaged by a wildfire, Isla Pajaros, Gulf of California, near Mazatlán, Sinaloa. APFF Islas del Golfo de California (27 March 2019). Photo by Comisión Nacional de Áreas Naturales Protegidas (CONANP) obtained via Instituto Nacional de Transparencia, Acceso a la Información y Protección de Datos Personales (INAI).

### 3.6.5. Tourism

According to Coccossis and Parpairis [201], the tourism-carrying capacity is "The maximum level of recreation use, in terms of visitor numbers and activities, that can be accommodated before a decline in ecological value sets in." Thus, the carrying capacity would be the point at which the demand of infrastructure requirements and natural resources become insufficient to meet the needs of the visitors and residents before environmental hazards appear. Considering this situation, it is evident that due to their isolation, reduced size, and vulnerability, the concept of carrying capacity is highly relevant when talking about the number of tourists an island can sustain.

Tourism is the third most important economic activity in Mexico. Despite the general perception of Mexico being an insecure country to visit, the truth is that tourism demonstrates a sustained growth year by year and Mexico usually ranks among the principal tourist destinations of the world. The region of the Yucatán Peninsula (and especially Quintana Roo) stands out as the principal sun and beach destination in the country. Much of this attraction is due to what is offered by the islands in the Mexican Caribbean, such Cozumel, Isla Mujeres, and Cancún. In the case of the latter, however, the insular ecosystem has been destroyed almost entirely by the tourism megainfrastructure and the physiography of the island has been modified to such a degree that it can hardly be considered an island anymore, but rather has been converted into a peninsula [57].

In the atoll of Banco Chinchorro there is a tourist attraction which began in 2013 that consists of swimming with American crocodiles (*Crocodylus acutus*) in the reef lagoon. This activity seems to be a good option for the sustainable use of crocodiles, however the methodology used seems to have caused a change in the behavior of the species a few years after its beginning [202]. Indeed, between 2017 and 2020 several human–crocodile incidents ($n$ = 6) occurred in Cayo Centro, which corresponds to a significant increase since only 2 incidents were recorded before 2017. However, the swimming with crocodile activity does not seem to be the only factor responsible for the change in crocodile behavior and the situation is currently being studied. In the meantime, the authorities of the Banco Chinchorro reserve have decided not to authorize the activity (Figure 31).

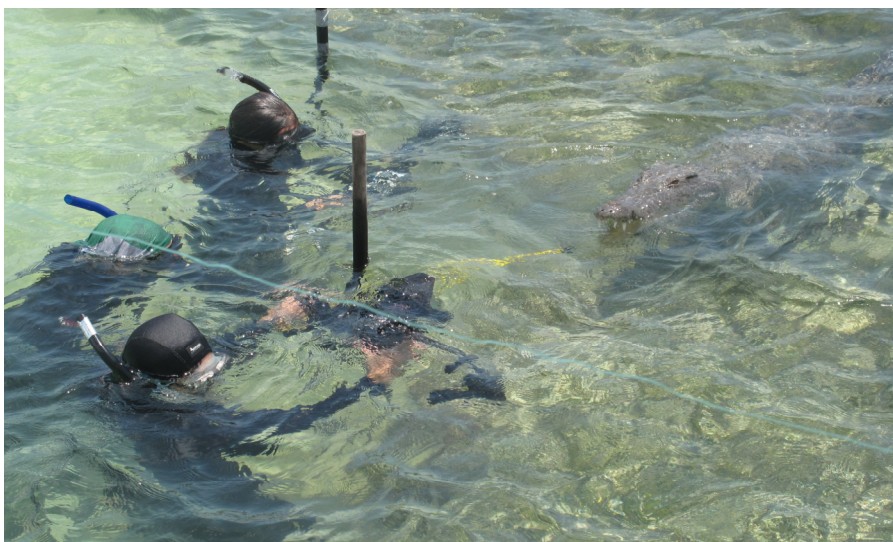

**Figure 31.** Tourist interacting with *Crocodylus acutus* in Banco Chinchorro. Insular species, due to their isolated habitats, often exhibit a sense of naïvety resulting from the absence of predatory pressures. This holds true for American crocodiles residing in the atoll, where their ethological adaptations are evident through decreased aggressiveness. As a result, Chinchorro's crocodiles have become accustomed to interacting with tourists and fishermen, displaying a habituation to their presence. Although this activity is not currently allowed by the CONANP, few touristic agencies continue to provide opportunities for observing or swimming with these crocodiles as part of their tours. Photo by Pierre Charruau.

Another popular touristic destination are the Islas Marietas, in where the whale watching, scuba diving, snorkeling, and primarily, the "Playa del Amor" (a semi-hidden beach located inside a cave within the island) makes this place one of the most attractive Mexican islands for tourists. The high demand, chaotic management, and excessive number of visitors led in 2016 to a controversial temporary closure of the island, to carry out ecological restoration labors and to redesign the visitors' policies, since the maximum carrying capacity of visitors for the island was calculated as 625. The average number of visitors, however, is over 1000, with peaks of more than 3000 on some days (https://www.informador.mx/Jalisco/Cierran-Islas-Marietas-por-dano-ecologico-20160414-0084.html accessed on 1 June 2020).

3.6.6. Invasive Species

The introduction of invasive species into insular ecosystems has occurred from prehistoric times and has been considerably accelerated during the last 50 years [65,203]. Subsequently, the impact of invasive species has been widely recognized since the 1950s [204] and is considered to be the second most important cause of global biodiversity loss [205], and the first in insular systems [4,7,65]. Usually, the species living in an isolated environment free of predators tend to become naïve and lose their anti-predatory behavior [5,206]. This is why the effects of invaders are much more accentuated in insular ecosystems, where a few mammal species are responsible for most of the insular diversity declines, including black rats (*Rattus* spp.), feral cats (*Felis catus*), goats (*Capra hircus*), pigs (*Sus scrofa*), donkeys (*Equus asinus*), and European rabbits (*Oryctolagus cuniculus*) [207]. The insular reptiles are particularly vulnerable, since populations are more affected by feral cats [208] and black rats [209]. Several declines of insular herpetofaunas due to invasive species has been documented across the world [210], and those mammals are among the most frequent insular invaders [203,208,211]. In Mexico, cat introductions are associated with the decline of the endemic *U. auriculatus* in Socorro [212] (Figure 32), and on Isla Isabel (prior to the eradication of cats), a survey estimated that at least 24% of cat scats contained remains of reptiles [213]. Venomous reptiles, too, are not free of cat predation;

Arnaud et al. [214] reported 13% of cat scats with remains of the endemic *Crotalus catalinensis* (prior to the eradication of cats on that island).

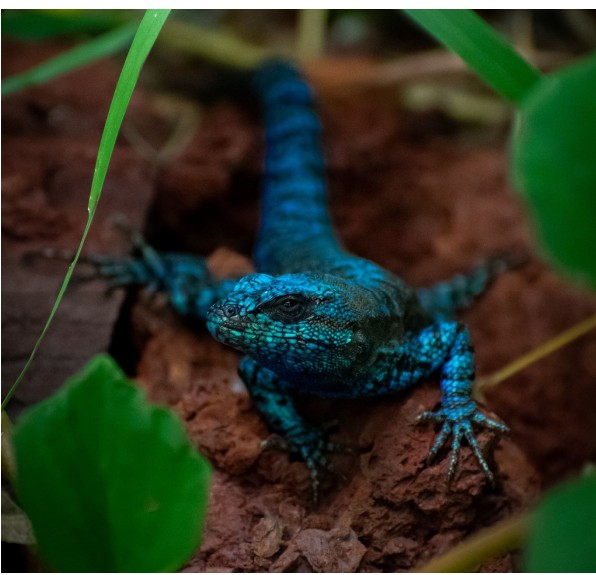

**Figure 32.** *Urosaurus auriculatus*, a Socorro Island Tree Lizad, Endemic of Isla Socorro although very conspicuous due to its bright blue color; females and juveniles are most frecuently of greyish color. After the eradication of feral cats in Socorro, its populations showed a growing tendency. Distributional Status = IE, EVS = H (16), IUCN = EN, NOM-059 = NS. Photo by Juan Diego Arias-Montiel.

The impact of invasive species on islands are not limited to direct predation, but also can involve change in land cover. The introduction onto islands of ungulate mammals, such as sheep, pigs, and goats, among others, as a source of food goes back to colonial times [191] and has resulted in severe losses of vegetal cover, soil erosion, and plant species extinctions on several Mexican islands, such as Guadalupe [49,215].

As the main source of impact, it is clear that the eradication programs of invasive species carry the greatest gains for biodiversity conservation; therefore, these programs have been increasing in scope, frequency, and complexity [8]. Since the end of the 20th century, many coordinated interinstitutional efforts involving governmental agencies, NGOs, and the academic community have successfully accomplished several eradication programs of invasive mammals on Mexican islands [207,216].

Most rodent eradications are carried out through the dispersion (areal or by hand) of brodifacoum, an anticoagulant rodenticide, which is known to have impacts on non-target species of mammals and birds. Little is known of its impact on reptiles, but large lizards, such as iguanas, might be susceptible due to their consumption of the baits (frequently they are made with a mixture of rodenticide and cereal or other attractants), or by secondary exposure as a result of eating poisoned dead rats, but the mortality rate seems to be low. Thus, reptiles might have a low risk of population-level declines through brodifacoum-induced mortality after rodent eradications [217]. Therefore, in a cost–benefit relation, reptiles can benefit highly from rodent eradication. Some examples of reptile recoveries after the extirpation of invasive species are the "reappearance" of *Lampropeltis californiae* (listed in the report as *L. getula nigrita*) on San Pedro Martir Island afther two years of rat eradication. *Phyllodactylus homolepidurus* went from "extremely rare" to "low abundance" in Farallon de San Ignacio (also after two years of rat eradication), and *Ctenosaura pectinata* on Isabel [218], whereas the Socorro Tree Lizard (*Urosaurus auriculatus*) showed an increase in population numbers since the implementation of programs to control the feral cats [219]. Also, an increase in sightings of boas (*Boa imperator*) has been observed on Cayo Centro in the atoll of Banco Chinchorro since the eradication of rats (*Rattus rattus*) and feral cats on the island (Pierre Charruau, pers. comm.). Major challenges for

future generations of conservationists, however, persist on the largest and most populated islands such as Cozumel (Figure 33) in the Caribbean and Cedros in the Pacific of Baja California.

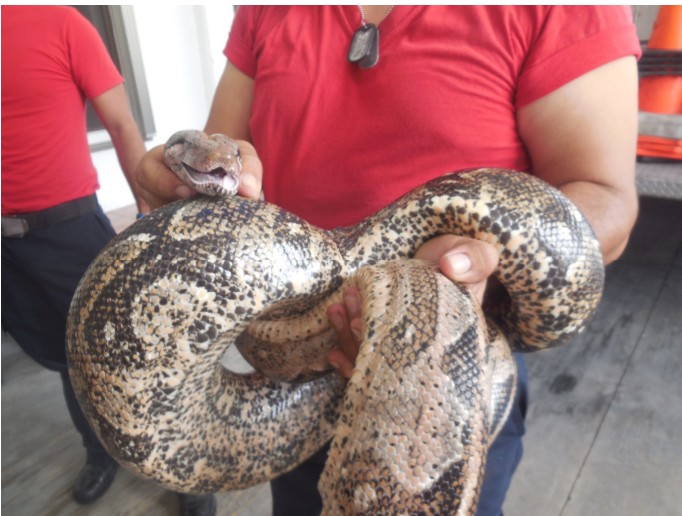

**Figure 33.** A large male of *Boa imperator*, rescued by members of the fire department from a hotel, in Cozumel, Quintana Roo, as mentioned by González-Sánchez, et al. [65]. This species was introduced in Cozumel by filmmarkers in 1971, also, the central American Boa Constrictor may be alien for Banco Chinchorro (Mexican Caribbean), and Venados (Gulf of California), but this must be confirmed or rejected by further studies. Photo by Lizbeth E. Lara-Sánchez.

3.6.7. Global Climate Change

Climate change is expected to become the principal cause of extinctions in future decades [220]. Also, it has been implicated as the main driver of population decline for both reptiles [210] and for amphibians [221]. A characteristic of insular populations is their reduced genetic variability, so these species are limited in their capacity to survive changing conditions [203]. One of the main consequences of climate change will be the shift in range distribution of species in order to find more suitable conditions to survive, along with alterations in community composition and changes in the interactions among species [220,222]. Obviously, for the great majority of insular species, range distribution shift is not an option for survival. Moreover, most islands are small-sized and, generally, their topography does not have the range of altitudes found in mainland territories.

Ectotherms are especially vulnerable to global warming, since activity periods [223] and processes such as spermatogenesis and sex determination are heavily influenced by temperature [222]. Not much is known of the effects of global warming in herpetofaunal insular populations in Mexico, but the existence of sex-biased ratio in the population of *Crocodylus acutus* (a species with temperature-dependent sex determination) (Figure 34) in Banco Chinchorro due to incubation temperature conditions is documented [66,195]. The main threat due to climate change, however, could be the destruction of habitat by two factors: (1) increase of severity and frequency of hurricanes and tropical storms, as discussed in previous paragraphs, and (2) increase of sea level. These factors are especially important on the islands of the Yucatán Peninsula that have low elevations, such Arrecife Alacranes and Banco Chinchorro. In the latter case, most of its surface is permanently emerged. A side-effect of the sea level rising and alterations of tides is erosion. This factor could be an important threat on sandy islands or those of coralline origin. Again, the islands of Banco Chinchorro, Arrecife Alacranes, and the islands of Veracruz are particularly susceptible to being damaged by these processes. An imperative measure is to annually map the perimeters of those islands, in order to ensure early detection of any change in shape and/or size and to take measures such restoring the beaches or the mangroves in the shore.

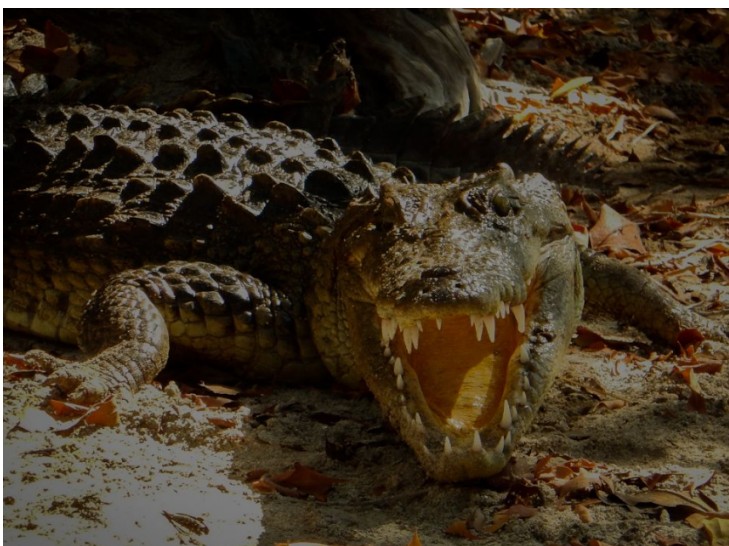

**Figure 34.** Specimen of *Crocodylis acutus* in Banco Chinchorro. Hybridization in Mexican crocodiles is a growing phenomenon due to the frequent translocations of swamp crocodiles (*Cr. Moreletti*) which successfully establish new population towards their invasion front. Since Morelett's crocodiles' haplotype is dominant, increasing hybridization events are a menace for American crocodiles due to genetic introgression. Banco Chichorro is crucial as an outpost for preserving *Cr. acutus* from genetic pollution, since its remoteness from the mainland constitutes an effective barrier to colonization by *Cr. moreletti*. However, the resilience of this population is under threat from global warming. The increased frequency and intensity of hurricanes, rising sea levels, and a sex-biased ratio (as crocodiles exhibit temperature-dependent sex determination) pose significant challenges to their long-term survival. Photo by Israel Sánchez-Ortega.

The effects of El Niño Southern Oscillation (ENSO) are particularly pernicious in insular ecosystems, and are important throughout the trophic chain, especially among seabirds, plankton, fishes, and marine mammals [224]. Evidently, those alterations might influence reptile populations, such as those of crocodiles [225] and/or marine turtles [226–228]. Cerdá-Ardura (2018) hinted that this phenomenon could make food scarcer, and result in the starvation of Chukwallas on Rasa Island. Still, how the most intense ENSO regimens would affect the Mexican insular populations of reptiles and amphibians is a matter which has not yet been explored.

As explained in the description of physiographic regions, most of the islands of the Pacific and the Gulf of California lack permanent freshwater sources, so the effects of drought are more serious on islands than on the mainland. Thus, intense droughts can lead to significant fluctuations and rapid population loss, as reported by Case (1982, in Lovich, et al.,[229]) for *Sauromalus hispidus* on Angel de la Guarda. Likely, this effect could be more accentuated in snakes, due to the decline of prey populations. Of course, droughts are even more serious among amphibians.

3.6.8. Oil and Gas Industries

The Gulf of Mexico is a semi-enclosed sea that connects in the east to the Atlantic Ocean through the Straits of Florida, and in the south to the Caribbean Sea through the Yucatan Channel. An important characteristic of this sea is the dominance of the Loop Current in the Yucatan Channel, and the formation swirls detached from that current (eddies) [230]. In the eventual case of hydrocarbon spill, it can be driven to several areas of the Gulf, including the insular systems. For example, the explosion in the shallow waters of the Bay of Campeche of the PeMex Ixtoc-I exploratory well occurred in 1979, resulting in the second major oil spill in the history of the Gulf of Mexico (just below the level of the 2010 Deepwater Horizon spill). Several months after the incident, the Ixtoc-I oil could be found all along the Gulf Coast from Texas to Yucatan [231]. Several insular systems such

as the Arrecife Alacranes, Sistema Arrecifal Veracruzano, and the islands of Laguna de Términos are in the path of the Ixtoc-I oil well.

In 2005, Chevron-Texaco's announcement regarding the installation of a liquefied Natural Gas Regasification Terminal just 600 m away from the Coronado islands, offshore of Tijuana, raised significant concerns (http://www.jornada.com.mx/2005/03/15/index.php?section=economia&article=025n1eco accessed on 1 June 2020) raised bitter controversy among conservationists, ONG's, environmental authorities, and the business sector of Baja California [232]. A decisive point was the numerous inconsistences detected in the process (in fact, the government initially authorized permission to build the terminal, even though the Environmental Impact Manifestation presented by Chevron–Texaco referred to a project located in Yucatán, not Baja California). The platform would increase the risk of catastrophic explosion, oil spill, and introduction of rats to the Coronado islands, among other problems. After two years of controversy, Chevron–Texaco finally stopped the construction of the terminal (P.L.F., 2007).

As a result of the discovery in 1971 of Cantarell in the bank of Campeche, the national oil industry experienced a bonanza. A collateral effect of the successful oil industry was the development of associated infrastructure, mostly in the 1970s, when a rapid urbanization process in the Laguna de Términos region led to the subsequent habitat degradation and land-use change. This process was not limited to the mainland, but also occurred with Isla del Carmen and Isla Aguada. These islands went from being towns, whose main activities were the fisheries, farming, and forestry, to becoming important urban centers, with important processes of land-use transformation [233]. Like the case of Cancún, the infrastructure development has transformed Isla Aguada into a peninsula [57].

Because of its strategic position, Cayo Arcas (a group of three sandy cays west of the coast of Campeche) is an important seaport for the charging of oil tankers and constitutes the most important port for the national oil industry). But it is also an important nesting site for marine turtles and seabirds. By 2 November 2019, researchers from Universidad Autónoma del Carmen detected via remote sensing two potential oil spills nearby Cayo Arcas, the first on October 4th with ~200 ha of extension, while the October 7th event was larger with an estimated affected area of 4500 ha, which probably damaged seabirds and hatchlings of *Chelonia mydas*. Despite the existence of a public statement from the Universidad Autónoma Del Carmen (https://www.pagina66.mx/wp-content/uploads/2019/12/DERRAME-CAYO-ARCAS.pdf accessed on 7 June 2023), several communication attemps with that university from the lead author of this paper remain unanswered. Even so, we could obtain photographic evidence (Figure 35) and a internal presentation facilitated by a reporter from a local media.An answer from PEMEX to an information request (number; 1,857,200,483,719, de fecha 08 de enero de 2020 en la Plataforma Nacional de Transparencia (PNT)) where they acknowledge two oil spills nearby Cayo Arcas (which they catalogue as having had an impact that is of a "minor" level), which constitutes confirmation that these spills really occurred. However, the whole extension and consequences of those spills remain unknown since the Agency for Security, Energy and Environment (Agencia de Seguridad, Energía y Ambiente "ASEA") maintains a three-year ban for the disclosure of any information (number of expedient: ASEA/USIVI/DGSIVEERC/AMB/0011/2019) related with that event.

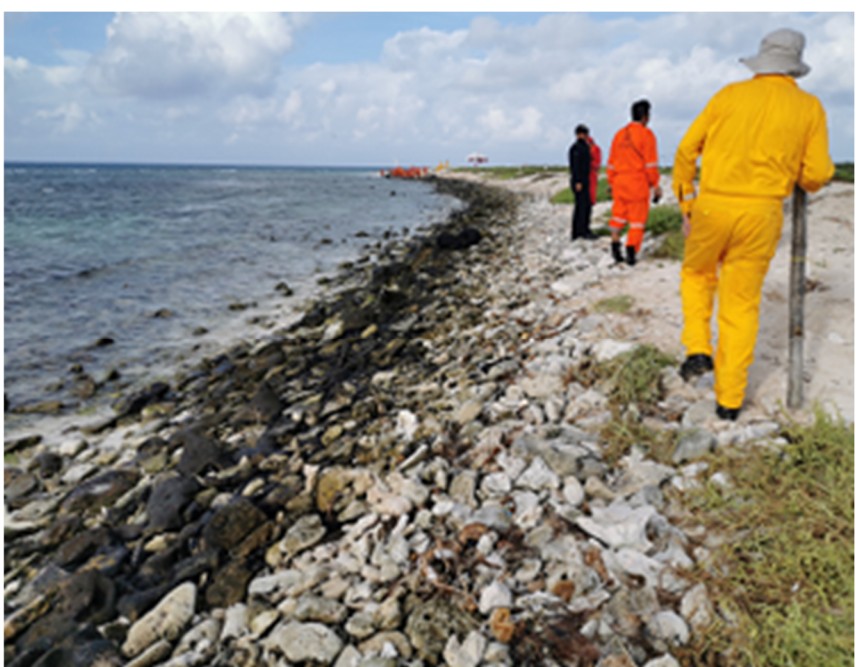

**Figure 35.** Brigades of PEMEX cleaning an oil spill in the beaches of Cayo Arcas (Gulf of Mexico) in late 2019. By its strategic location, Cayo Arcas is relevant as a logistical stepping stone for oil tankers. Since the oil industry is a matter of national security in Mexico, the information regarding these impacts is classified and not easily accessible, but it´s clear that oils spills are a constant menace for Mexican islands in the Gulf of Mexico. Photo by Secretaria de Marina (SEMAR) ceded by PAGINA 66 (www.pagina66.mx).

Even more serious is that the information closure extends to any oil spill incident on the Mexican islands: we made an information enquiry (331002522000635) to ASEA to require the relation of oil spills that had impacted (since 1950) on Mexican islands. However, that agency declared itself as incapable of providing an answer. Afterwards we filed a complaint and, subsequently, ASEA was ordered to answer our request. Nonetheless, they declared it would be impossible to provide an answer due to the "inexistence of records". The consequences of this blockade could be potentially harmful for the environment, since it makes it impossible to quantify the number and extention of those impacts and constitutes a major obstacle for conservation and/or restoration efforts. Finally, it contributes to the lack of awareness on the severity of the problem, since many of the oil spills would remain unknown to academics, conservationists, and the general public.

Unfortunately, the threat that oil and gas industries represent for Mexican biodiversity is far from being mitigated: in contrast with the rest of the world, which is moving towards alternative energy sources, the Mexican government has passed several legal reforms since 2018 that discourage investment in alternative energy sources and favors the oil industry, especially the highly contaminant parastatal PEMEX. There is little hope for a short- or mid-term change in this situation, as the production of petroleum and gas remains a deeply ingrained nationalist matter that most Mexicans are reluctant to challenge. Consequently, altering the existing legal framework for energy generation carries an immense political cost that no Mexican governors are willing to undertake.

### 3.6.9. Delinquency and Organized Crime

It is of public knowledge that Mexico has faced a crisis in matters of public security since the mid-2000s. How this crisis of insecurity affects the efforts of conservationists is something that is discussed in the informal talks among Mexican conservationists, but it is a topic that, with few exceptions, such as the problems involving the endemic porpoise known as the Vaquita (*Phocoena sinus*) on the High Gulf of California [234], is not discussed openly. It is even less possible to find any formal study focusing on that problem (but see

the compendium of Arroyo-Quiroz and Wyatt [235]), although subtle mentions of the problem can be found in few publications that discuss the many challenges of Mexican conservation [236,237]. Due to the obscure nature of these problems, their effects on biodiversity are not understood, but one of the main consequences of criminal activity is the displacement or abandonment of conservation efforts and/or research studies in sites that are considered too risky or dangerous for researchers and conservationists. This problem might be the principal effect in some insular territories; due to their remoteness, the islands are sites of difficult survelliance, and some of them are vulnerable to being used as a refuge for illegal fishermen (called "piratas" in the North and "pachocheros" in the Caribbean), or as steppingstones for drug dealers on their routes to the USA. Although it must be noted that the Mexican Navy is cooperative with the efforts of conservationists in the insular or oceanic territories, including sites where the activity of organized crime is intense, such as the High Gulf of California, where, otherwise, carrying out conservation programs could be much riskier [238].

3.6.10. Illegal Collecting

Illegal collecting is one of the activities that has had a great impact on reptile and amphibian populations [210]. In fact, reptiles are the second vertebrate group (just behind birds) most subjected to the pet trade, and only 8% of reptile species are regulated by CITES [239]. Additionally, the perceived rarity of a species increases the interest of collectors, which is reflected in an increase of collecting and an acceleration of extinction [240]. This problem also occurs with newly described species [241]. This problem is particularly worrisome with insular species, since they meet several risk factors, i.e., they are of restricted distribution, they are rare, and many of them are newly or yet to be described species; also, the characteristic naïvity of insular organisms makes them easy to collect.

Despite these considerations, the documentation of cases related to insular herpetofaunas in Mexico is scarce. The best-known example is the marine turtle, which is especially vulnerable when spawning on beaches, as they are exposed to poaching and egg harvesting [57]. This problem is especially evident on Cayo Lobos (Banco Chinchorro) in the Caribbean; this cay appears to be an important site for sea turtle nesting, but due to its remotenes and absence of surveillance, the site is frequented by illegal fisermen who collect turtle eggs [66]. In the Gulf of California, the hunting of sea turtles dates back for 12,000 years, with a peak in the mid-20th century, when this activity went from one of subsistence to one of marked economy. The regulation in the 1970s and the subsequent ban in 1990 reduced the exploitation of sea turtles [242], but it still occurs in the Baja California Peninsula [243]. We are unaware of the current situation of sea turtle hunting and/or egg collecting in the islands of Baja California, but certainly they are vulnerable sites for poachers.

The illegal collection of living specimens for the pet trade is one of the principal menaces for rattlesnakes [76]. Little, however, is known about this problem. Arnaud et al. [214] mentioned that illegal collecting of *C. catalinensis* occurred in the past, but also suggested a diminution of this activity, thanks to the environmental education of the local people. Mellink (1995, in Lovich, [229]) reported illegal collecting of *Lampropeltis herrerae* in Todos Santos islands by trapping (using living mice as bait), and spotted people searching for kingsnakes by turning rocks (Figure 36). He also mentioned another case of supposed scientists collecting the variety of Mexican Rosy Boas (*Lichanura trivirgata*) on Isla Cedros, who left the island when they were requested to show their scientific collecting permit [244]. Delibes et al. [245] found that populations of *Aspidoscelis hyperythra* showed a high ability to escape predators in the populations of seven islands in the south of Baja California. They considered that this conduct, uncharacteristic of isolated populations, might be the result of selection due to intense reptile harvesting.

The damage of collecting is not limited to the individual level, but it can extend to the population and community level as well. Often collectors use their hands, crowbars, hydraulic jacks, and other tools to break rocks in an attempt to collect reptiles. This activity

results in the permanent destruction of refuges or microhabitats that might have been used for decades by amphibians or reptiles. Thus, the continuous destruction of micro-habitats has detrimental effects on populations of herpetofauna [246]. Certanly, this can be pernicious on small-sized islands, since the area and habitat available are limited.

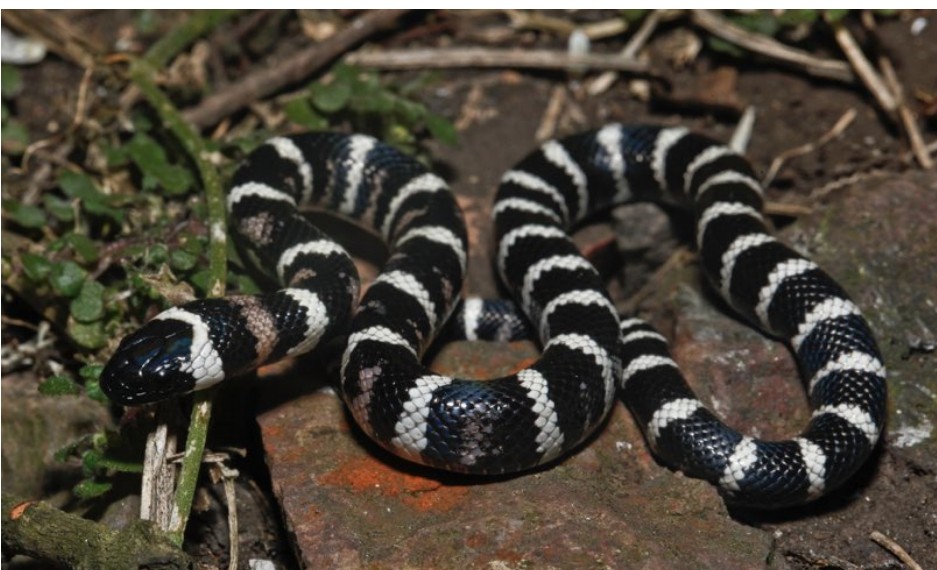

**Figure 36.** Todos Santos Island Kingsnake (*Lampropeltis herrerae*). The beauty, uniqueness, and rarity (endemic to Todos Santos Archipielago) of this Kingsnake make it an appealing trophy for unethical herpetoculturists. Furthermore, the Todos Santos islands are not far from Ensenada's (Baja California) port, so they are easily reachable to fishermen and boatmen. Furthermore, Todos Santos Sur is known wordwide for its tubular waves, which attract surfers from all over the world. So, the easy access, plus frequent and unregulated visitors, together with poor vigilance action, makes this Kingsnake highly vulnerable to extinction by poachers. Distributional Status = IE, EVS = H (17), IUCN = CR, NOM-059 = A. Matt Cage, 2018-Isla Todos Santos.

*3.7. Conservation Status*

3.7.1. The SEMARNAT System

The SEMARNAT system was developed by the Secretaria del Medio Ambiente y Recursos Naturales and is legally established through the NOM-059-SEMARNAT-2010, which has the objective to "identify the species or populations of wild flora and fauna threatened in the Mexican Republic" [247]. This norm is the principal reference frame for Mexican biologists to assign a risk category and encompasses four levels: probably extinct in the wild (E), endangered (P), threatened (A), and under special protection (Pr) (Table 13).

**Table 13.** SEMARNAT categorizations for the herpetofaunal species in the insular systems of Mexico, arranged by families. Non-native species are not included. Bolded lines are total/subtotals

| Families | Number of Species | SEMARNAT Categorizations | | | |
|---|---|---|---|---|---|
| | | Endangered (P) | Threatened (A) | Special Protection (Pr) | No Status (NS) |
| Bufonidae | 4 | — | — | — | 4 |
| Eleutherodactylidae | 1 | — | — | 1 | — |
| Hylidae | 5 | — | — | — | 5 |
| Leptodactylidae | 1 | — | — | — | 1 |
| Microhylidae | 1 | — | — | — | 1 |
| Scaphiopodidae | 1 | — | — | — | 1 |
| **Subtotals** | **13** | **—** | **—** | **1** | **12** |

| | | | | | |
|---|---|---|---|---|---|
| Plethodontidae | 2 | — | — | 1 | 1 |
| **Subtotals** | **2** | **—** | **—** | **1** | **1** |
| **Totals** | **15** | **—** | **—** | **2** | **13** |
| Crocodylidae | 2 | — | — | 2 | — |
| **Subtotals** | **2** | **—** | **—** | **2** | **—** |
| Anguidae | 3 | — | — | 2 | 1 |
| Anniellidae | 2 | — | — | 2 | — |
| Bipedidae | 1 | — | — | 1 | — |
| Corytophanidae | 1 | — | — | — | 1 |
| Crotaphytidae | 4 | — | — | 1 | 3 |
| Dactyloidae | 4 | — | — | — | 4 |
| Eublepharidae | 3 | — | 1 | 1 | 1 |
| Iguanidae | 13 | — | 6 | 3 | 4 |
| Mabuyidae | 2 | — | — | — | 2 |
| Phrynosomatidae | 31 | — | 10 | 5 | 16 |
| Phyllodactylidae | 15 | — | 1 | 3 | 11 |
| Scincidae | 2 | — | — | — | 2 |
| Sphaerodactylidae | 3 | — | — | 2 | 1 |
| Teiidae | 22 | 1 | 4 | 7 | 10 |
| **Subtotals** | **106** | **1** | **22** | **27** | **56** |
| Boidae | 2 | — | 2 | — | — |
| Colubridae | 36 | — | 8 | 3 | 25 |
| Dipsadidae | 19 | — | 1 | 7 | 11 |
| Elapidae | 3 | — | 1 | — | 2 |
| Leptotyphlopidae | 3 | — | — | — | 3 |
| Natricidae | 1 | — | 1 | — | — |
| Viperidae | 18 | — | 2 | 8 | 8 |
| **Subtotals** | **82** | **—** | **15** | **18** | **49** |
| Cheloniidae | 5 | 5 | — | — | — |
| Dermatemydidae | 1 | 1 | — | — | — |
| Dermochelyidae | 1 | 1 | — | — | — |
| Emydidae | 1 | — | — | — | 1 |
| Geoemydidae | 1 | — | 1 | — | — |
| Kinosternidae | 2 | — | — | 2 | — |
| Staurotypidae | 1 | — | 1 | — | — |
| Testudinidae | 1 | — | 1 | — | — |
| **Subtotals** | **13** | **7** | **3** | **2** | **1** |
| **Totals** | **203** | **8** | **40** | **49** | **106** |
| **Sum Totals** | **218** | **8** | **40** | **51** | **119** |

Of the 218 native species of herpetofauna occurring in the Mexican insular systems, 119 (54.6%) are not listed in the NOM-059 in any risk category, 51 (23.4%) are under special protection (Pr), 40 (18.3%) are in the threatened level (A), and just 8 (3.7%) are in the endangered (P) status. Of these endangered species, seven are turtles, six of which are marine turtles, and the other is the river turtle *Dermatemys mawii*. Only one teiid lizard, *A. rodecki*, appears in this category. The low incidence of endangered species under the NOM-059 is striking, and the omission of particularly relevant species such the Clarion Nightsnake (*Hypsiglena unaocularus*)[89], believed to be extinct, or the Santa Catalina Kingsnake (*Lampropeltis catalinensis*), which is known only from the holotype [71], is of concern. Hence, we assert that the NOM-059 underestimates the actual level of risk faced

by insular populations of herpetofauna. This underestimation becomes more worrisome if we consider the existence of subspecific taxa on islands that might be facing a higher risk of extinction than the continental lineages. Consequently, in the next revision to the NOM-059, special attention must be paid to addressing insular reptiles, since they are species with a high risk of extinction.

### 3.7.2. The IUCN System

The International Union for Conservation of Nature has been, since the 1950s, the principal institution for compiling lists of species at risk of extinction. The primary objective behind creating red lists is to heighten awareness and guide conservation efforts for various species. [248]. Initially, the red lists were heavily dependent of the opinion of experts, but since the mid-1990s the IUCN Red List has become based on a standard set of criteria in order to provide a useful listing and categorization for conservation, monitoring, and decision-making [249]. Therefore, the aims of the red list are: (1) to provide a global index of the state of degeneration of biodiversity; and (2) identify and document those species most in need of conservation attention if global extinction rates are to be reduced [248].

Although not exempt from criticism, the IUCN is the most-used system for assessing species risk; consequently, we use this list in this work to provide a frame of comparison with the EVS and SEMARNAT systems. This IUCN Red List encompasses eight categories, including Extinct (EX), Extinct in the Wild (EW), Critically Endangered (CR), Endangered (EN), Vulnerable (VU), Near Threatened (NT), Least Concern (LC), and Data Deficient (DD).

Of the 218 native species of herpetofauna, only 21 (9.6%) have been allocated to the three "threat categories," including six in the CR category, three in the EN category, and twelve in the VU category. The two "lower risk" categories include 141 species (64.7%), including 6 NT and 135 LC species, respectively. Finally, 56 species (25.7%) are placed in the DD and NE categories. No insular species is considered to be Exctinct (EX) or Extinct in the Wild (EW). Although, *Lampropeltic catalinensis* almost certainly is extinct (Table 14).

**Table 14.** IUCN Red List categorizations for the herpetofaunal families in the Insular systems of Mexico. Non-native species are excluded. The shaded columns on the left are the "threat categories," and the one on the right is the category summarizing those that have not been evaluated or are listed as Data Deficient (DD). Bolded lines are totals/subtotals.

| Families | Number of Species | IUCN Red List Categorizations | | | | | |
| --- | --- | --- | --- | --- | --- | --- | --- |
| | | Critically Endangered | Endangered | Vulnerable | Near Threatened | Least Concern | Not Evaluated/DD |
| Bufonidae | 4 | — | — | — | — | 4 | — |
| Eleutherodactylidae | 1 | — | — | — | — | — | 1 |
| Hylidae | 5 | — | — | — | — | 5 | — |
| Leptodactylidae | 1 | — | — | — | — | 1 | — |
| Microhylidae | 1 | — | — | — | — | 1 | — |
| Scaphipodidae | 1 | — | — | — | — | 1 | — |
| **Subtotals** | **13** | **—** | **—** | **—** | **—** | **12** | **1** |
| Plethodontidae | 2 | — | — | — | — | 2 | — |
| **Subtotals** | **2** | **—** | **—** | **—** | **—** | **2** | **—** |
| **Totals** | **15** | **—** | **—** | **—** | **—** | **14** | **1** |
| Crocodylidae | 2 | — | — | 1 | — | 1 | — |
| **Subtotals** | **2** | **—** | **—** | **1** | **—** | **1** | **—** |
| Anguidae | 3 | — | — | — | — | 2 | 1 |
| Anniellidae | 2 | — | 1 | — | — | 1 | — |
| Bipedidae | 1 | — | — | — | — | 1 | — |

| | | | | | | | |
|---|---|---|---|---|---|---|---|
| Corytophanidae | 1 | — | — | — | — | 1 | — |
| Crotaphytidae | 4 | — | — | — | — | 4 | — |
| Dactyloidae | 4 | — | — | — | — | 1 | 3 |
| Eublepharidae | 3 | — | — | — | — | 3 | — |
| Iguanidae | 13 | — | — | 1 | 1 | 4 | 7 |
| Mabuyidae | 2 | — | — | — | — | 1 | 1 |
| Phrynosomatidae | 31 | — | 1 | 5 | — | 24 | 1 |
| Phyllodactylidae | 15 | — | — | — | — | 6 | 9 |
| Scincidae | 2 | — | — | — | — | 2 | — |
| Sphaerodactylidae | 3 | — | — | — | — | 2 | 1 |
| Teiidae | 22 | — | — | 2 | 1 | 18 | 1 |
| **Subtotals** | **106** | **—** | **2** | **8** | **2** | **70** | **24** |
| Boidae | 2 | — | — | — | — | — | 2 |
| Colubridae | 36 | 2 | — | — | — | 22 | 12 |
| Dipsadidae | 19 | — | — | — | — | 13 | 6 |
| Elapidae | 3 | — | — | — | — | 2 | 1 |
| Leptotyphlopidae | 3 | — | — | — | — | — | 3 |
| Natricidae | 1 | — | — | — | — | 1 | — |
| Viperidae | 18 | 1 | — | — | 1 | 11 | 5 |
| **Subtotals** | **82** | **3** | **—** | **—** | **1** | **49** | **29** |
| Cheloniidae | 5 | 2 | 1 | 2 | — | — | — |
| Dermatemydidae | 1 | 1 | — | — | — | — | — |
| Dermochelyidae | 1 | — | — | 1 | — | — | — |
| Emydidae | 1 | — | — | — | — | — | 1 |
| Geoemydidae | 1 | — | — | — | 1 | — | — |
| Kinosternidae | 2 | — | — | — | 1 | 1 | — |
| Staurotypidae | 1 | — | — | — | 1 | — | — |
| Testudinidae | 1 | — | — | — | — | — | 1 |
| **Subtotals** | **13** | **3** | **1** | **3** | **3** | **1** | **2** |
| **Totals** | **202** | **6** | **3** | **12** | **6** | **121** | **55** |
| **Sum Totals** | **218** | **6** | **3** | **12** | **6** | **135** | **56** |
| **Category Totals** | **218** | **21** | | | **141** | | **56** |

### 3.7.3. The EVS System

The Environmental Vulnerability Score system is an algorithm developed by Wilson and McCranie [250] to assess the vulnerability of herpetofaunal species using measures such the extent of geographic range, extent of ecological distribution, and degree of specialization of reproductive mode in amphibians or, in the case of reptiles, the degree of human persecution. The utility and practicality of this system is discussed by Johnson et al. [37] and Wilson et al. [24,25]. The EVS values range from 3 to 20 and are grouped into three categories, i.e., Low (3–9), Medium (10–13), and High (14–20) vulnerability (Table 15).

**Table 15.** Environmental Vulnerability Scores (EVS) for the herpetofaunal species in the insular systems of Mexico, arranged by family. Shaded area on the left encompasses the low vulnerability scores, and the one on the right the high vulnerability scores. Bolded lines indicate totals/subtotals. Non-native and marine species are not included.

| Families | Number of Species | Environmental Vulnerability Scores | | | | | | | | | | | | | | | | |
|---|---|---|---|---|---|---|---|---|---|---|---|---|---|---|---|---|---|---|
| | | 3 | 4 | 5 | 6 | 7 | 8 | 9 | 10 | 11 | 12 | 13 | 14 | 15 | 16 | 17 | 18 | 19 |
| Bufonidae | 4 | 1 | — | 1 | 1 | — | — | — | — | — | 1 | — | — | — | — | — | — | — |
| Eleutherodactylidae | 1 | — | — | — | — | — | — | — | — | — | — | — | — | — | — | 1 | — | — |
| Hylidae | 5 | 2 | 2 | — | — | 1 | — | — | — | — | — | — | — | — | — | — | — | — |
| Leptodactylidae | 1 | — | — | 1 | — | — | — | — | — | — | — | — | — | — | — | — | — | — |
| Microhylidae | 1 | — | 1 | — | — | — | — | — | — | — | — | — | — | — | — | — | — | — |
| Scaphiopodidae | 1 | 1 | — | — | — | — | — | — | — | — | — | — | — | — | — | — | — | — |
| **Subtotals** | **13** | **4** | **3** | **2** | **1** | **1** | **—** | **—** | **—** | **—** | **1** | **—** | **—** | **—** | **—** | **1** | **—** | **—** |
| Plethodontidae | 2 | — | — | — | — | — | — | — | — | — | — | — | 2 | — | — | — | — | — |
| **Subtotals** | **2** | **—** | **—** | **—** | **—** | **—** | **—** | **—** | **—** | **—** | **—** | **—** | **2** | **—** | **—** | **—** | **—** | **—** |
| **Totals** | **15** | **4** | **3** | **2** | **1** | **1** | **—** | **—** | **—** | **—** | **1** | **—** | **2** | **—** | **—** | **1** | **—** | **—** |
| Crocodylidae | 2 | — | — | — | — | — | — | — | — | — | — | 1 | 1 | — | — | — | — | — |
| **Subtotals** | **2** | **—** | **—** | **—** | **—** | **—** | **—** | **—** | **—** | **—** | **—** | **1** | **1** | **—** | **—** | **—** | **—** | **—** |
| Anguidae | 3 | — | — | — | — | — | — | — | 1 | — | — | — | — | — | 2 | — | — | — |
| Anniellidae | 2 | — | — | — | — | — | — | — | — | — | 1 | 1 | — | — | — | — | — | — |
| Bipedidae | 1 | — | — | — | — | — | — | — | — | — | — | — | 1 | — | — | — | — | — |
| Corytophanidae | 1 | — | — | — | — | 1 | — | — | — | — | — | — | — | — | — | — | — | — |
| Crotaphytidae | 4 | — | — | — | — | — | — | — | — | 1 | — | 1 | — | — | 2 | — | — | — |
| Dactyloidae | 4 | — | — | — | — | — | 2 | — | 1 | — | — | 1 | — | — | — | — | — | — |
| Eublepharidae | 3 | — | — | — | — | — | — | 1 | — | 1 | — | — | — | — | — | — | 1 | — |
| Iguanidae | 13 | — | — | — | — | — | 1 | — | — | 1 | 1 | 1 | 1 | 1 | 4 | 2 | 1 | — |
| Mabuyidae | 2 | — | — | — | — | — | — | — | 1 | — | — | 1 | — | — | — | — | — | — |
| Phrynosomatidae | 31 | — | — | 1 | — | 2 | 1 | 1 | 2 | — | 5 | 3 | 2 | 1 | 5 | 8 | — | — |
| Phyllodactylidae | 15 | — | — | — | — | — | 1 | — | 1 | — | — | — | — | 4 | 6 | 3 | — | — |
| Scincidae | 2 | — | — | — | — | — | — | — | — | 2 | — | — | — | — | — | — | — | — |
| Sphaerodactylidae | 3 | — | — | — | — | — | — | — | 1 | — | 1 | 1 | — | — | — | — | — | — |
| Teiidae | 22 | — | — | — | — | — | 2 | — | 1 | 1 | 1 | — | 2 | 3 | 5 | 7 | — | — |
| **Subtotals** | **106** | **—** | **—** | **1** | **—** | **3** | **7** | **2** | **8** | **6** | **9** | **9** | **6** | **9** | **24** | **20** | **2** | **—** |
| Boidae | 3 | — | — | — | — | — | — | — | 2 | — | — | — | — | 1 | — | — | — | — |
| Colubridae | 36 | — | — | 2 | 6 | — | 1 | 6 | 5 | 4 | 2 | 1 | 1 | 1 | 2 | 5 | — | — |
| Dipsadidae | 19 | — | 1 | 2 | 2 | — | 4 | 2 | 1 | 2 | — | 2 | — | 1 | 2 | — | — | — |
| Elapidae | 2 | — | — | — | — | — | 1 | — | — | — | — | — | — | 1 | — | — | — | — |
| Leptotyphlopidae | 3 | — | — | — | — | — | 1 | — | — | 1 | 1 | — | — | — | — | — | — | — |
| Natricidae | 1 | — | — | — | — | 1 | — | — | — | — | — | — | — | — | — | — | — | — |
| Viperidae | 18 | — | — | — | — | — | 1 | 2 | — | 1 | 2 | 1 | — | 1 | 2 | — | 5 | 3 |
| **Subtotals** | **82** | **—** | **1** | **4** | **8** | **1** | **8** | **10** | **8** | **8** | **5** | **4** | **1** | **5** | **6** | **5** | **5** | **3** |
| Dermatemydidae | 1 | — | — | — | — | — | — | — | — | — | — | — | — | — | — | 1 | — | — |
| Emydidae | 1 | — | — | — | — | — | — | — | — | — | — | — | — | — | — | — | — | 1 |
| Geoemydidae | 1 | — | — | — | — | — | — | — | — | — | — | 1 | — | — | — | — | — | — |
| Kinosternidae | 2 | — | — | — | — | — | — | — | 1 | 1 | — | — | — | — | — | — | — | — |
| Staurotypidae | 1 | — | — | — | — | — | — | — | — | — | — | — | 1 | — | — | — | — | — |
| Testudinidae | 1 | — | — | — | — | — | — | — | — | — | — | — | — | 1 | — | — | — | — |
| **Subtotals** | **7** | **—** | **—** | **—** | **—** | **—** | **—** | **—** | **1** | **1** | **—** | **1** | **1** | **1** | **—** | **1** | **—** | **1** |
| **Totals** | **197** | **0** | **1** | **5** | **8** | **4** | **15** | **12** | **17** | **15** | **14** | **15** | **9** | **15** | **30** | **26** | **7** | **4** |
| **Sum Totals** | **212** | | | **56** | | | | | | | **62** | | | | | **94** | | |

We calculated the EVS for 202 of the 216 insular species; the non-native and the marine species are excluded. The highest score we obtained was 19 for four species: *Crotalus*

*catalinensis*, *C. estebanensis*, *C. lorenzoensis*, and *Trachemys venusta*. The species of *Crotalus* are of particular concern. Besides these four species, another seven have an EVS of 18: *Coleonyx gypsicolus*, *C. hemilopha*, *Crotalus angelensis*, *C. caliginis*, *C. polisi*, *C. thalassoporus*, and *C. tortuguensis*. Again, species of *Crotalus* are of significant interest. Two other rattlesnakes have an EVS of 16 (*C. cerastes* and *C. tigris*); thus, 10 of the 16 species (62.5%) of the rattlesnakes occupying insular systems in Mexico are allocated to the high vulnerability category (Table 10).

Regarding the amphibians, most of them have very low EVS values; in fact, of the eight lowest values we estimated (EVS = 3–4), seven were for amphibians. Only one toad, *Incilius mazatlanensis*, has a medium value (12) and only *Eleutherodactylus pallidus* has a high EVS (17). Additionally, the only two known salamanders occurring on Mexican islands, *Aneides lugubris* and *Batrachoseps major*, both have a high EVS value (14) (Figure 37).

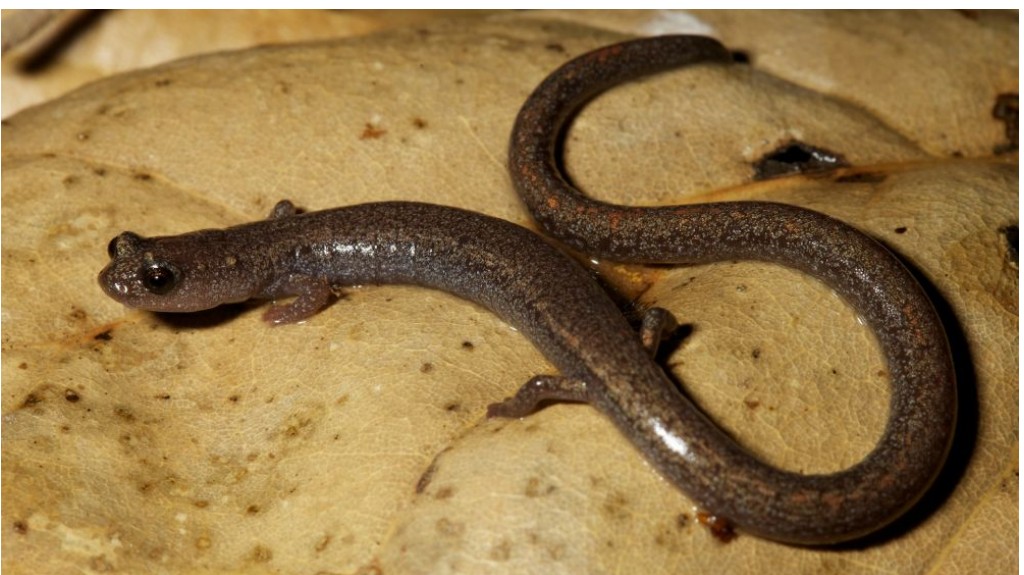

**Figure 37.** Batrachoseps major, a plethodonthid salamander inhabiting the Californian region. Although this specimen is from San Diego County (U.S.A.), The presence of this species in the Coronado Archipelago suggests the possibility of it being a relic population that recalls the islands' continental origin, but this asseveration must be proven by molecular studies. Distributional Status = NE, EVS = H (14), IUCN = LC, NOM-059 = NS. Photo from the Amphibian and Reptile Atlas of Peninsular California ([herpatlas.sdnhm.org], San Diego Natural History Museum), courtesy of Bradford Hollingsworth.

Reptiles as a group have EVS ranging from 4 to 19 (Table 15). The EVS for these creatures are arranged into the three categories of vulnerability as follows: low—46 species; medium—57 species; and high—84 species. Thus, it is evident that the number of species rise through the levels of vulnerability from low, through medium, to high.

In summary, of the 212 species, 56 (26.4%) have a low EVS, 62 (29.2%) have a medium EVS, and 94 (44.5%) have a high EVS. Thus, the same pattern is evident for the entire herpetofauna as for the reptiles alone, with the proportion of the low EVS more heavily represented for the total herpetofauna than for the reptile component, giving greater representation of low-EVS species among the amphibians (Table 15). Clearly, this pattern of EVS representation reflects the relatively high level of endemism, especially reptilian endemism, in insular systems. This same pattern exists in Mexican states with high levels of endemism, such as in Oaxaca [178] and Puebla [179], with the inverse pattern occurring in regions with limited endemism, such as the Yucatan Peninsula [57].

When comparing the EVS and IUCN categorizations (Table 16), the results indicate that only 15 belong to any of the three risk categories (Critically Endangered, Endangered, or Vulnerable) of the IUCN system. This figure is only 17.2% of the 87 species with high

EVS values. Alternatively, the 135 species within the LC category of the IUCN constitute 2.4 times the number of the low vulnerability species (57 species, which are the 42.2% of the IUCN LC). As can be seen, the results of the application of the EVS and the IUCN systems do not correspond well with each other (Table 16). This discrepancy is of concern if we consider that the IUCN Red List omits several endemic and/or conspicuous species, such as iguanas of the genera *Ctenosaura*, *Sauromalus*, and *Dipsosaurus*, notable endemics such *Elgaria cedrosensis*, and *Hypsiglena unaocularis* (Figure 38), abundant and common species like *Oxybelis aeneus*, and also has not evaluated species of recent description such as *Crotalus polisi* and *C. thalassoporus* (Figure 39).

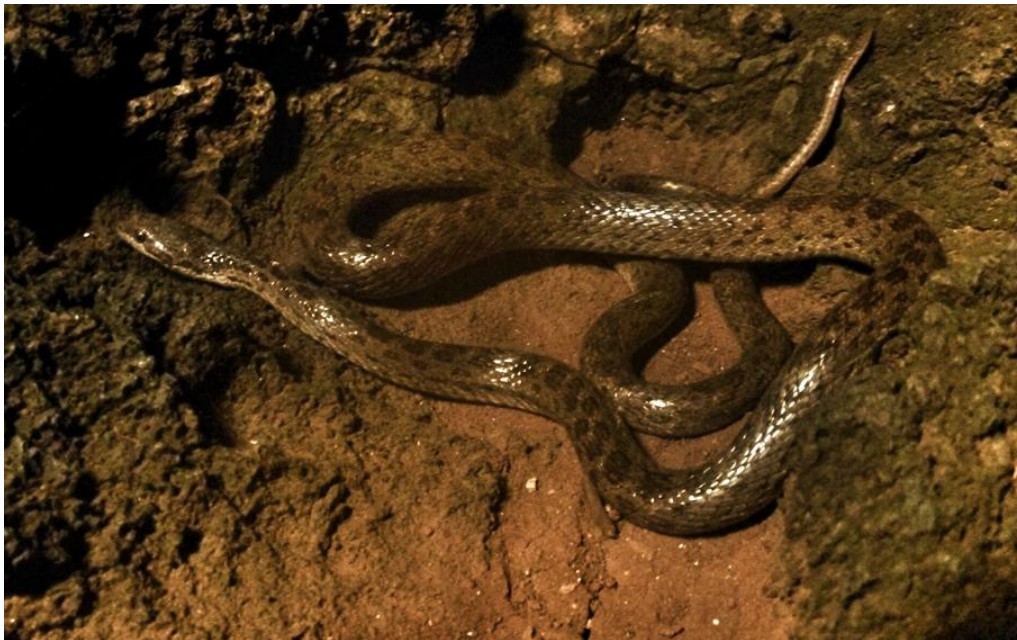

**Figure 38.** *Hypsiglena unaocularis* believed extinct for decades, its rediscovery allowed for molecular analysis, which concluded that it is a valid species (as *H. unaocularus*; [102]). EVS = H (16), UICN = NE, NOM-059 = NS. Photo by Humberto Almanza.

The lack of correspondence between the EVS and IUCN systems of conservation categorization is explicable for two principal reasons: first, there are several species that are in the category of not evaluated (NE) or data deficient (DD), as determined by the IUCN (Table 16). Of these 56 species, 4 are allocated to the DD category (Table 17) and 52 to the NE category (Table 18). The four DD species include one anuran and three squamates (colubrid snakes), with EVS ranging from 16 to 17. Thus, all four are high vulnerability species and the three species with an EVS of 17 most likely should be allocated to the CR category and the single species with an EVS of 16 to the EN category. The 52 NE species (Table 18) are all squamates, except for two turtles. Their EVS vary from 5 (L) to 19 (H); therefore, some of these 52 species that have not been evaluated by the IUCN are allocated to each of the three risk categories in the EVS system; 15 species in the low level, 12 in the medium level, and 25 in the high vulnerability level (Table 16). Remarkably, this means that 40.1% of the species unevaluated by the IUCN are of high risk according to the EVS system (Table 16). Our recommendation would be that the species with an EVS of 17 to 19 be allocated to the CR category (six species) and those with an EVS of 15 to 16 to the EN category (19 species). The remaining 27 species with an EVS of 5 to 12 can be best allocated to the LC category (Table 16).

**Table 16.** Comparison of the Environmental Vulnerability Scores (EVS) and applicable IUCN categorizations for members of the herpetofauna of the insular systems of Mexico. Non-native and marine species are excluded. The shaded area at the top encompasses the low vulnerability category scores, and the one at the bottom the high vulnerability category scores.

| EVS | IUCN Categories | | | | | | | Totals |
|---|---|---|---|---|---|---|---|---|
| | Critically Endangered | Endangered | Vulnerable | Near Threatened | Least Concern | Not Evaluated | Data Deficient | |
| 3 | — | — | — | — | 4 | — | — | 4 |
| 4 | — | — | — | — | 4 | — | — | 4 |
| 5 | — | — | — | — | 5 | 2 | — | 7 |
| 6 | — | — | — | — | 7 | 2 | — | 9 |
| 7 | — | — | — | — | 5 | — | — | 5 |
| 8 | — | — | — | — | 8 | 7 | — | 15 |
| 9 | — | — | — | — | 9 | 3 | — | 12 |
| 10 | — | — | — | 1 | 10 | 6 | — | 17 |
| 11 | — | — | — | 1 | 11 | 3 | — | 15 |
| 12 | — | — | — | — | 13 | 2 | — | 15 |
| 13 | — | 1 | — | 1 | 12 | 1 | — | 15 |
| 14 | — | — | 1 | 2 | 8 | — | — | 11 |
| 15 | — | — | — | — | 9 | 6 | — | 15 |
| 16 | — | 1 | — | 1 | 15 | 12 | 1 | 30 |
| 17 | 3 | — | 8 | — | 10 | 3 | 3 | 27 |
| 18 | — | — | — | — | 3 | 4 | — | 7 |
| 19 | 1 | — | — | — | 2 | 1 | — | 4 |
| Totals | 4 | 2 | 9 | 6 | 135 | 52 | 4 | 212 |

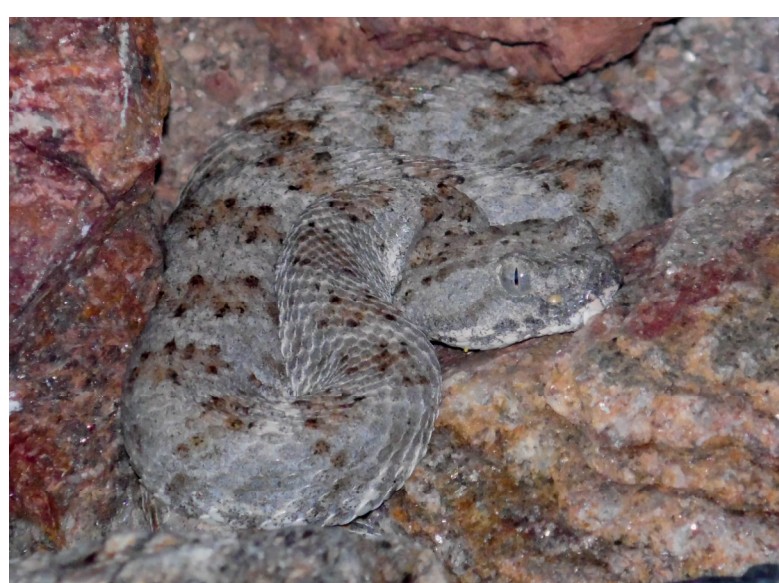

**Figure 39.** *Crotalus thalassoporus* (Isla Piojo Speckled Rattlesnake) is a rattlesnake of recent description (2018) and only known from its type locality (Isla Piojo, Gulf of California). An appealing advantage of EVS is that it gives a quick valoration of the vulnerability degree for species of recent description like this snake which, despite it being highly vulnerable, it still has not been listed neither by IUCN, nor NOM-059. Distribution status = IE, EVS = H (18), IUCN = NE, NOM-059 = NS. Photo by Tania Pérez-Fiol.

The second cause of the discrepancy is the possibility of an overuse of the LC category in the IUCN Red List. In this study, from the 212 native non-marine insular species, 135

(63.7%) correspond to the Least Concern level. Moreover, the EVS values for these 135 species ranges from L (3) to H (19), encompassing almost all the possible values for the EVS system (3–20). By looking at those values, as arranged in the three summary categories of EVS, it is interesting to see that they are distributed almost equally: 47 species in the High EVS level (38.5%), 46 in the Medium (34.1%), and 42 for Low EVS level (31.1%). It is tempting to think that this distribution, almost divided into thirds, suggests that either the IUCN or the EVS system could lead to random classifications of insular herpetofauna. An explanation could be that the IUCN might be biased by geographic scale issues, or the fact that the IUCN systems require enough knowledge of the species in order to weight a set of criteria [251,252], regarding the geographic scale, since the IUCN evaluates the risk of global extinction, this issue should not be a matter of discrepancy in endemic species, but might be in species with reduced distribution ranges [253]. With reference to the second point, insular herpetofaunal populations are comparatively less studied than their continental counterparts, mainly due to their limited accessibilityHowever, this is a primary speculation, but we think this is a topic worthy of exploration by a larger contrast of the IUCN against the EVS of insular populations in several countries.

Of the 212 species that can be provided with an EVS categorization, 94 species (47.2%) fall within the high vulnerability category. Of considerable significance is that of these 94 species, 77 (81.9%) are allocated to the LC, NE, and DD categories (Table 16). These 77 species are listed in Tables 17–19. A significant number of these 77 species are insular endemics (Like *A. ceralbensis*, Figure 40), including:

| | |
|---|---|
| *Elgaria nana* | *Aspidoscelis cana* |
| *Crotaphytus dickersonae* | *Aspidoscelis carmenensis* |
| *Crotaphytus insularis* | *Aspidoscelis celeripes* |
| *Coleonyx gypsicolus* | *Aspidoscelis ceralbelsis* |
| *Ctenosaura conspicuosa* | *Aspidoscelis danheimae* |
| *Dipsosaurus catalinensis* | *Aspidoscelis espiritensis* |
| *Sauromalus klauberi* | *Aspidoscelis franciscensis* |
| *Petrosaurus slevini* | *Aspidoscelis pictus* |
| *Sceloporus angustus* | *Lampropeltis catalinensis* |
| *Sceloporus grandaevus* | *Masticophis barbourin* |
| *Sceloporus lineatulus* | *Masticophis slevini* |
| *Uta nolascensis* | *Pituophis insularis* |
| *Phyllodactylus angelensis* | *Rhinocheilus etheridgei* |
| *Phyllodactylus apricus* | *Hypsiglena catalinae* |
| *Phyllodactylus bugastrolepis* | *Hypsiglena unaocularis* |
| *Phyllodactylus cleofasensis* | *Crotalus angelensis* |
| *Phyllodactylus coronatus* | *Crotalus caliginis* |
| *Phyllodactylus isabelae* | *Crotalus estebanensis* |
| *Phyllodactylus lupitae* | *Crotalus lorenzoensis* |
| *Phyllodactylus partidus* | *Crotalus polisi* |
| *Phyllodactylus tuberculosus* | *Crotalus thalassoporus* |
| *Aspidoscelis bacata* | *Crotalus tortuguensis* |

This list consists of 44 species, which is 78.6% of the insular endemics recorded for the insular systems of Mexico. Clearly, this group of species merits a much closer examination and likely upgraded IUCN categorization, which is among our recommendations (see below).

**Table 17.** Environmental Vulnerability Scores (EVS) for members of the herpetofauna of the Mexican insular systems assigned to the IUCN Data Deficient category. Non-native and marine taxa are not included.

| Taxa | Environmental Vulnerability Score (EVS) | | | |
|---|---|---|---|---|
| | Geographic Distribution | Ecological Distribution | Reproductive Mode/Degree of Persecution | Total Score |
| *Eleutherodactylus pallidus* | 5 | 8 | 4 | H (17) |
| *Lampropeltis catalinensis* | 6 | 8 | 3 | H (17) |
| *Masticophis barbouri* | 6 | 8 | 3 | H (17) |
| *Rhinocheilus etheridgei* | 6 | 7 | 3 | H (16) |

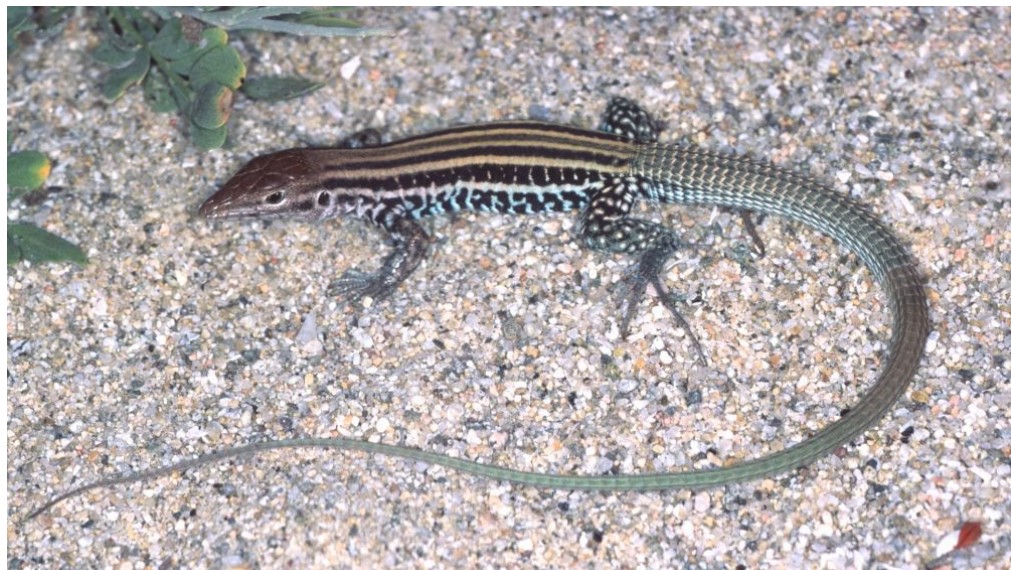

**Figure 40.** Cerralvo Island Whiptail (*Aspidoscelis ceralbensis*). With 22 species, Whiptal lizards contribute greatly to insular herpetofauna diversity (21 *Aspidoscelis* and 1 *Holcosus*). Eleven *Aspidoscelis* are insular endemics. Most of them are on the Sea of Cortes (13 species). Six are under IUCN's Low-Concern category, despite having High EVS values. Photo: Amphibian and Reptile Atlas of Peninsular California (herpatlas.sdnhm.org), San Diego Natural History Museum), courtesy of Bradford Hollingsworth.

**Table 18.** Environmental Vulnerability Scores (EVS) for members of the herpetofauna of the Mexican insular systems currently not evaluated (NE) by the IUCN. Non-native and marine taxa are not included. No asterisk = Non-endemic; * = country endemic and ** = insular endemic.

| Taxa | Environmental Vulnerability Score (EVS) | | | |
|---|---|---|---|---|
| | Geographic Distribution | Ecological Distribution | Reproductive Mode/Degree of Persecution | Total Score |
| *Elgaria cedrosensis* | 5 | 8 | 3 | H (16) |
| *Norops lemurinus* | 3 | 2 | 3 | L (8) |
| *Norops rodriguezii* | 4 | 3 | 3 | M (10) |
| *Norops ustus* | 4 | 1 | 3 | L (8) |
| *Ctenosaura conspicuosa* | 5 | 8 | 3 | H (16) |

| | | | | |
|---|---|---|---|---|
| *Ctenosaura hemilopha* | 5 | 7 | 6 | H (18) |
| *Ctenosaura pectinate* | 5 | 4 | 6 | H (15) |
| *Dipsosaurus catalinensis* | 6 | 8 | 3 | H (17) |
| *Sauromalus klauberi* | 6 | 7 | 3 | H (16) |
| *Sauromalus slevini* | 5 | 8 | 3 | H (16) |
| *Sauromalus varius* | 5 | 8 | 3 | H (16) |
| *Marisora aquilonaria* * | 5 | 3 | 3 | M (13) |
| *Phrynosoma cerroense* | 6 | 7 | 3 | H (16) |
| *Phyllodactylus angelensis* ** | 6 | 7 | 3 | H (16) |
| *Phyllodactylus apricus* ** | 7 | 7 | 3 | H (17) |
| *Phyllodactylus benedetii* * | 5 | 7 | 3 | H (15) |
| *Phyllodactylus cleofasensis* ** | 6 | 7 | 3 | H (16) |
| *Phyllodactylus coronatus* ** | 6 | 7 | 3 | H (16) |
| *Phyllodactylus isabelae* | 6 | 7 | 3 | H (16) |
| *Phyllodactylus nocticolus* | 2 | 5 | 3 | M (10) |
| *Phyllodactylus tuberculosis* | 1 | 4 | 3 | L (8) |
| *Phyllodactylus santacruzensis* ** | 1 | 4 | 3 | L (8) |
| *Sphaerodactylus continentalis* | 4 | 3 | 3 | M (10) |
| *Holcosus gaigeae* | 5 | 7 | 3 | H (15) |
| *Boa imperator* | 3 | 1 | 6 | M (10) |
| *Boa sigma* | 5 | 4 | 6 | H (15) |
| *Lampropeltis abnorma* | 1 | 3 | 5 | L (9) |
| *Lampropeltis californiae* | 3 | 4 | 3 | M (10) |
| *Lampropeltis polyzona* | 5 | 1 | 5 | M (11) |
| *Masticophis fuliginosus* | 2 | 3 | 4 | L (9) |
| *Oxybelis aeneus* | 1 | 1 | 3 | L (5) |
| *Oxybelis fulgidus* | 3 | 2 | 4 | L (9) |
| *Oxybelis microphtalmus* * | 5 | 2 | 4 | M (11) |
| *Spilotes pullatus* | 1 | 1 | 4 | L (6) |
| *Trimorphodon lyrophanes* | 4 | 2 | 4 | M (10) |
| *Hypsiglena catalinae* | 6 | 8 | 2 | H (16) |
| *Hypsiglena chlorophaea* | 1 | 5 | 2 | L (8) |
| *Hypsiglena ochrorhyncus* | 2 | 4 | 2 | L (8) |
| *Hypsiglena unaocularis* | 6 | 8 | 2 | H (16) |
| *Imantodes cenchoa* | 1 | 3 | 2 | L (6) |
| *Sibon nebulatus* | 1 | 2 | 2 | L (5) |
| *Micrurus apiatus* | 2 | 1 | 5 | L (8) |
| *Epictia bakewelli* | 5 | 6 | 1 | M (12) |
| *Epictia magnamaculata* | 3 | 7 | 1 | M (11) |
| *Rena humilis* | 4 | 3 | 2 | L (8) |
| *Agkistrodon russeolus* * | 4 | 6 | 5 | H (15) |
| *Crotalus polisi* | 6 | 8 | 5 | H (18) |
| *Crotalus Pyrrhus* | 4 | 3 | 5 | M (12) |
| *Crotalus thalassoporus* | 5 | 8 | 5 | H (18) |
| *Crotalus tortuguensis* | 6 | 7 | 5 | H (18) |
| *Trachemys venusta* | 3 | 4 | 6 | H (16) |
| *Gopherus morafkai* | 4 | 6 | 6 | H (15) |

**Table 19.** Environmental Vulnerability Scores (EVS) for members of the herpetofauna of the Mexican insular systems assigned to the IUCN Least Concern category. Non-native and marine taxa are not included. No asterisk = Non-endemic; * = country endemic and ** = insular endemic.

| Taxa | Environmental Vulnerability Score (EVS) | | | |
|---|---|---|---|---|
| | Geographic Distribution | Ecological Distribution | Reproductive Mode/ Degree of Persecution | Total Score |
| *Aneides lugubris* | 3 | 7 | 4 | H (14) |
| *Batrachoseps major* | 4 | 6 | 4 | H (14) |
| *Anaxyrus punctatus* | 1 | 3 | 1 | L (5) |
| *Incilius mazatlanensis* | 5 | 6 | 1 | M (12) |
| *Incilius valliceps* | 3 | 2 | 1 | L (6) |
| *Rhinella horribilis* | 1 | 1 | 1 | L (3) |
| *Dendropsophus microcephalus* | 3 | 3 | 1 | L (7) |
| *Hyliola regilla* | 1 | 1 | 1 | L (3) |
| *Scinax staufferi* | 2 | 1 | 1 | L (4) |
| *Smilisca baudinii* | 1 | 1 | 1 | L (3) |
| *Trachycephalus typhonius* | 1 | 2 | 1 | L (4) |
| *Leptodactylus fragilis* | 1 | 2 | 2 | L (5) |
| *Hypopachus variolosus* | 2 | 1 | 1 | L (4) |
| *Scaphiopus couchii* | 1 | 1 | 1 | L (3) |
| *Crocodylus moreletii* | 2 | 5 | 6 | M (13) |
| *Elgaria multicarinata* | 3 | 4 | 3 | M (10) |
| *Elgaria nana* | 5 | 8 | 3 | H (16) |
| *Anniella pulchra* | 3 | 8 | 1 | M (12) |
| *Bipes biporus* | 5 | 8 | 1 | H (14) |
| *Basiliscus vittatus* | 1 | 3 | 3 | L (7) |
| *Crotaphytus dickersonae* | 5 | 8 | 3 | H (16) |
| *Crotaphytus insularis* ** | 6 | 7 | 3 | H (16) |
| *Gambelia copeii* | 2 | 6 | 3 | M (11) |
| *Gambelia wislizenii* | 3 | 7 | 3 | M (13) |
| *Norops nebulosus* | 5 | 5 | 3 | M (13) |
| *Coleonyx elegans* | 2 | 3 | 4 | L (9) |
| *Coleonyx gypsicolus* | 4 | 6 | 8 | H (18) |
| *Coleonyx variegatus* | 4 | 3 | 4 | M (11) |
| *Ctenosaura similis* | 1 | 4 | 3 | L (8) |
| *Dipsosaurus dorsalis* | 4 | 4 | 3 | M (11) |
| *Iguana iguana* | 3 | 3 | 6 | M (12) |
| *Sauromalus ater* | 4 | 6 | 3 | M (13) |
| *Marisora lineola* | 4 | 3 | 3 | M (10) |
| *Callisaurus draconoides* | 4 | 5 | 3 | M (12) |
| *Petrosaurus mearnsi* | 4 | 5 | 3 | M (12) |
| *Petrosaurus repens* | 5 | 5 | 3 | M (13) |
| *Petrosaurus slevini* ** | 5 | 8 | 3 | H (16) |
| *Petrosaurus thalassinus* | 5 | 5 | 3 | M (13) |
| *Phrynosoma solare* | 4 | 7 | 3 | H (14) |
| *Sceloporus angustus* | 5 | 8 | 3 | H (16) |
| *Sceloporus chrysostictus* | 4 | 6 | 3 | M (13) |
| *Sceloporus clarkia* | 2 | 5 | 3 | M (10) |
| *Sceloporus cozumelae* | 5 | 7 | 3 | H (15) |

| | | | | |
|---|---|---|---|---|
| *Sceloporus grandaevus* | 6 | 7 | 3 | H (16) |
| *Sceloporus hunsakeri* | 5 | 6 | 3 | H (14) |
| *Sceloporus lineatulus* | 6 | 8 | 3 | H (17) |
| *Sceloporus magister* | 1 | 5 | 3 | L (9) |
| *Sceloporus occidentalis* | 3 | 6 | 3 | M (12) |
| *Sceloporus orcutti* | 2 | 2 | 3 | L (7) |
| *Sceloporus variabilis* | 1 | 1 | 3 | L (5) |
| *Sceloporus zosteromus* | 5 | 4 | 3 | M (12) |
| *Urosaurus bicarinatus* | 5 | 4 | 3 | M (12) |
| *Urosaurus nigricaudus* | 3 | 2 | 3 | L (8) |
| *Urosaurus ornatus* | 2 | 5 | 3 | M (10) |
| *Uta nolascensis* | 6 | 8 | 3 | H (17) |
| *Uta squamata* | 6 | 8 | 3 | H (17) |
| *Uta stansburiana* | 3 | 1 | 3 | L (7) |
| *Phyllodactylus bugastrolepis* | 6 | 8 | 3 | H (17) |
| *Phyllodactylus homolepidurus* | 5 | 7 | 3 | H (15) |
| *Phyllodactylus lanei* | 5 | 7 | 3 | H (15) |
| *Phyllodactylus lupitae* | 6 | 7 | 3 | H (16) |
| *Phyllodactylus partidus* | 5 | 8 | 3 | H (16) |
| *Phyllodactylus unctus* | 5 | 7 | 3 | H (15) |
| *Mesoscincus schwartzei* | 2 | 6 | 3 | M (11) |
| *Plestiodon skiltonianus* | 3 | 5 | 3 | M (11) |
| *Aristelliger georgeensis* | 3 | 7 | 3 | M (13) |
| *Sphaerodactylus glaucus* | 4 | 5 | 3 | M (12) |
| *Aspidoscelis bacata* | 6 | 8 | 3 | H (17) |
| *Aspidoscelis cana* | 5 | 8 | 3 | H (16) |
| *Aspidoscelis carmenensis* | 6 | 8 | 3 | H (17) |
| *Aspidoscelis celeripes* | 5 | 7 | 3 | H (15) |
| *Aspidoscelis ceralbensis* | 6 | 8 | 3 | H (17) |
| *Aspidoscelis communis* | 5 | 6 | 3 | H (14) |
| *Aspidoscelis costata* | 5 | 3 | 3 | M (11) |
| *Aspidoscelis cozumela* | 5 | 8 | 3 | H (16) |
| *Aspidoscelis danheimae* | 6 | 7 | 3 | H (16) |
| *Aspidoscelis deppii* | 1 | 4 | 3 | L (8) |
| *Aspidoscelis espiritensis* | 5 | 8 | 3 | H (16) |
| *Aspidoscelis franciscensis* | 6 | 8 | 3 | H (17) |
| *Aspidoscelis guttatus* | 5 | 4 | 3 | M (12) |
| *Aspidoscelis hyperythrus* | 2 | 5 | 3 | M (10) |
| *Aspidoscelis lineattissima* | 5 | 6 | 3 | H (14) |
| *Aspidoscelis maslini* | 4 | 8 | 3 | H (15) |
| *Aspidoscelis pictus* | 6 | 8 | 3 | H (17) |
| *Aspidoscelis tigris* | 3 | 2 | 3 | L (8) |
| *Lichanura trivirgata* | 4 | 3 | 3 | M (10) |
| *Bogertophis rosaliae* | 2 | 4 | 3 | M (10) |
| *Drymarchon melanurus* | 1 | 1 | 4 | L (6) |
| *Drymobius margaritiferus* | 1 | 1 | 4 | L (6) |
| *Leptophis diplotropis* | 5 | 5 | 4 | H (14) |
| *Leptophis mexicanus* | 1 | 1 | 4 | L (6) |
| *Masticophis bilineatus* | 2 | 5 | 4 | M (11) |

| | | | | |
|---|---|---|---|---|
| *Masticophis mentovarius* | 1 | 1 | 4 | L (6) |
| *Masticophis slevini* | 6 | 8 | 3 | H (17) |
| *Mastigodryas melanolomus* | 1 | 1 | 4 | L (6) |
| *Phyllorhynchus decurtatus* | 4 | 5 | 2 | M (11) |
| *Pituophis catenifer* | 4 | 1 | 4 | L (9) |
| *Pituophis insulanus* | 6 | 6 | 4 | H (16) |
| *Pituophis vertebralis* | 5 | 3 | 4 | M (12) |
| *Pseudelaphe flavirufa* | 2 | 4 | 4 | M (10) |
| *Salvadora hexalepis* | 4 | 2 | 4 | M (10) |
| *Sonora savage* | 6 | 7 | 2 | H (15) |
| *Sonora semiannulata* | 1 | 1 | 3 | L (5) |
| *Sonora stramineus* | 4 | 2 | 2 | L (8) |
| *Tantilla bocourti* | 5 | 2 | 2 | L (9) |
| *Tantilla calamarina* | 5 | 5 | 2 | M (12) |
| *Tantilla moesta* | 4 | 7 | 2 | M (13) |
| *Tantilla planiceps* | 4 | 3 | 2 | L (9) |
| *Coniophanes imperialis* | 2 | 3 | 3 | L (8) |
| *Conophis lineatus* | 2 | 3 | 4 | L (9) |
| *Conophis vittatus* | 2 | 5 | 4 | M (11) |
| *Diadophis punctatus* | 1 | 1 | 2 | L (4) |
| *Dipsas brevifacies* | 4 | 7 | 4 | H (15) |
| *Geophis annuliferus* * | 4 | 7 | 2 | M (13) |
| *Hypsiglena slevini* | 5 | 4 | 2 | M (11) |
| *Hypsiglena torquate* | 5 | 1 | 2 | L (8) |
| *Imantodes gemmistratus* | 1 | 3 | 2 | L (6) |
| *Leptodeira frenata* | 4 | 4 | 4 | M (12) |
| *Ninia sebae* | 1 | 1 | 2 | L (5) |
| *Rhadinaea Hesperia* | 5 | 3 | 2 | M (10) |
| *Tropidodipsas sartorii* | 2 | 2 | 5 | L (9) |
| *Micruroides euryxanthus* | 4 | 6 | 5 | H (15) |
| *Micrurus apiatus* | 2 | 1 | 5 | L (8) |
| *Thamnophis Proximus* | 1 | 2 | 4 | L (7) |
| *Crotalus angelensis* | 6 | 7 | 5 | H (18) |
| *Crotalus atrox* | 1 | 3 | 5 | L (9) |
| *Crotalus caliginis* | 6 | 7 | 5 | H (18) |
| *Crotalus cerastes* | 4 | 7 | 5 | H (16) |
| *Crotalus enyo* | 5 | 3 | 5 | M (13) |
| *Crotalus estebanensis* | 6 | 8 | 5 | H (19) |
| *Crotalus lorenzoensis* | 6 | 8 | 5 | H (19) |
| *Crotalus mitchellii* | 4 | 3 | 5 | M (12) |
| *Crotalus molossus* | 2 | 1 | 5 | L (8) |
| *Crotalus ruber* | 2 | 2 | 5 | L (9) |
| *Crotalus tigris* | 4 | 7 | 5 | H (16) |
| *Kinosternon integrum* | 5 | 3 | 3 | M (11) |

*3.8. Relative Herpetofaunal Priority*

The Relative Herpetofauna Priority (RHP) measure was developed with the intention to estimate the relative importance of the herpetofauna documented in a geographic area of interest for conservation, like a political entity or a physiographic region [180]. To assign conservation priority among the recognized geographical entities (in this case, the five

physiographic insular regions), the RHP ranks them in two ways: one as the absolute number of regional and country endemics found in the region (Table 20), and two, with higher rankings to regions with proportionally higher absolute numbers of high EVS category species (Table 21).

**Table 20.** Numbers of herpetofaunal species of four distributional categories among the five physiographic provinces of the insular systems of Mexico. Rank is determined by adding regional and country endemics. Note: Although *Ctenosaura pectinata* is native to Mexico, it is non-native for Clarion, so this iguana is counted as non-native for the Tropical islands of the Pacific. The same situation is true for *B. imperator*, since it is invasive in Cozumel; thus in the islands of the Mexican Caribbean it is listed as non-native. The total number of species inhabiting insular systems in Mexico is 226. Since some species are present in two regions, this number is not the sum of the column totals, nor do the total species file.

| Physiographic Provinces | Non-Endemics | Country Endemics | Regional Endemics | Non-Natives | Totals | Rank Order |
|---|---|---|---|---|---|---|
| Pacific of Baja California | 27 | 9 | 4 | 0 | 40 | 3 |
| Gulf of California | 49 | 15 | 44 | 0 | 108 | 1 |
| Tropical Pacific | 24 | 21 | 8 | 4 | 57 | 2 |
| Gulf of Mexico | 39 | 3 | 0 | 4 | 46 | 5 |
| Mexican Caribbean | 41 | 5 | 0 | 7 | 53 | 4 |
| Total Species | 180 | 53 | 56 | 15 | 226 | — |

**Table 21.** Number of herpetofaunal species in the three EVS categories among the five physiographic regions of the insular systems of Mexico. Rank is determined by the relative number of high EVS species. Marine and non-native species are not included.

| Physiographic Provinces | Low | Medium | High | Totals | Rank Order |
|---|---|---|---|---|---|
| Pacific of Baja California | 12 | 19 | 9 | 40 | 3 |
| Gulf of California | 19 | 25 | 58 | 102 | 1 |
| Tropical Pacific | 12 | 20 | 16 | 48 | 2 |
| Gulf of Mexico | 22 | 9 | 7 | 38 | 5 |
| Mexican Caribbean | 21 | 13 | 9 | 43 | 4 |
| Total Species | 86 | 86 | 99 | — | — |

To assign the RHP to the five physiographic regions of the Insular Systems of Mexico, we constructed two tables: Table 20, with the absolute endemicity values, and Table 21, with the high vulnerability scores (14–20). The data in Table 20 indicate that the islands of the Gulf of California constitute the physiographic region of greatest conservation significance, with 59 endemic species (country and regional) of a total of 108 species (54.6%). The second level is occupied by the Tropical Pacific region, with 29 endemic species of a total of 57 species (50.9%). The third region is the Pacific Baja California islands, with 13 endemics of a total of 40 species (32.5%). Thus, the three most conservation-significant regions are those on the Pacific side of the country. This area of the country is also the only one in which insular endemics (Like the Clarion Island Whipsnake, Figure 41) are found. The fourth and fifth areas of priority significance are located on the Atlantic side of Mexico. The endemic species found there are wholly country endemics and comprise three species (6.5%) of a total of 46 species in the Gulf of Mexico region and five species (7.5%) of a total of 53 species in the Mexican Caribbean region.

Based on the number of high-EVS vulnerability species per physiographic region, the islands of the Gulf of California support the largest number of such species by far (Table 21), with 58, which is higher than that for the remaining four regions put together (41). Next highest is the Tropical Pacific islands group, with 16 species. The remaining three regions differ from one another by double species numbers, i.e., nine in the Pacific islands

of Baja California and the islands of the Mexican Caribbean, and seven in the islands of the Gulf of Mexico.

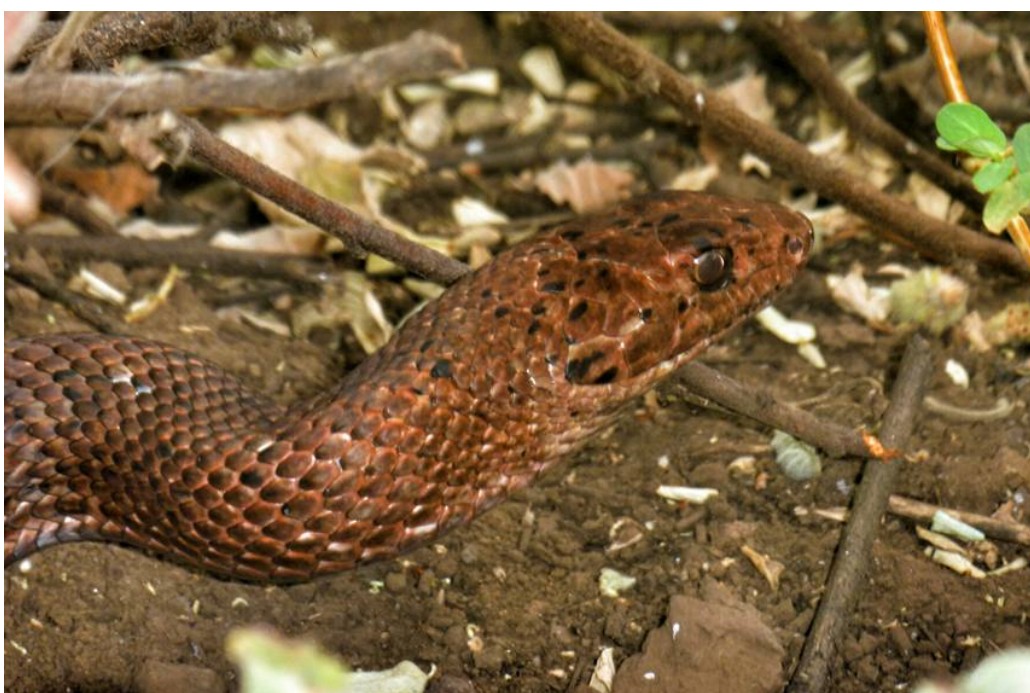

**Figure 41.** Mexican insular endemic herpetofauna species are found solely on Pacific Islands, such as the Clarion Island Whipsnake (*Masticophis anthonyi*). This particular snake species is exclusive to Clarion Island, which is part of the Revillagigedo Archipelago. It is one of several Mexican snakes that are uniquely confined to individual islands, making them highly vulnerable to extinction in the event of invasive mammal colonization on these islands.. EVS = H (17), IUCN = CR, NOM-059 = A. Photo by Humberto Almanza.

Comparison of the data in Tables 20 and 21 indicates that the ranking of the five physiographic regions is the same in each measure, as follows: first rank, Gulf of California islands; second rank, Tropical Pacific islands; third rank, Pacific islands of Baja California; fourth rank, Mexican Caribbean islands; and fifth rank, Gulf of Mexico islands.

Based on the results of the RHP analysis, it is obvious that the region of greatest conservation significance, by far, is that of the islands of the Gulf of California. This region is characterized both by the greatest number of country and insular endemic species and the most sizable number of high vulnerability species. The 52 endemics consist of all squamates, with 15 country endemics, and 39 insular endemics (like *C. tortuguensis*, Figure 42). These species are indicated by a single or double asterisk in Table 6. The Gulf of California island physiographic region also harbors 58 high vulnerability species, all of which are squamates (41 lizards and 16 snakes), except for one turtle. These 58 species and their respective EVS values are as follows (country endemic = *, insular endemic = **):

| | | | |
|---|---|---|---|
| *Crotaphytus dickersonae* ** | (16) | *Phyllodactylus unctus* * | (15) |
| *Crotaphytus insularis* ** | (16) | *Aspidoscelis bacata* ** | (17) |
| *Coleonyx gypsicolus* ** | (18) | *Aspidoscelis cana* ** | (16) |
| *Ctenosaura conspicuosa* ** | (16) | *Aspidoscelis carmenensis* ** | (17) |
| *Ctenosaura hemilopha* * | (18) | *Aspidoscelis catalinensis* ** | (17) |
| *Ctenosaura nolascensis* ** | (17) | *Aspidoscelis celeripes* ** | (15) |
| *Dipsosaurus catalinensis* ** | (17) | *Aspidoscelis ceralbelsis* ** | (17) |
| *Sauromalus hispidus* * | (14) | *Aspidoscelis danheimae* ** | (16) |

| | | | |
|---|---|---|---|
| *Sauromalus klauberi* ** | (16) | *Aspidoscelis espiritensis* ** | (16) |
| *Sauromalus slevini* * | (16) | *Aspidoscelis franciscensis* ** | (17) |
| *Sauromalus varius* * | (16) | *Aspidoscelis martyris* ** | (17) |
| *Petrosaurus slevini* ** | (16) | *Aspidoscelis pictus* ** | (17) |
| *Phrynosoma solare* | (14) | *Lampropeltis catalinensis* ** | (17) |
| *Sceloporus angustus* ** | (16) | *Masticophis barbourin* | (17) |
| *Sceloporus grandaevus* ** | (16) | *Masticophis slevini* ** | (17) |
| *Sceloporus hunsakeri* * | (14) | *Rhinocheilus etheridgei* ** | (16) |
| *Sceloporus lineatulus* ** | (17) | *Sonora 98avage* * | (15) |
| *Uta encantadae* ** | (17) | *Hypsiglena catalinae* ** | (16) |
| *Uta lowei* ** | (17) | *Micruroides euryxanthus* | (15) |
| *Uta nolascensis* ** | (17) | *Crotalus angelensis* ** | (18) |
| *Uta palmeri* ** | (17) | *Crotalus catalinensis* | (19) |
| *Uta squamata* | (17) | *Crotalus cerastes* | (16) |
| *Uta tumidarostra* ** | (17) | *Crotalus estebanensis* ** | (19) |
| *Phyllodactylus angelensis* ** | (16) | *Crotalus lorenzoensis* ** | (19) |
| *Phyllodactylus apricus* ** | (17) | *Crotalus polisi* ** | (18) |
| *Phyllodactylus bugastrolepis* ** | (17) | *Crotalus thalassoporus* ** | (18) |
| *Phyllodactylus coronatus* ** | (16) | *Crotalus tigris* | (16) |
| *Phyllodactylus homolepidurus* * | (15) | *Crotalus tortuguensis* ** | (18) |
| *Phyllodactylus partidus* ** | (16) | *Gopherus morafkai* | (15) |

Of these 58 species, eight (13.8%) are country endemics, forty-four (75.8%) are insular endemics, and six (10.3%) are non-endemics; their EVS values range from 14 to 19.

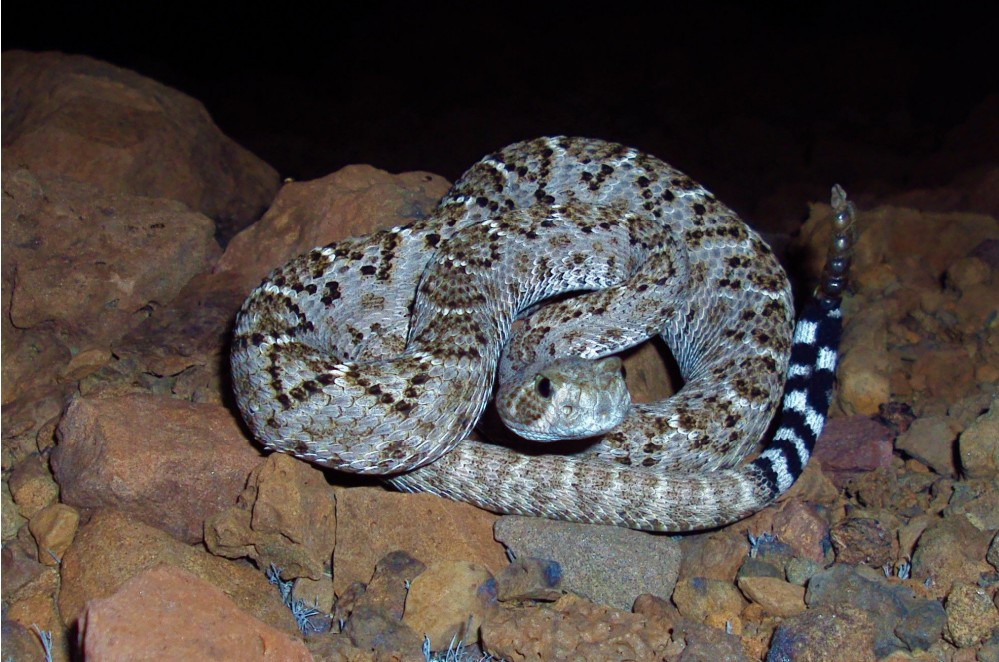

**Figure 42.** Tortuga Island Rattlesnake (*Crotalus tortuguensis*), endemic of Tortuga. Rattlesnakes constitute 16% of the insular herpetofaunal species with high EVS values. Status = IE, EVS = H (18), IUCN = NE, NOM-059 = Pr. Photo by Ruben A. Carbajal-Márquez.

The next most significant islands are the Tropical Pacific islands. The 29 endemic species comprise two anurans, twenty-six squamates, and one turtle, including 21 country endemics (Like *U. clarionensis* Figure 43) and eight insular endemics; these species are designated by a single or double asterisk in Table 7. In addition, the Tropical Pacific islands physiographic region supports 16 high vulnerability species. These species, and their respective EVS values, are as follows (country endemic = *, insular endemic = **):

| | | | |
|---|---|---|---|
| *Eleutherodactylus pallidus* * | (17) | *Phyllodactylus lanei* * | (15) |
| *Crocodylus acutus* | (14) | *Phyllodactylus lupitae* ** | (16) |
| *Ctenosaura pectinata* | (15) | *Phyllodactylus tuberculosus* | (17) |
| *Urosaurus auriculatus* ** | (16) | *Aspidoscelis communis* * | (14) |
| *Urosaurus clarionensis* ** | (17) | *Aspidoscelis lineatissima* * | (14) |
| *Phyllodactylus benedetii* * | (15) | *Leptophis diplotropis* * | (14) |
| *Phyllodactylus cleofasensis* ** | (16) | *Masticophis anthonyi* ** | (17) |
| *Phyllodactylus isabelae* ** | (16) | *Hypsiglena unaocularis* ** | (16) |

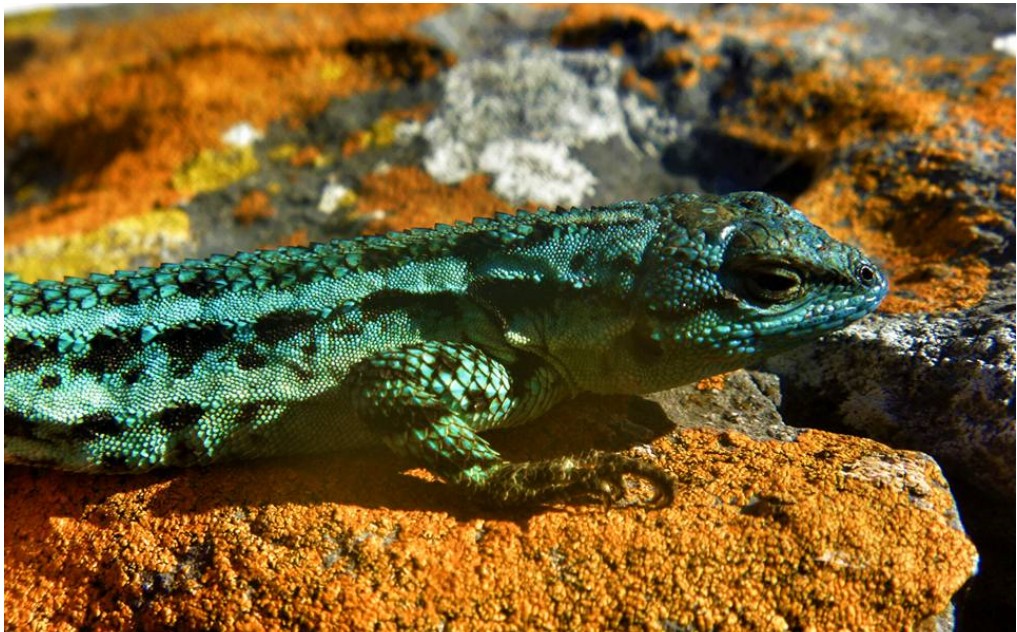

**Figure 43.** Urosaurus clarionensis, Clarion Island Tree Lizad, endemic of Isla Clarion, is easy to spot perched on rocks. As occurs with many insular species, this lizard shows naïve behavior, and it is easier to catch compared with continental *Urosaurus* (VHGS, pers. observ.). Thus, the introduction of an invasive mammal could be catastrophic for this lizard. Distributional Status = IE, EVS = H (17), IUCN = VU, NOM-059 = NS.

Of these 16 species, six are country endemics, eight are insular endemics, and two are non-endemics; their EVS values range from 14 to 17.

The third most significant region is the Pacific islands of Baja California. Thirteen endemic species inhabit these islands, all of which are squamates, and include nine country endemics and four insular endemics (Like *E. nana* Figure 44); these species are indicated by a single or double asterisk in Table 5. Additionally, the Pacific islands of Baja California harbors nine high vulnerability species; these nine species and their respective EVS values are as follows (country endemic = *, insular endemic = **):

*Aneides lugubris* (14)

*Batrachoseps major* (14)

*Elgaria cedrosensis* \* (16)

*Elgaria nana* \*\* (16)

*Bipes biporus* \* (14)

*Phrynosoma cerroense* \* (16)

*Lampropeltis herrerae* \*\* (17)

*Pituophis insulanus* \*\* (16)

*Crotalus calignis* \*\* (18)

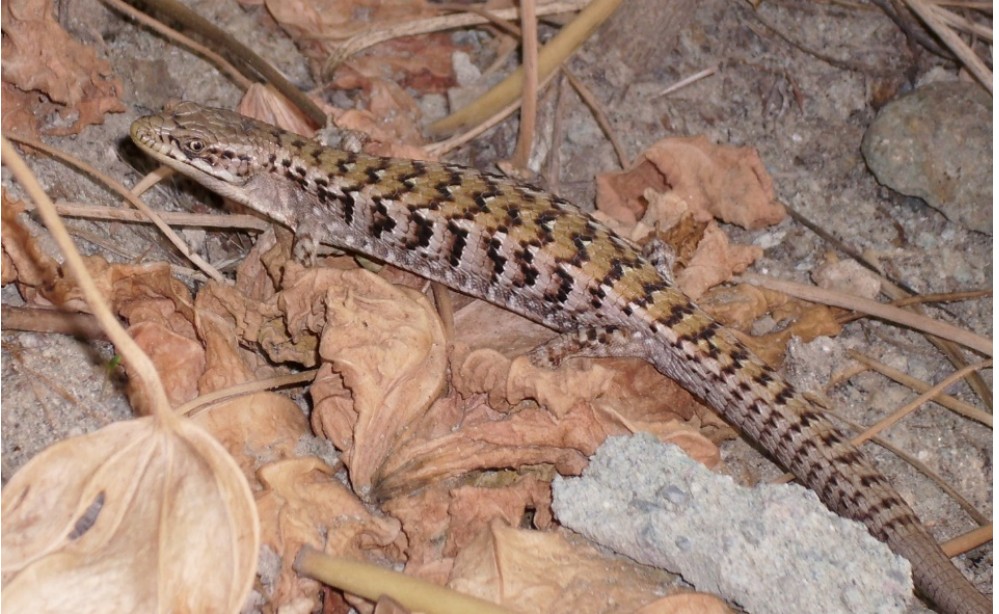

**Figure 44.** Los Coronados Alligator Lizard (*Elgaria nana*) from Coronado Norte (Archipielago Coronado). Distributional Status = IE, EVS = H (16), IUCN = LC, NOM-059 = Pr.

Of these nine species, three are country endemics, four are insular endemics, and two are non-endemics.

The fourth most significant region comprises the Mexican Caribbean islands; these islands harbor six endemic species, all of which are squamates and country endemics. The Mexican Caribbean islands also support eight high vulnerability species (nine with *Boa imperator*, but it is introduced in Cozumel and probably Chinchorro as well); these eight species and their respective EVS values are as follows (country endemic = \*, insular endemic = \*\*):

| | | | |
|---|---|---|---|
| *Crocodylus acutus* | (14) | *Aspidoscelis rodecki* \* | (16) |
| *Sceloporus cozumelae* | (15) | *Holcosus gaigeae* \* | (15) |
| *Aspidoscelis cozumela* \* | (16) | *Agkistrodon russeolus* \* | (15) |
| *Aspidoscelis maslini* | (15) | *Trachemys venusta* | (19) |

Of these eight species, five are country endemics and three are non-endemics. Their EVS values vary from 14 to 19.

Finally, the least significant physiographic area is the Gulf of Mexico islands. Only three endemic species occur on these islands. In addition, these islands support seven high vulnerability species; these five species and their respective EVS values are as follows (country endemic = \*, insular endemic = \*\*):

| | | | |
|---|---|---|---|
| *Aspidoscelis cozumela* \* | (16) | *Dipsas brevifacies* | (15) |
| *Aspidoscelis maslini* | (15) | *Dermatemys mawii* | (17) |

| | | | |
|---|---|---|---|
| *Holcosus gaigeae* * | (15) | *Staurotypus triporcatus* | (14) |
| *Boa imperator* * | (15) | | |

Of these five species, two teiids, and one *Boa imperator*, all are country endemics. None are insular endemic and the other four are non-endemics. Their EVS values range from 14 to 17.

A perusal of the lists of high vulnerability species above demonstrates that insular endemic species of this status (56) are located solely on the Pacific side of Mexico, with the highest number by far found on the islands of the Gulf of California (44) and significantly fewer on the Tropical Pacific islands (8) and the Pacific islands of Baja California (4). These three regions also contain a preponderance of country endemic species with high EVS values, with eight, six, and three species, respectively. On the Atlantic side of Mexico, where insular endemics do not figure into the tally, non-endemic species of high EVS values predominate, with two in the Mexican Caribbean islands and four in the Gulf of Mexico, as compared to country endemics six and three, respectively. Clearly, the islands on the Pacific side of Mexico provide the most significant conservation challenges.

### 3.9. Protected Areas

We identified 28 protected areas that encompass both portions of the coastal mainland as well as insular elements (Table 22). Two areas are of state jurisdiction. One of these is the Santuario del Manatí, which is a protected area in Quintana Roo, and comprises a coastal lagoon, where the island Tamalcab, along with other cays and islands of smaller sizes are found (such Cayo Venado, Cayo Violines, Dos Hermanos, Siete Mogotes, and several smaller unnamed islets). Independently of the jurisdiction of the protected area, Tamalcab Island is administrated by the Mexican federation [21]. The other of these state areas is the Reserva Estatal Selvas y Humedales de Cozumel. On this island several protected areas overlap, which is explained by González-Sánchez et al. [57].

**Table 22.** Characteristics of 28 Natural Protected Areas involving the Mexican insular systems. Abbreviations in Facilities available column as follows: A = administrative services; R = park guards; S = systems of pathways; and V = facilities for visitors. Other columns WHS = World Heritage Site, ND = No data available, NA = None applicable. Note: We list the WHS as informative, since they are not Natural Protected Areas under the Mexican legislation, and do not count it as a Natural Protected Area. Also, the "Componente del Complejo Insular Espíritu Santo" belongs to Area de Protección de Flora y Fauna Islas del Golfo de California. We list the component but do not count it as an NPA.

| Name | Category | Date of Decree | Area (ha) | Terrestrial and Inland Waters Area (ha) | Municipalities | Jurisdiction | Physiographic Regions | Facilities Available | Occupied by Landowners | Herpetofaunal Survey Completed | Management Plan Available |
|---|---|---|---|---|---|---|---|---|---|---|---|
| Laguna Madre y Delta del Río Bravo | Área de Protección de Flora y Fauna | 14 April 2005 | 572,808.60 | 572,808.60 | Matamoros, San Fernando, Soto la Marina | Federal | Islands of the Gulf of Mexico | A, R, S, V | Yes | No | Yes |
| Sistema Arrecifal Veracruzano | Parque Nacional | 24 August 1992 | 65,516.47 | 12.24 | NA | Federal | Islands of the Gulf of Mexico | A, R, V | No | No | Yes |
| Sistema Arrecifal Lobos—Tuxpan | Area de Protección de Flora y Fauna | 5 June 2009 | 30,571.15 | ND | NA | Federal | Islands of the Gulf of Mexico | A, R, V | No | No | Yes |
| Laguna de Términos | Area de Protección de Flora y Fauna | 6 June 1994 | 706,147.67 | 547,279 | Carmen, Champoton, Palizada | Federal | Islands of the Gulf of Mexico | A, R, S, V | Yes | No | Yes |
| Arrefice Alacranes | Parque Nacional | 6 June 1994 | 333,768.50 | 53 | Progreso | Federal | Islands of the Gulf of Mexico | A, R, S, V | No | No | Yes |
| Yum Balam | Área de Protección de Flora y Fauna | 6 June 1994 | 154,052.25 | 52,308 | Isla Mujeres, Lazaro Cardenas | Federal | Mexican Caribbean | A, R, S, V | Yes | No | No |

| | | | | | | | | | | | |
|---|---|---|---|---|---|---|---|---|---|---|---|
| Isla Contoy | Parque Nacional | 2 February 1998 | 5126.25 | 230 | Isla Mujeres | Federal | Mexican Caribbean | A, R, S, V | No | No | Yes |
| Costa Occidental de Isla Mujeres, Punta Cancún y Punta Nizuc | Parque Nacional | 19 July 1996 | 8673.06 | 0.6 | Benito Juarez, Isla Mujeres | Federal | Mexican Caribbean | A, R, S, V | NA. | NA. | Yes |
| La porción norte y la franja costera oriental, terrestres y marinas de la Isla de Cozumel | Área de Protección de Flora y Fauna | 25 September 2012 | 37,829.17 | 5733 | Cozumel | Federal | Mexican Caribbean | A, R, S, V | Yes | No | Yes |
| Sian Ka'an | Reserva de la Biósfera (WHS) | 20 January 1987 (1987) | 528,147.66 (Ídem) | 375,011.87 (Ídem) | Felipe Carrillo Puerto, Othon P. Blanco, Solidaridad | Federal (International) | Mexican Caribbean | A, R, S, V | Yes | Yes | Yes |
| Banco Chinchorro | Reserva de la Biósfera | 19 July 1996 | 144,360.00 | 586 | Othon P. Blanco | Federal | Mexican Caribbean | A, R, S, V | No | Yes | Yes |
| Santuario del Manatí | Zona sujeta a conservación ecológica | 8 April 2008 | 277,733.669 | 977,333.669 (180,000 of lagoon) | Othón P. Blanco | State. | Mexican Caribbean | A, S | Yes | Yes | Yes |
| Selvas y humedales de Cozumel | Reserva estatal | 1 April 2011 | 19,846.45 | 19,846.45 | Cozumel | States | Mexican Caribbean | A, S | Yes | No | Yes |
| Islas del Pacífico de la Península de Baja California | Reserva de la Biósfera | 7 December 2016 | 1,161,222.98 | 70,139.62 | NA | Federal | Pacific of Baja California Peninsula | A, S | No | Yes | No |
| El Vizcaíno (Whale Sanctuary of El Vizcaino) | Reserva de la biósfera (WHS) | 1 November 1988 (1993) | 2,549,790.25 | 2,259,002.95 (369,631) | Mulegé, BCS | Federal (International) | Pacific of Baja California Peninsula | A, R, S, V | Yes | Yes | Yes |
| Isla Guadalupe | Reserva de la Biósfera | 25 April 2005 | 476,971.20 | 26,276.97 | NA | Federal | Pacific of Baja California Peninsula | A, R, S | No | Note X | Yes |

| | | | | | | | | | | | |
|---|---|---|---|---|---|---|---|---|---|---|---|
| Islas del Golfo de California | Area de Protección de Flora y Fauna | 2 August 1978 | 374,553.63 | 374,553.63 | NA | Federal | Gulf of California | A, R | NA | No | Yes |
| Componente del Complejo Insular Espíritu Santo (Islas del Golfo de California) | APFF Islas del Golfo de California | 17 May 2007 | 48,654 | ND | NA | Federal | Gulf of California | A, R, S, V | No | Yes | Yes |
| Bahia de Loreto | Parque Nacional | 19 June 1996 | 206,508.75 | 21,692.08 | Loreto | Federal | Gulf of California | A, R, S, V | No | Yes | Yes |
| Alto Golfo de California y Delta del Río Colorado | Reserva de la Biosfera | 16 June 1993 | 934,756,25 | 407,147.55 | Mexicali, Puesto Peñasco y San Luis Río Colorado | Federal | Gulf of California | A, R, S, V | No | No | Yes |
| Isla San Pedro Mártir | Reserva de la Biosfera | 13 June 2002 | 30,165.23 | 126.98 | NA | Federal | Gulf of California | A, R, S, V | No | Yes | Yes |
| Archipiélago de espíritu santo | Parque Nacional exclusivamente de la zona marina | 17 May 2007 | 48,654–83-10.41 | ND | NA | Federal | Gulf of California | A, R, V | NA | Yes | Yes |
| Bahía de los Ángeles, canales de Ballenas y de Salsipuedes | Reserva de la Biosfera | 5 June 2007 | 387,956.88 | 483.20 | Ensenada | Federal | Gulf of California | A, R, S, V | NA | Yes | Yes |
| Islands and Protected Areas of the Gulf of California | World Heritage site | 2005 | 688,558 | NA | NA | International | Gulf of California, Tropical Islands of the Pacific | NA | NA | NA | NA |
| Archipiélago de Revillagigedo (ídem) | Parque Nacional (World Heritage Site) | 27 November 2017 | 14, 808,780.12 (636,685.375) | 15,518.22 | NA | Federal (International) | Tropical Pacific | A, R, V | NA | Yes | Yes |

| | | | | | | | | | | | |
|---|---|---|---|---|---|---|---|---|---|---|---|
| | | (6 June 1994 as Reserva de la Biósfera) (2015 as WHS) | | | | | | | | | |
| Islas Marietas | Parque Nacional | 24 April 2005 | 1383.01 | 71.16 | NA | Federal | Tropical Pacific | A, R, V | NA | Yes | Yes |
| Isla Isabel | Parque Nacional | 8 December 1980 | 194.17 | 194.17 | NA | Federal | Tropical Pacific | A, R, S, V | NA | Yes | Yes |
| Islas Marías | Reserva de la Biósfera | 27 November 2000 | 641,284.73 | 24,295.16 | NA | Federal | Tropical Pacific | A, R, S | No | No | Yes |
| Islas La Pajarera, Cocinas, Mamut, Colorada, San Pedro, San Agustin, San Andrés y Negrita y los Islotes Los Anegados, Novillas, Mosca y Submarino, situados en la Bahía de Chamela | Santuario | 13 June 2002 | 1981.43 | 1981.43 | NA | Federal | Tropical Pacific | A, R | No | No | Yes |
| Bahías de Huatulco | Parque Nacional | 24 June 1998 | 11,890.98 | 6374.98 | Santa María Huatulco | Federal | Tropical Pacific | A, R, S, V | Yes | No | Yes |

There are four world heritages sites (WHSs) with insular elements, but three of them are not concordant with the boundaries of the Mexican-protected areas, notably the Islands and Protected Areas of the Gulf of California. This WHS ranges from the Rio Colorado delta in the upper Gulf of California, through all the Sea of Cortes and far south of the Los Cabos region to the islands of Nayarit. This site includes all the protected areas we list for the Gulf of California (also those in that region we do not list due to their lack of insular elements, such as Cabo Pulmo) and the Marietas and Isabel islands, which we list here as located in the tropical Pacific region. The second site is the Whale Sanctuary of El Vizcaíno, which is considerably smaller than the Biosphere Reserve; the WHS only includes the Ojo de Liebre and San Ignacio lagoons with their respective interior islands. Thus, other islands from El Vizcaíno, such as Asunción and San Roque, and the ones in the Gulf of California, are not included in this WHS. The third WHS is the Sian Ka'an region; it includes a few islets and cays inside the Espiritu Santo and Ascension Bays (in Quintana Roo). Although Sian Ka'an is a site well-studied by herpetologists, few records exist for the insular elements in those bays. The fourth WHS is the Revillagigedo Archipelago, concordant with the homonymous biosphere reserve; this is the most recently established WHS site (2016). Banco Chinchorro is on the tentative list of proposals as a WHS site but has yet to be declared.

Of the federal administrated protected areas, nine are placed in the Biosphere Reserve category. Of these, Isla Guadalupe can be considered as the flagship ecological restoration program in Mexico. Amphibians and reptiles, however, are absent from Guadalupe and its associated islets. The islands of the Pacific region of Baja California constitute the most recently created biosphere reserve in Mexico (December 2016). The Gran Caribe Mexicano Biosphere Reserve is the largest protected area in the Mexican Caribbean, but it preserves mostly marine surface, and the polygon surrounds the extant protected areas in the region (such as Cozumel, Chinchorro, and Sian Ka'an). We do not include these as part of the Gran Caribe Mexicano, since those protected areas are still administratively independent.

Eight national parks exist that encompass insular elements. Especially important is the Revillagigedo Archipielago, also a WHS, because of its geological characteristics, its level of endemism, and its remoteness, which make this archipelago crucially important in extending the Mexican Exclusive Economic Zone. Decreed originally as a Biosphere Reserve in 1994, it was recategorized as a National Park in 2017. Importantly, however, this redesignation does not imply a downgrade in its level of protection, rather a change in what regulations apply and what land uses are allowed in the interior of the protected area. There are fewer activities, in fact, allowed inside a national park than in a biosphere reserve, which has more flexible land use regulations.

Another six sites comprise the categories of Area de Proteccion de Flora y Fauna and Sanctuary. Of importance is the APFF Islas del Golfo de California. This sole reserve includes an estimated ~900 islands and islets, with their surface area equaling 50% of the national insular territory. This area is so extensive and complex that it was necessary to create interior reserves that operate as components of the larger one; an example is the Componente del Complejo Insular Espíritu Santo (which we list). In the Parque Nacional Huatulco there are some islands in the bays, such as Cacaluta, San Agustín, and Blanca with a surface area of approximately 16, 12, and 2 has, respectively, and another eleven islands with an area of less than 1 ha. Despite the fact that the Huatulco Bays are a well-known place for tourism, we did not find any herpetofaunal records for any of these islands. Only one reserve has the Sanctuary categorization, i.e., "Islas La Pajarera, Cocinas, Mamut, Colorada, San Pedro, San Agustin, San Andrés y Negrita y los Islotes Los Anegados, Novillas, Mosca y Submarino, situated in la Bahía de Chamela" in the Chamela region of Nayarit.

The effort expended in protecting the Mexican insular systems is astonishing, since almost all of the Mexican islands have some degree of protection. The only exception we could identify is one in the Pacific coast of Baja California, i.e., Santa Margarita Island,

which was not included in the Islas del Pacífico de Baja California decree [47]. This island, however, is relatively safe since there is a Mexican Navy base located there, which imposes restrictions on accessibility and development on that island. The other examples are on the coast of Guerrero, i.e., the islands of Coral (peña), Ixtapa, and La Roqueta. Because of these efforts, of the 220 native Mexican insular amphibian and reptile species, only 3 are not protected within a Natural Protected Area. *Aspidoscelis guttata*, and *Epictia bakewelli*, from Isla Ixtapa, are both species whose insular population is not found within a Protected Area. *Aspidoscelis guttata* is a country endemic, but not exclusive to islands, since it is a common whiptail in southwestern Mexico, especially on the coastal regions of Guerrero, Oaxaca, and Chiapas.Same case for *Phyllodatylus lanei*, who inhabits islands on the coast of the southern Pacific, but it lives coastal areas of Pacific versant. Thus, we conclude that *P. lupitae* is the only insular endemic reptile not protected by any Natural Protected Area.

Unfortunately, we are not optimistic that the goal of protecting 100% of the offshore insular elements of the country will be reached in the short- or the mid-term, since Isla Ixtapa and La Roqueta are important touristic attractions at the Zihuatanejo and Acapulco ports, and any limitation on the development of tourism will face strong resistance. Also, the outgoing federal administration failed in implementing strong environmental policies, expressing only verbal support for the conservation of natural resources, but in reality enacted severe budget cuts to environmental and scientific institutions since its beginning [254–256]. In fact, before this administration started, it was very likely that two new federal reserves would be created in the Yucatan Peninsula, the APFF of Bacalar and the APFF Sayab Ha' [57], but, since then, that process has entered limbo. It was not until the second half of this presidential period that three federal protected areas were decreed [257–259], none of those were located in marine or coastal regions. Whether the current environmental policies approach will prevail in the following administration (which will begin in late 2024), is unclear and remains to be seen.

The region with the greatest number of protected areas is the Mexican Caribbean, with eight, six of federal administration and two at the state level. The Gulf of California has six, with the Componente del Complejo Insular Espíritu Santo as an integral part of the APFF Islas del Golfo de California, with its own management program. The Tropical islands of the Gulf have six protected areas, whereas the Gulf of Mexico has five and the islands of the Pacific of Baja California have only three, including Isla Guadalupe which, as mentioned before, has no herpetofaunal species. The Vizcaíno Biosphere Reserve is one of the largest of Mexico's protected areas, but its insular surface is limited to a few rocky islands of small size.

The oldest protected area is the APFF Islas del Golfo de California, which was decreed in December 1978, followed by the Islas Marías and Isla Isabel national parks, both decreed in 1980. Most protected areas for insular systems were created between the mid-1900s and 2011. The Islas del Pacífico de la Península de Baja California and the Gran Caribe Mexicano Biosphere Reserves were the last decreed in December 2016.

Herpetofaunal surveys are not complete in at least 14 of these protected areas. Significantly, the herpetofaunal listings of records are scarce or even lacking for some sites that, at a first glance, one might think are well known, such as the islands in the Bahias de Huatulco, Isla de Sacrificios (Sistema Arrecifal Veracruzano) of the islands on the Delta del Rio Colorado. Sian Ka'an is a region extensively surveyed by biologists, but this is not the case for its insular elements, whose herpetofauna is practically unknown. Only the Cuban anole (*Norops sagrei*) is known from its cays. The herpetofauna of the islands of the Gulf of California have been surveyed several times in the past and the knowledge of their herpetofauna is outstanding, but due to the complexity and number of insular elements in that region, there are still many islands that have not been surveyed as of yet. In addition, limited information exists about the herpetofauna of the islands of the Tamaulipecan coast.

In general, due to federal jurisdiction, the remoteness and inaccessibility of the islands, most of them are unpopulated, as mentioned earlier, so few important settlements

exist. This is why 13 of the protected areas with insular systems have no landowners within them. In nine cases, however, such is the case; examples are Cozumel, Isla Mujeres, Cedros, and Isla del Carmen. In most cases, the settlements consist only of settlements of fishermen, and almost always are occupied only seasonally, as occurs in the San Benito Archipelago.

Of the 28 protected areas, 26 have management plans. The APFF Islas del Golfo de California has two plans, including the specific management plan for the Espiritu Santo Component (thus, 27 management plans are available). Just two protected areas lack such plans, i.e., the APFF Yum Balam and the Islas del Pacífico de la Península de Baja California Biosphere Reserve. The latter, as mentioned previously, is of recent creation. Yum Balam is a reserve which has been in existence since 1994, and the lack of a management plan is of concern since the main island within that reserve (Holbox) is subject to pressure from tourist development.

Finally, the reserve or protected area that hosts the greatest number of species of reptiles and amphibians is the APFF Islas del Golfo de California, with 98. This single reserve protects 43.4% of the known insular species of Mexico; 43 of these 98 are insular endemics. This number represents 76.8% of the 56 insular endemics existing in the entire country. The next most important reserve in species richness is the APFF Laguna de Términos with 42 species, but it hosts only one country with endemic species, no insular endemics, 38 non-endemic species, and 3 exotic species. The Complex of Cozumel, the Islas del Pacífico de la Península de Baja California, The Bahía de los Ángeles, Islas Marías Biosphere Reserves, and the Archipielago de Revillagigedo National Park are also important, given the number of their country and/or insular endemics. The Sian Ka'an and El Vizcaíno Biosphere Reserves have low herpetofaunal species richness in their cays or islands, respectively. We wish to note that these data do not decrease in any sense the worth of these reserves; Sian Ka'an is important due to its biological richness in its mainland area and its reefs. Also, El Vizcaíno is important as a whale sanctuary and due to its desertic floristic and faunistic diversity (Tables 23 and 24).

**Table 23.** Distribution of amphibians and reptiles in the 29 Natural Protected Areas of the Mexican insular systems. Abbreviations are as follows: No asterisk = Non-endemic; * = country endemic and ** = species endemic to the Mexican insular systems. Non-native species are not included. Shaded cells only to facilitate reading.

| Taxa | Gulf of Mexico | | | | | | | | | Mexican Caribbean | | | | Pacific of BC | Tropical Pacific | | | | Gulf of California | | | | | Number of NPA in which the Species is Present |
| --- | --- | --- | --- | --- | --- | --- | --- | --- | --- | --- | --- | --- | --- | --- | --- | --- | --- | --- | --- | --- | --- | --- | --- | --- |
| | Laguna Madre y Delta del Río Bravo | Sistema Arrecifal Veracruzano | Sistema Arrecifal Lobos—Tuxpan | Laguna de Términos | Arrecife Alacranes | Yum Balam | Contoy | Sian Ka' an | Banco Chinchorro | Santuario del Manatí Bahía de Chetumal | Costa Occidental de Isla Mujeres, Punta Cancún y Punta Nizuc | Cozumel complex | Islas del Pacífico de Baja California | El Vizcaíno | Islas Marías | Isabel | Archipielago Revillagigedo | Islas Marietas | Islas La Pajarera, Cocinas, Mamut, Colorada, […]situados en la Bahía de Chamela | Bahía de Loreto | San Pedro Mártir | Bahía de los Ángeles, Canales de Ballenas y de Salsipuedes | Archipielago de Espíritu Santo | San Lorenzo | Islas del Golfo de Baja California | |
| **Amphibia (15 species)** | | | | | | | | | | | | | | | | | | | | | | | | | | |
| **Anura (13 species)** | | | | | | | | | | | | | | | | | | | | | | | | | | |
| **Bufonidae (4 species)** | | | | | | | | | | | | | | | | | | | | | | | | | | |
| *Anaxyrus punctatus* | | | | | | | | | | | | | | | | | | | | | | | + | | + | 2 |
| *Incilius mazatlanensis* * | | | | | | | | | | | | | | | + | + | | | | | | | | | | 2 |
| *Incilius valliceps* | | | | + | | + | | | | | | | + | | | | | | | | | | | | | 3 |
| *Rhinella horribilis* | | | | + | | | | | | | | | + | | | | | | | | | | | | | 2 |
| **Eleutherodactylidae (1 species)** | | | | | | | | | | | | | | | | | | | | | | | | | | |
| *Eleutherodactylus pallidus* * | | | | | | | | | | | | | | | + | | | | | | | | | | | 1 |
| **Hylidae (5 species)** | | | | | | | | | | | | | | | | | | | | | | | | | | |
| *Dendropsophus microcephalus* | | | | | | | | | | | | | + | | | | | | | | | | | | | 1 |
| *Hyliola regilla* | | | | | | | | | + | | | | | | | | | | | | | | | | | 1 |

| Species | | | | | | | | | | | | | Total |
|---|---|---|---|---|---|---|---|---|---|---|---|---|---|
| *Scinax staufferi* | | + | | | | | | + | | | | | 2 |
| *Smilisca baudinii* | | + | + | | | | | + | | + | | | 4 |
| *Trachycephalus vermiculatus* | | | + | | | | | + | | | | | 2 |
| **Leptodactylidae (1 species)** | | | | | | | | | | | | | |
| *Leptodactylus fragilis* | | + | | | | | | + | | | | | 2 |
| **Microhylidae (1 species)** | | | | | | | | | | | | | |
| *Hypopachus variolosus* | | + | | | | | | | | + | | | 2 |
| **Scaphiopodidae (1 species)** | | | | | | | | | | | | | |
| *Scaphiopus couchii* | | | | | | | | | | | + | + | 2 |
| **Caudata (2 species)** | | | | | | | | | | | | | |
| **Plethodontidae (2 especies)** | | | | | | | | | | | | | |
| *Aneides lugubris* | | | | | | | | | + | | | | 1 |
| *Batrachoseps major* | | | | | | | | | + | | | | 1 |
| **Reptiles (204 species)** | | | | | | | | | | | | | |
| **Crocodylia (2 species)** | | | | | | | | | | | | | |
| **Crocodylidae (2 species)** | | | | | | | | | | | | | |
| *Crocodylus acutus* | | | + | + | + | | + | + | | + | | | 6 |
| *Crocodylus moreletii* | + | + | ? | | | | | | | | | | 2 |
| **Squamata (189 species)** | | | | | | | | | | | | | |
| **Anguidae (3 species)** | | | | | | | | | | | | | |
| *Elgaria cedrosensis* * | | | | | | | | | + | | | | 1 |
| *Elgaria multicarinata* | | | | | | | | | + | | | | 1 |
| *Elgaria nana* ** | | | | | | | | | + | | | | 1 |
| **Anniellidae (2 species)** | | | | | | | | | | | | | |
| *Anniella geronimensis* * | | | | | | | | | + | | | | 1 |
| *Anniella pulchra* | | | | | | | | | + | | | | 1 |
| **Bipedidae (1 species)** | | | | | | | | | | | | | |
| *Bipes biporus* * | | | | | | | | | + | | | | 1 |
| **Corytophanidae (1 species)** | | | | | | | | | | | | | |
| *Basiliscus vittatus* | | + | | | | + | | + | | | | | 3 |
| **Crotaphytidae (4 species)** | | | | | | | | | | | | | |
| *Crotaphytus dickersonae* ** | | | | | | | | | | | | + | 1 |
| *Crotaphytus insularis* ** | | | | | | | | | | | + | + | 2 |
| *Gambelia copeii* | | | | | | | | | + | | | | 1 |
| *Gambelia wislizenii* | | | | | | | | | | | | + | 1 |

| | 1 | 2 | 3 | 4 | 5 | 6 | 7 | 8 | 9 | 10 | 11 | 12 | 13 | 14 | 15 | 16 | |
|---|---|---|---|---|---|---|---|---|---|---|---|---|---|---|---|---|---|
| **Dactyloidae (4 species)** | | | | | | | | | | | | | | | | | |
| *Norops lemurinus* | | | | | | + | | | | | | | | | | | 1 |
| *Norops nebulosus* * | | | | | | | | | | + | | + | + | | | | 3 |
| *Norops rodriguezii* | | + | + | | | + | | + | | | | | | | | | 4 |
| *Norops ustus* | | + | + | | | + | | | | | | | | | | | 3 |
| **Eublepharidae (3 species)** | | | | | | | | | | | | | | | | | |
| *Coleonyx elegans* | | | | | | | + | | | | | | | | | | 1 |
| *Coleonyx gypsicolus* ** | | | | | | | | | | | | | | | | + | 1 |
| *Coleonyx variegatus* | | | | | | | | | + | | + | + | + | | | + | 5 |
| **Iguanidae (13 species)** | | | | | | | | | | | | | | | | | |
| *Ctenosaura conspicuosa* ** | | | | | | | | | | | | | | | | + | 1 |
| *Ctenosaura hemilopha* * | | | | | | | | | | | | | | | | + | 1 |
| *Ctenosaura nolascensis* ** | | | | | | | | | | | | | | | | + | 1 |
| *Ctenosaura pectinate* | | | | | | | | | | + | + | + | + | | + | | 5 |
| *Ctenosaura similis* | + | + | + | + | + | + | | + | | | | | | | | | 8 |
| *Dipsosaurus catalinensis* ** | | | | | | | | | | | + | | | | | + | 2 |
| *Dipsosaurus dorsalis* | | | | | | | | | + | | + | + | + | | | + | 5 |
| *Iguana iguana* | + | + | | | + | | | + | | + | + | + | + | | | | 8 |
| *Sauromalus ater* | | | | | | | | | | | | | | | + | + | 2 |
| *Sauromalus hispidus* * | | | | | | | | | | | | + | | + | | + | 3 |
| *Sauromalus klauberi* ** | | | | | | | | | | | + | | | | | + | 2 |
| *Sauromalus slevini* * | | | | | | | | | | | + | | | | | + | 2 |
| *Sauromalus varius* * | | | | | | | | | | | | + | | | | + | 2 |
| **Mabuyidae (2 species)** | | | | | | | | | | | | | | | | | |
| *Marisora aquilonaria* * | | | | | | | | | + | | | | | | | | 1 |
| *Marisora lineola* | | + | + | + | + | | | + | + | | | | | | | | 6 |
| **Phrynosomatidae (31 species)** | | | | | | | | | | | | | | | | | |
| *Callisaurus draconoides* | | | | | | | | | + | | + | + | + | | | + | 5 |
| *Petrosaurus mearnsi* | | | | | | | | | | | | | | | | + | 1 |
| *Petrosaurus repens* * | | | | | | | | | | | + | | | | | + | 2 |
| *Petrosaurus slevini* ** | | | | | | | | | | | | + | | | | + | 2 |
| *Petrosaurus thalassinus* * | | | | | | | | | | | | | + | | | + | 2 |
| *Phrynosoma cerroense* * | | | | | | | | | + | | | | | | | | 1 |
| *Phrynosoma solare* | | | | | | | | | | | | | | | | + | 1 |
| *Sceloporus angustus* ** | | | | | | | | | | | | | | | | + | 1 |

| | | | | | | | | | | | | | | | |
|---|---|---|---|---|---|---|---|---|---|---|---|---|---|---|---|
| *Sceloporus chrysostictus* | | + | + | | + | | | | | | | | | | 3 |
| *Sceloporus clarkia* | | | | | | | | | | + | | | | + | 2 |
| *Sceloporus cozumelae* * | | | + | + | + | | | | | | | | | | 3 |
| *Sceloporus grandaevus* ** | | | | | | | | | | | | | | + | 1 |
| *Sceloporus hunsakeri* * | | | | | | | | | | | | + | | + | 2 |
| *Sceloporus lineatulus* ** | | | | | | | | | | + | | | | + | 2 |
| *Sceloporus magister* | | | | | | | | | | | | | | + | 1 |
| *Sceloporus occidentalis* | | | | | | + | | | | | | | | | 1 |
| *Sceloporus orcutti* | | | | | | | | | | + | | | | + | 2 |
| *Sceloporus variabilis* | + | | | | | | | | | | | | | | 1 |
| *Sceloporus zosteromus* * | | | | | | + | | | | + | | + | | + | 4 |
| *Urosaurus auriculatus* ** | | | | | | + | | | | | | | | | 1 |
| *Urosaurus bicarinatus* * | | | | | | | | | | + | | | | | 1 |
| *Urosaurus clarionensis* ** | | | | | | + | | | | | | | | | 1 |
| *Urosaurus nigricaudus* * | | | | | | + | | | | + | | + | + | + | 5 |
| *Urosaurus ornatus* | | | | | | | | + | | | | | | + | 2 |
| *Uta encantadae* ** | | | | | | | | | | | | | | + | 1 |
| *Uta lowei* ** | | | | | | | | | | | | | | + | 1 |
| *Uta nolascensis* ** | | | | | | | | | | | | | | + | 1 |
| *Uta palmeri* ** | | | | | | | | | | | | + | | + | 2 |
| *Uta squamata* | | | | | | | | | | + | | | | + | 2 |
| *Uta stansburiana* | | | | | | + | + | | | + | | + | + | + | 7 |
| *Uta tumidarostra* ** | | | | | | | | | | | | | | + | 1 |
| **Phyllodactylidae (15 species)** | | | | | | | | | | | | | | | |
| *Phyllodactylus angelensis* ** | | | | | | | | | | | | | | + | 1 |
| *Phyllodactylus apricus* ** | | | | | | | | | | | | | | + | 1 |
| *Phyllodactylus benedetii* * | | | | | | | | | | | | + | | | 1 |
| *Phyllodactylus bugastrolepis* ** | | | | | | | | | | + | | | | + | 2 |
| *Phyllodactylus cleofasensis* ** | | | | | | | | + | | | | | | | 1 |
| *Phyllodactylus coronatus* ** | | | | | | | | | | | | | | + | 1 |
| *Phyllodactylus homolepidurus* * | | | | | | | | | | | | | | + | 1 |
| *Phyllodactylus isabelae* ** | | | | | | | | | | + | + | | | | 2 |
| *Phyllodactylus lanei* * | | | | | | | | | | | | | | | 0 |
| *Phyllodactylus lupitae* ** | | | | | | | | | | | | | | | 0 |
| *Phyllodactylus nocticolus* | | | | | | + | | | | + | + | | + | + | 5 |

| Species | | Count |
|---|---|---|
| *Phyllodactylus partidus* ** | + + | 2 |
| *Phyllodactylus tuberculosos* | + | 1 |
| *Phyllodactylus santacruzensis* ** | + | 1 |
| *Phyllodactylus unctus* * | + + | 2 |
| **Scincidae (2 species)** | | |
| *Mesoscincus schwartzei* | + | 1 |
| *Plestiodon skiltonianus* | + | 1 |
| **Sphaerodactylidae (3 species)** | | |
| *Aristelliger georgeensis* | + + + | 3 |
| *Sphaerodactylus continentalis* | + | 1 |
| *Sphaerodactylus glaucus* | + + + | 3 |
| **Teiidae (22 species)** | | |
| *Aspidoscelis bacata* ** | + | 1 |
| *Aspidoscelis cana* ** | + + + | 3 |
| *Aspidoscelis carmenensis* ** | + + | 2 |
| *Aspidoscelis catalinensis* ** | + + | 2 |
| *Aspidoscelis celeripes* ** | + | 1 |
| *Aspidoscelis ceralbensis* ** | + | 1 |
| *Aspidoscelis communis* * | + | 1 |
| *Aspidoscelis costata* * | + | 1 |
| *Aspidoscelis cozumela* * | + | 1 |
| *Aspidoscelis danheimae* ** | + | 1 |
| *Aspidoscelis deppii* | + + | 2 |
| *Aspidoscelis espiritensis* ** | + + | 2 |
| *Aspidoscelis franciscensis* ** | + | 1 |
| *Aspidoscelis guttatus* * | | |
| *Aspidoscelis hyperythrus* | + + + | 3 |
| *Aspidoscelis lineattissima* | + + | 2 |
| *Aspidoscelis martyris* ** | + + | 2 |
| *Aspidoscelis maslini* | + + | 2 |
| *Aspidoscelis pictus* ** | + + | 2 |
| *Aspidoscelis rodecki* * | + | 1 |
| *Aspidoscelis tigris* | + + + + + + | 6 |
| *Holcosus gaigeae* * | + | 1 |
| **Boidae (2 species)** | | |

| Species | 1 | 2 | 3 | 4 | 5 | 6 | 7 | 8 | 9 | 10 | 11 | 12 | 13 | 14 | 15 | 16 | 17 | 18 | |
|---|---|---|---|---|---|---|---|---|---|---|---|---|---|---|---|---|---|---|---|
| *Boa imperator* * | + | + |  | + | + |  | ¨+ |  |  |  |  |  |  |  |  |  |  |  | 4 |
| *Boa sigma* |  |  |  |  |  |  |  |  | + |  |  | + |  |  |  |  |  |  | 2 |
| **Charinidae (1 species)** |  |  |  |  |  |  |  |  |  |  |  |  |  |  |  |  |  |  | |
| *Lichanura trivirgata* |  |  |  |  |  |  |  | + |  |  |  |  | + |  | + | + |  | + | 5 |
| **Colubridae (36 species)** |  |  |  |  |  |  |  |  |  |  |  |  |  |  |  |  |  |  | |
| *Bogertophis rosaliae* |  |  |  |  |  |  |  |  |  |  |  |  | + |  |  |  |  | + | 2 |
| *Drymarchon melanurus* |  |  |  |  |  |  |  |  | + |  |  |  |  |  |  |  |  |  | 1 |
| *Drymobius margaritiferus* |  | + |  |  |  |  |  |  |  |  |  |  |  |  |  |  |  |  | 1 |
| *Lampropeltis abnorma* |  | + |  |  |  |  |  |  |  |  |  |  |  |  |  |  |  |  | 1 |
| *Lampropeltis californiae* |  |  |  |  |  |  |  |  |  |  |  |  | + | + | + |  | + | + | 5 |
| *Lampropeltis catalinensis* ** |  |  |  |  |  |  |  |  |  |  |  |  | + |  |  |  |  | + | 2 |
| *Lampropeltis herrerae* ** |  |  |  |  |  |  |  | + |  |  |  |  |  |  |  |  |  |  | 1 |
| *Lampropeltis polyzona* * |  |  |  |  |  |  |  |  | + | + |  |  |  |  |  |  |  |  | 2 |
| *Leptophis diplotropis* * |  |  |  |  |  |  |  |  | + |  |  |  |  |  |  |  |  |  | 1 |
| *Leptophis mexicanus* |  |  | + |  |  | + |  |  |  |  |  |  |  |  |  |  |  |  | 2 |
| *Masticophis anthonyi* ** |  |  |  |  |  |  |  |  |  | + |  |  |  |  |  |  |  |  | 1 |
| *Masticophis barbouri* |  |  |  |  |  |  |  |  |  |  |  |  |  |  |  |  | + | + | 2 |
| *Masticophis bilineatus* |  |  |  |  |  |  |  |  |  |  |  |  |  |  |  |  |  | + | 1 |
| *Masticophis fuliginosus* |  |  |  |  |  |  |  | + |  |  |  |  | + |  |  |  | + | + | 4 |
| *Masticophis mentovarius* |  |  | + |  |  |  |  |  | + |  | + |  |  |  |  |  |  |  | 3 |
| *Masticophis slevini* ** |  |  |  |  |  |  |  |  |  |  |  |  |  |  |  |  |  | + | 1 |
| *Mastigodryas melanolomus* |  |  |  |  |  | + | + |  | + |  |  |  |  |  |  |  |  |  | 3 |
| *Oxybelis aeneus* |  |  |  | + |  |  |  |  |  |  |  |  |  |  |  |  |  |  | 1 |
| *Oxybelis fulgidus* |  |  | + |  |  | + | + |  |  |  |  |  |  |  |  |  |  |  | 3 |
| *Oxybelis microphtalmus* * |  |  |  |  |  |  |  |  | + |  |  |  |  |  |  |  |  |  | 1 |
| *Phyllorhynchus decurtatus* |  |  |  |  |  |  |  |  |  |  |  |  | + |  | + |  |  |  | 2 |
| *Pituophis catenifer* |  |  |  |  |  |  |  | + |  |  |  |  |  |  |  |  |  |  | 1 |
| *Pituophis insulanus* ** |  |  |  |  |  |  |  | + |  |  |  |  |  |  |  |  |  |  | 1 |
| *Pituophis vertebralis* * |  |  |  |  |  |  |  | + |  |  |  |  |  |  |  |  |  |  | 1 |
| *Pseudelaphe flavirufa* |  | + |  |  |  |  |  |  |  |  |  |  |  |  |  |  |  |  | 1 |
| *Rhinocheilus etheridgei* ** |  |  |  |  |  |  |  |  |  |  |  |  |  |  |  |  |  | + | 1 |
| *Salvadora hexalepis* |  |  |  |  |  |  |  | + |  |  |  |  |  |  |  |  | + |  | 2 |
| *Sonora savagei* * |  |  |  |  |  |  |  |  |  |  |  |  |  |  |  |  |  | + | 1 |
| *Sonora semiannulata* |  |  |  |  |  |  |  |  |  |  |  |  |  |  |  |  |  | + | 1 |
| *Sonora straminea* |  |  |  |  |  |  |  | + |  |  |  |  | + |  |  |  | + | + | 4 |

| Species | 1 | 2 | 3 | 4 | 5 | 6 | 7 | 8 | 9 | 10 | 11 | 12 | 13 | 14 | n |
|---|---|---|---|---|---|---|---|---|---|---|---|---|---|---|---|
| *Spilotes pullatus* | + | | | | | | | | | | | | | | 1 |
| *Tantilla bocourti* * | | | | | | | + | | | | | | | | 1 |
| *Tantilla calamarina* * | | | | | | | + | | | | | | | | 1 |
| *Tantilla moesta* | | | | | + | | | | | | | | | | 1 |
| *Tantilla planiceps* | | | | | | | | | | | + | | | | 1 |
| *Trimorphodon lyrophanes* | | | | | | | | | | | + | | | | 1 |
| **Dipsadidae (20 species)** | | | | | | | | | | | | | | | |
| *Coniophanes meridanus* | | + | | | | | | | | | | | | | 1 |
| *Coniophanes imperialis* | + | | | | | | | | | | | | | | 1 |
| *Conophis lineatus* | + | + | + | | | | | | | | | | | | 3 |
| *Conophis vittatus* | | | | | | | | | | + | | | | | 1 |
| *Diadophis punctatus* | | | | | | + | | | | | | | | | 1 |
| *Dipsas brevifacies* | + | | | | | | | | | | | | | | 1 |
| *Geophis annuliferus* * | | | | | | | + | | | | | | | | 1 |
| *Hypsiglena catalinae* ** | | | | | | | | | | | + | | | | 1 |
| *Hypsiglena chlorophaea* | | | | | | | | | | | | | | + | 1 |
| *Hypsiglena ochrorhyncus* | | | | | | + | | | | | + | + | + | + | 5 |
| *Hypsiglena slevini* * | | | | | | | | | | | + | | | | 1 |
| *Hypsiglena torquata* | | | | | | | + | | + | | | | | | 2 |
| *Hypsiglena unaocularis* ** | | | | | | | | + | | | | | | | 1 |
| *Imantodes cenchoa* | + | | | | | | | | | | | | | | 1 |
| *Imantodes gemmistratus* | | | | | | | + | | | | | | | | 1 |
| *Leptodeira frenata* | + | | | | + | | | | | | | | | | 2 |
| *Ninia sebae* | + | | | | | | | | | | | | | | 1 |
| *Rhadinaea hesperia* * | | | | | | | + | | | | | | | | 1 |
| *Sibon nebulatus* | + | | | | | | + | | | | | | | | 2 |
| *Tropidodipsas sartorii* | + | | | | | | | | | | | | | | 1 |
| **Elapidae (2 species)** | | | | | | | | | | | | | | | |
| *Micruroides euryxanthus* | | | | | | | | | | | | | | + | 1 |
| *Micrurus apiatus* | + | | | + | | | | | | | | | | | 2 |
| **Leptotyphlopidae (3 species)** | | | | | | | | | | | | | | | |
| *Epictia bakewelli* * | | | | | | | | | | | | | | | 0 |
| *Epictia magnamaculata* | | | | | + | | | | | | | | | | 1 |
| *Rena humilis* | | | | | | + | | | | | + | | | + | 3 |
| **Natricidae (1 species)** | | | | | | | | | | | | | | | |

| Species | 1 | 2 | 3 | 4 | 5 | 6 | 7 | 8 | 9 | 10 | 11 | 12 | 13 | 14 | 15 | 16 | 17 | Total |
|---|---|---|---|---|---|---|---|---|---|---|---|---|---|---|---|---|---|---|
| *Thamnophis proximus* | | | | | | | | | + | | | | | | | | | 1 |
| **Viperidae (18 species)** | | | | | | | | | | | | | | | | | | |
| *Agkistrodon bilineatus* | | | | | | | | | | | + | | | | | | | 1 |
| *Agkistrodon russeolus* * | | | | | + | | | | | | | | | | | | | 1 |
| *Crotalus angelensis* ** | | | | | | | | | | | | | | + | | | + | 2 |
| *Crotalus atrox* | | | | | | | | | | | | | + | | | | + | 2 |
| *Crotalus caliginis* ** | | | | | | | | | | + | | | | | | | | 1 |
| *Crotalus catalinensis* | | | | | | | | | | | | + | | | | | + | 2 |
| *Crotalus cerastes* | | | | | | | | | | | | | | | | | + | 1 |
| *Crotalus enyo* * | | | | | | | | | | + | | + | | | + | | + | 4 |
| *Crotalus estebanensis* ** | | | | | | | | | | | | | | | | | + | 1 |
| *Crotalus lorenzoensis* ** | | | | | | | | | | | | | | | | + | + | 2 |
| *Crotalus mitchellii* | | | | | | | | | | | | + | | + | + | + | + | 5 |
| *Crotalus molossus* | | | | | | | | | | | | | | | | | + | 1 |
| *Crotalus polisi* ** | | | | | | | | | | | | | | + | | | + | 2 |
| *Crotalus Pyrrhus* | | | | | | | | | | | | | | + | | | + | 2 |
| *Crotalus ruber* | | | | | | | | | | + | | | + | + | | | + | 4 |
| *Crotalus thalassoporus* ** | | | | | | | | | | | | | | + | | | + | 2 |
| *Crotalus tigris* | | | | | | | | | | | | | | | | | + | 1 |
| *Crotalus tortuguensis* ** | | | | | | | | | | | | | | | | | + | 1 |
| **Testudines (13 species)** | | | | | | | | | | | | | | | | | | |
| **Cheloniidae (5 species)** | | | | | | | | | | | | | | | | | | |
| *Caretta caretta* | + | + | + | + | + | + | + | + | + | + | | | + | | | | + | 12 |
| *Chelonia mydas* | + | + | + | + | + | + | + | + | + | + | | + | + | | + | | + | 14 |
| *Eretmochelys imbricata* | | + | | + | + | + | + | + | + | + | | + | + | + | + | | + | 13 |
| *Lepidochelys kempii* | + | + | + | + | | | | + | | | | | | | | | | 5 |
| *Lepidochelys olivácea* | | | | | | | | | | | | + | + | + | | | + | 4 |
| **Dermatemydidae (1 species)** | | | | | | | | | | | | | | | | | | |
| *Dermatemys mawii* | | | | + | | | | | | | | | | | | | | 1 |
| **Dermochelyidae (1 species)** | | | | | | | | | | | | | | | | | | |
| *Dermochelys coriacea* | | | + | | + | | + | + | + | + | | + | + | | + | | + | 10 |
| **Emydidae (1 species)** | | | | | | | | | | | | | | | | | | |
| *Trachemys venusta* | | | | | | | | | + | | | | | | | | | 1 |
| **Geoemydidae (1 species)** | | | | | | | | | | | | | | | | | | |
| *Rhinoclemmys areolata* | | | | | + | | | | + | | | | | | | | | 2 |

| | | | | | | | | | | | | | | | | | | | | | | | | | | |
|---|---|---|---|---|---|---|---|---|---|---|---|---|---|---|---|---|---|---|---|---|---|---|---|---|---|---|
| **Kinosternidae (2 species)** | | | | | | | | | | | | | | | | | | | | | | | | | | |
| *Kinosternon integrum* * | | | | | | | | | | | | | | + | | | | | | | | | | | | 1 |
| *Kinosternon scorpioides* | | | | + | | | | | | | | + | | | | | | | | | | | | | | 2 |
| **Staurotypidae (1 species)** | | | | | | | | | | | | | | | | | | | | | | | | | | |
| *Staurotypus triporcatus* | | | | + | | | | | | | | | | | | | | | | | | | | | | 1 |
| **Testudinidae (1 species)** | | | | | | | | | | | | | | | | | | | | | | | | | | |
| *Gopherus morafkai* | | | | | | | | | | | | | | | | | | | | | | + | | | | 1 |
| **Totals (219 species)** | 4 | 8 | 4 | 39 | 5 | 20 | 13 | 1 | 10 | 10 | 7 | 35 | 36 | 2 | 30 | 8 | 10 | 10 | 12 | 37 | 4 | 21 | 22 | 10 | 98 | - |

**Table 24.** Summary of the distributional status of the herpetofaunal species in protected areas in the Mexican insular systems (except for marine species). Totals = total number of species recorded in all of the listed protected areas. Non-endemic excludes the non-native species. *Boa imperator* and *Ctenosaura pectinata*, although natives to Mexico, are introduced to Cozumel and Revillagigedo, respectively, thus in these rows are enumerated as non-natives. Several species in the Gulf of California are translocated, but native to the physiographic region, not listed as non-native; see their respective entries for detailed discussion.

| Protected Areas | Number of Species | Distributional Status | | | |
| --- | --- | --- | --- | --- | --- |
| | | Non-Endemic (NE) | Country Endemic (CE) | Insular Endemic (IE) | Non-Native (NN) |
| Laguna Madre y Delta del Río Bravo | 4 | 4 | - | - | - |
| Sistema Arrecifal Lobos—Tuxpan | 4 | 4 | - | - | - |
| Sistema Arrecifal Veracruzano | 8 | 8 | - | - | - |
| Laguna de Términos | 42 | 38 | 1 | - | 3 |
| Arrefice Alacranes | 6 | 5 | - | - | 1 |
| Yum Balam | 22 | 17 | 3 | - | 2 |
| Contoy | 15 | 11 | 2 | - | 2 |
| Sian Ka'an | 1 | - | - | - | 1 |
| Banco Chinchorro | 13 | 10 | - | - | 3 |
| Santuario del Manatí Bahia de Chetumal | 11 | 10 | - | - | 1 |
| Costa Occidental de Isla Mujeres, Punta Cancún y Punta Nizuc | 7 | 7 | - | - | - |
| Cozumel complex | 41 | 33 | 2 | - | 6 |
| Islas del Pacífico de Baja California | 36 | 24 | 8 | 4 | - |
| El Vizcaíno | 2 | 2 | - | - | - |
| Islas Marías | 30 | 16 | 13 | 1 | - |
| Isabel | 11 | 3 | 3 | 2 | 3 |
| Archipielago revillagigedo | 12 | 6 | - | 4 | 2 |
| Islas Marietas | 10 | 6 | 3 | 1 | - |
| Islas La Pajarera, Cocinas, Mamut, […], situados en la Bahía de Chamela | 13 | 7 | 5 | - | 1 |
| Bahía de Loreto | 37 | 21 | 6 | 10 | - |
| San Pedro Mártir | 4 | 2 | - | 2 | - |
| Bahía de los Ángeles, canales de Ballenas y de Salsipuedes | 21 | 13 | 2 | 6 | - |
| Archipielago de Espíritu Santo | 22 | 14 | 6 | 2 | - |
| San Lorenzo | 10 | 5 | 2 | 3 | - |
| Islas del Golfo de Baja California | 98 | 42 | 13 | 43 | - |
| Totals | 219 | 117 | 46 | 56 | 7 |

## 4. Conclusions

A.　The herpetofauna of the Mexican insular systems consists of 226 species, including 16 amphibians, of which 14 are anurans and 2 are salamanders, and 210 reptiles, of which 2 are crocodilians, 195 are squamates, and 13 are turtles. This is the 16.2% of the Mexico's documented herpetofauna of 1397 species. These 226 species are arranged into 40 families and 95 genera. Of the 226 insular species, 118 (52.2%) are non-endemic species, 56 (24.8%) are insular endemics (Like the Nolasco Spiny Tailed Iguana, Figure 45), 45 (19.9%) are country endemics, and 7 (3.1%) are non-native species.

B.　Some genera appear to contain species complexes, such as *Phyllodactylus*, *Uta*, and *Hypsiglena*. Molecular revision of these insular populations could uncover new

　　　species, such as that which has happened recently with some insular populations of *Crotalus*. Additionally, although we do not recognize any infraspecific taxon, we are aware that some of them could constitute separate evolutionary lineages. Thus, it is very likely that the degree of endemism could rise if some of these lineages are recognized as full species based on new taxonomic revisions.

C.　We documented the herpetofauna of 141 insular elements in Mexico, but also detected areas with insular elements in which the composition of herpetofauna is unknown, such as the islands of the Bahias de Huatulco, in the Laguna Madre/Río Bravo basin in Tamaulipas, as well as sites where the herpetofaunal composition is not available for the insular elements, such as is the case of the islands of the Bahia de Chamela. Surveys must be carried in those sites in order to understand their herpetofaunal composition.

D.　Regarding the distribution of the Mexican insular herpetofauna within each of the five physiographic regions we recognized. The 17 islands of the Pacific Baja California physiographic region support from 1 to 19 species, the largest number of which is found on Isla Santa Margarita. The 70 islands of the Gulf of California physiographic region contain from 1 to 43 species, the greatest number of which lie on Isla Tiburón. The 16 islands of the Tropical Pacific physiographic region are occupied by from 1 to 22 species, with the greatest number found on Isla María Madre. The seven insular systems in the Gulf of Mexico host from 1 to 42 species, with the greatest value in the islands of Laguna de Términos (Isla del Carmen and Isla Arena). On the side of the Mexican Caribbean shore, we considered seven insular systems, unsurprisingly, Cozumel is the most herpetofaunistical-diverse island in the region, with 39 confirmed species, while the herpetofauna of the cays within Sian Ka'an is almost all unknown.

E.　Of the seven non-native species occurring on the Mexican islands, one is an anuran (*Eleutherodactylus planirostris*), and six are reptiles, including two anoles (*Anolis allisoni* and *Norops sagrei*), three geckos (*Gehyra mutilata*, *Hemidactylus frenatus* and *H. turcicus*), and one blindsnake (*Indotyphlops braminus*). In addition, some species are native to the country but alien to the Mexican islands, such as *Boa imperator* to Cozumel and, perhaps Banco Chinchorro and Isla Venados, and *C. pectinata* for Clarion.

F.　Our comparison of the distributional categorizations and the physiographic regional categorizations demonstrates that the greatest proportion of species in each physiographic region are non-endemics. Country endemic species are primarily represented on the Pacific side of Mexico, as opposed to those on the Atlantic side. The regional or insular endemics are mostly represented on the islands of the Gulf of California and not at all in the two Atlantic regions. The non-native species are not distributed on the two peninsular-associated regions, but rather occur to some extent in each of the three other regions.

G.　A comparison of the SEMARNAT, IUCN, and EVS systems indicates important discrepancies in assessment of the number of insular herpetofaunal species at risk. We believe that the SEMARNAT system underestimates the threat of herpetofaunal insular diversity, since it only lists eight insular species as "endangered," seven of those being turtles, but ignores an important number of insular endemic squamates, many of which are only known from the type locality and, in the case of *Lampropeltis catalinensis*, only from the holotype.

H.　The IUCN Red List, although considered as the standard system in global conservation status assessment, is of limited value for use with the Mexican insular herpetofauna, since only 21 species are placed in one of the three risk categories. As with the SEMARNAT system, seven of these species are turtles. As a result, 141 species are allocated to the Least Concern category and 6 species to the Near Threatened category; 56 species are not evaluated or are judged to be Data Deficient.

We believe that these omissions are driven primarily by the limited information available on species in insular environments.

I. The EVS system is more useful and practical for assessing the conservation risk for insular herpetofauna; it has easily allowed the assessment of threat level for species whose ecology and/or population numbers are poorly known. The results indicated that 56 species lie in the low vulnerability category, 62 in the medium category, and 94 in the high category, with the numbers increasing from the low, through the medium, to the high category. Simple calculations of the number of high EVS species would allow conservation managers to identify which islands are a priority for the conservation of the herpetofauna.

J. It is imperative to include the insular endemics into the threat levels of the NOM-059 and in the IUCN Red List, with particular emphasis on those species whose only known population is at the type locality.

K. The use of both of the Relative Herpetofauna Priority measures indicates that the islands of the Gulf of California are of the greatest conservation significance, followed in order by the Tropical Pacific islands region, the Pacific Baja California islands, the Caribbean islands, and the islands of the Gulf of Mexico.

L. The principal environmental threats for the insular herpetofauna are invasive species, urban and/or tourist development on the Caribbean islands, the risk of wildfires, which can be disastrous for insular ecosystems, and climate change as a major threat for the cays and islands of the Caribbean. Other threats include deforestation, agricultural development, cattle raising, hurricanes and other tropical storms, the oil and gas industries, public insecurity, and illegal collecting.

M. Protocols of biosecurity must be implemented in the Mexican islands in order to prevent the introduction of invasive species, with particular emphasis on rats and cats. These measures, in fact, are being carried out on some Mexican islands with various levels of efficiency, but there is still a long way to go for these measures to be widely adopted.

N. It is imperative to promote legislation that reinforces the need for government agencies to control and manage wildfires. The capacity of environmental institutions to manage wildfires must be reinforced, since a major wildfire on some islands could result in the extinction of endemic herpetofaunal species.

O. The efforts in decreeing protected areas in insular ecosystems are astonishing, with almost all of the offshore insular systems having some degree of legal protection. We did not identify any herpetofaunal insular endemic that is not represented in a protected area.

P. Mexico is very close to achieving the goal of protection of 100% of the insular elements. The creation of Natural Protected Areas on the islands of La Roqueta and Ixtapa should be encouraged; also, the island of Santa Margarita should be included in the Biosphere Reserve of Islas del Pacífico de Baja California. Efforts must be undertaken to to merge the statal reserves in Cozumel within a federal one, in order to facilitate their administration.

Q. Fortunately, most of the Mexican islands are administrated by the federal government and are unoccupied and unsettled. Also, the risk of privatization is nonexistent under current Mexican legislation. Measures for control of urban development are needed urgently for the Caribbean islands such Cozumel, Holbox, and Isla Mujeres.

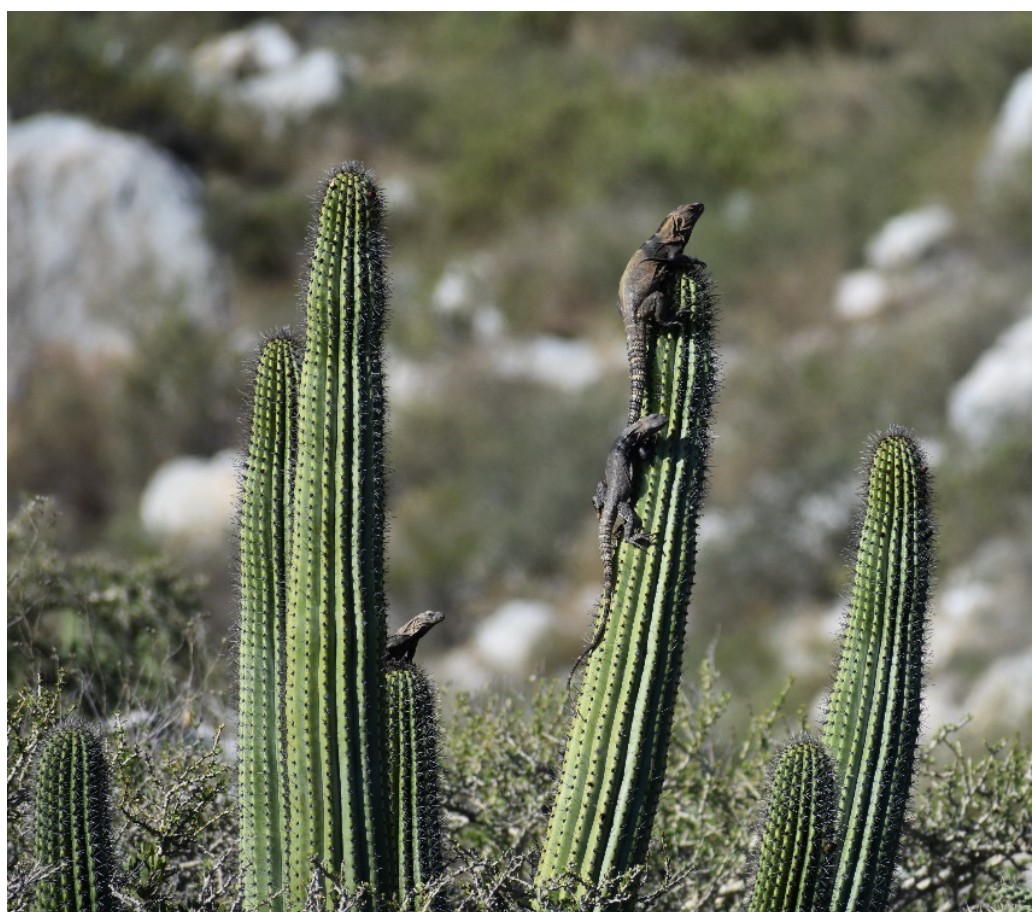

**Figure 45.** Nolasco spiny-tailed iguanas (*Ctenosaura nolascensis*) in San Pedro Nolasco (Gulf of California) taking a sunbath on top of organ pipe cacti (*Stenocereus thurberi*). The many islands of the Sea of Cortes are covered by cacti forests, the high number of endemism, not only for reptiles, but either cactus, shrubs, birds, and marine life, so each island has its own biomes characterized by their uniqueness. This makes the Gulf of California, or the Sea of Cortes, one of the most biodiverse regions of the world. Distribution status = IE, EVS = H (17), IUCN = VU, NOM-059 = NS. Photo by Fernanda Pérez-Alarcón.

**Author Contributions:** Conceptualization, V.H.G.-S.; methodology, V.H.G.-S. and J.D.J.; formal analysis, V.H.G.-S. and J.D.J.; investigation, V.H.G.-S. and J.D.J.; data curation, V.H.G.-S.; writing—original draft preparation, V.H.G.-S. and J.D.J.; writing—review and editing, C.A.N.-T., O.F.-M., L.M.M.O., A.P.-C., M.d.P.B.-P. and P.C.; supervision, C.A.N.-T. All authors have read and agreed to the published version of the manuscript.

**Funding:** This research received no external funding.

**Institutional Review Board Statement:** Not applicable.

**Data Availability Statement:** Data available through direct request to first author.

**Acknowledgments:** The lead author wishes to: Thank all my friends and colleagues that kindly shared their photographic material (in alphabetical order: Humberto Almanza, Edgar Alvarado-Rodríguez, Juan Diego Arias-Montiel, Ruben Alonso Carbajal-Márquez, Luis Díaz-Gamboa, Armando Escobedo-Galván, Bradford Hollingsworth, Brian Hubbs, Lizbeth Lara-Sánchez, Nysai Moreno, José Rafael Nolasco-Luna, Carlos Pavón-Vázquez, Fernanda Peréz-Alarcón, Tania Pérez-Fiol, Israel Sánchez-Ortega, and Rigel Sansores) and also to express my sincere apologies for not being able to use the material of some of them. All pictures were precious, and I thank you all. To Jessica Juan-Espinosa, for her help and patience checking one by one all the references and summations. To Dynorah González-Sánchez, for drawing the graphical abstract. Also, I am deeply in debt to Larry D. Wilson, who made valuable contributions, corrections, and editions to this paper.

**Conflicts of Interest:** The authors declare no conflict of interest.

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
