# Peer review of "The Herpetofauna of the Insular Systems of Mexico"

_diversity, doi:10.3390/d15080921_

Round 1

Reviewer 1 Report

This manuscript provides a review and summary of aspects of the herpetofauna found on the insular entities of Mexico. These islands constitute important habitats for Mexican amphibians and reptiles, especially the reptiles, and as such the study described in this manuscript is worthwhile even though another similar paper was published earlier by a different set of authors. Most of the comments below pertain to clarifying and improving the efficiency of presentation.

General comments:

1) There are several places in the manuscript where the authors provide extended tangents on topics I think are not necessary for advancing their “story”. It makes the manuscript much longer and harder to read than it needs to be. Examples include lines 98-120, 223-259, 407-414

2) There are numerous photographs included in the manuscript that are not cited in the text, nor are they even given a figure number (all are “figure X”). Many of these photos are not really needed and greatly add to the already lengthy manuscript.

3) I found much of the presentation of the summaries in the text quickly became tedious (and boring) to read. Is there any way these elements in the text could be more concisely and interesting presented? Perhaps figures could be used or more effective use of the tables.

4) I think some of the tables could be combined to make better use of space and provide the information effectively.

5) In the “threats” section, it would be useful to provide as much evidence that is available on the impacts of these threats on the herpetofauna of these insular elements. This is provided for some of the threat types but not all.

6) One of the key cautions of the study is outlined in lines 2387-2391. I have no problem with using information to date that is available even if it might be incomplete. However, it would be useful if some indication of how incomplete the lists for the islands might be. For example, it might be useful to examine species accumulation curves to see if they are approaching an asymptote.

7) The conclusion section is not really a conclusion but rather a summary. It would make the manuscript much more useful and valuable if conclusions were made that emphasize the value of the effort made to create these lists.

1) Throughout the manuscript there are several typos and other sloppy aspects to the presentation. For example, throughout the manuscript both numbers and author names are used in the in-text citations. Other examples include lines 125 (“Insertar la referencia”) and 1509 (“inserter cita de reporte interno”). Author contributions are not provided and Appendix A and Appendix B are listed but it is not clear if they actually exist.

2) It would be helpful if the authors had someone carefully go through the manuscript and check and improve the word choice. For example, “Remarkably” in lines 80 & 195 is not the appropriate word, nor is “abrasion” on line 147 (should be “erosion”) – there are many more examples.

Specific comments:

line 89: I think it would be useful to provide a specific definition of what you mean by “insular elements” early in the manuscript and make it clear, rather than have the definition appear later and more diffusely.

line 146: How small a portion? It would be helpful to be as precise as possible when you are defining the different insular elements.

line 154: What does “snout” mean here? I’ve never seen it used in this context.

lines 210-213: You’ve already said this multiple times. No need to repeat it again.

line 217: How do you determine these specifically, how shallow? how close is too close?

lines 273-276: How different are the taxonomies used? Is there a reason to use one taxonomy over the other beyond just personal preference?

line 284: define “INEGI”; what “ad hoc adaptations”?

lines 273-301: Several of these justifications seem trivial. Are they all needed to justify this new manuscript.

line 322: What is “GBIF”?

lines 353-357, 363-367: Are all these citations really needed here? Seems like citation padding. If you are going to cite all of these then you need to cite all the papers that make use of these methods, not just those in the “Mesoamerican Herpetology series”.

line 605: Why? Provide a better justification for this section. In addition, as with other aspects of the manuscripts this section seems overblown and needs to be much more concise.

line 857: As with the previous section, this section could be streamlined and made more concise. Too much detail is provided and is not needed in the context of the manuscript.

lines 962-963: The Mesoamerican Herpetology taxonomic list is a useful tool but is not peer-reviewed. A recent publication (possibly published after this manuscript was submitted) on the herpetofauna of Mexico has been published and should probably be cited and referred to (Ramírez-Bautista et al. 2023 Zookeys).

lines 1050-1052 and elsewhere: I don’t understand why this space-inefficient formatting is used. It seems like this information could easily be incorporated into the other existing tables. Also, no definition of the various * designations are provided.

lines 1471-1477: Citations for this?

line 1826: What does “insecurity” mean in this context?

lines 1968-1974: There are several other uncited papers that use this approach that are not part of the Mesoamerican Herpetology series.

See comments and suggestions to authors

Author Response

1) There are several places in the manuscript where the authors provide extended tangents on topics I think are not necessary for advancing their “story”. It makes the manuscript much longer and harder to read than it needs to be. Examples include lines 98-120, 223-259, 407-414

R=Removed or reduced.

2) There are numerous photographs included in the manuscript that are not cited in the text, nor are they even given a figure number (all are “figure X”). Many of these photos are not really needed and greatly add to the already lengthy manuscript.

R=Reduced,

3) I found much of the presentation of the summaries in the text quickly became tedious (and boring) to read. Is there any way these elements in the text could be more concisely and interesting presented? Perhaps figures could be used or more effective use of the tables.

R= Reduced.

4) I think some of the tables could be combined to make better use of space and provide the information effectively.

R= The way we present these tables is how we could optimize the data after several similar papers. This is the most effective way we have found to this day. We could try another approach, but in that case, it would take a major rewriting and restructuration of the paper.

5) In the “threats” section, it would be useful to provide as much evidence that is available on the impacts of these threats on the herpetofauna of these insular elements. This is provided for some of the threat types but not all.

6) One of the key cautions of the study is outlined in lines 2387-2391. I have no problem with using information to date that is available even if it might be incomplete. However, it would be useful if some indication of how incomplete the lists for the islands might be. For example, it might be useful to examine species accumulation curves to see if they are approaching an asymptote.

R=Yes, But the main focus of this paper is to address the diversity, endemism, vulnerability and threats for insular herpetofauna, the target are conservation biologists, natural resources managers and decision makers. However we are working in an upcoming paper that will deal with diversity and taxonomic index and oriented to biogeographical and diversity (alpha, beta, gamma) patterns, with other target audience in mind.

7) The conclusion section is not really a conclusion but rather a summary. It would make the manuscript much more useful and valuable if conclusions were made that emphasize the value of the effort made to create these lists.

R= Changes were made.

1) Throughout the manuscript there are several typos and other sloppy aspects to the presentation. For example, throughout the manuscript both numbers and author names are used in the in-text citations. Other examples include lines 125 (“Insertar la referencia”) and 1509 (“inserter cita de reporte interno”). Author contributions are not provided and Appendix A and Appendix B are listed but it is not clear if they actually exist.

R=Fixed

2) It would be helpful if the authors had someone carefully go through the manuscript and check and improve the word choice. For example, “Remarkably” in lines 80 & 195 is not the appropriate word, nor is “abrasion” on line 147 (should be “erosion”) – there are many more examples.

Specific comments:

line 89: I think it would be useful to provide a specific definition of what you mean by “insular elements” early in the manuscript and make it clear, rather than have the definition appear later and more diffusely.

line 146: How small a portion? It would be helpful to be as precise as possible when you are defining the different insular elements.

R= That’s a subjective definition, there is not standard definition of what a islet is (i.e. “any emerged land <1km2). A rock may be an island or islet, depending of the point of view. In this case is synonymous of “very Small Island”, often unnamed. But a technical, scientific or standard definition is inexistent in any scientific literature.

line 154: What does “snout” mean here? I’ve never seen it used in this context.

R= a vertical, prominent rock above the sea level. Since the term is barely used in the ms, we removed it.

lines 210-213: You’ve already said this multiple times. No need to repeat it again.

R= Ok. Removed-

line 217: How do you determine these specifically, how shallow? how close is too close?

R= There is not stablished definition of this. But in the examples provided the separation is less than 200m. This is subjective, since we refer to islets whose identity as true “independent” islands is neglected. Is the case of the providen examples; Cozumel is a very large island, La Pasión is a very small “island”, few meters separated from Cozumel, although separated by a shallow water channel from Cozumel, it’s clear that La Pasion is part of Cozumel. Other cases are islets associated to a larger island, in which one can say that those islets are just an extension of the large island. However, trying to provide a concise definition of this would require a very large explanation that is matter of a geology paper.

lines 273-276: How different are the taxonomies used? Is there a reason to use one taxonomy over the other beyond just personal preference?

R=The reason is the subspecies concept. However the Ramirez-Bautista parper (published after we submitted this paper) is highly concordant with the list of Mesoamerican herpetology.

line 284: define “INEGI”; what “ad hoc adaptations”?

R=Defined

lines 273-301: Several of these justifications seem trivial. Are they all needed to justify this new manuscript.

R=It’s necessary since it has been hard to explain why this paper is different to Pliego Snachez et al. Although we take into account your comment and reduced, the justificacion.

line 322: What is “GBIF”?

R=Global Biodiversity Information Facility (Defined).

lines 353-357, 363-367: Are all these citations really needed here? Seems like citation padding. If you are going to cite all of these then you need to cite all the papers that make use of these methods, not just those in the “Mesoamerican Herpetology series”.

R=Resumed, but these references are mentioned in several other parts of this paper.

line 605: Why? Provide a better justification for this section. In addition, as with other aspects of the manuscripts this section seems overblown and needs to be much more concise.

R= At the moment of writing this paper, we found many contradictions in different works, also some comments should be made regarding if a particular species is introduced or not, and so on. We believe it’s important to state why certain decisions were made.

line 857: As with the previous section, this section could be streamlined and made more concise. Too much detail is provided and is not needed in the context of the manuscript.

R=several islands have synonymies, even in scientific literature and/or collect records in zoological collections you can find these islands with different names, This is why is clarified in detail the most important names we use. I would really appreciate if you allow us to keep this section. Since this point is extremely confusing for any student or researcher who approaches the study of the Mexican insular systems for the first time. The nomenclature of the islands varies between authors and between periods, and makes any biogeographical investigation of these regions very difficult. We spent a lot of time solving this puzzle, so it is intended to save new generations days or weeks of work trying to solve the identity of certain islands. This two pages of text will save days of work to other students.

lines 962-963: The Mesoamerican Herpetology taxonomic list is a useful tool but is not peer-reviewed. A recent publication (possibly published after this manuscript was submitted) on the herpetofauna of Mexico has been published and should probably be cited and referred to (Ramírez-Bautista et al. 2023 Zookeys).

lines 1050-1052 and elsewhere: I don’t understand why this space-inefficient formatting is used. It seems like this information could easily be incorporated into the other existing tables. Also, no definition of the various * designations are provided.

lines 1471-1477: Citations for this?

Fixed.

line 1826: What does “insecurity” mean in this context?

R=Criminality, this is clear for any latinoamerican reader. However, the title of this subsection is changed, since readers from other countries may understand this word differently.

lines 1968-1974: There are several other uncited papers that use this approach that are not part of the Mesoamerican Herpetology series.

R=Rewritted

PS= We are aware of the in-text citations (example, year), we'll change that whe submiting the final version, since it's the necesary in this moment to find the citation in the EndNote. 

Reviewer 2 Report

The authors present relevant, interesting material that needs to be published. It is particularly necessary to analyze the existing environmental registers (SEMARNAT (NOM-059), the IUCN Red List and the Environmental Vulnerability Assessment (EVS). 

An interesting fact is that the invasive plant Mesembrianthemum cristallynum is a refuge for Aniella geronimensis. And also that the number of milk snakes can be an indicator of the trophic press of alien mammals, the black rat (Rattus rattus) and wild cats (Felis silvestris catus). I think that the authors need to supplement the information on individual islands or districts.

At the same time, I think that an additional role should be given:

1. Analysis and inventory of the taxonomic composition of herpetofauna by molecular genetic methods

2. Analysis of threats to diversity – taking into account the risks of population decline for each species of herpetofauna, including the impact of introduced species, the risk of forest fires, climate change and urban/tourism development.

1. Remark lines 950-952 "Except for Incilius and Eleutherodactylus, with two species each, all the other amphibian genera are monospecific for the islands of Mexico (however, one of the two species of Eleutherodactylus, E. planirostris, is an introduced species). Which one, please give more detailed information or make a separate section for Eleutherodactylus planirostris, taking into account the presence of the Eleutherodactylus pallidus species endemic to Mexico. And also with an assessment of the possible competition of these species and eels for an endemic species? I note that this information is given (line 1347) in the caption to Figure FigX. ).

2. For this review, detailed information is needed on the introduction of Ctenosaura pectinata to Clarion Island

I also think that it will improve the perception and understanding of the manuscript

- Statistical analysis of the diversity of herpetofauna, with the identification of islands with the highest proportion of endemic herpetofauna.

Analysis of the most similar herpetofauna of the islands 

To consider in more detail the concept of fauna formation taking into account the influence (duration) of isolation factors.

Author Response

At the same time, I think that an additional role should be given:

  1. Analysis and inventory of the taxonomic composition of herpetofauna by molecular genetic methods

R=Agree, but that’s beyond the scope of this paper which its main goal is to address the status of knowledge and conservation of the Mexican insular herpetofauna. However, an upcoming paper will discuss this matter.

  1. Analysis of threats to diversity – taking into account the risks of population decline for each species of herpetofauna, including the impact of introduced species, the risk of forest fires, climate change and urban/tourism development.
  2. Remark lines 950-952 "Except for Incilius and Eleutherodactylus, with two species each, all the other amphibian genera are monospecific for the islands of Mexico (however, one of the two species of Eleutherodactylus, E. planirostris, is an introduced species). Which one, please give more detailed information or make a separate section for Eleutherodactylus planirostris, taking into account the presence of the Eleutherodactylus pallidus species endemic to Mexico. And also with an assessment of the possible competition of these species and eels for an endemic species? I note that this information is given (line 1347) in the caption to Figure FigX. ).
  3. For this review, detailed information is needed on the introduction of Ctenosaura pectinata to Clarion Island

R= Clarified, also that is poined in the refered paper.

I also think that it will improve the perception and understanding of the manuscript

- Statistical analysis of the diversity of herpetofauna, with the identification of islands with the highest proportion of endemic herpetofauna.

Analysis of the most similar herpetofauna of the islands 

To consider in more detail the concept of fauna formation taking into account the influence (duration) of isolation factors.

R=Yes, that is part of another upcoming paper, oriented to diversity metrics.

Round 2

Reviewer 2 Report

The reviewer's comments are taken into account. The work on the article is visible, but even a cursory review reveals flaws and typos.

1. The signature is "Fig. 2" in Figure 12, and 11 points are indicated in the signature.

2. It is required to improve the design "Table 2. Distribution of the insular herpetofauna of Mexico by physiographic region. * = country endemic; ** regional endemic; *** = non-native." For example, instead of "+" fill the cell, there are empty lines "Amphibia (16 species)" - specify the number of species in the cells"

3. The isolation factor is described, but without specifying the distances between the islands, as well as the presence of shallow areas.

I recommend the manuscript for publication after removing the comments.

Author Response

Corrected typos, numerations, legends, signatures and desing of tables. Also reduced paragraphs and rearranged of text. which reduced the lenght of the ms by 20 pages less.
